



# The Multi-Compartment Hg Modeling and Analysis Project (MCHgMAP): Mercury modeling to support international environmental policy

Ashu Dastoor[1], Hélène Angot[2], Johannes Bieser[3], Flora Brocza[4,5], Brock Edwards[6], Aryeh Feinberg[7,8], Xinbin Feng[9,10], Benjamin Geyman[11], Charikleia Gournia[12], Yipeng He[13], Ian M. Hedgecock[14], Ilia Ilyin[15], Terry Keating[16], Jane Kirk[17], Che-Jen Lin[18,19], Igor Lehnherr[20], Robert Mason[21], David McLagan[22,23], Marilena Muntean[24], Peter Rafaj[4], Eric M. Roy[25], Andrei Ryjkov[1], Noelle E. Selin[25,26], Francesco De Simone[14], Anne L. Soerensen[27], Frits Steenhuisen[28], Oleg Travnikov[29], Shuxiao Wang[30,31], Xun Wang[9], Simon Wilson[32], Rosa Wu[33], Qingru Wu[30,31], Yanxu Zhang[34], Jun Zhou[35], Wei Zhu[36], Scott Zolkos[37]

[1]Air Quality Research Division, Environment and Climate Change Canada, Dorval, Quebec, Canada

[2]Univ. Grenoble Alpes, CNRS, INRAE, IRD, Grenoble INP, IGE, Grenoble, France

[3]Institute of Coastal Systems – Analysis and Modeling, Helmholtz- Zentrum Hereon, Geesthacht, Germany

[4]Energy, Climate and Environment Program, International Institute for Applied Systems Analysis, Laxenburg, Austria

[5]School of Chemical and Process Engineering, University of Leeds, United Kingdom

[6]Centre for Earth Observation Science, Department of Environment and Geography, University of Manitoba, Winnipeg, Manitoba, Canada

[7]Institute for Data, Systems, and Society, Massachusetts Institute of Technology, Cambridge, MA, USA

[8]Department of Atmospheric Chemistry and Climate, Institute of Physical Chemistry Blas Cabrera, CSIC, Madrid, Spain

[9]State Key Laboratory of Environmental Geochemistry, Institute of Geochemistry, Chinese Academy of Sciences, Guiyang, China

[10]University of Chinese Academy of Sciences, Beijing, China

[11]Harvard John A. Paulson School of Engineering and Applied Sciences, Cambridge, MA, USA

[12]Jožef Stefan International Postgraduate School, Ljubljana, Slovenia

[13]Applied Research Center, Florida International University, Miami, FL, USA

[14]CNR-Institute of Atmospheric Pollution Research, Rende, Italy

[15]Ecological Synthesizing Centre - East, Moscow, Russia

[16]U.S. Environmental Protection Agency, Washington DC, USA

[17]Aquatic Contaminants Research Division, Environment and Climate Change Canada, Burlington, Ontario, Canada

[18]Center for Advances in Water and Air Quality, Lamar University, Beaumont, TX, USA

[19]Department of Civil and Environmental Engineering, Lamar University, Beaumont, TX, USA

[20]Department of Geography, Geomatics and Environment, University of Toronto Mississauga, Mississauga, Ontario, Canada

[21]Department of Marine Sciences, University of Connecticut, Groton, CT, USA

[22]Department of Geological Sciences and Geological Engineering, Queen's University, Kingston, Ontario, Canada



[23]School of Environmental Studies, Queen's University, Kingston, Ontario, Canada

[24]European Commission, Joint Research Centre (JRC), Ispra, VA, Italy

[25]Department of Earth, Atmospheric and Planetary Sciences, Massachusetts Institute of Technology, Cambridge, MA, USA

[26]Institute for Data, Systems, and Society, Massachusetts Institute of Technology, Cambridge, MA, USA

[27]Department of Environmental Monitoring and Research, Swedish Museum of Natural History, Stockholm, Sweden

[28]Arctic Centre, University of Groningen, Groningen, the Netherlands

[29]Department of Environmental Sciences, Jožef Stefan Institute, Ljubljana, Slovenia

[30]State Key Joint Laboratory of Environment Simulation and Pollution Control School of Environment, Tsinghua University, Beijing, China

[31]State Environmental Protection Key Laboratory of Sources and Control of Air Pollution Complex, Beijing, China.

[32]Arctic Monitoring and Assessment Programme Secretariat, Tromsø, Norway

[33]Air Quality Research Division, Environment and Climate Change Canada, Toronto, Ontario, Canada

[34]School of Atmospheric Sciences, Nanjing University, Nanjing, China

[35]State Key Laboratory of Soil and Sustainable Agriculture, Institute of Soil Science, Chinese Academy of Sciences, Nanjing, China

[36]Department of Forest Ecology and Management, Swedish University of Agricultural Sciences, Umeå, Sweden

[37]Woodwell Climate Research Center, Falmouth, MA, USA

*Correspondence to*: Ashu Dastoor (ashu.dastoor@ec.gc.ca)

**Abstract.** The Multi-Compartment Hg (mercury) Modeling and Analysis Project (MCHgMAP) is an international multi-model research initiative intended to simulate and analyze the geospatial distributions and temporal trends of environmental
Hg to inform the effectiveness evaluations of two multilateral environmental agreements (MEAs): the Minamata Convention on Mercury (MC) and Convention on Long-Range Transboundary Air Pollution (LRTAP). This MCHgMAP overview paper presents its science objectives, background and rationale, experimental design (multi-model ensemble (MME) architecture, inputs and evaluation data, simulations and reporting framework), and methodologies for the evaluation and analysis of simulated environmental Hg levels. The primary goals of the project are to facilitate detection and attribution of recent
(observed) and future (projected) spatial patterns and temporal trends of global environmental Hg levels, and identification of key knowledge gaps in Hg science and modeling to improve future effectiveness evaluation cycles of the MEAs. The current advances and challenges of Hg models, emission inventories, and observational data are examined, and an optimized multi-model experimental design is introduced for addressing the key policy questions of the MEAs. A common set of emissions, environmental conditions, and observation datasets are proposed (where possible) to enhance the MME
comparability. A novel harmonized simulation approach between atmospheric, land, oceanic and multi-media models is developed to account for the short- and long-term changes in secondary Hg exchanges and to achieve mechanistic consistency of Hg levels across environmental matrices. A comprehensive set of model experiments is developed and prioritized to ensure a systematic analysis and participation of a variety of models from the scientific community.



## 1. Introduction

The presence and levels of mercury (Hg) in environmental matrices of concern are associated with its primary atmospheric emissions and land releases (from anthropogenic and geogenic sources) and environmental residence time. The latter is determined by physical and biochemical processes that govern the chemical and phase transformations and transport of Hg species in and across environmental matrices. Several recent studies have presented quantitative assessment of global Hg cycling (i.e., emissions and releases, concentrations, and exchange fluxes) using models and observations (e.g., Outridge et

al., 2018; Zhou et al., 2021; Jiskra et al., 2021; Feinberg et al., 2022; Sonke et al. 2023). Mercury emission mitigation policies reduce Hg levels directly and via slowing the build-up of legacy Hg in soils and oceans (Angot et al., 2018; Amos et al., 2013). The effects of changing climate (AMAP/ UN Environnement, 2019; Box et al., 2019; Saros et al., 2019; Sonke et al., 2023), environmental chemical composition (e.g., Parrella et al., 2012) and land use and land cover (Zhang et al., 2016a; Feinberg et al., 2023) on Hg cycling are, however, more complex because of interactive alterations of multiple Hg processes

and remobilization of legacy mercury (MacMillan et al., 2015; Yang et al., 2016; Chételat et al., 2022; Zhang, 2021). Moreover, climate change impacts can vary significantly across the globe (Chételat et al., 2022; Dastoor et al., 2022a, b; Wang et al., 2020b), due to regional differences in warming. Considerations of the effects of these changes on environmental Hg cycling are warranted to assess the effectiveness of emission mitigation policies on Hg levels. Changes in observed Hg levels exhibit combined influences of changes in multiple factors altering Hg cycling, and suitable methods are required to

isolate the impacts attributable to the emission policies from changes occurring due to other factors. The separate quantification of anthropogenic and natural contributions to observed changes in Hg levels is important both for understanding past changes in Hg levels and for constraining projections of future Hg cycling.

The 3D single- or multi-media mechanistic Hg models simulate spatiotemporally resolved environmental Hg concentrations and interfacial exchange fluxes via explicitly representing intervening processes, thus allowing direct quantitative attribution

of observed Hg levels and trends to emission sources and other drivers (UNEP, 2019; Zhang et al., 2016b). Furthermore, multi-media mass balance Hg models (Qureshi et al., 2011; Selin, 2014; Amos et al., 2013, 2014; Soerensen et al., 2016a) can be utilized to gain insight into the long-term fate of anthropogenic Hg emissions and releases in the biosphere reflected in the Hg records from environmental natural archives (Amos et al., 2015). To derive information from models for interpreting Hg monitoring data, reliable spatiotemporally varying anthropogenic emissions inventories of Hg species from

global sources are needed (Outridge et al., 2018). Equally important are the accuracy of process representations in models, and observations needed for their development and evaluation. Advances in monitoring of Hg levels (atmosphere: Aas and Bohlin-Nizzetto, 2019; Sprovieri et al., 2016; terrestrial surfaces: Lim et al., 2020; Zhou et al., 2021; Wang et al., 2019; oceans: Bowman et al., 2019; Liu et al., 2020) and fluxes (air-vegetation-soil: Zhou and Obrist, 2021; Zhou et al., 2021; Gerson et al., 2022; Schneider et al., 2023; air-cryosphere: Steffen et al., 2021; freshwater-ocean: Liu et al., 2021b; Zolkos et

al., 2022; air-ocean: DiMento et al., 2019; Osterwalder et al., 2021) and biochemical processes (Saiz-Lopez et al., 2020; Castro et al., 2022) continue to improve Hg models (atmosphere: Angot et al., 2016; Zhou et al., 2021; Shah et al., 2021;





Feinberg et al., 2022; ocean: Rosati et al., 2022; Bieser et al., 2023; Zhang et al., 2023a). Studies have shown that application of multiple models can increase the robustness of modeling results compared to the use of single models (Travnikov et al., 2017; Bieser et al., 2017; AMAP/ UN Environnement, 2019). Atmospheric multi-model ensemble (MME) simulations

together with field observations, for example, led to a comprehensive assessment of contemporary Arctic Hg levels, and its temporal changes, attribution and future projections (Dastoor et al., 2022a, b; Schartup et al., 2022).

The *2013 Minamata Convention on Mercury* (MC)*, a global multilateral environmental agreement (MEA)* to protect human health and the environment from Hg pollution, requires the party nations to reduce anthropogenic Hg emissions from point sources such as coal-fired power plants and certain non-ferrous metals production operations, and from the intentional use of

Hg in artisanal and small-scale gold mining (ASGM) and other industrial processes, as well as in products. The MC further requires the Parties to periodically evaluate its effectiveness "on the basis of available scientific, environmental, technical, financial, and economic information", including available data on "the presence and movement of mercury and mercury compounds in the environment as well as trends in levels of mercury and mercury compounds observed in biotic media and vulnerable populations" (MC Article 22). The effectiveness evaluation (EE) is intended to address four overarching policy

questions:

1.  Have the Parties taken actions to implement the Minamata Convention?

2.  Have the actions taken resulted in changes in mercury supply, use, emissions, and releases into the environment?

3.  Have those changes resulted in changes in levels of mercury in the environment, biotic media and vulnerable populations that can be attributed to the Minamata Convention?

4.  To what extent are existing measures under the Minamata Convention meeting the objective of protecting human health and the environment from mercury?

In 2021, the 4[th] Conference of the Parties adopted a framework for conducting the first EE (Decision MC-4/11, UNEP/MC/COP.4/28/Add.1), including creating an Open-Ended Science Group (OESG) to prepare a scientific report primarily addressing the last two overarching questions presented above. The OESG is charged with compiling and

analyzing available data to address a series of guiding questions that were identified in the "*Guidance on monitoring of mercury and mercury compounds to support evaluation of the effectiveness of the Minamata Convention*" (UNEP/MC/COP.4/INF/12). These guiding questions address the assessment of spatial patterns, temporal trends, exposures, and adverse impacts in environmental matrices, biota, and humans, and their attribution to sources and environmental processes, through the integration and analysis of observations using statistical techniques and mechanistic models. These

guiding questions have been mapped to data analysis questions in the OESG's draft data analysis plan (UNEP/MC/COP.4/INF/24; https://minamataconvention.org/en/intersessional-work-and-submissions-cop-5#sec1565). The OESG is to provide its report to an Effectiveness Evaluation Group who will provide findings and recommendations to the 7[th] Conference of the Parties (in 2027) (Decision 5/14, UNEP/MC/COP.5/25/Add.1).



The *1998 Heavy Metals Protocol* of the *1979 Convention on Long-Range Transboundary Air Pollution* (LRTAP), a regional
MEA, commits Parties to mitigate emissions of mercury (as well as cadmium and lead) from a variety of point sources and
provides guidance on mitigating emissions associated with heavy metal use in manufactured products. Article 10 of the
Protocol requires LRTAP's Executive Body to review periodically the progress towards meeting the obligations in the
Protocol and the sufficiency and effectiveness of those obligations and to evaluate whether additional emission reductions
are warranted. In 2012, the Executive Body's review led to a revision of the *Heavy Metals Protocol*, which included more
stringent emissions controls and updated guidance on best available technologies.

The effectiveness evaluations of both the MC and LRTAP entail assessing the significance of anthropogenic sources
influenced by the MEAs on Hg levels in the atmosphere and the receiving terrestrial and aquatic environments over time
compared to total anthropogenic, geogenic and secondary emissions and releases. This overview paper describes the science
background and experimental design of a new MME initiative, Multi-Compartment Hg Modeling and Analysis Project
(MCHgMAP), to simulate and analyze environmental Hg levels (concentrations and exchange fluxes) using state-of-the-
science mechanistic 3D atmospheric, land, oceanic and multi-media mass balance models for informing effectiveness of the
MC and LRTAP. The initiative supports the analysis of atmospheric and marine monitoring data by causally linking the
spatial distributions and temporal changes in environmental Hg levels to their drivers using mechanistic (or process-based)
models, thereby contributing to the first cycle of the MC EE as well as laying the groundwork for subsequent studies. This
paper has three goals: (a) to define the current state of process-based Hg models and their drivers (emissions; meteorological,
geophysical and biochemical conditions) and evaluation data (observations); (b) to discuss the challenges and selection of
models and their drivers and evaluation data; and (c) to present a strategy for MME simulations and analysis to conduct EEs
of the MEAs and identify key knowledge gaps to strengthen future EE cycles.

Appendix A provides a detailed list of guiding questions for the MCHgMAP activities, developed to address the policy
questions pertaining to environmental Hg in the MC OESG data analysis plan: (1) What is the fractional contribution of
contemporary anthropogenic emissions and releases to current Hg levels observed in air, biota, humans, and other media?,
(2) How have these contribution levels changed over the timeline of the MEA?, (3) How do the contribution levels and their
trends vary geographically at the global scale?, and (4) How have drivers other than changes in emissions and releases
contributed to the trend in observed Hg levels? A novel feature of the proposed MME design is an approach to couple
atmospheric, land, marine and multi-media mass balance models to perform a coordinated simulation of the Hg cycling
across environmental compartments using harmonized model drivers to capture the impact of changing secondary Hg
emissions and releases in attributing environmental Hg trends. The MCHgMAP activities do not address the EE objective -
*Estimation of exposure and adverse impacts*. However, the spatially distributed models, which simulate atmospheric and
marine Hg concentrations and fluxes, will provide valuable data that can be used to link Hg exposure and impacts to
emission sources (e.g., Giang and Selin, 2016; Zhang et al., 2021).

The overview topics are organized in the following order: Hg models and their drivers (Sect. 2); sources (Sect. 3);
observations (Sect. 4); MME simulation design (Sect. 5); model evaluation plan (Sect. 6); modeling analysis to inform EE





(Sect. 7); modeling analysis to improve Hg science and drivers (Sect. 8); modeling future scenarios (Sect. 9); and summary (Sect. 10). The sections are structured to include topic relevance, literature review, challenges, plans for the study, and current limitations and recommendations for future.

## 2. Models

A variety of mechanistic and empirical Hg models have been developed to trace the movement and concentrations of Hg in abiotic and biotic environments upon its primary emissions and releases to the atmosphere and land. These include large scale atmospheric and oceanic models, terrestrial-hydrological models, multi-media compartmental mass balance models, and watershed-scale or site-specific food web, bioaccumulation, and risk exposure models. These models are briefly discussed below including their drivers, applicability, and limitations.

The primary drivers of Hg cycling in surface environments are anthropogenic and geogenic emissions and releases and environmental (physical, chemical, and biological) conditions. The drivers for different types of Hg models are defined here as basic or derived quantities that induce variability or trend in Hg levels of the environmental system considered in the model. In a specific model, a given Hg driver might be treated as an external or internal forcing, depending on the model purpose and the environmental compartments and processes represented by the model.

### 2.1. Atmospheric models

Atmospheric models facilitate understanding of the main mechanisms governing Hg dispersion and cycling in the atmosphere (Travnikov et al., 2017). By tracing the link from emissions to deposition of Hg onto environmental surfaces, atmospheric models are used to evaluate the effectiveness of mitigation policies (e.g., Angot et al., 2018; Mulvaney et al., 2020).

Global 3D atmospheric models for Hg were developed in the late 1990s, including the GISS-CTM (Shia et al., 1999), GEOS-Chem (Selin et al., 2007), GRAHM (Dastoor and Larocque, 2004) and its later development GEM-MACH-Hg (Dastoor et al., 2021; Zhou et al., 2021), GLEMOS (Travnikov and Ilyin, 2009), ECHMERIT (Jung et al., 2009), and CAM-Chem/Hg (Lei et al., 2013) and its later development CAM6-Chem/Hg v1.0 (Zhang and Zhang, 2022), and WACCM (Saiz-Lopez et al., 2022). These atmospheric models (see Table F1) are built upon numerical models used to simulate climate, meteorology, and air quality.

Table F5 provides an overview of model drivers and their suggested setup for the MME. The models are driven by model-specific online or offline meteorological data. Chemical data (e.g., oxidant concentrations, photolysis rates) are needed to drive the Hg redox mechanisms and speciation (i.e., conversion between the different Hg species (Hg(0), Hg(II)$_g$, Hg(II)$_p$)). Finally, different types of emissions are needed: 1) natural emissions (geogenic), 2) anthropogenic emissions, and 3) secondary emissions (reemissions from land and oceans, and wildfires). Some of these drivers can be harmonized for the multi-model exercise; these are highlighted in Table F5 and described in Sect. 3.



All the aforementioned models simulate Hg(0), Hg(II)$_g$, and Hg(II)$_p$ concentrations (i.e., Hg speciation) along with wet and dry deposition of these species. Table F1 gives an overview of existing global and regional 3D atmospheric Hg models and highlights the main differences in parameterizations. The largest difference among models relates to the parameterization of atmospheric chemistry and surface-atmosphere gas exchange. While most 3D atmospheric models use prescribed emission fluxes or formulations based on prescribed surface concentrations of Hg from land and oceans (see Fig. 1), some models have attempted the use of 2D surface-slab ocean and 2D terrestrial reservoir (e.g., GEOS-Chem), or even fully coupled atmospheric and ocean components (Zhang et al., 2019c). All models use a similar resistance-in-series dry deposition approach but with different land cover types, which may affect dry deposition fluxes. Dry deposition velocities over various surface types are estimated through the resistance analogy including aerodynamic, soil, stomatal, and cuticle resistances (Wesely, 1989; Zhang et al., 2003, 2009). Simulated Hg exchange fluxes in canopy and underlying soils are highly sensitive to uncertain resistance parameters (Rutter et al., 2011; Zhang et al., 2019b). For example, based on micrometeorological measurements of Hg(0) fluxes, Khan et al., (2019) recommended that models increase resistances to reduce stomatal uptake of Hg(0) over grassland and tundra by a factor of 5–7 and increase ground and cuticular uptake by factors of 3–4 and 2–4, respectively. Feinberg et al., (2022) later recommended a higher dry deposition velocity for Hg(0) for the rainforest land category. Terrestrial Hg(0) evasion is parameterized empirically as a function of environmental conditions (i.e., temperature, solar irradiance, leaf area index) and legacy soil Hg content, and include a fraction of recently deposited Hg to soils/vegetation/snow as prompt reemission (Selin et al., 2008). Some regional models include mechanistic bidirectional atmosphere–terrestrial Hg(0) gas exchange parameterization (e.g., Bash, 2010; Wang et al., 2016c; see Sect. 3.4). The wet deposition of soluble Hg species typically includes in-cloud and below-cloud scavenging. The wet deposition flux is mostly driven by the atmospheric concentration of Hg(II) and precipitation amount (Travnikov et al., 2017). This flux is thus expected to differ among models as they use differing Hg oxidation schemes and input meteorological data. Gas exchange from the ocean surface to the atmosphere is typically parameterized based on the air-sea concentration gradient (prescribed or calculated online) and a simplified model-specific wind-speed dependent Hg transfer velocity (see Sect. 3.5). In the absence of direct measurements of Hg(0) fluxes, this air-sea Hg gas exchange remains largely unconstrained (Zhang et al., 2019c).

In parallel to global models, regional 3D Hg models have also been developed such as the Community Multiscale Air Quality (CMAQ-Hg) model (Bullock and Brehme, 2002) and its latest development CMAQ-newHg-Br that includes oxidation by bromine chemistry (Ye et al., 2018a) (see Table F1). They use initial and boundary conditions from global model output and allow simulations at much finer spatial resolution (typically 1–50 km grid spacing vs. 100–300 km) over limited areas of the globe. Hg chemistry was also implemented in other regional models including STEM-Hg (Pan et al., 2010) and WRF-Chem (Weather Research and Forecasting model coupled with Chemistry; Gencarelli et al., 2014; Ahmed et al., 2023). The global models GEOS-Chem and GEM-MACH-Hg can also be used for nested-grid simulations over specific regions (e.g., Wang et al., 2014a; Zhang et al., 2012; Fraser et al., 2018; Dai et al., 2023; Dastoor et al., 2021). A new regional model (WRF-GC-Hg v1.0; Xu et al., 2022), building on the WRF meteorological model and GEOS-Chem Hg





chemistry, was recently developed using a dedicated WRF-GC coupler and allowing extra flexibility for the choice of the model domain.

Despite notable improvement in our modeling capabilities over the past decade, several limitations and uncertainties in atmospheric Hg models can still influence our ability to evaluate source-receptor relationships and therefore limit application for policy evaluation (Kwon and Selin, 2016). Major uncertainties in atmospheric Hg modeling arise from: 1) anthropogenic emissions, including emission estimates, chemical speciation, spatial distribution, and temporal changes, 2) atmospheric chemistry and phase changes, e.g., redox reactions, and 3) biogeochemical cycling, e.g., the recycling of historically

deposited Hg in soils and oceans (via air-surface exchange of Hg(0)) of geogenic/anthropogenic origin in the present-day environment. As highlighted in Table F1, the chemical mechanisms and associated rate constants and air-surface Hg(0) exchange formulations used in atmospheric Hg models currently demonstrate significant variability. Sensitivity analyses and inter-model comparisons are thus required to assess uncertainties in atmospheric Hg modeling. Recent modeling assessments of Hg fluxes in global terrestrial and marine ecosystems still show large uncertainties (Zhou et al., 2021; Zhou and Obrist,

2021; Feinberg et al., 2022; Zhang et al., 2019c). Further observations and model development work in parameterizing the fundamental processes of air-surface Hg exchange is needed. The approach to assessing uncertainties in this modeling effort is further discussed in Sect. 6-9.

Distinguishing the relative contributions of historic and contemporary anthropogenic Hg emissions to atmospheric Hg concentrations (and its deposition) is important as the contemporary anthropogenic fraction informs how Hg levels are

affected by mitigation policies in the short term while the legacy fraction informs the long-term response of historically changing emissions. However, currently, most of the 3D atmospheric Hg models parameterize Hg exchange with planetary surfaces while assuming static levels of Hg in soils and oceans due to a limited understanding of processes and geospatial distributions of Hg concentrations in surface environments. Additionally, accounting for the contributions of historical changes in anthropogenic Hg emissions to recent Hg trends requires model simulations at long time scales (~centuries),

which is computationally challenging. Thus, the current 3D atmospheric models are best suited to examine the role of changes in contemporary Hg emissions and environmental variables on the distribution of Hg levels in air and deposition.

### 2.2. Ocean models

Development of ocean Hg models began later than for the atmospheric Hg models and only a limited number of models exist so far. There are several reasons for this. Firstly, ocean observations have until recently been scarce making it difficult to

evaluate models. During the last decade, global-scale oceanographic survey programs like CLIVAR and GEOTRACERS have increased the number of observations, but continuous measurements at fixed locations (as conducted for atmospheric Hg) are still missing. Secondly, there has been a limited mechanistic understanding of Hg chemistry in the ocean, which has made it difficult to model driving mechanisms. Thirdly, the ocean is one more step removed from the emission sources as Hg emitted into the ocean is first transported through the atmosphere or rivers, making it more difficult to model and assess the

effects of changing input loads.





As ocean observations and knowledge of chemistry and Hg inputs have increased in the past decades, the ocean Hg models have also become more complex (Fig. 1). In contrast to earlier treatment of oceanic Hg exchange as a boundary, most current atmospheric models now implement explicit air-sea exchange parametrization with surface ocean. Earliest marine Hg model development was the coupling of a 2D slab ocean model, including inorganic redox chemistry and vertical transport

between air, surface ocean and subsurface ocean, to the GEOS-Chem model (Selin et al., 2008; Strode et al., 2007; Soerensen et al., 2010). Next generation of ocean models, still limited to the inorganic Hg cycle, investigated the 3D marine Hg dynamics (Zhang et al., 2014a, 2015b; Bieser and Schrum, 2016). In addition, Soerensen et al., (2016a) developed a physical-biogeochemical multi-box ocean model including organic Hg chemistry to estimate Hg budgets in the Baltic Sea, and Pakhomova et al., (2018) developed a 1D hydrodynamic water column model with comprehensive Hg chemistry.

In recent years, more comprehensive 3D ocean Hg models including MeHg chemistry have been developed. Currently, four marine Hg models based on numerical 3D hydrodynamic modeling are in active use (Zhang et al., 2020; Kawai et al., 2020; Rosati et al., 2022; Bieser et al., 2023 – excluding models that are no longer actively being updated). Table F2 gives an overview of published 3D ocean Hg models. While all these models contain a complete marine Hg chemistry including MeHg, only Zhang (2020), Rosati et al., (2022), and Bieser et al., (2023) consider uptake to and release from marine biota.

This makes these three models the first hydrodynamic 3D Hg model to include the marine ecosystem. Of these, only the MITgcm (Zhang et al., 2020) and MERCY (Bieser et al., 2023) currently have the possibility to perform simulations at a global scale. Previously, Schartup et al., (2018) implemented Hg bioaccumulation in a complex marine food web model.

Table F6 provides an overview of drivers and their setup in the two primary global ocean models suggested for the MME (i.e., MITgcm and MERCY). The ocean models are driven by external sources (river input, deposition, and air-sea

exchange). These drivers will be harmonized for the model ensemble exercise. The models are further driven by physical, biological, and chemical input data, including data on the atmospheric boundary. Many marine Hg models are directly integrated into marine hydrological biogeochemical models, and thus generate their input variables internally. As they are intricately coded into a host model, typically only the physical forcing data can be harmonized. Temperature, salinity, and nutrient distribution are commonly initialized and nudged to data from the world ocean atlas (WOA) (Boyer et al., 2018) or

run completely free after initialization. For the atmospheric forcings, reanalysis data from NCEP or ECMWF are used either directly or as forcing for a coupled meteorological model. Given this, only a limited harmonization of physical input variables can be achieved. Therefore, the physical fields need to be part of an intercomparison.

With a modular model like MERCY (Table F2), it is possible to create a setup using the exact same physical model and forcing with two different ocean Hg cycling models using the FABM interface (Bruggeman and Bolding, 2014). This allows

for a quantification of the role Hg chemistry mechanisms and partitioning plays in the marine Hg cycle. Models that include Hg transfer through the food web rely on information on concentrations of individual biota species, their growth, feeding, and mortality rates in addition to DOC and POC concentrations from a biogeochemical ecosystem model. In these cases,





only the nutrient input fields can be harmonized. As with water chemistry, it is therefore relevant to compare primary productivity and carbon content as secondary variables when evaluating results from different models.

The 3D ocean models are useful for providing mass balance estimates of Hg between atmosphere and the global ocean, investigating the effect of Hg MEAs on the ocean and creating spatially resolved Hg air-sea exchange estimates for use in atmospheric model simulations. However, the models are still hampered by inadequate information on external sources and the uncertainty of internal chemistry and air-sea exchange parameterization. Of specific importance for simulating temporal trends are limited temporal Hg observations for model evaluation and limitations in our knowledge on temporal trends in

regional and global Hg inputs from rivers, coastal erosion, and hydrothermal vents in combination with a long response time of Hg concentrations in large parts of the ocean due to the slow turnover time of water masses.

## 2.3. Terrestrial-hydrological models

Terrestrial-hydrological models describe Hg fluxes and transformation across the land-freshwater-marine continuum and integrate Hg cycling among atmospheric, oceanic, and terrestrial reservoirs. Such models are – relative to atmospheric and

ocean Hg models (Sect. 2.1–2.2) – under-developed at global scales and are typically implemented at local to regional scales because parameters necessarily reflect physical, hydrological, and biogeochemical processes at the catchment level (Caruso et al., 2008; Jeong et al., 2020). Early efforts include the U.S. Environmental Protection Agency IEM-2M watershed Hg cycling model, which used atmospheric Hg concentrations and deposition rates as inputs to a simple mass-balance spreadsheet model that simulated Hg sinks, sources, and transformation in watershed soils and surface waters (Keating et al.,

1997). Subsequent efforts developed geographic information systems (GIS)-based models for improved simulation of Hg dynamics within the watershed distributary network and to assess relative contributions from direct and indirect atmospheric sources to Hg cycling (e.g., Ambrose et al., 2005).

Later generations of terrestrial-hydrological models probed Hg dynamics under changing future climate conditions. For instance, Golden et al., (2013) explored this in a coastal plain watershed of the mid-Atlantic United States using a multi-

model ensemble approach. The three watershed models included the spatially-explicit, process-based Grid-Based Mercury Model to simulate daily mass-balances of water, sediment, and mercury; the Visualizing Ecosystems for Land Management Assessment Model for Mercury (VELMA-Hg), a spatially-distributed mechanistic ecohydrological model that simulates surface and subsurface hydrology and nutrient and Hg dynamics; and the TOPography based constituent LOADing model for mercury (TOPLOAD-Hg), an empirical load model which utilizes a physically-based watershed model to simulate

hydrological fluxes (of Hg) from the land to fluvial network. While the ensemble approach helped to capture uncertainty associated with the different model conceptualizations of Hg cycling, terrestrial-hydrological models at the time were limited with respect to representation of biogeochemical processes underlying landscape Hg cycling (e.g., temperature-driven processes, availability of Hg species). Around the same time, Futter et al., (2012) published the process-based Integrated Catchments Model for Mercury (INCA-Hg), which included soil temperature and moisture to better simulate temperature-

dependent processes, and overall improved representation of Hg cycling (e.g., deposition, (de)methylation, reduction,





volatilization) in order to predict surface water Hg concentrations. Recently, Jeong et al., (2020) developed a mercury module for the Soil and Water Assessment Tool (SWAT-Hg), a GIS watershed model used to simulate climate and land use impacts on surface and groundwater hydrochemistry and contaminants, to address gaps in previous modeling efforts (e.g., INCA-Hg) including litterfall representation.

Development of terrestrial-hydrological Hg models over the past 25 years has borne significant improvements in the representation of Hg cycling dynamics. However, from the perspective of EE of the MEAs, the utility of these models is uniformly hindered by scaling issues. That is, no known terrestrial-hydrological model was developed for or can be readily applied at a global-scale. This is, in part, because model development historically focused on addressing local to regional watershed scale questions using GIS approaches (e.g., Ambrose et al., 2005; Jeong et al., 2020), which may be prohibitively

expensive (computationally) at broader scales. As computing power and understanding of Hg cycling continue to improve, future generations of terrestrial-hydrological models will allow the scientific community to tackle Hg cycling questions of global relevance, like those in support of EE.

In the absence of global terrestrial-hydrological Hg models, land-atmosphere Hg exchange can be estimated using two-dimensional terrestrial Hg models parameterizing air-terrestrial exchange. The Global Terrestrial Mercury Model (GTMM)

was developed over a decade ago within the carbon cycling framework of the Carnegie-Ames-Stanford approach (CASA) biosphere model (Smith-Downey et al., 2010) which covers the top 30 cm of soil and consists of four soil Hg pools. However, the GTMM has not been supported recently and cannot be coupled with available atmospheric models. Recently, Yuan et al. (2023) developed global air-land Hg model based on the Community Land Model within the Community Earth System Model (Lawrence et al., 2019) with an emphasis on Hg uptake and physiological transformations in vegetation for

different plant functional types.

### 2.4. Multi-media mass balance models

Multi-media mass balance models describe the exchange of mercury between media, providing a relatively simple approach to bottom-up attribution of trends across spatial and temporal scales. Mass balance models typically represent mercury cycling as a set of coupled ordinary differential equations, with mercury exchange between compartments parameterized as a

set of first-order rate coefficients (Qureshi et al., 2011; Amos et al., 2013, 2015). While retaining self-consistency, mass balance models provide a means to trace the fate of anthropogenic mercury emissions over decades or centuries, typically in a closed, mass conserving system. Because of these features, mass balance models have proven themselves to be useful tools for quantifying uncertainties in the global Hg budget (Qureshi et al., 2011), evaluating impacts of historical anthropogenic mercury emissions (Amos et al., 2013, 2015) and projecting the consequences of future activities (Angot et al., 2018; Selin,

2014, 2018; Chen et al., 2018).

The principal challenge in constructing a mass balance model is to define an appropriate number of physically meaningful compartments for which the mercury mass and fluxes can be constrained by observations and process-based models (Fig. 1). Existing global multi-media biogeochemical cycle mass balance models (also referred to as GBC box-model) typically





utilize between five (Qureshi et al., 2011) and seven (Amos et al., 2013, 2015) boxes representing a single global
atmospheric reservoir, and one or several soil and ocean reservoirs.  Significant effort has gone into compiling available
budget constraints from modern observations and environmental archives (Amos et al., 2015), though uncertainties remain in
the mobility and residence times of some reservoirs, including global soils and the large quantities of anthropogenic mercury
released to land and water in forms such as mine tailings, fly ash, calomel, and consumer and industrial waste (e.g., Outridge
et al., 2018; Guerrero, 2016; Streets et al., 2018, 2019a, b). A summary of published mass balance models is provided in
Table F3.

Mass balance models are typically defined by a fixed set of rates coupling mass exchange between a fixed number of
compartments. These models are then forced with time-varying inputs of geogenic and anthropogenic Hg which is mobilized
from the lithosphere by processes such as volcanism, mining, and fossil fuel combustion. Trends in mass balance can be
driven by (1) emission trends, (2) non-steady state variation in compartment magnitude, and (3) rate variability in response
to environmental change. Typically, the dependence of model rates on environmental drivers (*e.g.*, oxidants, meteorology,
land cover) is not explicit, so time-varying rates must be specified based on extrinsic calculations. Nonetheless, time-varying
rates can be easily incorporated into mass balance modeling frameworks.

Because available global mass balance models are similar in structure and composition, a fixed number of reservoirs are
recommended as the basis for the MME mass balance modeling activities. Furthermore, it is proposed that a fixed set of
time-invariant Hg exchange rates be compiled based on state-of-the-science estimates of contemporary reservoir masses and
fluxes, as well as associated uncertainties. These reservoirs levels and rates will be used as a "base case" which can be
compared to results from sensitivity analyses (see Sect. 5.3) in which emissions and rates are perturbed. Rates are calculated
as: $k_{ij} = F_{ij}/m_i$, in which $k_{ij}$ is the first-order rate coefficient describing mass transfer from reservoir $i$ to reservoir $j$, based on
the flux of mass from $i$ to $j$ ($F_{ij}$) and the mass in reservoir $i$ ($m_i$). An example of the modern mercury budget terms used to
calculate mass balance model rates can be found in Table F7, based on the work of Amos et al., (2014). Efforts to update the
budget estimates used for "base case" model configuration should be based on a community effort to reach a consensus on
contemporary global fluxes and reservoir magnitudes, reflecting results from both primary measurement and upscaling
techniques and from process model results.

Mass balance models will be useful for estimating time-dependent global long-term trends in secondary mercury emissions.
This may be important, since secondary emissions are currently the largest source of mercury to the atmosphere (Amos et al.,
2013). However, because existing mass balance models are generally composed of global, time-invariant rates, they provide
no information about regional differences or the effects of global change on mercury cycling. Additionally, these models
typically do not explicitly represent mercury speciation. It is recommended that the GBC box-model simulations be
performed to generate changes in secondary mercury emissions and releases and global reservoir concentrations over longer
time periods (decades - centuries); these values can in turn be used to scale boundary conditions for higher complexity single
medium mechanistic models such as the 3D atmospheric and marine models described above.


## 2.5. Emissions to exposure risk models

Major exposure pathway for Hg is the consumption of food that is contaminated with MeHg (Zhang et al., 2021; Selin et al., 2010). Other pathways include the direct inhalation of Hg(0) vapor such as for ASGM workers. One key step for emission-

exposure-risk models is the link between the environment and food (e.g., freshwater fish, seafood, marine mammals, rice) Hg levels. Existing studies often use the matrices that are most directly associated with the food Hg levels, e.g., atmospheric deposition/marine plankton MeHg concentrations for seafood, atmospheric deposition for freshwater fish, and soil concentrations for rice (Giang and Selin, 2016; Sunderland et al., 2018; Chen et al., 2019; Li et al., 2020b; Zhang et al., 2021; Pang et al., 2022). These environmental matrices can be linked with emissions via atmospheric and oceanic Hg

transport and transformation and food web and bioaccumulation models (Schartup et al., 2018). Different levels of complexity, e.g., involvement of different trophic levels/types of seafood and their geographical locations, have been considered in modeling efforts. Model parameters are generally site-specific for the food web and bioaccumulation models, which are often difficult to obtain (e.g., food web structure, water biogeochemistry) and challenging to extend to larger areas.


Exposure and human health risk assessment models represent the intake of mercury by humans, and require Hg concentrations in diet items, in occupational practices (e.g., ASGM, dentistry) and in certain products (e.g., skin-lightening creams, waste products). The human health risk of the exposure to different Hg chemicals involves studies in biological and health sciences such as epidemiology and pharmacokinetics that are beyond the scope of this whitepaper. Existing studies

often consider the neurological effects of developing brains and the fatal heart attack for adults (Giang and Selin, 2016). Simple pharmacokinetics are assumed between the total MeHg exposure and human biomarker Hg levels (e.g., hair, blood, and urine Hg). The latter is then applied to quantitative dose-response relationships to obtain the health risk. Some studies also further monetize the health risk by associating with socio-economic parameters such as value of a statistical life and expected lifelong production of newborns (Zhang et al., 2021; Pang et al., 2022).


Existing challenges include the direct modeling of food Hg levels. The lack of more comprehensive and process-based rice paddy Hg and MeHg models, as well as more comprehensive marine ecosystem model that is integrated with fishing activities hinder accurate risk assessment. In addition, there is incomplete information about dietary intake and the lag between environment and food Hg levels, although some studies simplified it by using a pre-specified constant lagging

period. Non-linear relationship also exists for response of food to environmental Hg levels, which is especially important in diagnostic studies (e.g., Pang et al., 2022). Human biomarker data, including hair, blood, and urine are needed to evaluate the estimated population MeHg exposure. For example, human hair samples are especially useful and can be collected in a relatively large scale as they are convenient, easy to measure, and representative of long-term exposure. More epidemiological studies to better quantify the health impact of Hg exposure are urgently needed, particularly for mid-low




doses for the cardiological diseases. The choices of risk monetization parameters are also important in policy analysis studies (e.g., Giang and Selin, 2016). Overall, we call for a comprehensive modeling approach that involves collaboration among Earth, medical, and social scientists to provide a much-needed tool to help parties to evaluate the MEAs.

## 2.6. Coordinated multi-media modeling

The primary focus of the MCHgMAP modeling plan is to simulate the mechanistic link between the primary emissions and
releases of Hg (from anthropogenic and geogenic sources) and Hg levels in large scale global abiotic environments (primarily, atmosphere and oceans) to detect and analyze their spatial patterns and temporal trends. Based on the availability and limitations of different Hg modeling frameworks described above, application of an ensemble of 3D atmosphere, 3D ocean, 2D air-land, and multi-media mass balance models is proposed. A coordinated multi-model approach, combining the strength of multiple single-medium and multi-media mass balance Hg models, is suggested to consistently simulate Hg
cycling among environmental compartments and to allow assessment of the impacts of both contemporary and historic changes in anthropogenic Hg emissions on environmental Hg levels. Current 3D mechanistic Hg models are suitable for analyzing the short-term responses of changes in both emissions and environmental conditions on spatially resolved Hg levels and trends. On the other hand, the influence of long-term changes in emissions and releases (over decades and centuries) on temporal trends of Hg levels in environmental media can be estimated by the mass balance Hg models; these
estimates can in turn be used in 3D models to assess their implications to spatially distributed contemporary Hg levels and trends. Currently, the mass balance Hg models are not equipped to examine the role of changing environmental variables on Hg levels.

The conceptual construction of the proposed coordinated MCHgMAP modeling framework including input sources, process representations and coupling of inter-media Hg exchange fluxes between different models is shown in Fig. 1. The Hg
sources and their proposed formulations for MCHgMAP are described in Sect. 3 followed by details of observations for model evaluation (Sect. 4). The technical details for the experimental design of the coordinated MME simulations including the consistency of external and inter-media fluxes across the modeling framework are described in Sect. 5.







**Figure 1. Conceptual design of large-scale environmental Hg models and their coordination in MCHgMAP. Summary of key drivers and coupling between 3D atmospheric, 3D marine, 2D terrestrial, and multi-media mass balance models. Different configurations are possible, ranging from prescribed to dynamic two-way surface–atmosphere gas exchange.**

## 3. Sources

Mercury is released to air, land, and water in earth's ecosystems from both geogenic and anthropogenic activities. Once released from primary sources, Hg accumulates as well as recirculates among major environmental reservoirs with wide-ranging temporal delays governed by differences in residence time of Hg in reservoirs. Burial in natural archives such as subsurface soils, lake and ocean sediments, and glacial ice ultimately immobilizes Hg from its circulation in global environments. Continuous biochemical transformations and physical processes result in residence times of Hg ranging from ~ 1 year in the atmosphere, surface waters and seasonal snowpack/sea ice, ~1-3 years in vegetation to $10^2$-$10^3$ years in soils,





oceans, and glaciers. Owing to its long residence time in soils and ocean (and resulting growth in Hg reemissions/releases),
contemporary Hg levels in the environment are well above those that can be explained by current emissions and releases.
Observations show that reemission of Hg from global soils and oceans is a major pathway of atmospheric Hg emissions.
Therefore, consideration of historic primary emissions, and its ongoing recycling among environments, is important to the
understanding of current and future environmental Hg levels and trends. Direct atmospheric deposition and land mobilization
(by rivers and coastal erosion) of contemporary and legacy Hg comprise marine Hg sources. Climate warming is anticipated
to enhance Hg remobilization due to higher surface temperatures, intensified wildfires, permafrost thaw, glacier and sea ice
melt, and increased river discharge.

Reemission of deposited Hg from planetary surfaces is often referred to as "legacy emissions" in the literature. In this paper,
we refer to the recycling of deposited Hg to air from both contemporary and historic primary emissions as "secondary
emissions" or "reemissions" and reserve the term "legacy emissions" to represent the component of reemissions that
originate from historic primary emissions that have occurred at least one year ago. For the effectiveness evaluation of the
MEAs, it is also useful to consider "legacy emissions" as originating from primary emissions prior to the implementation of
the MEAs.

The following sections describe current knowledge on the spatiotemporal distributions and budgets of Hg primary emissions,
reemissions and riverine exports, and options for their inputs to the atmospheric, marine, and mass balance multi-model
simulations proposed in this study.

### 3.1. Geogenic emissions

As is the case for many metal and metalloid elements, natural geogenic sources are potentially important contributors of Hg
to surface environments. Volcanoes, geothermal fields, hydrothermal vents and submarine spreading ridges release Hg from
direct mantle outgassing and the remobilization of crustal deposits (Fitzgerald and Lamborg, 2014). Natural chemical and
physical weathering of rocks can also release Hg from geological reservoirs, particularly within the global "mercuriferous
belts" of variably Hg-enriched sediments coinciding with plate boundaries (Gustin, 2003; Rytuba, 2003).

Quantifying Hg fluxes from geogenic sources has proven difficult due to the scarcity of direct emissions measurements as
well as due to uncertainties associated with the relatively few data that exist (Edwards et al., 2021). Volcanic and geothermal
systems in particular pose challenges for direct sampling of gas and aerosol emissions, since emission rates of Hg and other
species vary over time due to changes in volcanic activity and in magmatic source compositions (Pyle and Mather, 2003).
While volcanic systems differ in their "emission modes," it is widely accepted that the largest contribution of volcanic gases
to surface environments is from subaerial *persistently degassing* volcanoes with open vents, of which there are currently
~100 active on Earth (Carn et al., 2017; Fioletov et al., 2023). Infrequent explosive eruptions such as the 1991 Mt. Pinatubo
eruption may also release large "pulses" of Hg to surface environments, which are expected to perturb atmospheric and
terrestrial Hg reservoirs for several years, as has been observed indirectly for large igneous province events millions of years
ago (e.g., Grasby et al., 2015; Sanei et al., 2012). However, explosive eruptions are typically short-lived, contributing far less





than persistently degassing volcanoes when averaged over time (Fischer et al., 2019; Fitzgerald and Lamborg, 2014; Geyman et al., 2023). Meanwhile, closed-vent degassing from dormant volcanoes and active geothermal–hydrothermal systems appear to occupy the lower end of geogenic emission sources (Fischer and Chiodini, 2015; Werner et al., 2019),

although data from these sources are quite limited (Bagnato et al., 2015). The alteration and weathering of Hg-bearing rocks and soils constitutes an additional component of the geogenic flux, but Hg emissions from this nonpoint source occur over large areas and pose particular challenges to quantification.

Volcanic Hg fluxes are typically estimated using $Hg/SO_2$ mass ratios calculated from coincident Hg and $SO_2$ measurements taken within the volcanic plume. Hg is thereby indexed as an emission factor (EF) to the typically much better constrained

$SO_2$ flux from a given volcano, and the Hg flux is estimated on annual or longer timescales. However, the wide range of $Hg/SO_2$ ratios reported in the literature (spanning $10^{-3}$ to $10^{-7}$) and a lack of knowledge about these ratios' representativeness of source plume conditions contribute to the large uncertainties in most upscaled volcanic Hg fluxes. This is likely the main reason for the wide range among existing global volcanic Hg flux estimates, which span five orders of magnitude (0.6–7000 Mg per year; Bagnato et al., 2015; Ferrara et al., 2000; Geyman et al., 2023; Lamborg et al., 2006; Mason, 2009; Nriagu,

1989; Pyle and Mather, 2003; Varekamp and Buseck, 1981, 1986). By comparison, emission estimates from geothermal systems—while far fewer in number—span only two orders of magnitude (8.5-–60 Mg per year in total globally; Bagnato et al., 2015, 2018; Varekamp and Buseck, 1986). Because geothermal systems emit little to no $SO_2$, geothermal Hg fluxes typically use $CO_2$ as the EF; literature $Hg/CO_2$ mass ratios span ~$10^{-7}$–$10^{-9}$ (Aiuppa et al., 2007; Bagnato et al., 2009, 2013, 2014, 2015, 2018; Engle et al., 2006; Gagliano et al., 2019; Witt et al., 2008). Due to a lack of reliable data, time-averaged

contributions from infrequent explosive eruptions are more difficult to narrow down; most previous estimates are either derived from early data of questionable reliability (1000–7000 Mg per year; Nriagu, 1989; Varekamp and Buseck, 1986) or misattributed Hg peaks in an ice core from a glacier in Wyoming (600 Mg per year; Chellman et al., 2017; Pyle and Mather, 2003). Given the uncertainty in this component of the volcanic Hg flux, the total geogenic Hg flux estimate may be underestimated when explosive eruptive contributions are not considered. However, the discrepancy is not expected to be

great due to the larger time-averaged contributions of persistent degassing to volcanic gas release. Li et al., (2020a) recently formulated a relatively low explosive eruption flux of 20 Mg $y^{-1}$, only ~11% of their estimate of the global Hg flux from persistent degassing (179 Mg $y^{-1}$).

More recent estimates of the global volcanic flux (37–232 Mg per year; Bagnato et al., 2011, 2015; Fitzgerald and Lamborg, 2014; Geyman et al., 2023; Li et al., 2020a; Mason, 2009; Nriagu and Becker, 2003; Sonke et al., 2023) reflect

improvements in Hg analytical methods and a more critical use of EFs to estimate Hg emissions. Importantly, they are all lower than the more variable but generally higher estimates of past decades. This narrowed range suggests that on average, volcanic degassing contributes <5% of total emissions to the atmosphere from primary and secondary natural sources each year (~5000 Mg; AMAP/ UN Environnement, 2019). It should be noted that virtually all available Hg flux estimates data are from subaerial (surface) volcanic systems; virtually no data exist on submarine volcanic emissions, although estimates place

these figures at ≥100 Mg $y^{-1}$ (AMAP/ UN Environnement, 2019; Garrett, 2000; Rubin, 1997).





### 3.1.1. Formulating a global geogenic Hg emission inventory

As with another potentially important natural source (wildfires; see Sect. 3.4), the most pressing issue in geogenic Hg research is a lack of data. Importantly, the existing data are spatially imbalanced; there are no Hg plume measurements from some remote volcanic areas which have elevated fluxes of other volatiles, such as Papua New Guinea (summarized in Edwards et al., 2021). Nevertheless, estimates of the geogenic source component can be formulated for the purposes of MCHgMAP simulations. In the absence of a representative catalogue of direct Hg measurements, we propose to estimate volcanic and geothermal point Hg sources using recent high-quality volcanic flux data for $SO_2$ (Fioletov et al., 2023; Tournigand et al., 2020) and $CO_2$ (Werner et al., 2019) in combination with a compilation of emission factors (EFs) from the literature (i.e., $Hg/SO_2$ and $Hg/CO_2$ mass ratios) published since the year 2000 using more reliable Hg analytical methods (Edwards et al., 2021; Fitzgerald et al., 1998; Li et al., 2020a for explosive eruptions).

To estimate the contribution of geogenic emissions originating from weathered rocks, soils, and terrestrial sediments, we employ a scaling approach based on representative surface Hg flux measurements obtained from Hg-enriched zones. The spatial distribution of these emissions is determined by referencing the global mercury mineral belts (Rytuba, 2003). We will use a recent estimation of global emissions from Li et al. (2020a), which was derived from the comprehensive surface-atmosphere Hg fluxes database (Agnan et al., 2016; Biester and Scholz, 1997).

### 3.1.2. Limitations and recommendations

The major limitation of the above approach is the use of EFs to cover the large gaps in volcanic and geothermal emissions where Hg concentration data does not exist. Indeed, published volcanic EFs represent only 14 volcanoes, or ~13% of the total number of currently active degassing systems worldwide. This leaves many gaps in geographic coverage, where multiple factors specific to volcanic degassing may contribute to significantly higher or lower Hg fluxes than estimates derived from "one-size-fits-all" EFs.

An additional limitation of EF-based indexing approaches stems from the observation that available $Hg/SO_2$ measurements exhibit a "heavy-tailed" distribution. This implies that a small fraction of sites characterized by high $Hg/SO_2$ ratios contribute disproportionately to the cumulative global volcanic Hg flux relative to their $SO_2$ emissions (Geyman et al., 2023). This feature of the $Hg/SO_2$ distribution can be accounted for in global estimates of volcanic Hg flux (Geyman et al., 2023), but it fails to identify where and when these higher $Hg/SO_2$ sites occur. Improvements in understanding of the underlying drivers of variation in $Hg/SO_2$ ratios could therefore improve the accuracy of both the absolute magnitudes and the spatiotemporal distribution of volcanic Hg emissions.

Moreover, temporal changes in EFs require further attention. Previous research has shown that Hg/SO2 mass ratios can change over time due to decoupled degassing behavior of the different gas species (Aiuppa et al., 2007; Siegel and Siegel, 1984), although recent work has demonstrated Hg/SO2 remains conservative over at least several hundred meters from the volcanic source (Edwards et al., 2024). As the quality and temporal resolution of $SO_2$ and $CO_2$ flux data increasingly





improve, plume Hg measurements of "unmeasured" volcanoes are greatly needed to better understand the range of EFs and the representativeness of the single EF used for the respective geogenic Hg flux calculations of persistent degassing
volcanoes, geothermal systems and infrequent explosive eruptions. Measurements of eruption plume Hg using modern-day analytical techniques and aerial sampling platforms (e.g., drones) should be a priority for future volcanic Hg work.

One additional limitation concerns the speciation of volcanic Hg upon emission and downwind of the source, which has important implications for the atmospheric lifetime and environmental fate of Hg within the broader Hg cycle. While GEM appears to be the dominant Hg species in near-source volcanic plumes >95% (Bagnato et al., 2007; Martin et al., 2011; Witt
et al., 2008), some measurements of aged volcanic plumes show gaseous oxidized Hg ($Hg(II)_g$) and particulate-bound Hg ($Hg(II)_p$) concentrations well above background levels (Aiuppa et al., 2004; Ravindra Babu et al., 2022; Dedeurwaerder et al., 1982; Galindo et al., 1998). Further work focusing on the downwind physicochemical fate of Hg after emission is therefore essential to grasping the environment significance of geogenic emissions within the larger Hg cycle.

### 3.2. Anthropogenic emissions

Global anthropogenic Hg activities accounted for ~30% of total Hg emissions to the atmosphere in 2015 (Outridge et al., 2018). Anthropogenic Hg emissions can be from intentional Hg use activities, or the production and use of Hg-added products, as well as from the unintentional Hg emission sectors where Hg is an impurity in the raw materials or fuels. The history of atmospheric Hg emissions can be traced back over 4,000 years (Bebout et al., 2006). However, currently available emissions estimate only represent emissions during the last 500 years (Streets et al., 2019a), and comprehensive global
inventories only represent the most recent 50 years (AMAP/ UN Environnement, 2019; Wu et al., 2016; Liu et al., 2019a; Muntean et al., 2014, 2018; Zhang et al., 2016b; Pacyna et al., 2006a; Pacyna and Pacyna, 2002; Pacyna et al., 2010). Anthropogenic Hg emissions vary temporally and spatially and depend on the effects of economic, policy and human activities in different regions. Understanding the sectoral, temporal, spatial, and chemical speciation variations of current Hg emissions is essential to simulate the global transport and deposition of atmospheric Hg, and to evaluate ecological and
human benefit from Hg pollution control policies. In addition, Hg can take millennia to be returned to a secure and long-term (environmental) repository. Determining where and when Hg is released as a result of human activities allows better quantification of present-day secondary emissions and releases and future trajectories of environmental concentrations.

### *Mercury governance under the MEAs*

The mercury MEAs (MC and LRTAP) set out to protect human health and the environment by mitigating anthropogenic Hg emissions and releases. The Minamata Convention includes provisions to eliminate Hg mining and introduces trade-related provisions and targets to phase out or reduce Hg uses in consumer products, industrial processes, and ASGM. Such actions will benefit Hg emission reduction from intentional Hg use sectors. In addition, the Convention sets out emission control obligations for five source categories responsible for the largest proportion of point-source atmospheric Hg emissions: coal-
fired power plants, coal-fired industrial boilers, nonferrous metal smelting (zinc, lead, copper and large scale-gold





production), waste incineration, and cement clinker production. To formulate emission control policies and to evaluate the Convention's effectiveness, each Party shall establish, as soon as practicable and no later than five years after the date of entry into force of the Convention, and maintain thereafter, an inventory of emissions from relevant sources (MC Article 8, Para 7). However, it should be noted that Parties are only required to maintain information on emissions from the five

convention-related point source categories and are only required to identify sources that comprise at least 75% of the estimated emissions from each of those categories (MC Article 8 Para 2(b)). LRTAP requires Parties to maintain and submit an annual inventory for anthropogenic Hg sources, but these are also often limited to specific sources covered by the agreement (ECE/EB.AIR/115. https://unece.org/sites/default/files/2021-10/ECE.EB_.AIR_.115_ENG.pdf)

Therefore, emission datasets from other sources are essential to have a comprehensive estimate of anthropogenic emissions
for purposes of simulating Hg levels in the environment. In the following section, we review currently available emission data and identify plans for compiling model-ready emission inventories that can be used in MCHgMAP MME simulations.

### 3.2.1. Historical emissions

### 3.2.1.1. Overview of global and regional emission inventories

Currently, there are four main datasets of global anthropogenic emission inventories, identified here as Arctic Monitoring
and Assessment Programme / UNEP Global Mercury Assessment (AMAP/GMA; Steenhuisen and Wilson, 2019), Emissions Database for Global Atmospheric Research (EDGAR; Muntean et al., 2018), STREETS (Streets et al., 2019b) and WHET (Zhang et al., 2016b). Table E1 summarizes available global and regional anthropogenic emission inventories, including their methodologies, years of emission inventory, chemical speciation and uncertainties. Previously, the 1990, 1995, 2000, and 2005 AMAP emission datasets have been compiled by using sector-specific emission factors and sectoral activity data
by nations (Pacyna and Pacyna, 2002; AMAP/UNEP, 2008). The improved methodology employed to produce the AMAP/GMA 2010 and 2015 emission datasets adapted this approach to also include multiple fuel/raw material-specific emission factors for a given sector as well as emissions control technology factors to derive emissions estimates that consider: activity data, the associated Hg content of fuels and raw materials and the types of process involved and technology applied to reduce emissions to air (through technology profiles that reflect the degree of application and the
degree of effectiveness of air pollution control) (AMAP/UNEP, 2013; AMAP/ UN Environnement, 2019). We refer to the emissions methodology that traces mass conservation of Hg through the process (such as used for AMAP/GMA 2010 and 2015 emissions) as "mass-flow approach" in the remainder of this paper. Thus, the AMAP 1990–2005 and AMAP/GMA 2010–2015 inventories would not be consistent with each other. EDGAR datasets were compiled using mass-flow approach, similar to AMAP/GMA 2010-2015 (Muntean et al., 2014, 2018). A trend of global anthropogenic emission of mercury to the
atmosphere from 1510-2010 was developed by Streets et al. (2019a), which is the available emission inventory at the longest timescale. Subsequently, Streets et al. (2019b) developed emissions inventory covering the years 2010-2015 with slight adjustment of emission sectors. Some of the sectoral activity data were only available at the regional (multi-country) scale.



Default settings in WHET dataset (Zhang et al., 2016b) mainly originated from the Streets et al. (2011), but key national emissions such as the emissions in China (Zhang et al., 2015a) were updated according to regional inventories.

The mass-flow approaches are also used in national/regional emission inventories, such as the Abacas-EI-Hg in China (Wu et al., 2016), and the emission inventories for Thailand and Korea (Wongsoonthornchai et al., 2016; Sung et al., 2018). These national studies adopted local parameters and provided the possibility to compile more refined emission maps than the global inventories. For example, more than 90% of mercury emissions in Abacas-EI-Hg are from point sources. Some national emission inventories adopted the default sector-specific emission factors provided by the UNEP toolkit (Pudasainee

et al., 2014) or estimated from local field tests, especially in the Minamata Initial Assessment (MIA) reports (Back et al., 2019). Until now, the number of countries submitting MIA report has reached 70. Facility reporting is also a commonly used method to compile national emission inventories. Currently, approximately 44 countries have established a Pollutant Release and Transfer Register (PRTR) system; most of these are for countries located in Europe (https://environment.ec.europa.eu/topics/industrial-emissions-and-safety/european-pollutant-release-and-transfer-register-e-

prtr_en). These PRTR systems provide the basic facility data to compile a national emission inventory; however, they may only include facilities with emissions above a specified threshold.

Sectors responsible for the majority of atmospheric Hg emissions are largely accounted for in most global inventories. But source categories in these inventories are not identical; not only do the sectors included differ, but also the way the sectors are defined differ, e.g., in some inventories cement emissions may appear under a general category of stationary combustion

that could include power plant or other industrial sector emissions. Even in the emissions compiled by one group, the source categories can change in the inventories of different years for many reasons, such as the new understanding of emission sources or more available data. Compared to AMAP datasets before 2005, additional sectors such as the artisanal and small-scale gold mining and waste from mercury-containing products were added in AMAP/GMA inventories for 2010 and 2015 (AMAP/UNEP, 2013; AMAP/ UN Environnement, 2019). International shipping is included as a new Hg emission source in

EDGAR datasets, with a contribution of less than 1% (Muntean et al., 2014, 2018). Cremation is only discussed in the AMAP/GMA inventory and had a low contribution. The assessed years of these inventories also varied.

Uncertainty range is an important index to reflect the accuracy of estimated emissions. The reported uncertainties of the estimated total global anthropogenic Hg emissions are: −10 to +27% for the AMAP/GMA dataset for 2015 (AMAP/ UN Environnement, 2019), −26 to +33% for EDGAR dataset 2012 (Muntean et al., 2018), −20% to +44% for STREETS dataset

2015 (Streets et al., 2019b), and −33% to +60% for WHET dataset 2010 (Zhang et al., 2016b). However, different methods are used to quantify uncertainties and any direct comparison of these uncertainty ranges could result in misleading conclusions; the subjectivity of assigning errors to the parameters themselves also affects the uncertainty ranges of the final emissions estimates. A semi-quantitative approach of grading all the parameters in Hg emission factors is one of the most common methods used to evaluate the uncertainty of emissions. Such methods are simple, arguably only applicable to the

deterministic emission factor model, and are of relatively lower reliability. The AMAP/GMA 2015 work applied a 'propagation of errors' approach to quantify uncertainty, similar to the approach applied in some IPCC work. Quantitative





approaches based on Monte Carlo simulations to produce probabilistic emissions estimates by taking into account the probability distribution of key parameters have been widely used in emission inventories in the past decade. Such methods are believed to be more objective, but a more sophisticated scheme usually needs more data and more assumptions to support

its application.

### Historical trends and sectoral contributions

EDGAR datasets reported anthropogenic Hg emissions for the period 1970-2012, with the emissions increasing from 860 to 1889 Mg y[-1] during this period (Fig. 2). According to STREETS datasets, emissions grew from 1971 to 2390 Mg y[-1] during 2000-2015. AMAP/GMA indicated that atmospheric Hg emissions at 1810 Mg y[-1] in 2010 and 2220 Mg y[-1] in 2015. WHET

assessed Hg emissions at 2890, 2160 and 2280 Mg y[-1] in 1990, 2000, and 2010, respectively. It is obvious that both the emission trends and emission amount in typical years from current studies are not comparable. Global annual anthropogenic emissions in 1990 reached as high as approximately 2890 Mg y[-1] in the WHET dataset while the emissions were only half of this in EDGAR datasets. According to WHET datasets, global emissions were suggested to decrease during 1990-2000, whereas the converse trend was suggested in EDGAR datasets. As to sectoral emission trends after 2000, both STREETS

and EDGAR datasets indicated an upward emission trend in industrial and fossil fuel combustion sectors.

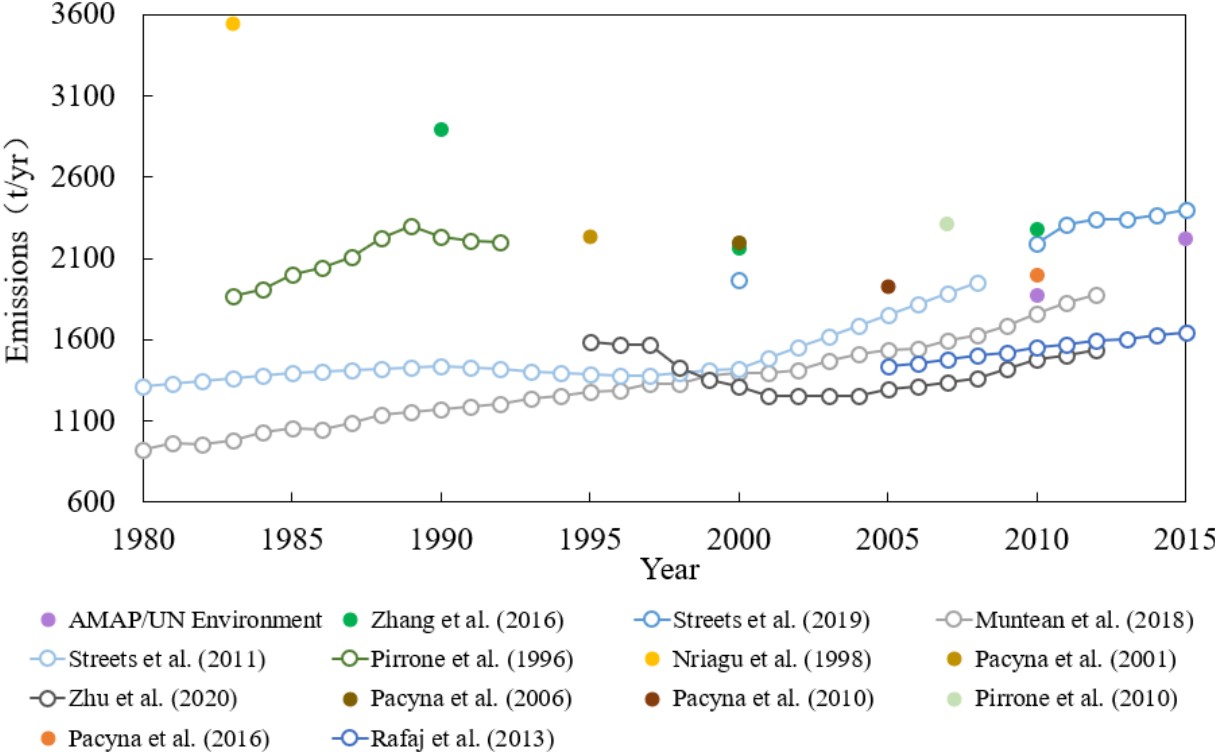

**Figure 2. Temporal trends of global anthropogenic Hg emissions to air based on available inventories (AMAP/ UN Environnement, 2019; Streets et al., 2011; Zhu et al., 2020; Pacyna et al., 2016; Zhang et al., 2016b; Pacyna et al., 2006a; Rafaj et al., 2013; Streets et al., 2019b; Pacyna et al., 2010; Muntean et al., 2018; Pacyna et al., 2001; Pirrone et al., 2010, 1996, 1998).**



The differences in the estimates of total Hg emissions between different inventories before 2010 are particularly significant.
Some explanations for these differences can be hypothesized. However, reasons for the apparent differences need to be better
understood. Different source classification criteria (Table E2) are certainly a part of the explanation for differences between
different inventories. For example, STREETS 2015 estimated Hg from pesticides and fertilizer which were not included in
AMAP/GMA 2015. Conversely, the AMAP/GMA 2015 inventory included Hg emission from biomass burning, cremation
and gas combustion, which reached 56.3 Mg y$^{-1}$ in total. Emissions from waste streams of mercury-added products were
included in AMAP/GMA 2010 and 2015 inventories but not in earlier AMAP inventories (1990-2005). Even for one
emission data source, the source coverage may therefore vary over time.

In Fig. 3, we present the emissions of dominant sources, mainly the MC-related sectors associated with five large point
sources, from different inventories. In 2010, the differences between emissions estimates from the coal-fired sector and from
cement production were within 12%. However, emissions from nonferrous metal smelting and waste incineration were
subject to much larger differences. Hg emissions from nonferrous metal smelting in EDGAR were approximately 50% lower
than the estimation by AMAP/GMA and STREETS. It should be noted that the specific types of nonferrous metals studied
were not identical, although their emissions estimates were similar between AMAP/GMA and STREETS. In AMAP/GMA,
emissions from aluminum smelting were included where it was not included in STREETS datasets. Emission differences
from waste incineration between different studies were even larger. The emissions from STREETS were approximately 8
times that from EDGAR and 3 times that from AMAP/GMA. Such differences were mainly due to the inclusion of emissions
caused by disposing of wastes from commercial Hg use sectors in the STREETS dataset. In AMAP/GMA 2015, emissions
from the disposal of Hg-added products were also included. However, the estimated Hg emissions from waste disposal were
still only half that of the STREETS dataset. Therefore, atmospheric mercury emissions from waste incineration sector are
still under large uncertainty, which can be attributed to a large variation of mercury content in the wastes, various waste
types and waste disposal method.

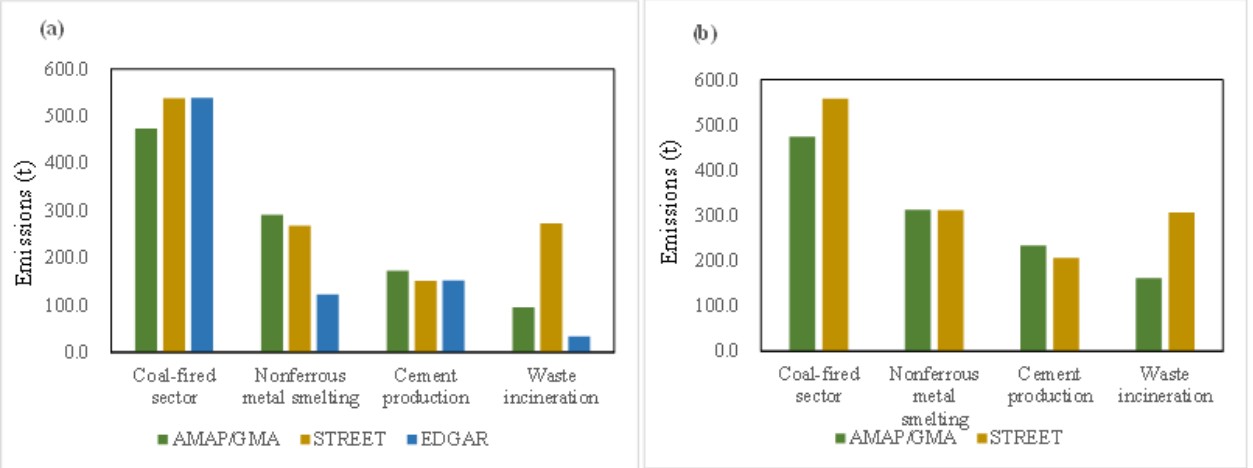

**Figure 3. Hg emissions of dominant sources from different inventories in (a) 2010, and (b) 2015.**



Nonferrous metal smelting included zinc, lead, copper, aluminum, and large-scale gold production in AMAP/GMA and
EDGAR, whereas aluminum was excluded in STREETS.

*Spatial distribution*

Methodologies used to produce spatially distributed emission datasets have evolved over the last two decades. Earlier work distributed all anthropogenic emissions on a 1°×1° or 0.5°×0.5° latitude/longitude grid using a population distribution (population density) proxy. Recent datasets, which are producing more refined resolution, incorporate more information on
point source emissions (e.g., Steenhuisen and Wilson, 2019) and use different proxy data to estimate the spatial distribution for different source sectors (see Table E3). The spatial resolution of the STREETS 2010 and WHET 2010 inventories is 1°×1°, whereas EDGAR provided emissions with the resolution of 0.1°×0.1°. The spatial resolution of AMAP/GMA 2015 dataset in UNEP (2019) was 0.25°×0.25° as default resolution (but 0.1°×0.1° resolution is also available) (Steenhuisen and Wilson, 2019). However, high spatial resolution is only more accurate when point source coordinates or the proxy data
themselves are available at high resolution. In general, proxy data resolution tends to be much lower than would justify higher precision in spatial distribution.

Table E3 shows the proxy rasters applied to the various sectors in the geospatial distribution of emissions. Generally, there is a desire to obtain facility-level emission data based on point source activity level and air pollution control information, especially for fuel combustion and industrial sources associated with large point sources. However, such data is seldom
available except for a very few large point source facilities. Improvements in facility information, and in some cases emissions quantification at the facility-level, is a feature of both global and regional emission inventories. Emission inventories compiled based on point source information are supposed to be subject to lower emission uncertainty and higher spatial distribution accuracy. This is true when available point source coordinates and emissions, or proxy data are at high resolution. However, a more sophisticated scheme usually needs more work, especially to compile a global emission
inventory and maintain up-to-date facility level information.

Therefore, point source information and spatial proxies are used to distribute emissions that are by their nature associated with diffuse sources, or where the location and relevant characterization of point sources is unknown. The distribution intensity of the proxy is assumed to be a reasonable surrogate for the intensity of the emissions within the studied geographical extent. Different proxies were applied to emissions from different sectors based on available data. In the
AMAP/GMA work, for example, for the stationary combustion sector or industrial sector, an industrial activity mask derived from an inventory of carbon emission point sources produced by the Carbon Monitoring for Action (CARMA) organization, facility capacity, or revenue based on industrial enterprises were generally used as surrogates. Population remains a commonly used proxy for geospatial distribution of emission from sectors such as waste disposal, residential heating and some intentional uses of mercury, which are essentially diffused sources by nature.

Due to the application of different proxy rasters, the spatial distribution of emissions varies by inventory. Here, the gridded emissions are normalized to compare the emission distribution differences in different inventories (Fig. 4). A relatively large proportion of Hg was distributed in South Africa and Southeast Asia in STREETS 2010. In addition, the emission in the



STREETS 2010 dataset is more scattered whereas more high-emission grids can be observed in EDGAR 2010 and AMAP 2010. Such phenomenon is especially obvious in East and South Asia, Europe, and America. Moreover, Hg emissions

calculated for international shipping by multiplying the fuel consumption with emission factors led to emissions on oceanic surface in EDGAR 2010 dataset, which are absent in STREETS 2010 and AMAP/GMA 2010.

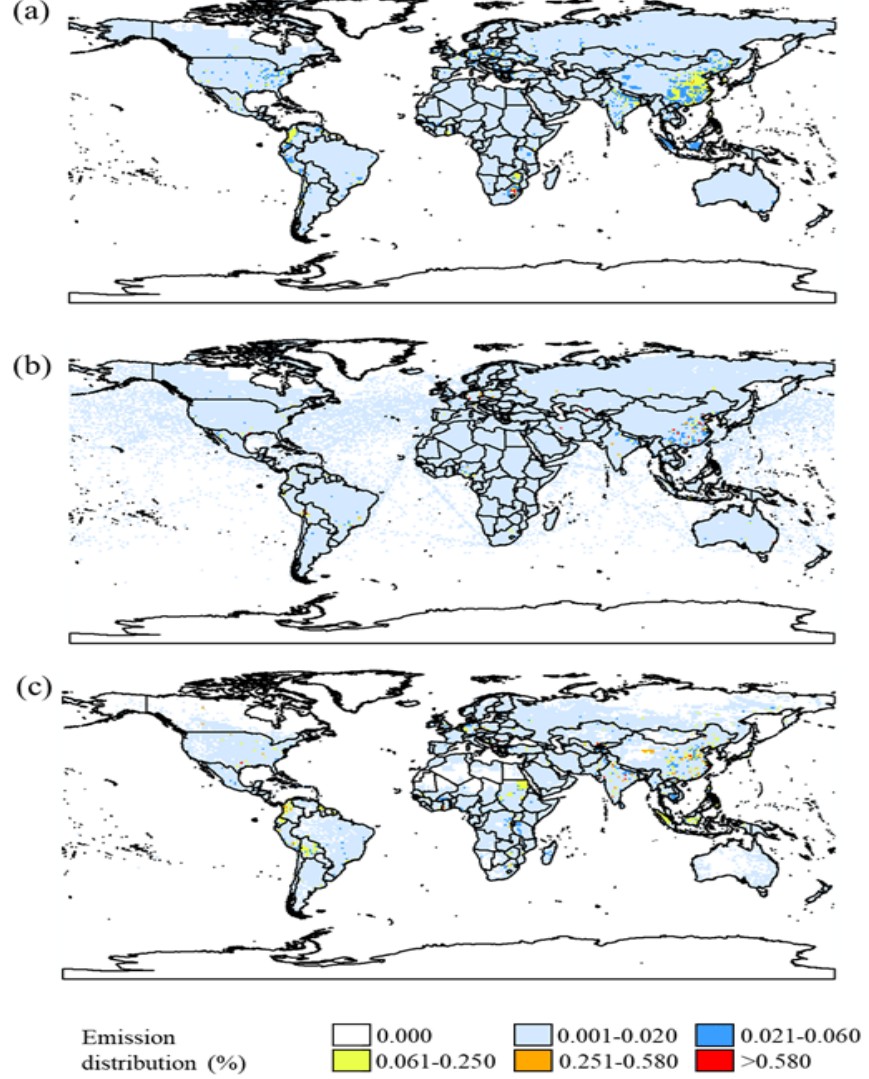

**Figure 4. Spatial distribution of anthropogenic Hg emissions in different inventories: (a) STREETS dataset 2010 (Streets et al., 2019c); (b) EDGAR dataset 2010 (Muntean et al., 2018); (c) AMAP/GMA dataset (AMAP/UNEP, 2013).**

*Chemical Speciation*

Together with spatial distribution, characterization of the emitted forms of Hg has been identified as a major challenge and recognized in the AMAP/GMA work as an area requiring attention in the future. Both subjects are of importance to Hg transport and fate modelling. In several of the global inventories, the emissions of different forms of Hg (Hg(0), Hg(II)$_g$ and





Hg(II)$_p$) have been derived by multiplying total Hg emissions with 'Hg speciation profiles' (i.e., percentages of the mercury

in each of the three forms emitted). Hg speciation profiles can differ between sectors due to differences in fuel consumption and fuel type, control devices for air pollutants, and operating procedures. Table E4 compares the speciation profiles (%) used in AMAP/GMA (2005 and 2015), WHET and EDGAR inventories. AMAP/GMA (2005) and WHET used sector-specific speciation profiles. Many recent publications have reported results of measurements of Hg species under various plant-operating conditions and for different applied technologies, including air pollution control technologies. These studies

provide the possibility to apply technology-specific speciation profiles, such as that used in EDGAR datasets. The functionality of the spatial distribution application employed in AMAP/GMA 2015 to generate a speciated global inventory has been developed to make it more flexible, e.g. with respect to applying more complex speciation schemes based on country technology levels. As part of the AMAP/GMA work, Steenhuisen and Wilson (2019) investigated the effect of applying different speciation schemes for emissions from China, utilizing the information presented in Zhang et al., (Zhang

et al., 2015a) for all sectors other than ASGM. However, it was not considered justified to adopt this scheme to the entire global inventory as this would imply that all countries apply similar (and in the case of newly constructed large point sources, relatively advanced) control technology as China. The limitations of continuing to use the older default speciation scheme for much of the AMAP/GMA 2015 inventory work was however fully acknowledged.

Due to the differences in speciation profiles, sectoral emissions contributions, and spatial proxies in different studies, the

proportion of the three Hg forms in the same grid cell varied by inventory, as shown in Fig. 5. Overall, the emission share of Hg(0) from AMAP/GMA 2010 was generally higher than those from STREETS 2010 and EDGAR 2010. In all these three datasets, there were cells with Hg(0) proportion higher than 80% in middle Africa and South America. Whereas overall Hg(0) emission proportion higher than 80% in North America is only found in STREETS 2010. Hg(II)$_g$ proportion in EDGAR datasets was mainly located in the range of 20% to 60%, which is much higher than the proportion in other two

datasets. The emission proportion of Hg(II)$_p$ in all three inventories was lower than 20%.



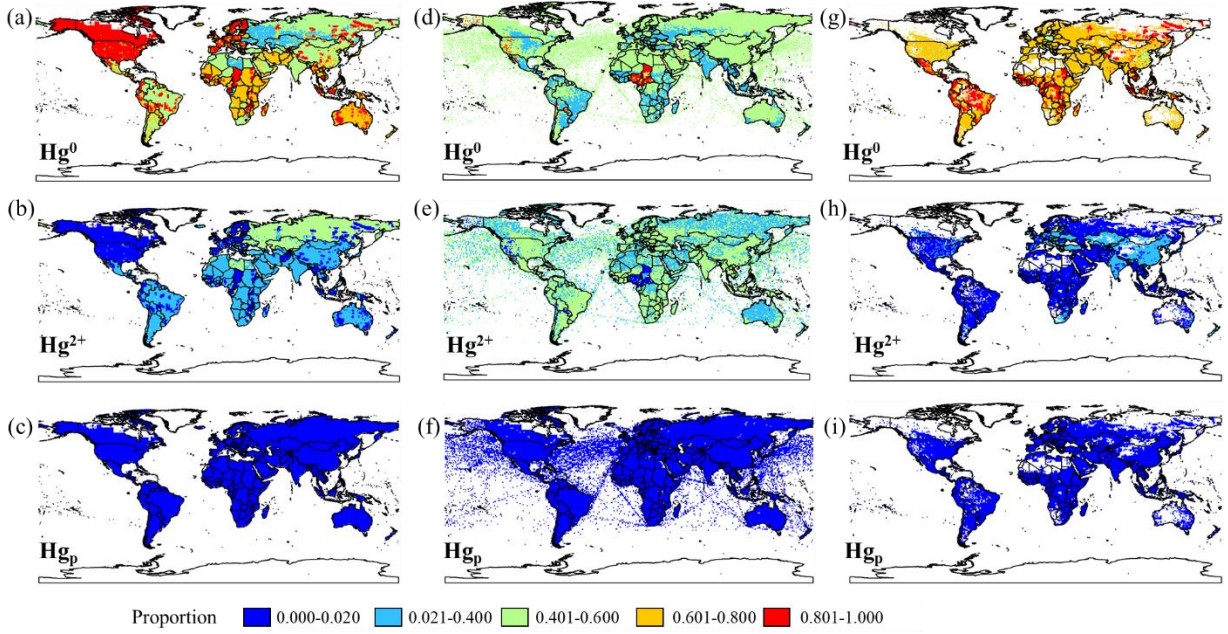

**Figure 5. The proportion of speciated Hg emissions in total emissions by grid: (a-c) STREETS dataset 2010 (Streets et al., 2019c); (d-f) EDGAR dataset 2010 (Muntean et al., 2018); (g-i) AMAP/GMA 2010 dataset.**

### 3.2.1.2. Emission inventories for MCHgMAP simulations

The MC was signed in 2013 and entered into force in 2017. To evaluate the effectiveness of the MC, the proposed timescale for the emission inventory should cover at least the two years mentioned above. As discussed above, currently several global anthropogenic Hg emission inventories are available for 2010, and two inventories are available for 2015 (AMAP/GMA and STREETS). Given that UNEP launched the negotiations on a global mercury instrument in 2009 and actions have been taken since the Convention negotiations by some parties, using 2010 as the baseline year for the EE analysis is recommended for

MCHgMAP simulations. Furthermore, an extension of current Hg emission inventories to at least until 2020 is recommended to analyze the effects of changes in anthropogenic Hg emissions as a result of the MC on environmental Hg levels using model simulations. However, it should be noted that 2010 may be an anomalous year in the longer time trend of Hg emissions as it was probably affected by lower activity levels following the global economic downturn in 2008 as may be the case also in 2020 due to COVID-related economic impacts.

The EDGAR global anthropogenic speciated mercury emissions inventory is planned to be extended until 2022. It will provide emissions time series for all world countries at 0.1 x 0.1-degree resolution for the period 1970-2022 together with a comprehensive trend analysis by sector and by region/top emitters, including the impact of pandemic on mercury emissions. The updated EDGAR mercury inventory release, based on the EDGARv4.tox2 (Muntean et al., 2018) mercury emissions for the period 1970-2012, is primarily focused on updating the global mercury emissions for the last decade. However, in the

case of the availability of new and better input data sources, activity data in particular, the EDGAR emissions will be updated retrospectively as well. The technologies and end-of-pipe, for power generation sector in particular, are from the





new edition of Platts database except for China for which the information in Zhang et al. (2023b) is used. EDGAR emissions time series are aggregated using both IPCC 2006 and IPCC 1996 classifications for each country. The revised EDGAR Hg inventory release will also provide global monthly sector-specific gridded emissions. Regarding spatial
distribution, the proxy data for point, linear and areal sources are planned to be updated continuously in EDGAR, thus the new version of EDGAR Hg emissions will benefit from the latest updated proxy data; e.g. the new population proxy used in EDGAR is based on the JRC GHSL (https://ghsl.jrc.ec.europa.eu/https://ghsl.jrc.ec.europa.eu/), which combines the latest population census data (CIESIN GPWv4) with satellite imagery of built-up areas (Landsat 8). Speciation of mercury emissions will be based on the literature review presented and evaluated in Muntean et al., (2018) through model simulations
using GEOS-Chem mercury model including nested simulations and comparison with measurements. In addition to mercury emissions, EDGAR (a multipurpose and independent global database) also provides consistent emissions of greenhouse gases (GHGs), unintentional persistent organic pollutants (UPOPs) and air pollutants (APs).

Plans are also under consideration for updating AMAP/GMA 2015 emissions to 2020 estimates for at least some parts of the inventory based on the AMAP/GMA methodology, and potentially provide estimates for intervening years since 2010.


### 3.2.1.3. Limitations of available emissions inventories and recommendations

As presented above in this section, Hg emission inventories compiled by different groups varied in source classification, emission amounts, inventory compilation and spatial distribution method, as well as Hg speciation profiles. Therefore, when using existing emission inventories for temporal trend modelling, it is necessary to ensure the comparability of emission
inventories over the time period considered. Even if the emission inventory was compiled by one group, the emission data from different years may not be fully comparable due to changes in emissions estimation methods, sectors included, or e.g., sources of activity data used. Inventories are in some cases only prepared at intervals (typically 5-years) so annual estimates would involve interpolation.

The EDGAR dataset (1970-2012) annual estimates are internally comparable, and gridded speciated Hg emission inventories
by sector are publicly available. Currently, the emission amounts estimated in EDGAR are significantly lower than those estimated in other emission inventories, especially for the earlier part of the period. The AMAP/GMA emission inventories for 2010 and 2015 are consistent and comparable due to consistent compilation methods; the emission sectors addressed in 2010 were updated in connection with the work on the 2015 inventory. However, the AMAP/GMA gridded data for 2010 and 2015 apply the same (2015) spatial distribution proxies and point source information and therefore do not reflect any
changes in these between 2010 and 2015. The advantages of the STREETS dataset are its centennial scale coverage. However, to achieve such a long-term emission dataset, the spatial resolution and the fineness of emission source classification have been more or less sacrificed. In addition, the emission factor changes over time were generated based on a transformed normal distribution function. Therefore, the accuracy may be impacted to some extent when using this method to evaluate the effectiveness of control measures.





The gaps in understanding Hg speciation profiles in emissions to the atmosphere are considered to be the largest uncertainty source of speciated Hg emissions. The criteria used to determine the speciation profiles have varied in different inventories. For example, WHET datasets recommended the emission profiles for coal-fired power plants to be 80:15:5 for $Hg(0):Hg(II)_g:Hg(II)_p$ based on the assumption that oxidized Hg will be reduced to Hg(0) in the smoke plume. It is important to also recognize that improving information on speciation associated with different control technologies is only part of the

puzzle. A further complication (and probably more important when it comes to developing inventories for tracking changes over time) concerns the extent to which different technologies are applied in different countries and (if possible) individual facilities, and how this has changed over time. Speciation of ASGM emissions is an area requiring particular attention. The AMAP/GMA inventories assume that all ASGM emissions are in the form of Hg(0) (i.e., the form evaporated when heating amalgam). It is therefore assumed that any subsequent oxidation to Hg(II) species is not an intrinsic part of the 'emission'

profile associated with the ASGM process but would need to be addressed in the modelling (as part of the air chemistry) even if, due to emission at ground level, this may take place very soon after emission due to interaction with, e.g. vegetation or surfaces. Further, the global dispersion and deposition of Hg is seasonally dependent. Some information is available on seasonal distribution of Hg emissions from regional inventories, but current global inventories provide no such information. Seasonal distribution of anthropogenic emissions depends on sectoral distribution; therefore, it is likely to vary spatially and

possibly change over time as emissions controls are being applied to various sectors. In addition to specific Hg control measures, air pollution control devices aimed to remove $SO_2$, $NO_x$, $PM_{2.5}$ or dioxins can also synergistically remove Hg. Therefore, consideration of changes in fuel-use driven by economic factors or pollution control and climate change mitigation policies is important to attribute the observed changes in Hg emissions and environmental levels over the MC/LRTAP 'effectiveness' analysis period.

One of the key questions of EE is to address whether the measures taken have changed Hg emissions, releases and usage. Therefore, to evaluate the impact of control measures, the mass-flow approach is the most suitable method to be applied. By using this method, the Hg removal effect of control measures can be quantified. To promote EE cycles, mass-flow approaches should be applied for emission inventory compilation. To understand the impact of future emission reductions on the environment and humans, control measure-specific Hg speciation profiles should be applied instead of sector-specific

speciation profiles. Many field experiments have been conducted in the past two decades to characterize emissions, especially addressing exhaust flue gas emissions from large point sources. Although it can be argued that the information on speciation of emissions is very specific to measurements at a particular plant or facility, these studies provide an opportunity to improve the generic speciation schemes employed in global emission inventories. It is timely therefore to review existing field experiments, including the test methods, results, and facility information, and develop new, standardized Hg speciation

profiles based on pollution control device information to replace the existing speciation profiles applied in different emission inventories to date.

Where possible, the source classification criteria should be harmonized between inventories. Given that both air pollution control and measures to reduce carbon will impact Hg emissions, using consistent source classification criteria for sources





which emit air pollutants, carbon and Hg simultaneously are recommended. As to spatial distribution, distribution methods
as well as the proxy applied should also be unified to ensure the comparability of different studies. Point source information
should be used to the greatest extent possible for locating emissions for convention-related sectors: coal-fired power plants,
coal-fired industrial boilers, cement clinker production, waste incineration, and nonferrous metal smelting (zinc, lead,
copper, and large-scale gold production). Several data sources can aid in the compilation of point source emissions. First,
according to the requirement of MC, parties should compile the emission inventories of point sources before 2022 and
update them periodically. This will provide information for future inventory compilation. Second, PRTR systems can be
used to develop indicators for Hg emission point sources. Although the methods and reporting systems applied in a given
country/system may vary with another, the facility information such as facility location will help to locate the calculated
emissions. Third, the resolution and time scale of regional emission inventories have been improved in the past decade. The
participation of national experts can help considerably both the development of emission estimates and the compilation of
gridded emission inventories that could be used to evaluate the effectiveness of control measures. Updating of global
emission inventories prepared under the EDGAR and AMAP/GMA inventory initiatives should be continued and supported,
including efforts to understand and reconcile the differences in historical emission estimates.  Finally, the information
applied to compile anthropogenic Hg emission inventories to support the MC should be transparent, reviewed and fully
documented.

**3.2.2. Future emissions**

Scenario modelling is a valuable tool to understand the effects of different policies and abatement measures on future
anthropogenic mercury emissions. For each scenario, Hg emissions are projected into the future based on a set of
assumptions and projections on energy use, industrial activity and macroeconomic development combined with a narrative
on climate, air pollution and mercury control policy. Such emissions scenarios provide vital input for atmospheric models to
assess the effects of mitigation policies on global Hg transport and deposition (see Table E6).
To project mercury emission inventories into the future, a large number of considerations need to be addressed and
reconciled with available data sources, both for present emissions and emission projections (see Table E7). Frequently,
emission projections are based on trajectories of different economic activities linked to Hg emission, computed by integrated
assessment and energy systems models (e.g., IEA 2022 Chen et al., 2022). The future emission intensity of each sector is
driven by activity levels, fuel mix changes and the impacts of emission control measures. The activity projections can
incorporate underlying assumptions on climate policy, creating a possibility for evaluating the co-benefits of different GHG
mitigation strategies on mercury emissions. Furthermore, air pollution control policy for particulate matter (PM), $SO_2$ and
$NO_x$ can be distinguished from Hg-specific policies to identify where Hg reduction materializes due to co-benefits or direct
Hg abatement.  Scenarios of emission control strategies (i.e., the application rates of technical pollution control) are included
in future global Hg emission projections either implicitly or explicitly.





### 3.2.2.1. Overview of global and regional future emission scenarios

Tables E8 and E9, respectively, summarize global and regional future Hg emission scenarios published in the literature, including their base and last projection year, source of underlying activity and energy projections, as well as assumptions on Hg control measures, where applicable. The emission sectors covered by the published global emission projections are
presented in Table E10.

*Sectoral coverage*

The sectoral coverage of the published global emission projections varies widely. The projections differ in the choice and level of details of anthropogenic emission sources by which the sectors are represented, as well as in their inclusion or exclusion of sectors such as biomass combustion (e.g., burning of crop residues or residential biomass combustion for
cooking and heating), different industrial processes using Hg, other sectors with unintentional emission such as iron and steel production, caustic soda or vinyl chloride monomer (VCM) production.

For the representation of power plants, industrial and residential combustion, some models account for different types, sizes and ages of power plants, while others simply consider an average emission factor for all coal combustion processes. Similarly, emission factors of power plants vary depending on the quality and type of fuel, as well as fuel trading patterns,
which may change in the future. Industrial processes such as iron and steel making, cement production or non-ferrous metal (NFME) production are also highly complex and may be aggregated or sub-divided into several processes, all with distinct Hg emission intensity. For example, copper, zinc and lead are reported separately per metal in most Hg emission scenarios (Streets et al., 2009; Pacyna et al., 2010, 2016; Lei et al., 2014), but are aggregated with all other NFME in Rafaj et al., (2013). On the other hand, iron and steel production are separated into sub-sectors in some (Rafaj et al., 2013) but
aggregated into one sector for other studies. Processes such as cement production, where it may be difficult to separate emissions from fuel combustion and those emitted from the raw materials, also require close attention. For the waste sector, activity projections need consideration of the changing Hg content in waste as intentional Hg use is phased out from different products. Some studies apply own calculations regarding consumption of Hg-added products (e.g., Pacyna et al., 2016; Burger Chakraborty et al., 2013) while others assume an average Hg content in waste and the emission factor scaled to
complete waste generation statistics (e.g., Rafaj et al., 2013).

*Source of energy projections*

Various integrated assessment models for energy and activity projections have been used for energy-intensive emission sources such as fossil fuel and biomass combustion, industrial activity and transport, as well as for activities driven by population growth, i.e., cremation. Frequent use has been made of climate pathways from IPCC reports, such as the IPCC
Special Report on Emission Scenarios (IPCC, 2000; Streets et al., 2009; Lei et al., 2014; Giang et al., 2015) or the IEA World Energy Outlook scenarios (Zhao et al., 2015). Others have made use of regional energy models (e.g., ReMIX for Europe, Rafaj et al., 2014; C-REM for China, Mulvaney et al., 2020). Activities of sectors not covered by energy or integrated assessment models (e.g. ASGM, some Hg-specific sectors such as chlor-alkali production, cremation, generation





of mercury-bearing wastes, VCM production) are derived from other sources and drivers. For the waste sector(s),

Chakraborty et al., (2013) considered the build-up of stocks of mercury-added products and its delay in entering the waste system.

*Emission controls*

Lei et al., (2014) use implicit assumptions on emission control technologies while others include explicit emission control strategies (Rafaj et al., 2013; Ancora et al., 2016; Pacyna et al., 2016; Wu et al., 2018b). Most scenarios compare a base case,

such as a "current legislation" (CLE) scenario, to the maximum technologically feasible reduction of Hg (HgMFR). Air pollution control policies for particulate matter (PM), $SO_2$ and $NO_x$ can be distinguished from Hg-specific policies to identify where Hg reduction materializes due to co-benefits or direct Hg abatement. In well-studied sectors like coal combustion or industrial processes like cement production, iron and steel and NFME production, a range of pollution control options, mostly relating to co-benefits from installing PM, $SO_2$ or NOx control technologies (Rafaj et al., 2013; Ancora et

al., 2016; Pacyna et al., 2016; Wu et al., 2018b), may be applied to divert Hg emission to air into solid or liquid waste streams. In other sectors, the applied measures may be more qualitative, such as good practices, bans of activities, improved storage and handling, or an assumption on the decreasing emission intensity of an activity, without explicit reference to a control measure. Mercury-specific control technologies are also entering the market. While they are not widely applied at the current moment except for very specific applications (e.g., cleaning of $H_2SO_4$ from acid plants, waste incinerators, selected

coal power plants), they may become more prominent in the future. The best-documented CLE control strategy for Hg can be found in the GMA 2018 (AMAP/ UN Environnement, 2019), where assumptions on technology efficiency and application for the year 2015 were applied to each sector-activity combination and each country – however, no assumptions on future technology application are included in this document.

*Scenario design*

Available future emission scenarios also differ in their chosen base and projection years, whether projections are available over the time series or for one end point year, and whether emissions are spatially distributed or aggregated. The GAINS model produces emission projections in 5-year steps from 2010 to 2050, but Pacyna et al., (2016) and Streets et al., (2009) project only the last year, i.e., 2035 and 2050, respectively. While Pacyna et al., (2016) and Streets et al., (2009) provide gridded emissions, Hg emissions from GAINS are not spatially distributed yet.

**3.2.2.2. Formulation of future emission scenarios**

Currently available published future emission scenarios of Hg are mostly projected based on 2010 inventories or older data on anthropogenic emissions. Several atmospheric modeling studies have estimated future Hg transport and deposition based on these projections (e.g., Schartup et al., 2022). It is recommended that the new modeling assessment of the future emission scenarios be conducted starting projections from 2015 emission inventories, the most recent year of currently reported

anthropogenic emissions. Presently, two such efforts are ongoing that are aimed at producing up-to-date future global Hg emission scenarios, summarized in Table E11:





   i.    An update of the Hg implementation in the most recent version of the **Greenhouse Gas – Air Pollution Interactions and Synergies (GAINS) model** (http://gains.iiasa.ac.at/models/gains_models4.html) at the International Institute for Applied Systems Analysis (IIASA) (Amann et al., 2011). The updated model is calibrated to GMA 2018 emissions, using 2015 as the base year, with a mixture of the most up-to-date emission factors derived from GMA 2018, supplemented by regionalized emission factors derived from coal Hg content and trading patterns. The update also includes the introduction of full range of Hg-specific control options for all relevant sectors and extends the calculation to co-benefits from PM and $SO_2$ control policies in the model. Activity data input stems from different global and regional statistics and energy systems models, notably the World Energy Outlook Scenarios which are annually published by the IEA and are fully implemented in the modeling framework. GAINS is freely available to the public, although access to Hg emissions is currently restricted due to ongoing development. Hg-GAINS 4 is fully functional as of December 2023 and publication of first results is in progress.

   ii.   Future Hg emission scenarios are being added in the **TAPS model** (Massachusetts Institute of Technology (MIT); Joint Program on the science and policy of global change) in collaboration with the Italian National Research Council, Institute of Atmospheric Pollution Research (CNR-IIA) and the European Commission Joint Research Council (JRC)). The Tool for Air Pollution Scenarios (TAPS) is a flexible and public tool that enables users to combine and study a wide range of future emissions under climate strategic objectives and air pollution control policies, using the most up-to-date emission factors (Atkinson et al., 2022). It can efficiently assess a wide range of climate and air quality policy pathways – from broad to specific at the regional, sectoral, and fuel-based levels. TAPS has three components: historical global anthropogenic emissions inventories, activity data for future scenarios from the MIT Economic Projection and Policy Analysis model (EPPA) (Chen et al., 2022), and emissions intensity trends based on recent scenario data from the GAINS model. Spatially gridded emissions outputs can be used as input variables in chemical transport models (CTMs). The Hg extension in TAPS will calculate Hg emission projections by combining the EDGAR Hg inventory, EPPA climate policy scenarios, and time-dependent Hg emission factors derived from different future Hg emission scenarios (Atkinson et al., 2022; Chen et al., 2022).

Updated future emissions should include a set of scenarios spanning policy-relevant conditions from no additional control, technical scenarios based on Hg control policy and co-benefits from other air pollutant abatement, as well as a range of plausible scenarios on climate pathways. A selected list of scenarios is outlined in Table E12. The output should be spatially gridded emissions per sector and region up until 2050 or beyond. This full set of requirements is to be available in future in both the GAINS v4 and TAPS.

There are several limitations in projecting Hg emissions into the future. Accuracy of future emissions scenarios is limited by the large uncertainties in Hg emissions in many sectors such as waste or ASGM. Many available Hg control technologies are still not widely used, as their cost is prohibitively high for many industrial applications. Cost considerations are not (yet)





included in this modelling plan but are in development by different research groups. Furthermore, assumptions are taken on future emission control technologies, their efficiency, and their impact on emission speciation. As data is scarce for such technologies, large uncertainty within speciation profiles of future emissions occurs. The future spatial distribution of emissions is also important, e.g., for modelling Hg deposition, but may be hard to project depending on the location of future

Hg emission sources. Furthermore, the emission intensity of emission sources in different regions might change over time in complex ways: Trading patterns of fuels and ores with varying Hg contents may shift or technology learning may lower emission factors of a particular emission source in a particular region.

### 3.3. Wildfire reemissions

The uptake of gaseous elemental mercury (Hg(0)) by plant stomata represents the major global sink of mercury (Hg) from

the atmosphere to terrestrial systems (Zhou et al., 2021; Feinberg et al., 2022). Combustion of Hg containing vegetation has the effect of reducing oxidized forms of Hg and releasing the highly volatile Hg(0) to the atmosphere (Biester and Scholz, 1997; DeBano, 2000; Biswas et al., 2008). Therefore, it is critically important that we can produce constrained estimates of Hg emissions from wildfires, particularly with predicted increases in fire frequency and intensity associated with climate change driven extreme weather events (McKenzie and Kennedy, 2012; Walker et al., 2019).

Without question, the greatest obstacle in the development of effective Hg wildfire emissions modelling is the scope (or the lack thereof) of existing observational data (De Simone et al., 2017b; Fraser et al., 2018; McLagan et al., 2021). From our review of the literature there are only five near-source data sources (i.e., measurements taken <100 km from fires): (1) Brunke et al., (2001) – ground-based measurements of a fynbos shrubland fire in Cape Point, South Africa; (2) Friedli et al., (2003a) – aircraft-based measurements of a boreal forest fire in Northern Ontario, Canada; (3) Friedli et al., (2003b) –

aircraft-based measurements of a temperate forest fire in Washington State, USA; and (4) McLagan et al., (2021) – aircraft-based measurements of a boreal forest fire in Northern Saskatchewan, Canada; and (5) Desservettaz et al., (2017) - ground-based measurements of North Australian savanna fires. Hg accumulation rates within vegetation vary by a factor of ≈25 across different species/biomes; accordingly, Hg concentrations in biomass also vary considerably by biome, vegetation type, and tissue type (Zhou and Obrist, 2021). With existing data limited to few biomes, large assumptions (and associated

high uncertainties) are required to estimate wildfire emissions of Hg across broad spatial scales.

All other studies that have examined Hg emissions from wildfires have involved large source-receptor distances and the emissions ratios (ERs; ratio of concentration enhancements between Hg and a reference species; typically, Hg(0):CO (Carbon monoxide)) of these studies are elevated compared to those of the four near-fire studies (for summary of these data see Wang et al., 2015; McLagan et al., 2021). McLagan et al., (2021) highlight the concerns associated with large source-

receptor distances associated with the differing atmospheric lifetimes of Hg(0) and reference compounds (i.e., CO). While ERs can be used to produce estimates of wildfire emissions by simple conversions of emissions estimates of the reference compound at the scale of interest (single fire, regional, global biome, or total global emissions), this method is highly uncertain and full uncertainties are difficult to characterize [BE1] (McLagan et al., 2021). Emissions factors (EFs) quantify





the unit mass of the target species (i.e., Hg(0)) released per unit mass of fuel combusted and are the preferred empirical

method used to generate emissions estimates. Hg EFs require the enhanced (above background) mixing ratio data of Hg (Hg(0), Hg(II)$_p$, and/or Hg(II)$_g$) and the major carbon species emitted from fires: CO, carbon dioxide (CO$_2$), methane (CH$_4$), and non-methane organic gases (NMOGs). Friedli et al., (2003b, a) measured CO and/or CO$_2$ and derived the Hg EFs from their respective fires using an assumed ratio of 90:10:0:0 for CO$_2$, CO, CH$_4$, and NMOGs. However, the study by McLagan et al., (2021) measured all four carbon species and determined a ratio of 76.5 : 13.0 : 1.3 : 9.2 (± 3.4 : 6.1 : 0.4 : 3.5). The

ratios of these co-emittants matter for the estimation of Hg emissions from wildfires, and (where possible) they should be measured in tandem with Hg species when assessing Hg releases from wildfires.

Data is further limited by the fact that the one near source study (Friedli et al., 2003b) that did measure particulate bound Hg used a less-robust bag-collection system with *ex post facto* laboratory analysis (Hg(II)$_p$). Furthermore, none of these near source studies have measured gaseous oxidized mercury. While it is assumed that any Hg(II)$_g$ species formed within wildfire

plumes would rapidly partition to the abundance of wildfire released particulates and aerosols as evidenced in laboratory burns (Obrist et al., 2008), there is no field data to validate this assumption. Previous modelling efforts have utilized the 3.8% fraction of Hg(II)$_p$ (out of total atmospheric Hg: Hg(0) + Hg(II)$_p$ + Hg(II)$_g$) that was measured in the Washington State fires (Friedli et al., 2003b) with the addition of other (assumed) Hg(II)$_p$ ratios (up to 50% Hg(II)$_p$) to provide sensitivity analyses (De Simone et al., 2017b; Fraser et al., 2018; Kumar and Wu, 2019; McLagan et al., 2021). However, we have

little-to-no knowledge of how the ratio of Hg(II)$_p$ to Hg(0) might vary by biome or fire intensity. The atmospheric lifetime of Hg(II)$_p$ (and Hg(II)$_g$) are much lower than Hg(0); altering the fraction of Hg(II)$_p$ released results in large increases in local and regional Hg deposition (De Simone et al., 2017b; Fraser et al., 2018; Kumar and Wu, 2019). Full atmospheric Hg speciation measurements within wildfire plumes are desperately needed to constrain the fraction of Hg(II)$_p$ (and potentially Hg(II)$_g$) released from wildfires and how this varies by fire intensity, vegetation type, meteorology, and duff conditions will

be critical to improving estimates of Hg emissions from wildfires.

Due to the conversion of Hg(II) to Hg(0) at temperatures observed in fire, we assume complete volatilization of Hg from combusted biomass. Yet this is not an assumption that can be made down whole soil profiles, which are a major storage pool of Hg within vegetated ecosystems (Schwesig and Matzner, 2000; Friedli et al., 2007; Obrist et al., 2012). The fraction of Hg that will be released from soil depends on the temperature the soil reaches during a fire, which in turn is dependent on fire

intensity and soil properties (i.e., texture, initial water content, chemical properties, water repellency, wet aggregate stability, and soil organic matter content) (Stoof et al., 2010; Mataix-Solera et al., 2011; Martínez et al., 2022). More data on Hg releases from laboratory-controlled burns that include reconstructed soil profiles and pre- *versus* post-fire site assessments of soil profile Hg mass balances would be greatly beneficial to characterize the contribution of Hg released from soil when modelling Hg releases from wildfires. "Bottom-up"-based approaches that utilize soil + biomass Hg concentrations to

estimate EFs and total Hg emissions from fires do exist (Biswas et al., 2007; Cinnirella and Pirrone, 2006; Turetsky et al., 2006; Engle et al., 2006; Woodruff and Cannon, 2010) and many of these are incorporated into the Hg EFs for different biomes provided by Andreae (2019). However, compared to the near-source, aircraft-based, top-down EFs in boreal and





temperate forests (Friedli et al., 2003b, a; McLagan et al., 2021), it would appear most of the "bottom-up" methods also overestimate EFs (and consequently Hg emissions) from wildfires by a factor of 3.1 ± 2.7 for these biomes (based on the studies listed in the previous sentence).

The interannual variability in the burned area of wildfires both between and within different ecosystems can be vast due to the complexities of factors that control large-scale wildfire (de Groot et al., 2013; Clarke et al., 2019). The magnitude of this variability is related to fire return interval, fuel load, meteorology, forest composition, and longer-term trends in climate (*i.e.*, seasonal, annual, decadal) that drive drought/flood cycles (de Groot et al., 2013). Moreover, there is large variability between the remote sensing-based biomass burning emissions products (i.e., GFED, FINN, GFAS, QFED, FEER) used to estimate spatially and temporally varied distributions of burned area and the mass of fuel burned with very large differences shown between inventories on regional and global scales (Giglio et al., 2013; Forkel et al., 2019; Humber et al., 2019; Pan et al., 2020). This highlights the uncertainty of remote sensing estimations of burned area and burned fuel on the coarser resolution typically used for global estimates – many small fires are missed at this spatial scale and small fires make up the majority of the total number of fires that occur (Ramo et al., 2021).

Existing global emissions estimates incorporate many of the uncertainties and limitations discussed in the previous section; and hence, they have large uncertainty terms that may not include all or fully propagated uncertainty components (McLagan et al., 2021). Friedli et al., (2009) estimated global wildfire emissions at 675 ± 240 Mg y$^{-1}$ using biome specific wildfire EFs (based off large assumptions), and Kumar et al., (2018) produced an estimate of 612 Mg y$^{-1}$ (no overall uncertainty provided) using a similar method albeit with some adjustments to certain biome specific EFs. De Simone et al., (2017b) produced an alternate estimate of 400 Mg y$^{-1}$ (no overall uncertainty provided). All these estimations include the mean of all measured ERs or EFs including in studies with "bottom-up" approaches and studies large source-receptor distances, with the exception of the most recent and robust study examining wildfire emissions (McLagan et al., 2021) that was published after all previous modelling efforts. ERs and EFs in these studies are all considerably larger than the values determined from near source measurements (McLagan et al., 2021), which has likely resulted in an overestimation of wildfire emissions. This is highlighted by contrasting the emissions estimate made for all boreal forests by McLagan et al., (2021) (18 ± 14 Mg y$^{-1}$; based on near-source measurements of a boreal forest wildfire) to Kumar et al.'s (2018) estimates for boreal forests (≈200 Mg y$^{-1}$; including EFs measured at large source-receptor distances and bottom-up assessments), which differ by an order of magnitude.

### 3.3.1. Formulating wildfire emissions

Logically, this and future modelling efforts should use the 14 fire regions typically utilized by biomass burning emissions products as the basis for the different biomes that should be assessed. For MCHgMAP, wildfire Hg emissions estimates will be developed using latest GFED4 and FINN2 biomass burning products (both products allow EF inputs rather than the more uncertain ER approach); the inherent variability between these products' estimates will provide enhanced uncertainty assessment and model sensitivity to the critical burned area and burned fuel amount parameters for the years 2010 – 2020.





EFs for each biome will be taken from Andreae (2019). As discussed, "bottom-up" approaches and studies with large source-receptor distances likely overestimate EFs in temperate and boreal biomes. Thus, we plan to utilise the mean value of EFs from the three "top-down", near-source, aircraft-based studies ($1.08 \pm 0.08$ x 10-4 g kg-1; Friedli et al., 2003b, b; McLagan et al., 2021) for temperate and boreal biomes. Lacking near-source EF data for other biomes (grassland/savanna, tropical forests, and croplands), we will use the EF estimates from Andrea (2019), unadjusted. Additionally, we will apply a range of $Hg(II)_p$ fractions (out of the total atmospheric Hg emitted) ranging from 0% (100% Hg(0) emissions) to 25% $Hg(II)_p$.

### 3.3.2. Limitations and recommendations

While we anticipate this assessment of global Hg emissions from wildfire will represent the best assessment to date, large uncertainties and limitations remain. First and foremost is the very large (and difficult to fully assess) uncertainties of EFs applied to the 14 global fire regions. The good agreement between the calculated EFs for the three aircraft study from wildfires in boreal North America and temperate North America lead us to conclude that the applied EFs based on these studies are the most constrained biome-specific EFs available. How well these values apply to similar fire regions such as Europe or boreal Asia we cannot accurately estimate without further in-depth study. Moreover, EFs for other biomes are highly uncertain; they are based on highly variable data and include estimates from very different ecosystems (i.e., EFs for Savanna/grasslands/shrublands vary by close to two orders of magnitude and include estimates from South African fynbos shrubland, South African Savanna, Mediterranean shrubland, Australian savanna, and Nevada (USA) brushland). Generalising highly variable EFs for such broad and variable ecosystems/biomes is a clear limitation of past studies and remains so for any estimates generated from this work. Future work should focus on (1) expanding the number of measurement-based EF estimates we have available (i.e., more near-source, top-down estimates), and/or (2) examining methodologies to adjust and constrain these biome specific EFs using proxy data from more spatially and ecosystem diverse measurements/estimates such as species/ecosystem specific biomass Hg concentrations or accumulation rate data (Feinberg et al., 2022). Future efforts to improve upon the rather limited top-down measurements should also look to include speciated atmospheric Hg measurement. As stated, the fraction of $Hg(II)_p$ emitted has a substantial impact on local re-deposition vs long-range transport potential of the emitted Hg. With so little speciated atmospheric Hg data measured from wildfire plumes in the literature to date, this not only adds uncertainty to the estimates of Hg emitted from wildfires, but also to the cycling and fate of that Hg after it is emitted.

### 3.4. Soil reemissions

Mercury emissions from natural surfaces include the evasion of Hg derived from the primary geogenic sources (e.g., Hg release from rock weathering processes) (see Sect. 3.1) and legacy Hg stored in the terrestrial surfaces from historic atmospheric deposition and land releases. Reemissions (or secondary emissions) of legacy Hg from terrestrial biosphere are an important source of emissions of Hg. Global Hg(0) reemission from terrestrial ecosystems has been estimated to range from 1000 to 2500 Mg y$^{-1}$ (Pirrone et al., 2010; Agnan et al., 2016; Outridge et al., 2018), based on scaling up from field flux





measurements. The relatively large uncertainty of this natural Hg emission limits the understanding of global and regional Hg cycling budgets (Pirrone et al., 2010; Wang et al., 2014b; Song et al., 2015). Using the air-surface exchange models for

estimating Hg reemission from terrestrial ecosystems provide a more systematic estimate compared to scaling up field measurements because of parameterization of physicochemical factors that influence Hg reemission fluxes at various temporal and spatial scales.

Two categories of modeling approaches have been applied to estimate Hg reemissions from soil surfaces. Table 1 lists the modeling approaches reported in the literature. One is the regression-based scheme that relates emission flux to the time

dependent environmental factors (e.g., soil Hg, temperature and solar radiation) (Xu et al., 1999; Bash et al., 2004; Lin et al., 2005; Gbor et al., 2007; Selin and Jacob, 2008; Shetty et al., 2008; Smith-Downey et al., 2010; Eckley et al., 2016; Khan et al., 2019). These statistical relationships were obtained from controlled experiments that measure air-soil exchange fluxes subject to changes in air and soil Hg concentration, soil moisture and organic content, air and soil temperature, solar irradiance, and other site-specific parameters. Field measurements suggest that soil Hg concentration and solar irradiance are

the dominant driving factors for Hg evasion from bare land and polluted soils, while other factors can influence the Hg fluxes at various degrees depending on the environmental conditions at the measurement sites (Carpi and Lindberg, 1997; Lindberg et al., 2007; Gustin et al., 2008; Agnan et al., 2016; Zhu et al., 2016). Since the soil flux data were collected at a limited number of sites and duration, such regression-based schemes may not be generally applicable across the globe because of existing data gaps in site-specific characteristics and the heterogeneous nature of soil surface.

The other approach is the bidirectional flux scheme based on state-of-the-science understanding on Hg transformation in soil and on cuticular surface under the influence of meteorological and environmental parameters (Bash, 2010; Wang et al., 2014b; Wright and Zhang, 2015; Wang et al., 2016b). This mechanistic approach depicts spatial soil flux variability more realistically and has the advantage of incorporating global land use data and model-generated meteorology in estimating the flux, although it is still limited by the availability of required soil property data and other physicochemical parameters such

as Hg(II) reduction kinetics and characteristics of interfacial exchanges (Bash, 2010; Wang et al., 2014b). Recent advances in the understanding of Hg(II) reduction kinetics provide new opportunities to enhance existing air-soil exchange modeling schemes. These include constrained $10^{-11}$ to $10^{-10}$ $s^{-1}$ pseudo-first-order rate constant of Hg(II) reduction in soil (Scholtz et al., 2003; Qureshi et al., 2011) and 0.2-1.0 $h^{-1}$ rate constant in natural water (O'Driscoll et al., 2006; Qureshi et al., 2010). In these reactions, the UV-band of actinic light has been shown to be an important driver for Hg(II) photo-reduction in soil and

water (Moore and Carpi, 2005; Si and Ariya, 2011). The role of functional substructures (e.g., -COOH, -SH, -OH) of dissolved organic matter during photoreduction has been determined by kinetic studies using model compounds (Si and Ariya, 2011; He et al., 2012; Si and Ariya, 2015). In dry soil, the first-order rate constants of Hg(II) photo-reduction are 0.007-0.028 $h^{-1}$ for $HgCl_2$ coated over sand and 0.003-0.006 $h^{-1}$ for Hg(II) in a natural soil (Quinones and Carpi, 2011). In the absence of light, Hg(II) reduction in soil is also observed at a rate of 0.001-0.002 $h^{-1}$ at 293 K (Pannu, 2013).



Parametrization of Hg reemission from natural surfaces in global Hg models have primarily used the regression-based scheme in the form of $F = aC^bR^c$ (where C is air concentrations and R is resistance). For example, the Global Terrestrial Mercury Model (GTMM) simulates Hg(0) soil emission using soil Hg concentration (C) and select environmental parameters including temperature, solar radiation, and leaf area index (Smith-Downey et al., 2010) with reemission fluxes coupled into the global chemical transport model (e.g., GEOS-Chem-Hg) and other Earth System model components (Zhang et al., 2021; Zhang and Zhang, 2022). This modeling methodology requires relatively few readily available parameters for estimating spatially and temporally resolved reemission but does not account for the heterogeneities of air-surface interface and the interplays among multiple environmental factors. Recent implementation of soil chemistry and kinetic parameters in bidirectional resistance schemes had been demonstrated for regional modeling (e.g., Bash, 2010; Wang et al., 2016c). Wang et al., (2016c) applied their mechanistic bi-directional terrestrial flux model and updated soil Hg data to estimate 565.5 Mg $y^{-1}$ of emission from soil and 100.4 Mg $y^{-1}$ uptake by vegetation in China, improving previous estimates and spatial distribution (Shetty et al., 2008). The revised Hg air-surface exchange quantity as well as its spatial and seasonal pattern agree favorably with the measured fluxes at field sites in China.

The increasing availability of global datasets, specifically the regional (China, US, and Europe) and global soil Hg distributions (Olson et al., 2022; Panagos et al., 2021; Wu et al., 2016; Wang et al., 2019), forms a basis for reducing the uncertainty of soil Hg reemission parameterizations and estimates. Currently, there is a general lack of mechanistic processes implemented in global Hg models for estimating air-vegetation-soil exchange, although spatially and temporally resolved dry and wet deposition of atmospheric Hg is calculated in the models using species-specific (for elemental, oxidized and particulate Hg) deposition schemes. Modeling work to incorporate updated process representations using more recent observational data is needed (Eckley et al., 2016; Wang et al., 2016c; Khan et al., 2019).

For the MCHgMAP MME simulations (see Sect. 5), spatially and temporally varying high-resolution global soil Hg reemissions for 2010-2020 will be developed based on the mechanistic bi-directional air-land Hg exchange model by Wang et al., (2014b, 2016c) using global soil Hg distribution (Wang et al., 2019) and spatiotemporally varying environmental physical conditions simulated by the model GEM-MACH-Hg (Zhou et al., 2021). Harmonized soil reemissions will be considered as the default option for the multi-model study, and sensitivity simulations will be performed using model-specific parameterizations of soil reemissions. In addition, influences of long-term changes in Hg emissions from soils (reflecting historic changes in anthropogenic emissions) will be investigated using mass balance modeling (see Sect. 5.3).

**Table 1. A comparison of natural surface mercury flux models. Extended from a previous review in Zhu et al., (2016), summarizing data from literature (Xu et al., 1999; Bash et al., 2004; Shetty et al., 2008; Gbor et al., 2006; Zhang et al., 2009; Bash, 2010; Wang et al., 2014b; Wright and Zhang, 2015; Selin and Jacob, 2008; Lin et al., 2010; Kikuchi et al., 2013; Eckley et al., 2016; Khan et al., 2019).**

| | General models | Description | References |
|---|---|---|---|
| Foliage | S1: $F = EC_s$ | E: transpiration rate (g m$^{-2}$ s$^{-1}$) | Xu, Yang et al. (1999); Bash, Miller et al. (2004); |





| | | | Shetty, Lin et al. (2008); Gbor, Wen et al. (2006) |
|---|---|---|---|
| | | Cs: Hg$^0$ in soil water (ng g$^{-1}$) | |
| | S2:$F_{\text{st/cu}}=\dfrac{\chi_{\text{st/cu}}-\chi_{\text{c}}}{R_{\text{st/cu}}}$ | $\chi_{\text{st/cu}}$: stomatal/cuticular compensation point (ng m$^{-3}$) | Zhang, Wright et al. (2009) |
| | | $F_{\text{st/cu}}$: air-cuticular/stomatal flux (ng m$^{-2}$ s$^{-1}$) | Bash (2010); |
| | | $\chi_{\text{c}}$: compensation point at the air-canopy (ng m$^{-3}$) | Wang, Lin et al. (2014); |
| | | $R_{\text{st/cu}}$: resistance between air-cuticular/stomatal (m s$^{-1}$) | Wright and Zhang (2015) |
| Soil | S1:$\log F=-\dfrac{\alpha}{T}+\beta\log(C)+\gamma R+\varepsilon$ | $T$: soil temperature (∘) | Xu, Yang et al. (1999);Bash, Miller et al. (2004); |
| | | $C$: soil Hg concentration (ng g$^{-1}$) | Gbor, Wen et al. (2006); Shetty, Lin et al. (2008); |
| | | $R$: solar radiation (W m$^{-2}$) | Selin, Jacob et al. (2008) |
| | S2:$\dfrac{F}{C}=\alpha T+\beta R+\delta\Theta+\delta TR+\dots$ | $T$: soil temperature (∘) | |
| | | $C$: soil Hg concentration (ng g$^{-1}$) | Lin, Gustin et al. (2010); |
| | | $R$: solar radiation (W m$^{-2}$) | Kikuchi, Ikemoto et al. (2013) |
| | | $\Theta$: soil moisture (%) | |
| | S3:$F=\dfrac{\chi_{\text{s}}-\chi_{\text{c}}}{R_{\text{g}}+R_{\text{ac}}}$ | $\chi_{\text{s}}$: soil compensation point (ng m$^{-3}$) | Zhang, Wright et al. (2009) |
| | | $\chi_{\text{c}}$: compensation point at the air–soil (ng m$^{-3}$) | Bash (2010); |
| | | $R_{\text{g}}$: resistance between air–soil (m s$^{-1}$) | Wang, Lin et al. (2014); |
| | | $R_{\text{ac}}$: In-canopy aerodynamic resistance (m s$^{-1}$) | Wright and Zhang (2015) |
| | S4: $\log F = 5.932 + 0.326 \log C + 0.210 \log R$ | $C$: soil Hg concentration (ng g$^{-1}$) | Eckley, Tate et al. (2016) |
| | | $R$: solar radiation (W m$^{-2}$) | |
| | $F = 10^{(0.709+0.119 \log C+0.137 \log R)} \times a^{-1} \sin\dfrac{\pi t}{D}$ | $C$: soil Hg concentration (ng g$^{-1}$) | Khan, Obrist et al. (2019) |
| | | $R$: solar radiation at the soil surface (W m$^{-2}$) | |
| Water | $F=\dfrac{\chi_{\text{w}}-\chi_{\text{c}}}{R_{\text{w}}+R_{\text{a}}}$ | $\chi_{\text{w}}$: water compensation point (ng m$^{-3}$) | Xu, Yang et al. (1999); Bash, Miller et al. (2004); |
| | | $\chi_{\text{c}}$: air Hg0 concentration (ng m$^{-3}$) | Gbor, Wen et al. (2006), |
| | | $R_{\text{w}}$: liquid side resistance (m s$^{-1}$) | Shetty, Lin et al. (2008); |
| | | $R_{\text{a}}$: air side resistance (m s$^{-1}$) | Bash (2010); Wang, Lin et al. (2014) |





## 3.5. Oceanic reemissions

Ocean Hg reemissions (or evasion) are driven by a combination of Hg natural (geogenic) and anthropogenic sources. Oceanic evasion is a major source of Hg to the atmosphere. Therefore, the development of spatiotemporally resolved surface ocean Hg(0) concentrations and air-ocean exchange datasets that can be used as forcing for atmospheric and multi-media

model simulations are important to the MCHgMAP EE objectives. As Hg emissions and removal of Hg from active biogeochemical cycling is not in equilibrium, the Hg concentrations in the world's oceans have slowly been increasing but with varying speed driven by their circulation time in different basins and at different depths (Amos et al., 2013). The lifetime of Hg in the surface ocean is less than a year after which the Hg will either evade to the atmosphere or be transported to deeper water layers (Soerensen et al., 2010). Atmospheric deposition and riverine inputs are external sources

entering the surface ocean but Hg from deeper ocean layers can also return to the surface waters due to, for example, thermocline circulation and seasonal mixed layer deepening (Soerensen et al., 2010). Thus, the surface ocean contains Hg that recently entered the ocean through deposition or river discharge and Hg that has cycled in the ocean for decades to millennia before reaching the surface ocean again. In addition to naturally released Hg, oceanic Hg reemissions therefore contain a large fraction of Hg originally emitted from anthropogenic sources, both recently and centuries ago (Amos et al.,

2013; Outridge et al., 2018).

Oceanic Hg evasion, a diffusive process at the air-sea interface, results in the flux of Hg(0) across this interface. The gross diffusion is bi-directional, and the net direction of the flux is determined by the relative degree of saturation of the gas Hg(0) in water compared to the air-concentrations (Soerensen et al., 2010). If the dissolved Hg(0) concentration in the surface water is undersaturated relative to equilibrium, the net Hg(0) flux is from the atmosphere to the ocean and if Hg(0) is

supersaturated, the opposite will occur. Both spatially and temporally, the surface ocean is thought to be supersaturated in terms of Hg(0) most of the time resulting in a net flux to the atmosphere (e.g., Soerensen et al., 2013, 2014; Mason et al., 2017; Huang and Zhang, 2021). The supersaturation is caused by the continuous unidirectional input of other Hg species (mainly Hg(II)) through deposition and river input, and the redox processes in the ocean resupplying the Hg(0) reservoir from that of Hg(II). The atmosphere–ocean Hg exchange is still largely unconstrained, with estimated Hg(0) evasion raging

from 4800-8300 Mg y$^{-1}$, Hg(0) deposition ranging from 1950 to 3900 Mg y$^{-1}$, and Hg(II) deposition from 2350 to 3900 Mg y$^{-1}$ (Sonke et al., 2023).

It is currently not possible to determine the origin (anthropogenic/geogenic) or period of origin of the primary emission of Hg that are currently being reemitted from the ocean based on traditional measurements of Hg concentrations (for an explanation on how the Hg(0) evasion flux is determined see Sect. 4.4 and Appendix B). However, both box and 3D models

can be used to gain insights into the extent of the Hg(0) evasion flux from the ocean, the fraction that different sources or historic periods contribute, and how concentrations will change in response to changes in Hg ocean inputs as a result of measures under the Hg MEAs. Early box model estimates of gas exchange emissions from the ocean were initially estimated using limited data, primarily from the equatorial Pacific Ocean, as summarized in Sect. 4.4. These box models provided an





early estimate of the relative change in emissions from the ocean due to anthropogenic impacts (e.g., Mason et al., 1994).
Numerical models of ocean Hg gas exchange fluxes have been developed more recently in conjunction with the observational data (see Sect. 4.4 and Appendix B).

For the MCHgMAP MME simulations, we propose to develop spatiotemporally resolved surface ocean Hg(0) concentrations using ocean models as forcings for atmospheric models when calculating ocean reemissions to the atmosphere (see Sect. 5.2 on 3D atmosphere and ocean simulations). Further, we propose to use atmospheric deposition estimates from atmospheric
models to constrain the ocean models by using the coupling approach described in Sect. 5.1. Currently, various 3D atmosphere and ocean models use slightly different parameterizations to calculate the air-sea exchange flux but for the MCHgMAP a harmonized parameterization will be used to achieve consistency of ocean evasion flux as outlined in Appendix B.

Accurate determination of air-sea exchange fluxes is currently a major limitation for the simulation and analysis of
environmental Hg cycling, given the limited observational data to constrain these estimates (discussed in Sect. 4.4: Air-sea flux exchange). The net flux determination also relies on the choice of gas exchange models, with current gas exchange models deviating especially for high wind speed estimates (Osterwalder et al., 2021). For recommendations on improving air-sea exchange estimates see Sect. 4.4.

### 3.6. Riverine export

Rivers are important for ocean Hg concentrations due to their export of Hg, nutrients and organic material to the costal ocean. Accurate representation of spatiotemporal distribution of the export of Hg, nutrient and TOC (total organic carbon) from terrestrial landscape to the ocean is needed to describe the ocean Hg levels and their environmentally driven changes. Riverine inputs include direct anthropogenic releases and Hg that has travelled through the terrestrial landscape for shorter or longer (centuries) periods potentially being stored in the process (for example in permafrost) (Campeau et al., 2022). River
inputs of Hg are controlled both by the water volume (which changes over time for example due to climate change) and the Hg concentrations in the river discharge. Furthermore, as most of the riverine Hg discharge is in the particulate phase, the amount of total suspended solids (TSS) is also an important driver of riverine Hg discharge.

There are many studies of export from single rivers and river basins, and even global bulk estimates (Fig. 6). However, only a few global spatially resolved inventories have been assembled (Amos et al., 2014; Liu et al., 2021b). Amos et al., (2014)
estimated river Hg export to be $5500\pm2700$ Mg y$^{-1}$. The more recent study by Liu et al. (2021b) arrived at a lower estimate with the input from rivers to coastal areas at 1000 (890-1220) Mg y$^{-1}$. Both studies estimated that ~30% (28-35%) of river exports make it to the open ocean. Riverine inputs thus dominate in coastal areas (75% of total inputs) and the Arctic (30% - erosion being an additional source to deposition and river export in this area) but are less important in the open ocean (<10%; Liu et al., 2021a; Dastoor et al., 2022a). The Liu et al. (2021b) inventory also, for the first time, estimated MeHg
discharge (9656 (8405-11350) kg y$^{-1}$).




For the MCHgMAP simulations, we do not recommend any new development of river export schemes. Instead, the schemes for river discharge of Hg and environmental drivers (water volume, nutrients and TOC) already integrated into the ocean models will be used (Table F6). However, we suggest a harmonization towards using the Liu et al. (2021b) Hg export inventory for all ocean models.

As seen in Fig. 6, there is still a large uncertainty in the global riverine Hg export with estimates from the past decade spanning from <300 Mg y$^{-1}$ to 5500 Mg y$^{-1}$ (Outridge et al., 2018; Amos et al., 2014). Only two studies have published a spatially resolved dataset that can be used in 3D model simulations, and these showed a factor of 5 difference in the total load. Furthermore, there is a lack of understanding of the temporal development of riverine export at the global scale. More studies on riverine Hg export resulting in spatially-resolved export datasets should be conducted with a focus on capturing

temporal trends. The export range found in current publications should further be narrowed down in the coming years. While the end goal is a fully coupled ocean-land-atmospheric model, such datasets can be used to drive ocean models until the coupled models are developed and will be of great value for evaluating coupled models once these are developed.

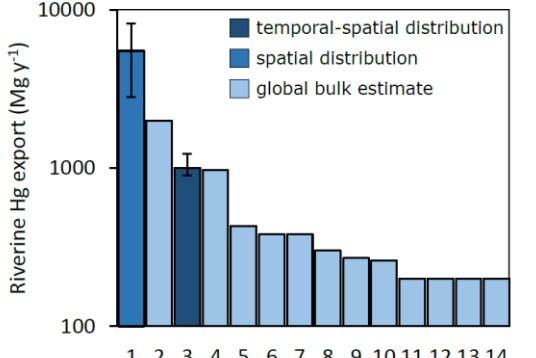
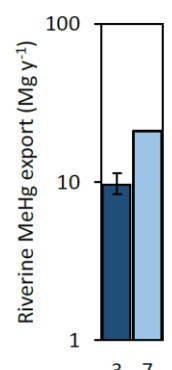

**Figure 6. Previously published riverine estimates for Hg and MeHg. 1) Amos et al. (2014), 2) Sunderland and Mason (2007), 3) Liu**
**et al. (2021b), 4) Cossa et al. (1996), 5) Amos et al. (2013), 6) Mason et al. (2012), 7) Driscoll et al. (2013), 8) Outridge et al. (2018),**
**9) Semeniuk and Dastoor (2017), 10) Zhang et al. (2014b), 11) Mason et al. (1994), 12) Mason and Sheu (2002), 13) Fitzgerald et al.**
**(2007), 14) Selin (2014).**

## 4. Observations

Observational data may provide essential model inputs or provide a basis for comparison to and evaluation of model outputs.
The following sections review the current availability of key observational data sets that may be useful for defining necessary model inputs or evaluating modeling outputs for a multi-model exercise. Specific methodologies for model evaluation are discussed in Sect. 6.

### 4.1. Air concentrations and wet deposition

The atmosphere is a key component of the Hg biogeochemical cycle, acting as a reservoir, efficient transport mechanism,
and facilitator of chemical reactions. The chemical and physical behavior of atmospheric mercury determines how, when,





and where emitted mercury pollution impacts earth's ecosystems through deposition processes. To support the EE of the Minamata Convention, the guidance report UNEP/MC/COP.4/INF/12 recommends focusing on Hg(0) concentrations and wet deposition monitoring though automated, manual or passive sampling. Monitoring of Hg(II)$_g$ and Hg(II)$_p$ is currently not prioritized for the EE due to analytical challenges (e.g., Lyman et al., 2010; Gustin et al., 2013; Jaffe et al., 2014; Huang et al., 2013). However, as noted in the guidance report, existing Hg speciation measurements can still be helpful in answering questions for the EE and can be used as first estimates.

Atmospheric Hg levels and Hg(II) wet deposition fluxes have been successfully monitored for decades through dedicated regional/global networks including e.g., the European Monitoring and Evaluation Program (EMEP; e.g., Tørseth et al., 2012), the US Atmospheric Mercury Network (AMNet; e.g., Gay et al., 2013) and National Atmospheric Deposition Program (NADP; e.g., Risch et al., 2017), the Canadian Atmospheric Mercury Measurement Network (CAMNet; e.g., Temme et al., 2007), the Global Mercury Observation System (GMOS; e.g., Sprovieri et al., 2016, 2017), or more recently through the South African Mercury Network (SAMNet) and the Asian-Pacific Mercury Monitoring Network (APMMN; Sheu et al., 2019). Hg(0) measurements primarily rely on automated active air sampling techniques, but the use of comparably inexpensive and easy-to-use passive samplers is becoming more common (e.g., McLagan et al., 2018b, 2019) progressively expanding the global coverage of atmospheric Hg(0) monitoring. The performance of passive samplers has been tested at several sites worldwide with satisfactory results (McLagan et al., 2018a). However, strict deployment and storage protocols must be followed to ensure collection of reliable results (Hoang et al., 2023). To support the MC EE, currently there is an ongoing effort to develop an overview of existing platforms managing monitoring of Hg levels in air and compile associated datasets which will serve as the basis for the MCHgMAP MME evaluation (see Sect. 6).

Building on the existing literature, Sonke et al., (2023) provided a revised modern day global Hg budget. The atmosphere is estimated to contain around 4 Gg of Hg while global total (wet and dry) Hg deposition to land includes 1600 Mg y$^{-1}$ Hg(II) deposition and 2850 ± 500 Mg y$^{-1}$ Hg(0) deposition. It is worth mentioning that despite current monitoring efforts, clear data gaps still exist, most notably in the Southern Hemisphere (Schneider et al., 2023). Recent research also shows that gaseous Hg(0) deposition is more important than Hg(II) deposition (e.g., Enrico et al., 2016; Jiskra et al., 2021) as discussed in the next section. This major deposition pathway is, however, still unmonitored due to technical challenges (Sonke et al., 2023). The advent of new analytical techniques, such as gradient or eddy covariance flux methods that allow whole-ecosystem net Hg(0) exchange measurements (e.g., Obrist et al., 2017; Osterwalder et al., 2016), will allow the scientific community to better constrain deposition fluxes and support the EE.

## 4.2. Air-land flux exchange

Air-land Hg vapor exchange fluxes have been reported at global and regional sites summarized in several studies (Agnan et al., 2016; Zhu et al., 2016; Eckley et al., 2016; Wang et al., 2016c). The Hg(0) flux data reported at global sites imply that vegetative surfaces act as a net sink of atmospheric Hg(0), while soil surfaces act as a net source of Hg to air (Cooke et al., 2020; Grigal, 2003; Obrist et al., 2012; Zhou et al., 2020). Due to the diverse land surface characteristics and the limitations



associated with the measurement methods, estimates of global air-land Hg fluxes remain highly uncertain. This section
summarizes the observations of air-land exchange of Hg reported in the literature for model evaluation and provides
recommendations for future studies. Observational data is invaluable to evaluating model performance and understanding
how soil Hg exchange has varied historically and will change in the future.

Hg stored in vegetative biomasses originates primarily from uptake of atmospheric Hg followed by assimilation in plant
tissues. Mercury uptake by vegetation is the largest sink of atmospheric Hg in the terrestrial ecosystems (60–90% of
terrestrial Hg deposition), when above-ground biomasses are removed through litterfall, plants die-off (biomass turnover),
and washout by precipitation (i.e., throughfall) (Zhou et al., 2021). Pathways of Hg uptake in vascular plants include uptake
of atmospheric Hg via surface adsorption to foliage (Stamenkovic and Gustin, 2009), stomatal and/or cuticular uptake in
foliage (Arnold et al., 2018; Greger et al., 2005; Peckham et al., 2019b), passive uptake to bark (Arnold et al., 2018;
Chiarantini et al., 2016), and uptake from soils through roots (Greger et al., 2005; Juillerat et al., 2012; Laacouri et al., 2013;
Obrist et al., 2012). Approximately 90% of Hg in foliage is derived from uptake of atmospheric Hg(0) vapor, while the Hg in
root mainly originates from the soil uptake. Translocation of root Hg uptake from below-ground to above-ground tissues in
various plant species is limited (Yuan et al., 2022). Laboratory and field measurements confirm that foliage is net sink of
atmospheric Hg and that Hg accumulation in foliage is influenced by the air Hg concentration (Assad et al., 2016; Fay and
Gustin, 2007; Millhollen et al., 2006; Niu et al., 2014); several studies including Hg stable isotopes suggest that stomatal
uptake is the dominant pathway of foliage Hg (Blackwell and Driscoll, 2015; Laacouri et al., 2013; Obrist et al., 2021; Rutter
et al., 2011; Olson et al., 2019; Yuan et al., 2019a). Other above-ground plant tissues including the outermost bark,
characterized by a high porosity and relative biochemical inertness, lack metabolic processes and most likely absorb air Hg
in elemental, oxidized and particulate forms via non-physiological adsorption (Arnold et al., 2018; Chiarantini et al., 2016).

The total Hg mass assimilated in vegetative biomasses has been estimated to be 1180±710 Mg y$^{-1}$ based on the measured
litterfall quantity and litter Hg concentration measured at global sites (Wang et al., 2016b). Whereas aboveground global
annual Hg assimilation in vegetation is estimated at 2433±483 Mg y$^{-1}$ based on measurements of Hg concentration in plant
tissues and annual biomass production in major global biomes (Zhou and Obrist, 2021). The bottom-up estimate of
atmospheric Hg uptake in vegetation is significantly higher than litterfall-based estimates, largely because of the
consideration of Hg uptake in lichens, mosses, and plant woody tissues (Melendez-Perez et al., 2014; Richardson and
Friedland, 2015; Yang et al., 2018; Zhou et al., 2017); Hg deposition mediated by non-vascular vegetation (e.g., moss and
lichen) contributes significantly to total deposition flux with a global mean Hg assimilation of 630-798 Mg y$^{-1}$ (Wang et al.,
2020c; Zhou and Obrist, 2021). The vegetation Hg deposition tends to decrease with biomass productivity from tropical to
temperate to boreal zones with approximately 70% of global litterfall-based deposition estimated to occur in tropical and
subtropical regions (Wang et al., 2016b). Hg deposition in throughfall is of similar magnitude as via litterfall at about 1,340
Mg y$^{-1}$ (Wang et al., 2020c; Zhou et al., 2020), yielding an upper estimate (i.e., via plant turnover + throughfall) of global
vegetation-derived Hg deposition of ∼3773 Mg y$^{-1}$ (Zhou and Obrist, 2021); isotope studies suggests that 34% to 82% of Hg





in throughfall is derived from adsorbed atmospheric Hg (Wang et al., 2020c), remaining being wet deposition. Hg dry deposition in terrestrial ecosystems has been projected to increase by up to 20% in the northern mid-latitudes by 2050 due to increases in plant productivity associated with $CO_2$ fertilization (Zhang et al., 2016a).

In contrast to net uptake of Hg by vegetation, air-soil Hg exchange is an important pathway of reemissions flux of legacy Hg from natural systems. Evasion of Hg from soil surfaces is derived from the geogenic sources (e.g., Hg release from rock weathering processes) and the recycling of legacy Hg deposited and stored in the terrestrial surfaces. However, Hg release from soil primarily comes from the atmospherically deposited Hg in vegetated ecosystems and in regions influenced by anthropogenic emissions (Wang et al., 2019; Zhu et al., 2016). Recent isotopic evidence suggests that ~65% of Hg mass

stored in global surface soil originated from atmospheric Hg deposition (Obrist et al., 2017; Wang et al., 2019). In forest ecosystems, the contribution of atmospheric Hg(0) deposition to soil Hg is 57% to 94% in North America (Demers et al., 2013; Zheng et al., 2016), 70% in Arctic tundra, Alaska (Obrist et al., 2017), 79% in a high-altitude peatland in the Pyrenees (Enrico et al., 2016), 90% in a boreal forest of Sweden (Jiskra et al., 2015), and 26% in the surface soil of Tibetan wetlands (Wang et al., 2019). In areas where glacial retreat has occurred, increasing soil Hg with increasing soil age was observed

along a chronosequence, attributed to atmospheric Hg deposition (Wang et al., 2020c, b).

Table 2 shows the range of up-to-date measured air-land fluxes based on compilation of documented datasets in the literature. Since Zhu et al., (2016) additional air-land Hg flux measurements have become available such as data from Australia (Schneider et al., 2023; MacSween et al., 2020), yet there are significant data gaps in many regions (e.g., Africa,

South Asia and Middle East). Legacy Hg emission fluxes from soil surfaces are highly heterogeneous depending on soil Hg concentration and environmental factors driving the reemission. Typically 1–2 orders of magnitude higher reemission fluxes have been reported from natural Hg-enriched and anthropogenic polluted sites than the fluxes from unpolluted terrestrial forest and grassland surfaces (Agnan et al., 2016; Zhu et al., 2016). The MCHgMAP MME simulated air-land flux exchange will be evaluated against site-specific observations, and the budgets will be constrained with biome-specific data on Hg in

vegetation, litterfall and throughfall.

**Table 2. An updated summary of literature reported Hg(0) flux from terrestrial land surfaces obtained from field *in situ* observations. Updated by supplement literature published post Zhu et al., (2016).**

| Landscapes | Hg flux (ng m$^{-2}$ h$^{-1}$) | | | | $N$ | References |
| --- | --- | --- | --- | --- | --- | --- |
| | Mean | Median | Min | Max | | |
| Background soil | 2.2 | 1.3 | -51.7 | 33.3 | 170 | (Lei et al., 2021; Ma et al., 2018; Cizdziel et al., 2019; Ci et al., 2016a; Fu et al., 2016; Yuan et al., 2019a) |
| Urban settings | 16.5 | 6.3 | 0 | 129.5 | 30 | (Osterwalder et al., 2016) |
| Agricultural fields | 23.9 | 10.8 | -4.1 | 183 | 63 | (Cizdziel et al., 2019; Zhu et al., 2018; Sommar et al., 2016; MacSween and |





| | | | | | | |
|---|---|---|---|---|---|---|
| | | | | | | Edwards, 2021; Wang et al., 2021a; Gao et al., 2020; Xia et al., 2021) |
| Forest foliage and canopies | -8.2 | -0.2 | -311 | 44.4 | 29 | (Fu et al., 2016; Yu et al., 2018; Luo et al., 2016; Yuan et al., 2019b; Obrist et al., 2021; Wang et al., 2022) |
| Grasslands | 4.6 | 0.4 | -18.7 | 41.5 | 48 | (Cizdziel et al., 2019; Castro and Moore, 2016; Obrist et al., 2017; Howard and Edwards, 2018; Osterwalder et al., 2020; MacSween et al., 2020; Feng et al., 2022b) |
| Wetlands | 9.1 | 1.0 | -1.9 | 85 | 32 | (Osterwalder et al., 2017; Cesário et al., 2021; Haynes et al., 2017) |
| Natural enriched surfaces | 5612 | 239 | -5493 | 239200 | 332 | (Cabassi et al., 2021) |
| Anthropogenic contaminated surfaces | 444 | 107 | -59 | 13700 | 117 | (Lei et al., 2021; Zhu et al., 2018; Osterwalder et al., 2019; Tao et al., 2017; Li et al., 2018a, b; Zhang et al., 2019a; Floreani et al., 2023) |

Currently, no long-term station measurements of Hg(0) fluxes from various terrestrial surfaces are available to evaluate modeled temporal trends; these are needed to improve the air-land Hg exchange budget estimates and to understand the kinetics of Hg(II)-Hg(0) redox chemistry and microbial processes in soils, particularly in the areas of global land use changes due to climate change and human activities. We recommend creating and maintaining an updated database of air-vegetation-soil Hg exchange measurements, specifically network-based measurements using standardized methods, to

understand the long-term trends and associated drivers. Since air-land Hg(0) exchange is bi-directional, flux measurement data cannot decouple the contribution of the deposition and evasion processes. Stable Hg isotope techniques are capable of quantifying the contribution of individual biogeochemical processes during air−soil Hg(0) exchange (Yuan et al., 2019a, 2023); their application are recommended to quantify the deposition and reemission of Hg(0) in terrestrial ecosystems.

**4.3. Ocean concentrations**

The ocean links atmospheric concentrations with those of the food web. Currently, there is no systematic monitoring of marine Hg water concentrations, but an increasing number of cruise data is being published. This provides a possibility for the development of an observational based picture of the large-scale spatial Hg variability. Ocean Hg models are important to fill the gaps in our knowledge on Hg spatiotemporal trends. However, to create reliable models, all available observational data should be used for model evaluation. There is therefore a need for the development of a seawater Hg database (UNEP,

2022 - UNEP/MC/COP.4/INF/25) including data from as many expeditions as possible. Synthesizing the ocean concentration data would help describe spatial trends and play an important part for biogeochemical model evaluation.





A range of studies have presented observation-based basin and global ocean Hg concentration estimates for total or speciated Hg concentrations (e.g., Mason et al., 2012; Lamborg et al., 2014; Soerensen et al., 2016a) but Bowman et al., (2020) does so based on the largest number of observations (over 200 high resolution, full-depth profiles of speciated Hg). The authors

take advantage of recent global-scale oceanographic survey programs (CLIVAR and GEOTRACERS) focusing on offshore waters. Still, a large fraction of data is collected outside of these two programs, many of these in pollution impacted regions like downwind of Asia (Yang et al., 2017; Marumoto et al., 2018; Liu et al., 2020). A large subset of existing sample locations is indicated in Fig. 7 to give an idea about the global geographical extent of current ocean Hg observations.

A simple database in SQL, R or another appropriate tool is recommended to be created based on the extraction of cruise

information (period 2007-2022) from a few large data storage facilities (GEOTRACERS portal, BCO-DMO, Pangea, BODC) and integration of other publicly available data sets (for example available in the supporting information of published papers). Furthermore, authors who have not yet made their datasets publicly available may be contacted to augment the ocean Hg observation coverage. The database will facilitate the analysis of observational patterns of oceanic Hg and MCHgMAP model evaluation.

For analysis of observational patterns, we suggest building on the methodology from Bowman et al., (2020) but with an extended number of observations. For MCHgMAP, a software package will be developed to 1) perform simple data analyses with focus on investigating spatial trends by aggregating and computing standard statistics on data within user defined timeframes, depths, and areas (for example based on FOA major fishing areas; Table C2) similar to the model receptor regions (Sect. 6.2) or other divisions like ocean basin, provinces, coastal boundaries:

https://www.marineregions.org/sources.php) and 2) create an observational based global mass budget for speciated Hg in the ocean.

The major limitation of ocean Hg observations is linked to the estimation of temporal trends, which is currently not feasible, neither long term (pre-2005) nor for the past decade. The lack of long-term stations measuring Hg in the ocean means that we must rely on sporadic cruises (different time of year, different ship tracks). Furthermore, for deeper offshore water the

ocean response time can be decades to centuries (Amos et al., 2013), while the variability in Hg observations for coastal and surface water are likely to be higher than anthropogenically driven decadal trends due to the sporadic observational coverage. There is also uncertainty related with observations performed prior to the introduction of clean measurements technique, which were established in the 1980s but not uniformly used until later (Gill and Fitzgerald, 1987; Soerensen et al., 2012). This makes it difficult to know if/how to include older observations (that could otherwise help explore multi-decadal temporal trends).

temporal trends).

To inform future EE efforts of Hg MEAs, the proposed database may be further developed by creating a formal network as suggested in UNEP (2021; UNEP/MC/COP.4/INF/12) and by creating an official way to submit new data to the database.



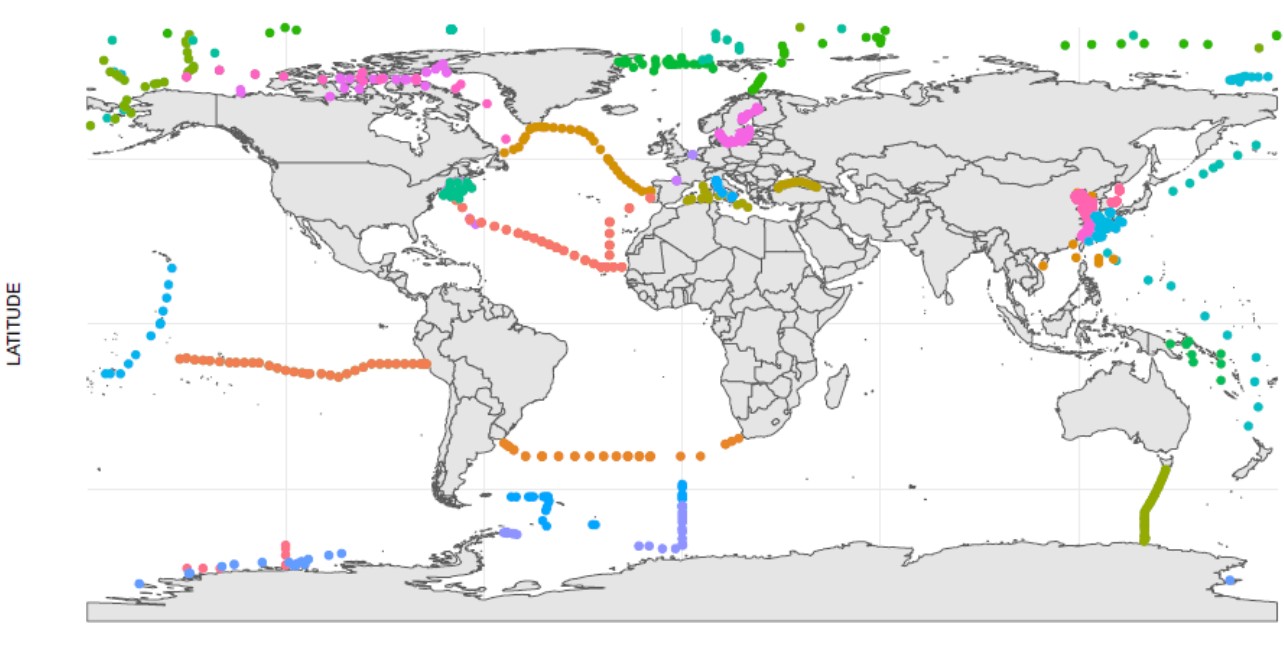

**Figure 7. THg and MeHg marine sampling locations conducted with rosette during the past ~15 years showing a global coverage (except for the Indian Ocean and parts of the Pacific Ocean). Note that this is a subset of existing published datasets. References: Agather et al., (2019), Bowman et al., (2015, 2016), Bratkič et al. (2016), Ci et al., (2016b), Cossa et al., (2018), Hammerschmidt et al., (2013), Jonsson et al., (2022), Kim et al., (2017, 2020), Marumoto et al., (2018), Munson et al., (2015), Nerentorp Mastromonaco et al., (2017b, a), Perrot et al., (2023), Petrova et al., (2020), Sharif et al., (2014), Soerensen et al., (2013, 2016a, 2018), Yang et al., (2017), Yue et al., (2023), Wang et al., (2016a, 2020a). GEOTRACES cruises downloaded from the repository [www.bodc.ac.uk](www.bodc.ac.uk).**

## 4.4. Air-sea flux exchange

The ocean can exchange Hg with the atmosphere via diffusion of Hg(0) providing a sink for Hg in the ocean and prolonging the active cycling of Hg in the environment (Mason et al., 2017; Amos et al., 2013). There is currently no systematic monitoring of Hg fluxes and only a single study (Osterwalder et al., 2021) has measured the Hg(0) air-sea exchange flux directly with the use of micrometeorological methods in a coastal area. In all other cases, the Hg(0) flux is estimated through the use of thin film gas exchange models such as Liss and Merlivat (1986) and Wanninkhof (1992). These models use in situ measurements of Hg(0) in surface water and air together with wind speed dependent parameterization of the gas transfer velocity to calculate a flux. Flux estimation using the gas exchange model is dependent on especially Hg(0) measurements in surface water (Hg(0) in the atmosphere is less variable and therefore not as important in the calculation). However, while past evasion estimates were based on few Hg(0) observations collected on research cruises, in the last two decades, the amount of information has increased dramatically through the use of underway continuous sampling systems and the involvement in the GEOTRACES program (e.g., Andersson et al., 2008b; Mason et al., 2017; DiMento et al., 2019). The Hg air-sea exchange data (i.e., Hg(0) concentrations) from previous published papers (examples from recent years are Yue et al., 2022; Kuss et al., 2011; Soerensen et al., 2014; DiMento et al., 2019; Marumoto et al., 2018) cover nearly all oceans but on





different time scales. These datasets have not been systematically stored anywhere and there is therefore a need to develop a
database including all information needed to calculate air-sea exchange based on gas exchange models. Synthesizing the
Hg(0) ocean concentration data and auxiliary variables will help describe spatial trends and play an important part in the
biogeochemical model evaluation.

While the major Hg species degassing from the surface ocean to the atmosphere is Hg(0) as described above, the current
understanding is that both MMHg and DMHg can also evade from the surface ocean as a gas or within sea spray
(Hammerschmidt et al., 2007; Weiss-Penzias et al., 2016). While such a flux can be important for distributing MeHg from
the ocean to land, this flux is considered to be much smaller than that of Hg(0) and estimates are still so few that it may not
be possible to include consideration of this flux at this time.

For the current multi-model study, we propose to include observations relevant for calculating air-sea exchange (i.e., Hg(0)
concentrations in surface air and water and associated environmental variables such as wind speed, air and water
temperature, UV radiation, ozone, other gas concentrations) into a similar format as the Hg Ocean Observational Database
described in Sect. 4.3. In the cases where raw data does not exist or is not available, the database could keep the averaged or
ranged data for future reference. This would provide a way to review spatial and temporal trends in the context of the Hg
biogeochemical cycling. The compiled data could be used to 1) create the best possible observationally based map that
spatially resolves air-sea exchange on a global scale, 2) evaluate ocean models (see Sect. 6.2) and 3) create an observational
based global flux budget for Hg(0) across the air-sea interface (using the harmonized air-sea exchange formulation for the
MCHgMAP described in Appendix B).

Currently, the patchiness of observations (geographical and temporal) in coastal and remote regions is a significant
limitation. Furthermore, there are still a limited number of research groups that make high resolution surface water Hg(0)
measurements and intercomparison between continuous analyzers and batch sample analysis are generally missing. To
inform the future EE efforts of Hg MEAs, repeated observations of Hg(0) at one or more ocean locations to better evaluate
seasonal and annual trends are needed. Opportunities for such an endeavor could be the deployment of continuous dissolved
gaseous mercury analyzers on cruises at stations where routine repeat cruises occur (e.g. BATS (Bermuda Atlantic Time
Series) or HOTS (Hawaii Ocean Time Series) cruises, or on ferries or other vessels (research ships, as done for $CO_2$) that
cover a specific path routinely or are at sea continuously. The use of micrometeorological methods for ocean studies should
also be further developed to provide an additional approach to constrain the Hg(0) evasion flux. For recommendation on the
Hg Ocean Observational Database see Sect. 4.3.

### 4.5. Environmental Archives

One challenge in assessing the effectiveness of Hg MEAs is the lack of well-defined pre-impact and pre-industrial baselines
for Hg in the environment. Anthropogenic activities have caused the release of significant quantities of Hg to air, water and
soil beginning around 1550 (Streets et al., 2017, 2019a). To determine a true pre-impact baseline, long-term temporal records
of Hg that not only date back to pre-industrial (pre-1850) but also pre-1550 are needed. Direct measurement of atmospheric





Hg can be logistically challenging, particularly in remote locations, and has only been possible since the 1990s. Very few monitoring stations exist around the world, and therefore the instrumental record is temporally and spatially limited. In the absence of direct long-term Hg measurements in the environment, natural archives such as peat bogs (Bindler, 2003; Enrico et al., 2017), lake sediments (Engstrom et al., 2014; Muir et al., 2009), ice cores (Beal et al., 2015) and tree-rings (Clackett et al., 2018; Ghotra et al., 2020) provide a means to reconstruct long-term trends in atmospheric Hg concentration and deposition, and hence provide valuable information on long term changes in anthropogenic Hg emissions and its environmental cycling via reemission and soil mobilization. Thus, natural archives provide important constraints for developing and evaluating Hg models (especially mass balance models). Mass balance models, in turn, provide an understanding of the lifetime of Hg in various environmental matrices, which is important to MC. Natural archives show increase in Hg deposition to remote locations by a factor of 3–5 since industrial revolution (1850) and peak in the 1970s followed by 2-fold decline, but also a lack of decline in some remote lake sediment archives likely due to continued mobilization of legacy Hg from watershed soils (Sonke et al., 2023; Zhang et al., 2014b; UNEP, 2019; Li et al., 2020a; Faïn et al., 2009; Enrico et al., 2017).

Lake sediment records are produced by the annual sinking of particulate-bound Hg to lake bottoms and reflect Hg that has entered lakes both directly via atmospheric deposition as well as from catchment contributions. Lake sediment records have shown to be reliable for reconstructing historical trends in total deposition of Hg. For example, sediment derived Hg deposition estimates have been shown to compare well to those obtained from wet precipitation instruments for modern times (Wiklund et al., 2017; Roberts et al., 2019, 2021), to those derived from the GEM-MACH-Hg model (Muir et al., 2009; Roberts et al., 2021) and to reliably track changes in reported Hg emissions from various industrial operations, such as smelters, Hg and gold mines, and chlor-alkali plants (Lockhart et al., 2000; Wiklund et al., 2017; Roberts et al., 2019). Unlike lake sediments, ice caps and glaciers receive Hg inputs solely from atmospheric sources, and thus can be useful for reconstructing temporal trends in atmospheric Hg deposition. Because ice mass accumulation rates are known to impact rates of Hg deposition and accumulation, either by increasing Hg deposition or the efficiency that this Hg is buried and accumulated (as opposed to remitted to the atmosphere following photoreduction), ice core records are more useful for reconstructing temporal trends in atmospheric deposition, rather than for determining flux magnitude (Lemire, 2021). Similar to ice cores, tree-rings can be used to determine the pre-anthropogenic impact baseline for Hg in the environment as tree-ring chronologies can extend back 1000 years or more (Clackett et al., 2018; Ghotra et al., 2020). Tree-rings have the distinct advantage of being annually resolved and dated to the calendar year using standard dendrochronology methods (Fritts, 1976) and are not subject to large radiometric dating uncertainties and low temporal resolution that commonly apply to sedimentary archives. Both experimental and field data suggest that trees actively assimilate mainly Hg(0) species from the atmosphere via stomatal uptake, followed by translocation through the phloem and accumulation in annual growth rings (Peckham et al., 2019a; Arnold et al., 2018; Stamenkovic and Gustin, 2009; Laacouri et al., 2013; Maillard et al., 2016; Zhang et al., 1995; Siwik et al., 2010; Odabasi et al., 2016; Navrátil et al., 2019; Jung and Ahn, 2017; Hojdová et al., 2011; Becnel et al., 2004; McLagan et al., 2022). Thus, tree-rings can serve as biomonitors for atmospheric Hg(0).





Due to the ease of collecting "short cores" using gravity corers and the availability of $^{210}$Pb dating to derive ages and sediment accumulation rates (g m$^{-2}$ y$^{-1}$), there are numerous published Hg sediment records spanning the past 100-200 years, with $^{137}$Cs dating often also used as an independent tracer to validate the $^{210}$Pb chronology (Blais et al., 1995). Cores which span the pre- and post-industrial periods (pre- and post-1850) enable levels of post-industrialization anthropogenic Hg

enrichment and depositional fluxes (µg m$^{-2}$ y$^{-1}$) to be calculated. The comparison between sediment derived Hg fluxes from multiple lakes/locations is commonly conducted by applying a sediment focusing factor based on a comparison of $^{210}$Pb fluxes in lake sediment and catchment soils, which accounts for in-lake sedimentation processes (Blais et al., 1995; Perry et al., 2005; Muir et al., 2009). Anthropogenic versus geogenic origin of Hg in lake sediments can be isolated by applying geochemical normalization to Hg concentration data given a strong correlation between Hg and lithogenic elements such as

aluminum (Al), lithium (Li) and titanium (Ti) (Loring, 1991; Kersten and Smedes, 2002; Wiklund et al., 2017; Roberts et al., 2019). Furthermore, catchment Hg runoff to lakes can also be subtracted from the sedimentary record using a "catchment effect" correction when sedimentary Hg accumulation and the catchment area to lake area ratio are linearly correlated, allowing a reconstruction of atmospheric deposition (Drevnick et al., 2012; Wiklund et al., 2017; Roberts et al., 2019). There are several recent papers that summarize spatial and temporal trends Hg fluxes over large geographical areas using 30-138

lake sediment cores; however, deposition Hg fluxes in these papers have been corrected to varying degrees (Drevnick et al., 2016; Muir et al., 2009; Kirk et al., 2011; Roberts et al., 2021). Longer term records covering the pre-industrial period are less common as they often require longer cores collected using a percussion corer and a combination of $^{210}$Pb and $^{14}$C dating methods; however depositional Hg fluxes spanning the last 500-2000 years are available from cores collected in remote regions of North America, South America, and an equatorial crater lake from Mount Kilimanjaro, Africa (Engstrom et al.,

2014; Biester et al., 2018).

Ice core dates, an age-depth scale, and ice mass accumulation rates are derived from $\delta^{18}$O record and ion chemistry analyzed over the length of the core. There are several Hg deposition records inferred from ice and firn cores from the Arctic (Greenland; Ellesmere Island including Mount Oxford, Agassiz ice cap, and Prince of Wales icefield; Devon Island: Devon ice cap; Baffin Island: Penny ice cap) and high elevation lower latitude sites (Yukon: Mt. Logan; Wyoming: Upper Fremont

Glacier; Central Asia: Belukha glacier) (Beal et al., 2015; Eyrikh et al., 2017; Zheng et al., 2014; Zheng, 2015; Schuster et al., 2002; Chellman et al., 2017; Zdanowicz et al., 2013, 2015, 2016, 2018; Lemire, 2021); however few of these capture the pre-industrial period (Beal et al., 2015; Eyrikh et al., 2017; Zheng, 2015; Jitaru et al., 2009).

The development of tree-ring as a tool to monitor trends in atmospheric Hg is an area of active research, therefore, tree-ring records of Hg are still subject to a number of limitations. For example, tree-ring Hg measurements provide information on

the relative change in atmospheric Hg(0) over time but cannot be converted to absolute concentrations in air, at least not until a number of tree-ring records are calibrated against co-located direct measurements of Hg(0) in air. Because of the mechanism of Hg(0) uptake from air into plants via stomata, tree-ring Hg concentrations reflect growing season atmospheric Hg(0) concentrations and not annual mean Hg(0) concentrations. To date, the ability of tree-rings to faithfully record atmospheric mercury has only been investigated in a small number of species, and some of those species were shown to be





inadequate biomonitors due to translocation causing a shift in the temporal dendrochemical record (Nováková et al., 2021;
        Wang et al., 2021c). The applicability of tree-ring Hg records to spatial coverage might be limited due to the geographical
        range of suitable tree species, similarly to how ice core records of Hg deposition and accumulation are limited to high
        latitude/high elevation sites.

        To inform the MCHgMAP multi-model study, temporal trends of Hg concentrations and depositional flux magnitude derived
from sediment cores, ice cores, and tree ring records for both the pre- and post-industrial periods will be compiled and
        compared to constrain Hg mass balance modelling. Both total/uncorrected and fully corrected (for sediment focusing and
        catchment contributions) sediment-derived Hg fluxes may be compiled, with the latter likely better reflecting wet plus dry
        atmospheric contributions.

        Since lakes and forested areas are widely distributed around the world, sediment core and tree-ring Hg data allow for the
potential development of a global network of multi-century Hg timeseries (Eccles et al., 2020) to evaluate global
        atmospheric Hg emissions, as well as the spatial footprint and environmental impact of local Hg(0) emissions from point
        sources (Clackett et al., 2021; Navrátil et al., 2017, 2018; Perone et al., 2018; Schneider et al., 2019; McLagan et al., 2022).
        Because temporal changes in atmospheric Hg(0) concentrations are not globally synchronous (Dastoor et al., 2015; Steffen et
        al., 2015), sediment cores and tree-ring records collected from different geographic locations can elucidate the spatial
variability in these trends. A globally distributed network of sediment-derived fluxes, including a subset of long cores
        spanning pre-1550, and corrected for sediment focusing and catchment or geogenic Hg inputs could be particularly useful for
        quantifying both baseline and changes in Hg deposition over time. Ice core data is useful for assessing changes in Hg
        deposition at high elevation, high latitude locations and efforts should be made to collect these data prior to further
        degradation of the Hg deposition record due to climate change-driven melting of glacial ice. Furthermore, records of Hg
concentrations and deposition across time and space should be further leveraged and used to assist with the validation of
        mechanistic process-driven mercury models. Finally, the quantification of Hg stable isotope ratios in sediment and cores and
        tree rings holds promise for source attribution, including resolving Hg local industrial sources from regional/global sources
        (Lee et al., 2021; Sun et al., 2022; Lepak et al., 2020; Scanlon et al., 2020).

## 5. Coordinated multi-modeling design

The available models and data described in the preceding sections can provide essential information to the MC and LRTAP
        EE efforts, addressing several of the guiding questions outlined in the MC monitoring guidance (UNEP/MC/COP.4/INF/12;
        see Appendix A). Mechanistic models are essential tools for distinguishing the influences of temporal trends in emissions
        and releases from other environmental drivers on observed trends in Hg levels in air, biota, and humans. This section
        describes the MCHgMAP experimental design for a coordinated set of multi-media multi-model ensemble simulations
between atmospheric, oceanic, and mass balance models to develop EE-relevant insights into global and regional Hg
        cycling.





3D atmospheric and oceanic models require geospatially and chemically distributed time series of emissions to drive and observational data to evaluate the models. Preparation of global anthropogenic Hg emissions inventories is a complex exercise (see Sect. 3.2), and the methods have been continuously updated since the development of initial inventory for 1990 (AMAP/UNEP, 2008). Currently, most up-to-date global anthropogenic Hg emissions inventories are available starting ca. 2010 (e.g., UNEP, 2019; Streets et al., 2019b). The global anthropogenic Hg emissions inventories are planned to be extended to 2020 to facilitate the effectiveness evaluation of the MC (see Sect. 3.2.1). In addition, development of future scenarios of Hg emissions up to 2050 spanning policy-relevant conditions ranging from no additional control, MC Hg control policy, co-benefits from other air pollutants' abatement, and to a range of climate scenarios are planned (see Sect. 3.2.2). Regarding Hg observations for the MME evaluation, a literature survey of Hg observations from various environmental compartments suggests a more comprehensive coverage in the last decade; these datasets will be prepared in coordination with other ongoing efforts to support the MC EE (see Sect. 4).

Since anthropogenic emission inventories as well as observations are best defined starting around 2010, detailed 3D atmospheric and oceanic multi-model simulations are proposed for the years 2010 – 2020 to analyze the spatial patterns and temporal trends of Hg levels from the period beginning prior to the MC (circa 2010) and extending to as close as possible to the present (referred to as "recent levels and trends"). Future projections of Hg levels in air and ocean up to 2050 will be simulated using the proposed set of future Hg emissions scenarios (see Sect. 3.2.2). In addition to 3D modeling of recent historic and near future Hg levels, examination of the long-term changes in earth's biogeochemical cycling of Hg from the pre-anthropogenic period to the present and future is proposed using multi-media mass balance model simulations. Mass balance models are driven by all-time changes in geographically aggregated primary emissions and releases of Hg estimated from information on historic anthropogenic activities (Guerrero and Schneider, 2023; Streets et al., 2019a; Amos et al., 2015), and are evaluated against Hg levels from natural environmental Hg archives (see Sect. 4.5).

## 5.1. MCHgMAP simulation approach

Several Hg multi-model ensemble intercomparison studies have been carried out during the past decade on global (AMAP/UNEP, 2013; UNEP, 2019; Travnikov et al., 2017; Bieser et al., 2017) and regional (Angot et al., 2016; Dastoor et al., 2022a, b; Gencarelli et al., 2017) scales. These multi-model studies were mainly focused on the assessment of atmospheric Hg, its spatial distribution, source-receptor relationships, chemical mechanisms, and other processes. Furthermore, the model simulations in these multi-model studies lacked full accounting of changes in secondary Hg emissions from land and ocean, making them inadequate to analyze temporal trends of environmental Hg. In MCHgMAP, we expand the multi-model Hg assessment to global oceans and propose a harmonized MME simulation approach between 3D atmospheric and marine models, multi-media mass balance models, and a mechanistic air-land Hg exchange model to account for the short- and long-term influences of changes in primary anthropogenic Hg sources and environmental conditions on Hg cycling between land, atmosphere, and ocean.



The harmonized modeling approach is designed to simulate the evolution of primary and secondary sources of Hg emissions
        and releases in air and ocean: (1) to generate spatially-resolved maps of recent Hg levels (concentrations and fluxes), filling
        monitoring gaps; (2) to detect their spatial gradients and temporal trends; (3) to attribute the levels and trends to emission
        sources and environmental drivers; (4) to quantify the impacts of MC and LRTAP implementations on Hg levels and trends;
        (5) to quantify uncertainty and sensitivity of modeled Hg levels and trends; (6) to develop insights into future Hg cycling

under different scenarios of implementation of the Minamata Convention and other MEAs and changing environmental
        conditions across global regions; and (7) to improve the understanding of environmental Hg processes and their
        parameterizations in models. The simulations address the MC and LRTAP EE relevant policy questions listed in Appendix
        A.

        To accomplish the above objectives, we broadly define three types of model simulations:

*Baseline simulation*: a state-of-the-art simulation of Hg levels and trends over the historical or future period,
        capturing the range of variability between models.

        *Perturbation simulations*: a simulation either zeroing out a model driver, fixing a model driver to a certain year's
        values, or fixing all drivers but one to a certain year's values, conducted to diagnose the contribution of the driver to
        Hg levels or temporal trends.

*Sensitivity simulations*: a simulation using a modified driver dataset or parametrization, conducted to probe the
        sensitivity of model results to that driver or process.

        Additional details of the simulations design are provided in the next two sections. Below we describe the proposed approach
        for the coordination of MME simulations of atmospheric, oceanic, and mass-balance models in the context of analyzing

recent Hg levels; an analogous strategy will be adopted to conduct coupled atmosphere-ocean simulations for future
        scenarios. Close collaboration to equilibrate the ensemble of atmospheric and oceanic models is proposed during the baseline
        and perturbation simulations (Table F8-10), which is important to produce consistent estimates of Hg levels and fluxes
        between atmosphere and oceans. This approach consists of the following four stages, illustrated in Fig. 8:

        *Stage 1: Initialization of atmospheric and oceanic models with harmonized drivers*

The atmospheric modeling groups will conduct year 2015 simulations with the updated set of harmonized model
        drivers (Table F5) and the Hg(0) sea surface concentration dataset from Horowitz et al., (2017). Ensemble monthly
        mean deposition and concentration data for the year 2015 from the atmospheric models will be transferred to ocean
        modeling groups. Ocean models will use the year 2015 Hg(II) deposition and Hg(0) air concentration fields as
        inputs for a spin up simulation long enough to equilibrate Hg concentrations in the ocean adopting drivers proposed

in Table F6.

        *Stage 2: Baseline simulations in atmospheric and oceanic models*

        For consistent baseline simulations between atmospheric and oceanic models to reproduce environmental Hg levels,
        we will follow an iterative procedure, previously used to couple MITgcm and GEOS-Chem (Horowitz et al., 2017).






Simulations for the period 2010–2020 will be initiated in atmospheric models, using the Hg(0) sea surface concentration dataset from Horowitz et al., (2017). Hg(II) deposition and Hg(0) air concentration fields from the individual atmospheric models will be averaged into ensemble monthly means. The 2010–2020 ensemble mean atmospheric fields will be passed to oceanic Hg models, to be used as inputs for their 2010–2020 simulations. Sea surface concentrations of Hg(0) from the oceanic models will be averaged into ensemble monthly means for 2010–2020, which can then be provided back to the atmospheric models for input into their simulations. We will conduct


2–3 iterations of this approach to reach convergence of the ensemble mean of atmospheric and oceanic baseline model simulations. This procedure will produce 2010–2020 ensemble mean sea surface Hg(0) concentrations (which will be the standard inputs for further atmospheric simulations) and ensemble mean deposition and Hg(0) air concentration fields (which will be the standard inputs for further oceanic simulations).

*Stage 3a: Mass balance simulations with updated rate constants*


Mass balance models require globally averaged estimates of rate constants for processes that transfer Hg between different environmental compartments (see Table F7). The ensemble mean data for the atmospheric and oceanic baseline simulations (2010-2020) will be used to calculate updated rate constants needed for the mass balance simulations. Rate constants for terrestrial pools will be estimated based on past literature (Amos et al., 2013, 2015; Qureshi et al., 2011). In doing so, we will update the GBC box models to be consistent with the state-of-the-art 3D


atmospheric and oceanic models for the purposes of this study. Uncertainty bounds for the rate constants will also be estimated from the variability of individual atmospheric and oceanic models within the ensemble, for use in sensitivity simulations of the mass balance model. The legacy reemissions trend and magnitude predicted by the GBC box model can be used as input for further sensitivity simulations in the 3D atmospheric and oceanic models.

*Stage 3b: Perturbation simulations in atmospheric and oceanic models*


Atmospheric models will run perturbation simulations for 2010–2020 (Table F8), for example turning off all anthropogenic Hg emissions. The individual atmospheric models' deposition and concentration fields will be averaged into ensemble monthly mean fields. The atmospheric ensemble mean fields will be passed to the oceanic models, as inputs for their perturbation simulations. As opposed to Stage 2, a selection of perturbations experiments will be considered for multiple iterative rounds of simulations for the atmosphere-ocean coupling, depending on the


computational expense and time constraints. For example, a coupled atmosphere-ocean "nature simulation" for 2010-2020, where primary anthropogenic emissions are kept fixed at 2010 level and the Hg trend is solely driven by natural factors, will be performed using the iterative approach.

*Stage 4: Further sensitivity analysis in models*

Atmospheric, oceanic, and mass balance models will conduct further sensitivity simulations, e.g., testing the


uncertainty of model simulations to different driver datasets, conducting idealized emission trend experiments, and testing the uncertainty in different model parametrizations (e.g., air-sea exchange parametrizations) and input fields





(e.g., Hg and MeHg loads from rivers in the ocean models). During this phase, atmospheric and oceanic models do not require coupling for their individual experiments.

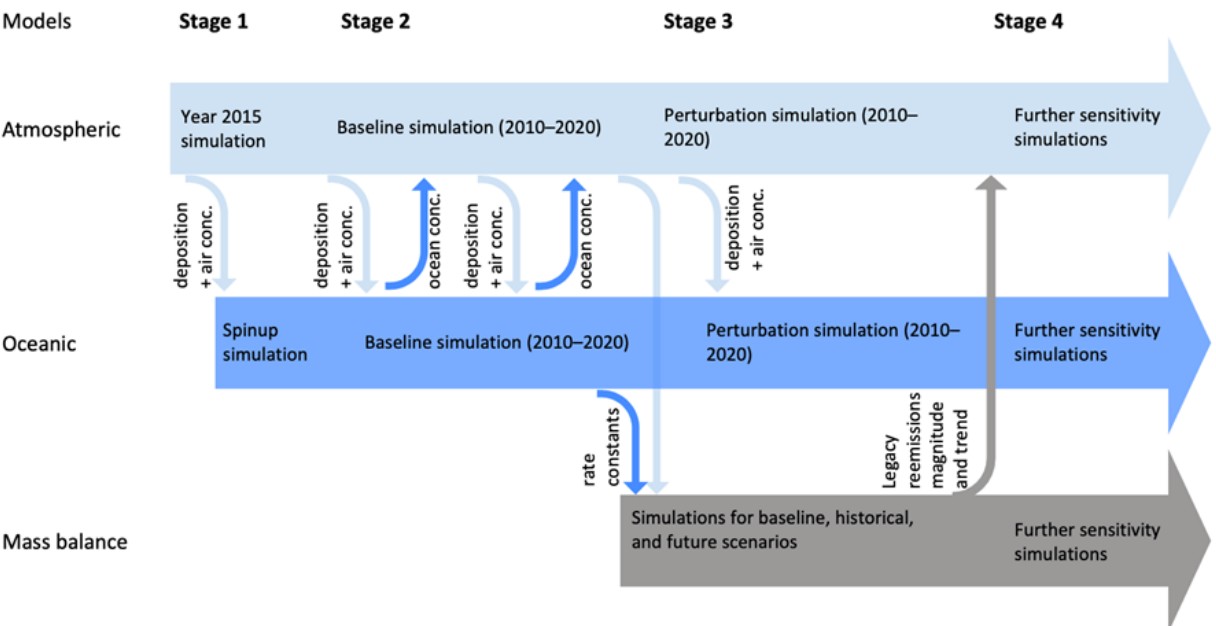

**Figure 8. Diagram showing how the coupling between atmospheric, oceanic, and mass balance models will be achieved by conducting the work in stages and exchanging information between model types in MCHgMAP.**

## 5.2. 3D atmosphere and ocean simulations

The proposed coordinated multi-model atmospheric and oceanic model simulations for the MCHgMAP are presented in Table F8-9. Most of the model simulations will be performed for the period 2010-2020 for historical trends and from 2020 to

2050 for the future scenarios except for several optional sensitivity simulations, which can be conducted for individual years. Appropriate model spin up to reach steady-state conditions is required; a consecutive simulation from 1980 to 2020 starting from semi-realistic initial conditions is recommended for ocean models, while spin up of approximately 5 years is needed for atmospheric models. The primary and secondary emissions for atmospheric models over land are provided by emission inventories and mechanistic air-land Hg exchange model simulations, respectively, (Sect. 3) and over oceans the boundary

conditions are exchanged between atmospheric and ocean models as described in the previous section (Fig. 8). The model simulations in Table F8-9 are divided into five groups in accordance with their purposes:

i. The '*Baseline*' simulation (A1; O1) aims to obtain global patterns and temporal trends of atmospheric and oceanic Hg concentrations and fluxes utilizing state-of-the-art model configurations for model evaluation (Sect. 6) and analysis (Sect. 7 & 8) of observed Hg levels based on the modelling. The simulation will be performed using

temporally varying default set of drivers (marked in bold in Table F5-6) that are harmonized as much as possible





between the models (using a combination of updated harmonized and model-specific input datasets and parametrizations).

ii. The '*Source contributions*' group (A2.1-A2.6; O2.1-O2.6) is focused on assessment of the relative contributions of various primary and secondary emissions and releases from anthropogenic and natural sources to Hg levels in the air and ocean. It consists of perturbation simulations with zeroed emissions from specific source types.

iii. The '*Trend analysis*' group (A3a.1-A3c.12; O3.1-O3.14) is intended to evaluate contributions of various anthropogenic and environmental drivers to recent changes in Hg levels over the considered period (2010-2020). This group is divided into three subgroups: '*Contributions of various drivers to trends*' (A3a.1-A3a.4), '*Effects of changes in various drivers on trends*' (A3b.1-A3b.7), and '*Contributions of HTAP regions anthropogenic emissions changes to trends* (A3c.1-A3c.12)*. HTAP regions are the geographical world regions used in previous modeling experiments organized by the LRTAP Task Force on Hemispheric Transport of Air Pollution (Galmarini et al., 2017). These regions are defined in Appendix C. The perturbation simulations in the first and third subgroups are carried out with one driver fixed at the values corresponding to 2010 for the whole period, with other drivers varying in accordance with the Baseline simulation. The perturbation simulations in the second subgroup are carried out with one driver varying as in the Baseline simulation, and all other drivers fixed at values corresponding to 2010. The comparison of the perturbation simulations with the results of the Baseline simulation provides contributions or effects of the considered drivers to the trend in Hg levels.

iv. The '*Sensitivity analysis*' group (A4.1-A5.10; O4.1-O4.7) is divided into two subgroups: "*Uncertainty experiments*" (A4.1-A4.8) and "*Idealized experiments*" (A5.1-A5.10). The first subgroup of simulations diagnoses modeling uncertainties using alternative driver datasets when available (e.g., multiple anthropogenic emissions inventories may be available for the time period 2010–2020) or alternative parametrizations (e.g., multiple air-sea exchange parametrizations could be compatible with available measurements). In the "*Idealized experiments*" subgroup, sensitivity of Hg levels and trends to a driver or parametrization is assessed by replacing it with a hypothetical option. For example, to test the importance of accurate speciation in emissions inventories, an extreme scenario could be simulated where all anthropogenic Hg is emitted as Hg(0).

v. The '*Future scenarios*' group (A6.1-A6.3, O5.1-O5.3) estimates future changes of Hg levels over the considered period (2020-2050) using available scenarios of anthropogenic emissions and projected environmental conditions and ascribes the estimated changes to various anthropogenic and environmental drivers.

Changes in primary anthropogenic emissions and releases and environmental conditions (physical and biochemical) form the two groups of forcings (or drivers) responsible for changes or trends in environmental Hg levels through their influences on Hg processes (i.e., Hg transport, biochemistry, and flux exchanges). In the proposed experimental design, the contributions of these two groups of drivers are first separated via the "nature simulation" (using coupled A3a.1 and O3.1), where primary anthropogenic emissions are kept fixed at 2010 level and the Hg trend is solely driven by natural factors. The anthropogenic contribution is then further divided between contributions of ASGM (A3a.2), Minamata Annex D (A3a.3), and other





sources. It is important to note that the contribution of changes in anthropogenic emissions is itself modified by the concurrently occurring environmental changes, which is reflected in these estimates. Isolating individual contributions of changes in various environmental conditions (such as meteorological, land, and ocean parameters) is difficult due to their inter-dependencies. Nonetheless, the individual effects of changes in a suite of environmental drivers on Hg levels can be examined by selectively allowing temporal variations of these drivers or their impacted Hg processes (e.g., Hg transport and

secondary emissions and releases), while keeping all other drivers fixed in perturbation experiments (A3b.1-A3b.7). In addition, proposed idealized experiments (A5.1-A5.10), despite their simplicity, can reveal fundamental response characteristics of the model drivers or parameterizations and inform their potential impacts on future Hg cycling.

Tables F8-9 represent an extensive list of potential simulations, yet one must account for time and resource restrictions of participating models. Therefore, the model experiments are ranked into three priority levels (Tables F8-9): (1) "Core

simulations", expected to be performed by all global models; (2) "Optional simulations", expected to be performed by as many models as possible; and (3) "Future simulations", expected to be conducted as a third set of model simulations following the modeling and analysis of historical Hg levels. Both global and regional models can take part in the simulations program. The time variable boundary conditions for regional modeling experiments will be first extracted and stored from multi-model simulations of global models. The model output variables (Tables F11-12) will be stored in an open data format

(netCDF) with appropriate metadata. This includes the definition of new official CF convention tags for marine Hg variables (Davis et al., 2020) (see Appendix D).

### 5.3. Multi-media mass balance simulations

Given the dominant role of secondary emissions in contemporary Hg levels, quantifying its changes is important to the attribution of observed temporal trends of Hg in recent decades. Currently, it is difficult to fully account for the temporal

trend in secondary emissions and releases based on observations or 3D Hg models alone. Multi-media mass balance box models have been used to trace the movement of all-time anthropogenic mercury emissions and releases and its influence on Hg levels in environmental matrices (Amos et al., 2015).

Multi-media mass balance simulations (see Sect. 2.4) will serve three primary MCHgMAP objectives: (1) attribution of all time anthropogenic Hg sources to secondary Hg emissions and releases and Hg levels and trends in environmental

compartments; (2) determination of sensitivity of Hg levels and trends to uncertainty in emission histories and global cycling; (3) response of global Hg cycling to future anthropogenic emission scenarios. Mass balance models serve an additional purpose for the proposed MCHgMAP activities, which is to (4) provide the long-term changes in secondary emissions and releases (i.e., legacy emissions and releases) and/or environmental levels driven by historic changes in anthropogenic Hg emissions and releases as inputs to 3D atmospheric and ocean models. Mass balance model simulations

represent a global geographic domain and will be generally performed for a period of greater than 500 model years (e.g., 1500 – 2020). Simulations will produce annual arrays of multi-media reservoir Hg levels and fluxes that can be stored as tables for analysis.





To account for uncertainties of Hg in emissions, reservoirs, and inter-compartmental fluxes, a mass balance model ensemble
will be produced (see Fig. 9). Sensitivity simulations can be evaluated against available independent constraints (see below)
in a similar manner as in Amos et al., (2015). The list of simulations to be performed with the model ensemble is given in
Table F10. The legacy component of environmental Hg will be developed and evaluated following the methodology of
Angot et al., (2018).

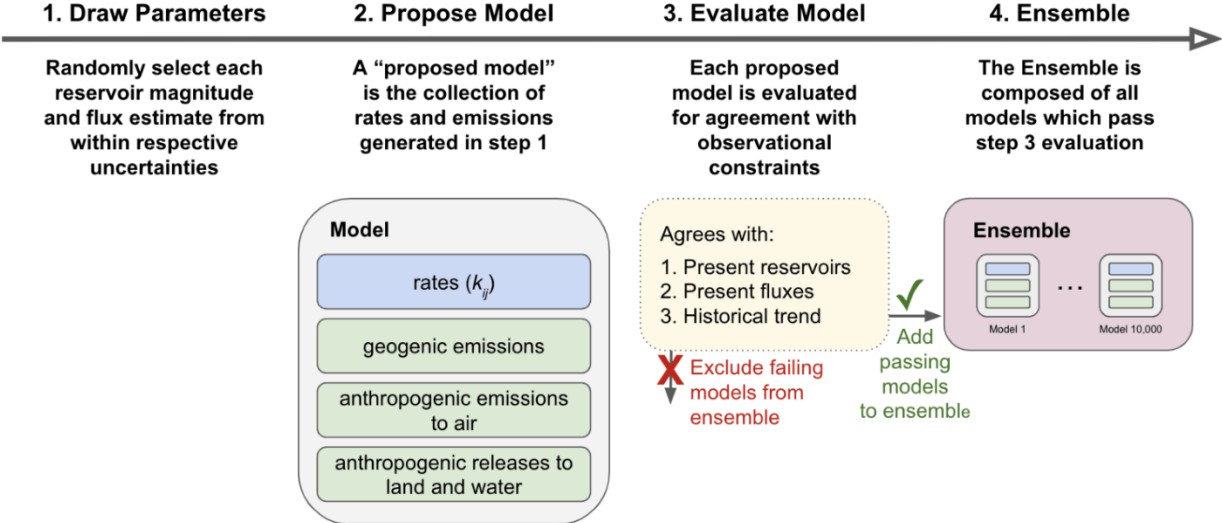

**Figure 9. Schematic of process for constructing mass balance model ensemble.**

The plan for the construction of revised mass balance model ensemble (steps 1 – 4 below) and its simulations (steps 5 – 7
below; Table F10) is as follows:

    i.    ***Update rate coefficients:*** Update central estimates for rate coefficients based on state-of-the-science review of
contemporary Hg reservoir magnitude and exchange flux estimates (from models and observations). Rate
coefficients used in baseline steady-state simulations from 3D air and ocean models will be considered.

ii.    ***Develop base model\*:*** Use a perturbational approach to define the set of rates and historical emissions and releases
that are consistent with evaluation metrics (see below).

    iii.    ***Evaluate Model:*** The combination of feasible rate coefficients and emissions and releases trajectories will be
constrained by model ability to reproduce:

        a.    Contemporary reservoir magnitudes (e.g., Sect. 4.1, 4.3)

b.    Contemporary fluxes (e.g., Sect. 4.2, 4.4)

        c.    Historical trends in environmental archives (Sect. 4.5)

Model evaluation metrics should reflect global magnitudes/trends, which may require rescaling/ rebalancing due to
differences in regional sampling density of observations.





 

iv. ***Model Ensemble:*** All box models (i.e., combinations of rate coefficients and emissions and releases trajectories) that successfully reproduce global magnitudes/trends (step 3) will compose the model ensemble (i.e., Baseline simulation M1; Table F10) used in the following analysis simulations.

v. ***Characterize secondary emissions trend\*:*** Quantify contributions of legacy and recent primary emissions to recent temporal trend (i.e., 2010 – 2020) in global Hg levels and secondary emissions from the land and oceans (simulations M2 to M11 in Table F10). Uncertainties in contributions will be bounded using the model ensemble.

vi. ***Perform attribution of temporal trend to primary sources\*:*** Quantify the fraction of all-time trend in global Hg levels and secondary emissions and releases attributable to specified primary source categories (simulations M12 to M17 in Table F10).

vii. ***Project future effects of policy interventions\*:*** Quantify temporal trend in Hg levels and secondary emissions and releases under future anthropogenic emission scenarios (simulation M18 in Table F10).

\*Steps involving model simulation

Further development of spatially resolved mass balance models (e.g., hemispheric- or ocean basin-resolved compartments) including Hg speciation would improve the analysis of all-time Hg trends. Importantly, accounting for changes in inter-media exchange rates due to changes in environmental conditions such as oxidant concentrations or precipitation regimes, or changes in terrestrial Hg export to coastal ocean due to accelerated permafrost thaw and erosion is needed to perform sensitivity simulations to quantify response of secondary reemissions and reservoir Hg burdens under future climate change scenarios.

## 6. Model evaluation

Evaluation of the MME simulations against observational data is of key interest and relevance both scientifically and to the MC and LRTAP effectiveness evaluations. The primary goal of this model evaluation is to assess the robustness of baseline historical simulations to facilitate observationally constrained attribution of observed changes in global Hg levels to anthropogenic and natural forcings and to predict future Hg levels. The discrepancies between modelling and measured data indicate areas of increased uncertainty and provide directions for further improvement of the models, model inputs (e.g., emissions data) or measurement data. Additionally, in the regions where measurements are absent or for the parameters and processes which are not directly measured, the inter-comparison of results from different models can provide additional information on the level of uncertainties. To facilitate the inter-model comparison and the evaluation of the modelling results against observations, the spatially resolved variables from 3D models will be interpolated from the model-specific grids to a unified latitude-longitude grid and to the points of measurement sites' location, respectively. The evaluation of mass balance model historical simulations is discussed in the preceding section (Sect. 5.3). The following sections present the MME approach for the evaluation of recent Hg levels and trends simulated by the 3D atmosphere and ocean models.



## 6.1. Atmospheric models

Comparison of model simulations with measurement data is a standard evaluation procedure applied for various purposes – model development, integrated analysis and evaluation of the modelling results, and process studies. A variety of single- and multi-model studies have used measurements to support research of Hg dispersion on a global scale (Horowitz et al., 2017; Shah et al., 2021; Zhang and Zhang, 2022) or in the Polar regions (Angot et al., 2016; Dastoor et al., 2022a, b), to evaluate emission inventories and their trends (De Simone et al., 2016, 2017a; Zhang et al., 2016b; Giang et al., 2018), and to study Hg chemical mechanisms (Shah et al., 2016; Song et al., 2018; Saiz-Lopez et al., 2018, 2020) and other processes in the atmosphere (Gencarelli et al., 2017; Travnikov et al., 2017; Bieser et al., 2017).

A variety of available measurement data will be used for evaluation of the multi-model atmospheric simulations, including ground-based measurements from global and regional networks, aircraft measurements, ship and terrestrial flux measurements. In addition, the simulated concentrations of Hg species, wet and dry deposition, air-surface exchange fluxes and relative contribution of various emission sources and sectors will be compared between models. Table F11 summarizes the output variables to be used in the model evaluation. Different characteristics of atmospheric Hg will be evaluated in both the model-to-measurement and model-to-model comparisons, namely:

*Spatial distributions* (comparison with ground-based measurements): The comparison will include temporally averaged Hg levels, spatial correlation, and gradients. It is important to evaluate different measured parameters (e.g. Hg(0) / Hg(II) concentrations, Hg wet deposition). This comparison can evaluate the models' atmospheric chemistry and global mass balance of sources/sinks.

*Temporal trends* (comparison with ground-based measurements): The comparison will include temporal changes at individual sites or average values over different regions. It is important to combine different measured parameters (Hg(0) concentration, Hg wet deposition) since they characterize the relative contributions of global and regional sources. This comparison can provide evaluation of the modelled changes driven by emissions (anthropogenic, legacy) and effects of other drivers (e.g., meteorology, land cover, LAI, sea ice). Besides, the analysis of seasonal variations provides insights on major sources and sinks of atmospheric Hg.

*Vertical profiles* (comparison with aircraft measurements): The comparison will be focused on vertical distribution of Hg species (Hg(0), Hg(II)) at certain locations. The analysis provides additional insights into free tropospheric processes (Hg chemistry) and the effect of emission sources (e.g., power plants, volcanoes etc.).

*Air-surface exchange* (comparison with ship and terrestrial flux measurements): It will consist of comparison of Hg concentrations (Hg(0), Hg(II)) and estimated air-sea exchange fluxes at locations of ship measurements and comparison of Hg dry/throughfall deposition and exchange fluxes at terrestrial sites.


*Spectral analysis* (comparison with ground-based measurements, model intercomparison): In addition to the traditional model-to-measurement comparison using scatter plots and statistical metrics (e.g., bias, correlation, $R^2$), spectral analysis will be used to examine and compare model and observed data (Sirois et al., 1995; Bowdalo et al., 2016). This will further help in determining which data characteristics (e.g., seasonal cycles or diurnal cycles) are
best reproduced by the models.

The model evaluation is limited by the sparse geographical and temporal coverage of surface observations (particularly, for wet deposition and speciated Hg). Moreover, vertical profiles are only available from limited aircraft observations, mostly of total gaseous Hg (THg) or Hg(0). The model-observation comparison can be used to analyze spatial representativeness of atmospheric Hg observations and improve their networks by identifying the regions of important gaps.

**6.2. Ocean models**

There are few marine Hg model studies evaluating individual models against observations (Zhang et al., 2019c; Kawai et al., 2020; Rosati et al., 2022; Bieser et al., 2023), and so far, no ensemble studies have been performed. Here, we propose a framework for the first marine Hg model ensemble study. The study will focus on evaluating individual models and ensemble simulations against observations. For this, we will create a harmonized database including surface measurements
and vertical profiles of marine Hg species (the Hg Ocean Observational Database; see Sect. 4.3 and 4.4). Table F12 summarizes the output variables considered for the model performance evaluation. Air-sea exchange fluxes will be investigated together with those calculated by the atmospheric model (Sect. 6.1). Through repeated offline coupling of atmosphere and ocean models, we will investigate whether the flux estimates of the compartmental models converge. For areas and variables where observations are sparse or not available (see for example Fig. 7), we will conduct an
intercomparison evaluation of the individual models to identify the uncertainty range of model results. For this, hydrodynamic, biogeochemical, and mercury variables will be investigated independently. Identifying disagreements between the models will help to identify major sources of uncertainty and to discover processes that are currently not resolved or unknown.

The ensemble evaluation will be based on established performance criteria for model and ensemble analysis Solazzo and
1865 Galmarini (2016) as proposed for use in ocean model evaluation by Bieser et al., (2023). The central metric for model evaluation is the mean squared error (MSE) (Eq.1).

$$\frac{1}{N}\sum_{i=1}^{N}(mod_i - obs_i)^2 \qquad \text{(Eq. 1)}$$

Based on the method proposed by Solazzo and Galmarini (2016), the MSE can be broken down into its components: bias (accuracy), variance (precision) and covariance (unexplained error) allowing for a more detailed model assessment (Eq.2 -
1870 Eq.5). The bias represents the systematic error. It is strongly influenced by model drivers and is the quantity models are typically optimized for. The variance and covariance parts of the error are the most relevant for model developers as these


are direct results of the model itself. Especially the amplitude error (variance) can be an indicator for missing or non-resolved processes in the model. While the remaining unexplained error (mMSE) can be interpreted as unsystematic fluctuations. This is the least troublesome error as it represents unresolved fluctuations in the natural system not resolved by the model.

$$MSE = bias^2 + var(mod) + var(obs) - 2\,cov(mod, obs) \qquad \text{(Eq. 2)}$$

$$MSE = (mod - obs)^2 + (\sigma_{mod} - \sigma_{obs})^2 + 2(1-r)\sigma_{mod}\sigma_{obs} \qquad \text{(Eq. 3)}$$

$$MSE = (mod - obs)^2 + (\sigma_{mod} - r\sigma_{obs})^2 + mMSE \qquad \text{(Eq. 4)}$$

$$mMSE = \sigma_{obs}^2 (1 - r^2) \qquad \text{(Eq. 5)}$$

Finally, we will use the model quality objective (MQO) (Thunis et al., 2013; Carnevale et al., 2014). The MQO is a statistical measure for the possibility to improve a model given the available observations. This is achieved by including the uncertainty of the observations with which the model is compared. Especially for areas with spare observations and large measurement uncertainties, even a model with a large error might be in a range where it cannot be reasonably improved upon unless more abundant or more precise observations become available. This is relevant as for some Hg species the measurement uncertainty can be in the same order of magnitude as the overall model uncertainty. As no ocean model ensemble studies have been conducted so far, the MCHgMAP exercise will be used to test and further develop these methods into a robust framework that can be used during future EE efforts.

For the MCHgMAP ocean Hg simulations, the initial step will be to identify to what degree Hg species concentrations across regions and times agree with observations. Starting from this base case scenario, the models will be used to investigate the impact of individual drivers on marine Hg cycling and air-sea exchange. We will compare the baseline scenario (O1) to surface observations and vertical profiles for total Hg and individual Hg species to evaluate the ability of the models to reproduce observed spatiotemporal variations. Air-surface exchange will be investigated in unison with the atmospheric models, including the impact of model simulations with alternative air-sea parametrizations (O4.5, O4.6).

For the ocean model evaluation, there are currently some important limitations. Marine observations for evaluation are sparse (Sect. 4.3 and 4.4). This is especially true for Hg and MeHg in the bottom of the food web (phyto- and zooplankton), but also with regards to water, sediment and fish. While this is a limitation for this study, the result of the model evaluation will help to identify regions of interest that should be targeted for additional observational campaigns. Further, for the water, plankton and biota observations that exist, there is little to no information on the seasonality of Hg concentrations.

Further development of a framework for the evaluation and intercomparison of ocean model simulations is recommended to facilitate standard methodologies across EE cycles of the MEAs. Also, results from coupled multi-compartment models





should be integrated into the intercomparison evaluation once they become available. Regions of particular interest, identified during the MCHgMAP work, where no or insufficient observations are currently available, should be prioritized
when planning research cruises. Further, as marine food-web models develop further, more focus should be added to evaluate and compare Hg and MeHg levels in marine biota. This could include the integration of phyto- and zooplankton into the Ocean Observational Database (see Sect. 4.3) and the use of Hg concentrations in fish from existing databases for observation-model evaluation (Ammar et al., 2024; Bieser et al., 2023).

## 7. Modeling analysis: effectiveness evaluation

The following sections discuss the analysis of proposed MCHgMAP MME simulations to address the key questions relevant to the MC and LRTAP effectiveness evaluations (Appendix A, questions 7 to 17). Sect. 7.1 and 7.2 describe the analysis approach for the detection and attribution of global and regional spatiotemporal patterns and trends of Hg in air and ocean. Sect. 7.3 discusses the development of global mass balance of environmental Hg levels and movements based on coordinated multi-model simulations.

### 7.1. Atmosphere

### 7.1.1. Current levels

Levels of Hg concentration and deposition substantially differ from region to region over the globe, being affected by a number of anthropogenic and environmental drivers. These include remoteness from significant anthropogenic and natural sources, prevailing atmospheric flows and circulation, spatial variability of the oxidation potential of the atmosphere,
precipitation patterns, and properties of the earth's surface including land cover type and characteristics of vegetation (Obrist et al., 2018). The highest Hg concentration and deposition levels occur in the proximity to emission sources in industrial and highly populated regions as well as in regions with Hg intentional use (ASGM) (Pacyna et al., 2016; UNEP, 2019). Air concentrations are generally higher in the Northern hemisphere because of the predominant location of anthropogenic emissions north of the equator (Travnikov et al., 2017; Horowitz et al., 2017; Shah et al., 2021). Hg deposition is elevated
over forest-covered land, in locations with intensive chemical transformations of Hg(0) to oxidized Hg forms, and areas with large precipitation amounts (Dastoor et al., 2022a; Zhou et al., 2021; Zhang and Zhang, 2022). Given the limited geographical coverage of existing measurement networks, which are globally inhomogeneous and mostly restricted to the continents, the model simulations can provide a more complete picture of Hg levels distribution over the globe. The MME simulations can be used to develop spatial patterns of Hg concentration and deposition and their changes over the period
2010-2020 and explain these in terms of contributions of different emission sources (anthropogenic, geogenic, secondary) and influences of environmental drivers on Hg processes.

Global distributions of atmospheric Hg concentrations and deposition have been studied previously using both monitoring (e.g., Sprovieri et al., 2016, 2017) and modelling (Chen et al., 2014, 2015; De Simone et al., 2016; Horowitz et al., 2017;





Shah et al., 2021; Zhou et al., 2021; Zhang and Zhang, 2022). Detailed model analysis of Hg levels has been also performed
on regional scales (Gencarelli et al., 2014; Wang et al., 2014a; Ye et al., 2018a; Xu et al., 2022). Along with single-model
studies, several multi-model assessments and intercomparisons have been carried out to provide more reliable information on
Hg levels, spatial distribution and source apportionment (Angot et al., 2016; Travnikov et al., 2017; Dastoor et al., 2022a, b).
The Global Mercury Assessment 2018 (UNEP, 2019) provides a comprehensive analysis of Hg atmospheric pathways,
transport and fate on a global scale including multi-model estimates of Hg concentrations and deposition to various
terrestrial and aquatic regions in 2015, evaluation of the model ensemble results against available observations, attribution of
Hg deposition to different source regions and emission sectors. Uncertainties of the model derived source-receptor
relationships were estimated by De Simone et al., (2017a) using statistical processing of an ensemble of multiple
simulations. In addition, a simplified statistical emulator has been developed based on the output of a global transport model
to simulate changes in anthropogenic Hg fluxes in a source-receptor manner for evaluation of various emission scenarios (De
Simone et al., 2017a, 2021, 2022).

#### 7.1.1.1. Spatial pattern detection

The spatial patterns of Hg levels on a global scale and across remote and affected regions will be identified using the
Baseline simulation of the multi-model ensemble as well as perturbation simulations (Table F8). Simulated maps of Hg
concentration and deposition from the Baseline simulation (A1) will be analyzed and explained through comparison with
spatial patterns of emissions, meteorological parameters (e.g. precipitation), chemical reactants (Br, $O_3$, OH, etc.) and land
cover types. This analysis will be quantitatively aided by modeled spatial distributions of individual contributions from wet
and dry deposition processes and sources contributions from various emissions to the spatial patterns of air concentrations
and/or total deposition (simulations A2.1-A2.6). The analysis will be performed for different years of the period 2010-2020
to reveal their temporal changes and causes. The simulated distributions of the near-surface air concentrations of Hg(0) and
Hg wet deposition fluxes will be evaluated against available observations on the annual basis and for different seasons. The
global maps of Hg levels will also be compared between individual models, particularly, for Hg cycling characteristics,
which are sparsely measured, e.g., Hg(II) dry deposition and the air-surface Hg(0) exchange flux. The model evaluation and
inter-model discrepancies will provide quantitative information on the uncertainty of simulated spatial patterns. Additional
insight into the uncertainties of modelled spatial patterns will be gained from the sensitivity simulations with alternative or
idealized sets of emissions, environmental drivers and processes parameterizations (Table F8, simulations A4.1-A5.10).

#### 7.1.1.2. Attribution of levels

Mercury emitted to the atmosphere is subject to long-range transport and exchange with other environmental compartments.
Therefore, Hg levels in the atmosphere and other media consist of contributions of various emission sources distributed
worldwide. Source attribution of Hg concentrations and deposition will be performed with a set of the multi-model
perturbation simulations "Source contributions" (Table F8, simulations A2.1-A2.6). These simulations will be carried out for
the period 2010-2020 under the same conditions as the Baseline (A1), but with emissions of a certain source type zeroed.
Subsequent comparison of the results of the "source contribution" perturbation simulations with the Baseline will provide




individual contributions of particular source types to Hg levels in each year of the period. Furthermore, the influences of inter-annual variability in meteorology and other environmental conditions on source attribution of Hg levels can be examined by replacing the Baseline simulations for 2011-2020 in the attribution analysis with trend perturbation simulations where various emissions are kept fixed at 2010 level but meteorology and other environmental variables are allowed to vary (A3a.1-A3a.4).

The source types considered include all anthropogenic emissions (A2.1), sources regulated by MC (Annex D) (A2.6), emissions from ASGM (A2.5), geogenic emissions (A2.2), wildfire emissions (A2.4), and secondary emissions from land (A2.3) and the ocean. Contribution of the ocean secondary emissions will be estimated by subtraction of all other contributions from the baseline simulation. Direct estimation of the oceanic reemission source contribution by zeroing its emissions is not feasible because of the bi-directional character of the air-water exchange of Hg(0), which depends on Hg concentrations in both air and seawater and cannot be decoupled. For this reason, the method applied does not allow tagging contributions of different reemission sources through the air-water interface but calculates Hg(0) evasion from aquatic surfaces as the leftover difference after attributing all other emission sources.

Intercomparison of the source attribution results obtained by different models will provide information on the uncertainty level of performed estimates. In contrast to the previous multi-model assessment (UNEP, 2019), the proposed simulations will be conducted for the multi-year period that will allow analysis of the temporal dynamics of relative contributions of emission sources to Hg levels in different geographical regions. Besides, following a similar approach to the existing HERMES emulator (De Simone et al., 2020), the output of these simulations could be used to create publicly available statistical emulators of Hg deposition for the multi-model ensemble. Thus, the model data could be flexibly extended to further policy scenarios relevant to the Minamata Convention.

### 7.1.1.3. Limitations and recommendations

Anthropogenic emission inventories play a dominant role in the model estimated spatial patterns and source attribution of Hg levels. Therefore, the uncertainties in anthropogenic emissions data (i.e., geospatial distribution, temporal variation and chemical speciation) is a major factor in limiting the reliability of the modelling results. Additional uncertainties are also introduced by model parameterizations of atmospheric transformations and exchange processes including atmospheric chemistry and air-surface exchange fluxes. Limited spatial coverage of available monitoring data complicates evaluation of the simulated global patterns of Hg levels. Additionally, limitations of time and computational resources might not allow tracking of emission sources through the air-ocean interface in source attribution experiments using a coupled atmospheric and ocean modeling approach (see Sect. 5.1). Similarly, the MCHgMAP experimental design does not fully track the contributions of various primary emissions across air-soil interface. Therefore, the source attribution analysis lacks the influence of contemporary emissions on the modification of recycling of Hg between the atmosphere, soil and ocean. This error is likely insignificant for the air-soil exchange due to relatively long residence time of Hg deposited in soils (Amos et al., 2013) but might be important across air-ocean interface.





Continuous updates and refinement of anthropogenic Hg emissions inventories including geospatial, seasonal, and annual variations and chemical speciation are recommended to improve the future EE of the MEAs. Model parameterizations of atmospheric Hg chemistry and air-surface exchange fluxes also require further refinement to reduce uncertainties of spatial patterns and source attribution estimates. More precise estimation of contributions of contemporary primary emission sources relative to legacy sources requires iterative coupling of atmospheric, ocean and terrestrial models or fully coupled multi-media Earth system models. Evaluation of the source attribution estimates can also benefit from development and use of the Hg isotopic fractionation modeling tools (Song et al., 2022).

### 7.1.2. Temporal trends

Numerous drivers can affect the trends of atmospheric Hg concentrations and deposition measured at surface stations globally. For example, previous studies have highlighted the role of drivers including changing regional and global emissions trends (Olson et al., 2020; Zhang et al., 2016b), decreasing ocean legacy re-emissions (Soerensen et al., 2012), meteorological variability (Dastoor et al., 2022b), and land use change (Jiskra et al., 2018). By designing simulations that investigate the impact of these individual drivers on atmospheric Hg trends in multiple Hg models, we aim to improve the detection and attribution of Hg trends. The proposed spatially-resolved simulations for 2010–2020 can be used to assess the causes of trends at different monitoring stations and quantify the impact of MC policies on observed Hg trends. In addition, the impacts of non-Minamata Convention factors on observed Hg trends (e.g., meteorology, legacy re-emissions, biomass burning, land use change) will be identified.

There have been an array of observational studies investigating past atmospheric trends of Hg. At measurement stations in North America and Europe, atmospheric Hg concentrations have generally declined from the 1990s until the present (Custódio et al., 2022; Roberts et al., 2021; Olson et al., 2020; Zhang et al., 2016b; Cole et al., 2014). The trend magnitude varies depending on the station; for example, total gaseous mercury (TGM) declined at Mace Head (1996–2020) by $1.6 \pm 3.9\%$ y$^{-1}$ (Custódio et al., 2022), for North Atlantic ship cruises (1990–2009) by $2.5 \pm 0.5\%$ y$^{-1}$ (Soerensen et al., 2012), and at Alert (1995–2011) by $0.9 \pm 0.3\%$ y$^{-1}$ (Cole et al., 2013). Trends in North America and Europe have been nonlinear over time, with several studies reporting stagnated trends for a period around 2008 (Custódio et al., 2022; Weiss-Penzias et al., 2016; Lyman et al., 2020a). In other regions, atmospheric Hg measurements began later (generally after 2005) and there are fewer stations, leading to more uncertain regional trends. Available atmospheric Hg trends in East Asia show declines over the last decade in the Yangtze River Delta (Tang et al., 2020), Xiamen (Shi et al., 2022), Okinawa (Marumoto et al., 2019), and Taiwan (Nguyen et al., 2019), and the magnitude of the trends is similar to those observed in North America and Europe. On the other hand, constant or increasing trends in atmospheric Hg have been observed recently the Southern Hemisphere sites Cape Point (2007–2017) and Amsterdam Island (2012–2017) (Slemr et al., 2020) and in Siberia (2011–2020) (Mashyanov et al., 2022).

Atmospheric Hg models have been useful tools to assess the drivers of Hg trends and evaluate the accuracy of emission assumptions and biogeochemical cycle parameterizations. In the most recent evaluation of model and measurement trends on





the global scale, Zhang et al., (2016b) conducted GEOS-Chem simulations between 1990 and 2010 and compared with
available observations globally. The model-measurement comparison enabled the authors to refine assumptions about
changes in both the magnitude and speciation of anthropogenic Hg emissions. Since that time, more Hg observations have
been collected, continuously providing us with a better picture of global Hg trends. At the same time, trend detection and
attribution methods for atmospheric compounds have become more sophisticated (Chang et al., 2021).

### 7.1.2.1. Trend detection

Robust trend analysis methods will be applied to both observed and modeled Hg concentrations and deposition. Multiple
trend analysis methods will be considered to maximize the information obtained from observed and modeled time series
(Table 3). The most recent modelling study to consider observed trends on the global scale used linear regression methods
(Zhang et al., 2016b), while the recent AMAP report applied the seasonal Mann-Kendall Test and the Theil-Sen trend
estimator to analyze Arctic trends (MacSween et al., 2022). There are more complex approaches available (Table 3) to: 1)
evaluate nonlinear changes in trends (Kernel regression and Generalized Additive Models, GAMs); 2) quantify trends across
the statistical distribution of Hg measurements, for example the trends in extreme values (quantile regression); and 3) include
meteorological covariates in regressions (e.g., temperature and humidity) to remove the influence of meteorological
variability from the trend analysis (e.g., Shi et al., 2022), which can be achieved with multiple approaches (multiple linear
regression, GAMs, quantile regression) (Chang et al., 2023).

We would analyze trends in the Baseline simulation A1 (Table F8) from available atmospheric Hg models, comparing these
to observations of atmospheric Hg concentrations and deposition for 2010–2020. The Baseline simulation would be
conducted with drivers (i.e., anthropogenic emissions, legacy re-emissions, oxidant concentrations, meteorology) that vary
temporally over 2010–2020, representing the state-of-the art knowledge for processes driving Hg cycling. Simulated trends
at the grid cells of measurement stations will be extracted to compare with available measurements. We would evaluate the
accuracy of the Baseline models for capturing measured trends for 2010–2020, using an array of metrics ($R^2$, mean absolute
and relative biases, fraction of sites at which a model predicts correct sign of trend). The comparison of trends can be
summarized for available observation media (Hg(0), THg, and Hg wet deposition) and specific regions where measurements
are available.

Trends would also be compared between individual Hg models in the Baseline simulation. In this analysis, we would not be
limited to analyzing only areas where measurement stations are available. Modelled trends in atmospheric Hg concentrations
will be calculated globally, over each hemisphere, and for all HTAP regions. Trends in Hg fluxes (e.g., dry deposition over
land, wet deposition, and ocean-air Hg exchange) will be compared between models. Since all models will conduct
simulations with harmonized emission datasets, differences between modelled trends will be attributed to other
environmental drivers which will not be harmonized between models (i.e., meteorology, land surface parameters) and the
individual parametrizations of Hg cycling. In addition, each individual model can conduct sensitivity simulations (A4.1–
A5.10 in Table F8) using alternative anthropogenic and secondary emissions and parametrizations to diagnose the impact of
these drivers on trends. This combination of inter-model and inter-simulation analysis will demonstrate the sensitivity of Hg





trends to structural and parametric uncertainties in available Hg models. By comparing model temporal trends across global regions, uncertainty estimates in modeling results will be calculated and scientific gaps in modeling will be identified.


**Table 3. Trend analysis methods that could be applied to observed and simulated Hg time series.**

| Method | Description | Advantages | Example study for Hg |
|---|---|---|---|
| (Multi-) Linear regression | Assumes a linear relationship between dependent variable (e.g., Hg(0)) and independent variable(s), e.g., time and/or other covariates. The ordinary least squares loss function is minimized to calculate mean estimates for regression coefficients. | -Simple to interpret<br>-Can consider covariates | Zhang et al., (2016b) |
| (Seasonal) Mann-Kendall test and Theil-Sen trend estimator | Robust non-parametric method to calculate median trend, by considering the median difference between each pair of observations. | -Does not require normally distributed data<br>-Simple to interpret | MacSween et al., (2022) |
| Generalized additive models (GAM) | The observations are modeled as a sum of smooth functions of dependent variables, with regularization of functions used to avoid overfitting. | -Flexible for nonlinear trends<br>-Can consider covariates | Shi et al., (2022); Wu et al., (2020) |
| Quantile regression | The trend in different parts of the observed distribution (i.e., quantiles) can be calculated | -Analyzes trends across distribution (e.g., extreme or baseline values)<br>-Can consider covariates | Feinberg et al., (2024) |
| Kernel regression | The observations are modeled non-parametrically using smoothing kernel functions | -Can calculate nonlinear trends | Custódio et al., (2022) |

### 7.1.2.2. Attribution of trends

The proposed list of multi-model ensemble simulations would allow for attribution of Hg trends to individual emissions
sources and environmental drivers (Simulations A3a.1–A3b.7, see Table F8). For example, we will conduct a 2010–2020 simulation with fixed anthropogenic emissions at 2010 values (Simulation A3a.1). By comparing this simulation to the Baseline simulation (A1, where anthropogenic emissions vary for 2010–2020), the influence of anthropogenic emissions changes on Hg trends can be quantified. This attribution analysis can be conducted at different spatial scales (globally, hemispheric, regionally, and at locations of measurement stations) to better quantify the role of anthropogenic emissions in
observed Hg trends.





Similar to the exercise proposed above for Simulation A3a.1, other drivers can be investigated by conducting simulations with 2010–fixed values. Simulations could be conducted to quantify the impacts of ASGM emissions (A3a.2), Annex D anthropogenic emissions sources (A3a.3), and geogenic emissions (A3a.4). These sensitivity simulations would illustrate the roles of these diverse drivers for Hg trends between 2010 and 2020. Among other uses, the model products could be used to map the principal driver influencing Hg trends for each individual grid cell globally.

Effects related to the environmental conditions are difficult to disentangle, as many of the individual drivers (meteorology, land surface conditions, secondary emissions) are intertwined. Additional optional simulations will probe the effects of changes in various environmental drivers on trends for diagnostic purposes. For these simulations, we will vary one driver between 2010–2020 and keep all other drivers fixed at 2010 levels. Proposed simulations will consider the effect in 2010–2020 trends of varying transport (A3b.1), wildfire emissions (A3b.2), terrestrial secondary emissions (A3b.3), ocean Hg concentrations (A3b.4), land surface conditions (A3b.5), chemical reactants (A3b.6), and anthropogenic emissions speciation (A3b.7). The contribution of anthropogenic emissions changes from different HTAP regions will be explored with simulations A3c.1–A3c.11.

The model intercomparison will highlight whether the influence of drivers is comparable between the different models. As well, by analyzing the role of different trend drivers, we will be able to better understand differences in the modeled Baseline trends and discrepancies between the models and observations. The previous global multi-model Hg study focused on an individual simulation year (Travnikov et al., 2017). Therefore, the transient 2010–2020 intercomparison simulations present a novel opportunity to evaluate the variety of model responses to changing Hg emissions and environmental drivers. In this way, MCHgMAP initiative will refine our understanding of the drivers of recent Hg trends and help with designing further simulations in subsequent EE efforts to isolate the impacts of MC and LRTAP activities on Hg trends.

### 7.1.2.3. Limitations and recommendations

Currently, the key limitations adding uncertainties to modeling trend analysis are the scarce information available regarding the trend in anthropogenic emissions from certain sources (e.g., for ASGM), geographically limited observations (mainly from North America and Europe) that cover the full 2010–2020 time period, and a lack of field or lab studies investigating the response of Hg fluxes (deposition and secondary emissions) to changes in environmental drivers. For example, the impact of land cover and vegetation properties on Hg uptake depends on model parametrizations of Hg(0) dry deposition calibrated to available field measurements (Zhou et al., 2021; Feinberg et al., 2022). However, there are few measurements of Hg vegetation uptake in tropical forests with high biological productivity, including no measurements in tropical regions of Asia or Africa. Although the proposed 10-year simulations would be longer than any previous Hg multi-modeling activity, the relatively short time period presents challenges to trend analysis owing to the role meteorological interannual variability in driving trends over shorter time periods as shown in studies using mechanistic modeling (Dastoor et al., 2022b) and statistical approaches (Shi et al., 2022; Wu et al., 2020).

A long-term Hg measurement record that is both regionally- and process-representative and improved reporting of Hg emissions from various sectors can improve trend analysis in future EE efforts. Inverse models have been used in different





atmospheric chemistry applications to calculate temporally and spatially varying emission fields using Bayesian inference and available observations (Bousquet et al., 2011; Park et al., 2021). Inverse modeling could aid in diagnosing changes in anthropogenic Hg emissions driven by the Hg MEA policies. However, until now, inverse modeling applications for Hg cycling have been limited in scope spatially and temporally (Song et al., 2015; Denzler et al., 2017). In future, a diverse set of inverse Hg modeling methods could be applied to available atmospheric observations, in order to produce estimates of Hg

emissions trends and their uncertainties.

### 7.2. Ocean

#### 7.2.1. Current levels

The principal source of oceanic Hg is atmospheric deposition, air-sea exchange of gaseous elemental Hg(0), river input, and coastal erosion (Dastoor et al., 2022a; Amos et al., 2014; Mason and Fitzgerald, 1996). Much of oceanic Hg input is of

anthropogenic origin (Driscoll et al., 2013), since the anthropogenic influence on global Hg cycling is significant. However, the response (time and level) of oceanic Hg levels to ongoing changes in anthropogenic emissions is currently uncertain. In addition to ongoing anthropogenic emissions, the oceanic response time is highly dependent on contributions from natural (geogenic) and historical (anthropogenic legacy emissions) sources (Zhang et al., 2014b). Thus, the design of MME simulation experiments needs to include ocean models that can track the contributions of different anthropogenic and

geogenic sources of Hg emissions to water column Hg levels at observational sites and across global regions. The attribution of Hg emissions from different sources can be traced in several classes: geogenic, anthropogenic and secondary (separated by land, ocean, and wildfires); different anthropogenic Hg emission sectors; and anthropogenic Hg emission sources influenced by the MEAs.

Several previous modeling studies have focused on the impact of anthropogenic emissions on oceanic Hg levels (Chen et al.,

2018; Corbitt et al., 2011). However, no 3D biogeochemical ocean model has been used to source-track Hg emissions over historic periods. Corbitt et al., (2011) used a tagged version of the GEOS-Chem model to track the fractional contribution of sources of atmospheric deposition to the surface ocean. Several previous modeling studies have examined the trends in evasion of Hg(0) from the oceans and their controlling factors: i) global numerical models (Soerensen et al., 2010; Zhang et al., 2019c; Huang and Zhang, 2021), ii) regional models (e.g., Fisher et al., 2013; Bieser and Schrum, 2016), and iii) global

box models (Mason and Sheu, 2002; Sunderland and Mason, 2007; Amos et al., 2013; Chen et al., 2018). It is estimated that the surface ocean Hg concentrations and its evasion to the atmosphere has increased by a factor of 3-5 since preindustrial times (Outridge et al., 2018; Zhang et al., 2014a; Huang and Zhang, 2021). Zhang et al., (2014b, 2016b) tracked the perturbation by anthropogenic Hg emissions in the past 600 years using a 3D ocean model and found that in surface waters 77% of the Hg is of anthropogenic origin. Overall, modeling suggests that most of the Hg(0) being evaded comes from Hg

that was recently added to the ocean (60-70%), and that deeper water Hg sources are therefore a smaller component of the flux (Soerensen et al., 2010). Sunderland and Mason (2007) found that the contemporary ocean is not in a steady state with



present-day atmospheric Hg inputs. Hg concentrations will therefore continue to increase, on average, if present-day emissions remain constant. If the ocean were to reach a steady state with present-day atmospheric Hg inputs, anthropogenic Hg enrichment would increase to ~80% in the surface ocean and >150% in the deep ocean. Two-thirds of the present Hg fluxes (such as deposition on land and the ocean) are directly or indirectly of anthropogenic origin (primary or legacy anthropogenic emissions). Elimination of the anthropogenic load in the ocean and atmosphere will take hundreds of years if not more after the termination of all anthropogenic emissions (Amos et al., 2014).

### 7.2.1.1. Spatial pattern detection

The global distribution of aqueous mercury species in the ocean is determined primarily by the global patterns of a) deposition, governed by location of emissions, atmospheric transport and reduction potential, b) primary productivity, which affects the conversion of aqueous Hg(II) to Hg(0) and MeHg, c) settling of Hg sorbed to organic particles, and d) upwelling/resuspension of sedimented Hg. Gaining a comprehensive understanding of Hg levels and spatial patterns in the global ocean, especially in sensitive regions, is critical for accurately assessing the risk of human exposure and the reliability of the MC EE.

Moreover, deposition plays an important role in the distribution of Hg. Deposition is high in the western North Atlantic and western North Pacific, which are downwind of large industrial regions of the eastern U.S. and East Asia (Strode et al., 2007). Due to increasing emissions from Asia and decreasing emissions from the US, deposition to the Atlantic has decreased while increasing in the pacific over the last decades. These key regions, including the tropics, western North Atlantic and western North Pacific, should therefore be treated as "affected" by anthropogenic emissions. Also, for the tropics not located downwind of source areas, model simulations of Hg in the open oceans show elevated concentrations because of the prominent atmospheric deposition due to high precipitation in the Inter-Tropical Convergence Zone (ITCZ) in combination with increased atmospheric oxidation rates producing Hg(II) for dry deposition (Zhang et al., 2019c; Soerensen et al., 2014). While there are no large point sources of Hg in the Arctic, atmospheric transport of Hg(0) to this region still results in a significant anthropogenic impact to the Arctic Ocean (Schartup et al., 2022). Although considered as a "remote" region, Hg levels in the Arctic Ocean are therefore very sensitive to anthropogenic Hg emissions (Dastoor et al., 2022a). In the future, large quantities of Hg from thawing permafrost might be released into the Arctic.

The "Baseline" simulation (O1, Table F9) aims to obtain global levels and spatial patterns of oceanic Hg utilizing state-of-the-art model configurations to perform global-scale simulations of oceanic Hg transport. The oceanic models incorporates boundary conditions (i.e., atmospheric Hg(0) concentrations, Hg(II) deposition fluxes, and Hg from rivers) from atmospheric and mass balance model results and present-day ocean biogeochemistry fields from ocean models such as Zhang et al., (2020) and Bieser et al., (2023). Due to the sparse nature of marine observations in general, and the lack of cruises during the stormy season and an absence of repeated (inter and intra-annual) measurements in particular, a comprehensive evaluation of spatial patterns can only be achieved by combining observations and models. Here, the ensemble approach will help to





reduce the uncertainty introduced by the models through the identification of regions of (dis-)agreement between different
models.

### 7.2.1.2. Attribution of levels

To comprehensively evaluate the contributions of various Hg sources on oceanic Hg levels and spatial patterns, simulations
will be conducted using available ocean Hg models (O2.1-O2.6; Table F9). The contributions of geogenic and anthropogenic
Hg emission sources (O2.1-O2.2) to oceanic total Hg concentrations at observation sites and across global regions will be
determined by performing oceanic simulations using Hg deposition inputs from the respective atmospheric MME
experiments (see Sect. 7.1.1). The oceanic models will be further used to identify the relative contributions of different
anthropogenic emission sectors to oceanic Hg levels such as the contributions of anthropogenic emissions from sources
influenced by the MC (O2.5-O2.6). To account for non-atmospheric emission sources, the relative contribution of riverine
input will be assessed using simulations with riverine emissions from Liu et al., (2021) and without riverine export (O2.3).
Additional experiment (O2.4) will investigate the contribution of Hg and MeHg bioaccumulation in marine biota in the
lower food web (phytoplankton, zooplankton) to oceanic Hg distribution and air-sea exchange by disabling food web
interactions.

### 7.2.1.3. Limitations and recommendations

A major limitation is the sparse availability of marine measurements of speciated Hg with large parts of the ocean still
unsampled. From a model perspective, the chemical mechanisms for methylation and demethylation processes are still not
understood. In the coastal ocean, the complex interactions between Hg and other components in river water and sediment,
such as dissolved organic matter and suspended particles and biota have yet to be fully reproduced by a model. Assumptions
made for key parameters in oceanic Hg biochemistry models introduce errors in the estimation of Hg cycling in the transition
zone from freshwater to marine waters (e.g., change in pH).

Another limitation of current global ocean models is that they do not accurately represent coastal dynamics. Eddies are
dissipated at much smaller scales than the grid resolution in the turbulent boundary layers and advection schemes governed
by turbulent mixing cannot be precisely captured by these coarse-resolution ocean models (Sarmiento et al., 2004; Wunsch
and Ferrari, 2004). Furthermore, these models cannot adequately reflect transport and mixing conditions in the continental
shelfs as global model usually lack tides which are a major driver of coastal mixing. Higher resolution or unstructured grid
models are necessary to capture the complex processes at the land-sea-air interface. Here, we will perform additional runs
evaluating the impact of model resolution. Models with unstructured grid will be used to selectively increase model
resolution in coastal areas (Logemann et al., 2021). Furthermore, for the estimation of different source origins of oceanic Hg
levels and its reemission there are computational problems for conducting historic (decennia or millennia) 3D model
simulations including source-receptor relationship tracking. For future EE efforts, we recommend that model simulations
(mass balance models and fully coupled 3D models) with a focus on tracking Hg source origin (geographical and time
dependent) are further developed building on the work of Corbitt et al., (2011), Amos et al., (2013) and Chen et al., (2018).





### 7.2.2. Temporal trends

The relevance of detecting and attributing changes in the concentration and distribution of Hg in the marine environment over time to the MC on Hg is significant. This can be accomplished using a combination of observational data, modeling studies, and statistical analyses. Estimates of temporal trends from observational data is not feasible due to the lack of long-term stations and sporadic cruising activity. However, ocean Hg models can be effective tools for addressing these gaps in our understanding of temporal trends in Hg levels in the ocean. Understanding the temporal trends provides a way to directly monitor progress towards reducing exposure to MMHg and other public health and the UN Sustainable Development Goals related to Hg pollution, particularly in coastal and indigenous communities that rely on fish and seafood as a primary food source. It is also essential for identifying changes in sources and pathways of ocean Hg levels, which can inform strategies for reducing anthropogenic Hg emissions and protecting human and ecosystem health.

Ocean Hg models have emerged as a powerful tool to provide valuable insights into the drivers and impacts of Hg pollution in the ocean over time. For example, 2D slab models of the oceanic mixed layer employed to mainly focus on air-sea exchange (Strode et al., 2007; Soerensen et al., 2010), only include inorganic Hg chemistry and transportation and use GEOS-Chem model results of atmospheric Hg. These are mainly concerned with seasonal temporal variability. Multi-box biogeochemical model (Soerensen et al., 2016a; Amos et al., 2013) are also developed to simulate Hg budget and response to emission inventory. Combined with projections of future anthropogenic Hg emission scenarios, Giang and Selin (2016) used Amos et al., (2013) and Schartup et al., (2022) used Soerensen et al., (2016a) to assess future changes in Hg levels. Three-dimensional biogeochemical models including ecosystem interactions are increasingly applied to simulate the spatiotemporal dynamics of Hg species in the ocean including bioaccumulation in the marine food web (Zhang et al., 2020; Rosati et al., 2022; Bieser et al., 2023; Zhu et al., 2023).

### 7.2.2.1. Trend detection

It is currently not possible to estimate temporal trends of oceanic Hg from observational data due to the lack of long-term monitoring and only sporadic cruising activity. Ocean Hg models are potent tools for filling the gaps in our knowledge on temporal trends of Hg levels in the ocean. Some models perform offline simulations for a certain amount of time with repeated external forcing from rivers and atmospheric deposition (Zhang et al., 2020), or fixed field (Rosati et al., 2022) or archived field (Bieser et al., 2023). Others are coupled to an atmospheric model using a monthly, weekly, or even hourly (Zhang et al., 2014b, 2019c) interval for data exchange. To capture the temporal trends associated with changes in emissions, the atmosphere-ocean coupled model simulations (discussed in Sect. 5) might provide a more justifiable time-dependent simulation than previous offline ocean model simulations.

Trends in the Baseline simulation (O1; Table F9) of oceanic Hg models will be examined and compared to available measurements of Hg levels (e.g. THg, MMHg) in the ocean for 1980-2020 (when data is available and keeping in mind that these are sparse especially for the early decades and that clean sampling techniques had not always been introduced in early



observation campaigns: see Sect. 4.3 and 4.4). The Baseline simulation will be forced with upper boundary conditions (i.e., atmospheric Hg concentrations and deposition fluxes) from atmospheric and mass balance models and ocean biogeochemistry fields from present-day situation to capture short- and long-term changes in oceanic Hg levels. Ocean physics field that is used to drive the transport of Hg in the models varies temporally over 1980-2020. We will extract simulated trends for those measurement locations where observational data spanning several years have been built up and compare them with the available measurement data. Statistical metrics, such as $R^2$ in regression analysis, and model performance evaluation criteria described in Sect. 6.2 will be applied to assess whether the Baseline simulation can recapitulate the observed trends direction for the period 1980-2020.

Similar analysis will be applied to compare trends across multiple oceanic Hg models in the Baseline simulation. The analysis will not be constrained to areas where measurements are available. Global trends will be calculated, along with trends for different regions and all oceans, rather than specific locations with observations. Differences in trends for Hg fluxes, such as sea-air exchange, particle sinking, and sedimentation will also be compared across different models. Through this comparative analysis, any differences or similarities in trends will be identified, providing insights into the underlying processes that drive these trends.

### 7.2.2.2. Attribution of trends

The proposed trend analysis perturbation simulations (O3.1–O3.14, Table F9) will enable the attribution of Hg trends to emissions sources and environmental conditions in the ocean. The trend attribution analysis can be performed at different spatial scales (globally, hemispheric, regionally, and at observation locations). Perturbation simulation O3.1 allows for the separation of the impacts of changes in anthropogenic emissions and environmental factors. A 2010–2020 simulation will be conducted to examine the effect of changing meteorological conditions on oceanic Hg by keeping all other factors fixed to 2010 while changing the meteorological conditions. Furthermore, simulation O3.3-O3.14 will estimate the influences of changes in anthropogenic emissions in various global regions on oceanic Hg trends. Comparison across multiple models will indicate whether the impact of drivers is similar among the models. Resolving the role of different drivers also provides an opportunity to better interpret biases between simulated baseline trends and observations, as well as model differences. Through this attribution exercise, we aim to improve our understanding of the drivers that are responsible for the observed Hg trends in the ocean. This will aid in the design of future EE cycles, which will focus on isolating the impact of Minamata Convention activities on Hg trends.

### 7.2.2.3. Limitations and recommendations

As the ocean has a much slower turnover time compared to the atmosphere, a time span of 10 years might be insufficient to detect certain trends at a significant level. This is especially true for Hg(II) methylation and bioaccumulation trends. Several Hg biogeochemical processes such as methylation/demethylation and sorption/desorption to organic material remain highly uncertain. Without proper parameterizations of these processes, it is difficult to accurately capture and attribute temporal





trends. Finally, in this study models for atmosphere and ocean will be coupled offline. In future EE efforts fully coupled models should be used if possible.

### 7.3. Environmental mass balance

The mass balance of budgets between environmental Hg sources (anthropogenic and natural), burdens, movements, and
2275 burial provide a quantitative summary of the global Hg cycling. The mass balance implies that changes in Hg levels in any environmental compartment of the biosphere over time would result in compensating changes in other interacting compartments. Thus, assessment of mass balance is useful for explaining the linkages between temporal trends of Hg in environmental matrices. Further, the knowledge of relative magnitudes of Hg budgets provides insight into how curbing anthropogenic Hg emissions under the MEAs is likely to affect Hg levels in different media.

Given the biogeochemical complexity and sparseness of Hg observations, global Hg budgets have been commonly reported based on 3D and multi-media mass balance model simulations using compilations of anthropogenic and natural Hg emissions (Outridge et al., 2018). However, increased observations during the last decade have enabled some observational budgets, notably for terrestrial-atmosphere Hg exchange (Sommar et al., 2020; Zhou et al., 2021; Zhou and Obrist, 2021) and marine Hg burden (Bowman et al., 2020). With increasing number and accuracy of models, modeling estimates of global
atmospheric Hg budgets have been successively revised (Selin et al., 2008; Amos et al., 2013; Song et al., 2015; Cohen et al., 2016; Horowitz et al., 2017; Shah et al., 2021; Kawai et al., 2020; Zhou et al., 2021; Feinberg et al., 2022; Zhang and Zhang, 2022; Sonke et al., 2023). For example, updated understanding of vegetation and soil uptake of Hg(0) led to the revision of Hg(0) deposition budget to land up from 1200 (Horowitz et al., 2017) to $2850 \pm 500$ Mg y$^{-1}$ (Sonke et al., 2023). The Hg mass balance for global ocean has been developed using 3D and mass balance models (Sunderland and Mason,
2007; Semeniuk and Dastoor, 2017; Zhang et al., 2019c), including interactions with marine biota (Zhang et al., 2020). On the regional scale, a combination of observations and models have been used to estimate Hg mass balance (Arctic: Soerensen et al., 2016a; Sonke et al., 2018; Petrova et al., 2020; Dastoor et al., 2022a; Mediterranean: Cossa et al., 2022; China: Feng et al., 2022a).

Reported global Hg budgets thus far are mostly based on individual modeling estimates. However, a combination of MME
and observed estimates have been used to develop Arctic Hg mass balance (Dastoor et al., 2022a), which led to improved understanding of Arctic Hg cycling and key knowledge gaps in the region. In this study, global environmental budgets of total Hg (THg) (i.e., Hg anthropogenic and natural emissions and releases, loadings, and inter-media exchange and burial fluxes) will be developed based on the coordinated baseline MME simulation; budget details such as exchange fluxes for different land cover types and Hg speciation will be added. Since human exposure to Hg toxicity is dominated by the
consumption of seafood, mass balance budgets of both THg and MeHg (the Hg form that biomagnifies in food chains) in the global ocean will be developed. Range of uncertainties in global Hg budgets will be determined based on updated synthesis





of monitoring data and multi-model experiments examining sensitivity to uncertainties in emissions and physical and biogeochemical processes.

The Hg mass balance uncertainty estimates developed in this study will help prioritize Hg monitoring and research areas. While there has been progress on observations of global air-vegetation-soil flux exchange, air-cryosphere and air-ocean-sediment Hg exchanges remain understudied. Hg stable isotopes studies have led to novel constraints on speciated Hg pathways in terrestrial ecosystems (Enrico et al., 2016; Obrist et al., 2017; Zhou et al., 2017; Yuan et al., 2019b); these methods can be further utilized to reduce errors in flux estimates of other matrices. Understanding the role of legacy Hg recycling is important to evaluating the recovery of environmental Hg under the MC implementation (Amos et al., 2013, 2015). Global change induced alterations in processes such as net primary production, wildfires, permafrost and sea ice cycles and river discharge can significantly shift the global Hg mass balance (Denkenberger et al., 2012; Hsu-Kim et al., 2018; Schaefer et al., 2020). Diagnosing recent perturbations of global Hg mass balance due to climate change stressors is currently difficult due to lack of long-term observations in key environmental matrices. The development of past Hg records from environmental archives and mechanistic 3D atmosphere–land–ocean biogeochemical Hg models would help in constraining the contemporary Hg mass balance and model its future trajectory under global change.

## 8. Modeling analysis: process understanding

Mechanistic models integrate available knowledge of Hg sources and biogeochemical processes of environmental matrices to estimate Hg levels. Spatiotemporal inconsistencies of modeled Hg levels with observations suggests inaccuracies or knowledge gaps in primary drivers (emissions and releases and environmental conditions) and processes of Hg and their model representations. Sensitivity of modeled Hg levels to these inaccuracies or knowledge gaps can be diagnosed using model simulations and statistical analysis, guiding further development of Hg models, emission inventories, and field and lab studies investigating environmental Hg processes.

### 8.1. Atmospheric models

Previous modelling studies have explored the influence of anthropogenic forcing on the simulation of atmospheric Hg concentration and deposition. Uncertainties in atmospheric Hg emissions inventories impact the prediction of source-receptor relationships of Hg (De Simone et al., 2016, 2017a), which are important for estimating the impacts of MC policy on the global distribution of Hg deposition. Previous studies have used models to evaluate Hg emissions inventories against observational evidence (Hg air concentrations and deposition). The impacts of Hg speciation (Hg(0) vs. Hg(II)) in emission inventories was analyzed by Bieser et al. (2014), Zhang et al. (2016b), and Giang et al. (2018), finding that accurate representation of emissions speciation is crucial for capturing the impacts of Hg emissions reductions on atmospheric Hg concentrations and deposition. The impacts of other types of anthropogenic forcing (e.g., land use change) have been less



frequently investigated. Previous generations of models showed limited impacts of future land cover and vegetation change on atmospheric Hg (Zhang et al., 2016a), yet these models likely underestimated the importance of Hg(0) dry deposition in the global Hg budget (Zhou et al., 2021; Feinberg et al., 2022). A recent modeling study implied that land use change

impacts on atmospheric Hg deposition can be on par with direct anthropogenic emission impacts (Feinberg et al., 2023). The choice of other input driver datasets can play a role in simulated Hg cycling, including the choice of meteorological input data (Dastoor et al., 2015; Giang et al., 2018).

The prediction of atmospheric Hg cycling depends not only on the input driving datasets, but also on the design of model parametrization of Hg biogeochemistry. There are several published studies investigating the impact of the Hg chemical

scheme on the spatial distribution and seasonality of Hg concentrations and deposition (Saiz-Lopez et al., 2020, 2018; Ye et al., 2018b; De Simone et al., 2017a; Travnikov et al., 2017). The parametrizations of air-surface exchange also play an important role in the emissions and fate of atmospheric Hg. Zhang et al. (2019c) tested five parametrizations of air-sea Hg(0) exchange, determining that global net Hg(0) emissions range between 2840–3710 Mg y$^{-1}$, with strong regional differences between parametrizations. The choice of Hg(0) dry deposition parametrization can strongly impact the magnitude of Hg(0)

uptake by vegetation (Zhou et al., 2021; Feinberg et al., 2022). Even with the same model parametrizations, the horizontal resolution of Hg models can impact the simulation of wet deposition due to finer resolution models being better able to capture deep convection (Xu et al., 2022; Zhang et al., 2012).

By constructing simulations in a diverse set of atmospheric Hg models with updated harmonized boundary conditions, we can identify current knowledge gaps in the biogeochemical Hg cycle and test key uncertainties in available models.

Sensitivity and uncertainty simulations will provide further insight into the drivers of Hg trends and spatial distribution and inspire further developments to improve cause-effect relationships in Hg cycling.

MCHgMAP will explore the impacts of several anthropogenic and environmental forcing factors on Hg cycling. In addition to the previously discussed Baseline (A1) and trend analysis (A3a.1–A3b.7) simulations for 2010–2020, an array of atmospheric model simulations is proposed to test the influence of alternative emissions datasets and parametrizations

(A4.1–A4.8). Additional idealized sensitivity simulations (A5.1–A5.10) are proposed to investigate the sensitivity of Hg model budgets in a single year (2015) to individual processes. This set of simulations is idealized in the sense that rough perturbations are imposed on drivers (e.g., A5.1: all Hg emissions are released as Hg(0)) or model parameters (e.g., A5.3: enhancing Hg oxidation rates by +50%). These experiments can probe whether existing models differ in their sensitivity to individual drivers or biogeochemical parameters, enabling the prioritization of key uncertainties for future EE efforts. The

idealized experiments can also explore potential impacts of future global change, e.g., the increases in LAI due to global greening (A5.8) or the disappearance of sea ice (A5.10).



Currently, there are limited options for anthropogenic Hg emissions inventories, especially continuous time series using a consistent methodology covering the years 2010-2020 or longer (see Sect. 3.2.1), restricting the ability to investigate emission uncertainties and sensitivity to modeling results. Chemical mechanisms differ among the models because of knowledge gaps and uncertainties in atmospheric Hg and Br chemistry, but the interpretation of differences in model simulated chemistry is made more difficult due to limited observations of speciated Hg and potential biases in these observations (Lyman et al., 2020b). Finally, the sensitivity analysis performed with atmospheric models decoupled from the aquatic and terrestrial compartments does not account for changes of Hg levels in other environmental media affecting secondary emissions to the atmosphere. The internal consistency of this analysis would be improved by Hg simulations in Earth system models (e.g., Zhang and Zhang, 2022), yet the computational expense of dynamically coupled models imply that atmosphere-alone sensitivity simulations are worthwhile for probing process uncertainties.

## 8.2. Ocean models

Previous single model process studies have investigated (1) the impact of riverine Hg loads (Fisher et al., 2012; Soerensen et al., 2012; Zhang et al., 2015b), (2) the role of organic matter in marine Hg cycling and methylation (Soerensen et al., 2016a), (3) air-sea exchange (Bieser and Schrum, 2016; Zhang et al., 2023a), (4) the impact of nutrients on methylation and bioaccumulation (Soerensen et al., 2016b), and (5) the impact of adding interactions with the marine ecosystem (Zhang et al., 2019c; Rosati et al., 2022; Bieser et al., 2023). Yet, so far, no systematic model intercomparison or ensemble study has been performed for oceanic Hg cycling. MCHgMAP will, for the first time, use a set of marine Hg models driven by harmonized input data and model parameterizations. Thus, the Baseline (O1) scenario (Table F9) will be used to quantify the range of currently available models (Table F2) based on harmonized input data and process parameterizations (Table F6). So far it is unknown how marine Hg models compare and whether spatial or seasonal patterns are in model (dis-)agreement.

The initial evaluation will focus on comparing the Baseline scenario model simulations to identify regions, times, and species for which the model's baseline results differ (see Sect. 6.2). The models will be evaluated against observations with a focus on regions and species where models are not in agreement with each other. The initial evaluation will also include hydrodynamic (e.g., circulation patterns, overturning time, regional stratification and upwelling) and biogeochemical (e.g., nutrients, phytoplankton and zooplankton concentrations and seasonality) variables so as to understand the discrepancies of non-harmonized internal model processes. Further, a range of simulations (O4.1-O4.7) attributing uncertainties to individual external and internal modeling factors are proposed.

*Anthropogenic emissions*: The model uncertainty due to emission inventories and speciation will be estimated using different atmospheric Hg boundary conditions from the atmospheric models. For this, there are three scenarios (O4.1 - O4.3) using alternative anthropogenic emission inventories as model drivers. This will help in understanding the uncertainty due to Hg total emission, and their spatial distribution and speciation. The latter is of special importance, as Hg speciation in emission, especially from the largest source sector coal combustion, has





been changing due to new technologies such as desulphurization. Depending on the fraction of Hg emitted in oxidized or particulate form, even the identical total Hg emissions can lead to vastly different spatial distributions of Hg deposition to global oceans since the atmospheric lifetime of elemental Hg(0) is an order of magnitude higher than that of the other species.

*Methylation-demethylation:* The exact processes that govern methylation and demethylation are still not fully understood. To better understand the model uncertainty due to different chemical mechanisms for Hg transformation, scenario O4.4 tests the impact of the chemical mechanism by harmonizing reactions and rates for Hg transformations.

*Air-sea exchange:* With an annual Hg net evasion of 3000-4000 Mg from the ocean, it is the largest Hg flux into the atmosphere. Yet, this process is not well understood and only indirectly observable. Estimates for global air-sea exchange are based on either co-located Hg(0) measurements in air and surface ocean or from global atmospheric models. Simulations O4.5 and O4.6 will use alternative parametrizations for the two-layer air-sea exchange model to estimate the uncertainty due to this process. Moreover, by coupling to the atmospheric models, we will create consistent multi-compartment estimates taking into account atmospheric and marine processes and quantify the model uncertainty due to the air-sea exchange parametrization.

*Model resolution:* Simulation O4.7 examines the impact of marine model resolution. This is of interest as low resolution global models cannot resolve the coastal shelf seas. However, these regions have the largest biological productivity and are thus potentially both an important sink for Hg and a region of enhanced methylation. The impact of the shelf oceans and the necessity of high-resolution ocean models to resolve this process for carbon cycling has recently been shown by Mathis et al. (2022).

*Secondary emissions:* A central question in the global Hg cycle is what fraction of marine Hg evasion into the atmosphere stems from previously deposited anthropogenic Hg emissions. This secondary emission fraction will be estimated based on simulations O2.1 and O2.2 where contemporary anthropogenic and geogenic emissions are zeroed out respectively.

*Bioaccumulation:* The central reason for nations to agree on the MC is the threat to human health posed by Hg and MeHg in sea food. With biogeochemical marine Hg models, it is possible to quantify methylation and uptake of Hg in the marine ecosystem and the impact of external stressors like anthropogenic emissions and climate change thereon. Understanding the influence of biotic Hg is important to the distribution of oceanic Hg. For example, a recent study (Amptmeijer and Bieser, 2023) shows that Hg uptake by phytoplankton can lead to higher MeHg concentrations in the surface ocean as well as increased Hg(II) sedimentation and Hg(0) air-sea exchange. Simulation O2.4 investigates the role of Hg bioaccumulation in the lower food web (phytoplankton, zooplankton) and sedimentation of Hg sorbed to organic particles on marine Hg.



## 9. Modeling future scenarios

Within the limitations and uncertainties outlined above, and quantified through the proposed simulations, models are essential tools for projecting future trajectories of Hg levels in air and ocean. This includes projecting the influence of scenarios of implementation of the MC and other MEAs, as well as changes in Hg levels that result from changing

environmental conditions (due to climate and other global changes).

Appropriate methods for modeling changes in atmosphere and ocean Hg levels depend strongly on the timescale of interest. With respect to the immediate timeframe of the first MC EE, projecting changes over a few years requires consideration of changes in the contemporary cycle (see Fig. 1). For changes over several years to decades, the long-term cycle becomes increasingly relevant (Amos et al., 2013). It is important to note that changes to the long-term cycle that affect atmospheric

and ocean Hg concentrations on timescales of several years or more are driven not only by past emissions, but also by the cycling of ongoing and future Hg emissions (Selin, 2018).

A number of previous studies have used atmospheric models to project atmospheric concentrations of Hg, as well as changes in wet and dry deposition, under future conditions, both regionally and globally. Pacyna et al., (2016) used two global chemical transport models (GLEMOS and ECHMERIT) to project changes in future Hg concentrations and

deposition at global scale under policy scenarios relating to current pollution control and climate policies. Angot et al., (2018) simulated the same emissions scenarios using the GEOS-Chem model. Other efforts have also drawn upon global scenarios to quantify potential changes in source-receptor relationships (Chen et al., 2018; Corbitt et al., 2011; Zhang, 2021) and regional impacts (Giang et al., 2015; Lei et al., 2014; Schartup et al., 2022). Relating to the ocean, there are fewer existing studies that project future ocean Hg concentrations under different scenarios (consistent with the more general

limitation in the number of models that exist, as noted in Sect. 2.2). Zhang et al., (2021) simulated five different future anthropogenic emissions scenarios (ranging from a >80% reduction to a 2.5-fold increase) using a coupled three-dimensional atmosphere-ocean model. A recent assessment of the Arctic environment simulated likely changes in both atmospheric and oceanic Hg there under future emissions scenarios (Schartup et al., 2022).

A few attempts have used various degrees of model coupling to address the combined impact of future and legacy changes,

accounting for the influence of future emissions scenarios on the long-term cycle. One study calculated that extending emissions reductions over 5 years lead to a 14% decrease on local-scale Hg deposition (Angot et al., 2018). These estimates are affected by the uncertainties in timescales of inter-compartmental cycling and overall mercury budget: Zhang et al., (2023a) recently calculated a lower sensitivity of atmospheric levels to anthropogenic emissions than had been estimated in previous models, due to a larger role for legacy emissions from the terrestrial and marine environment.

Sect. 3.2.2 (Future emissions) outlines progress towards developing future emissions scenarios for Hg that leverage best-available information on trajectories of different economic activities (including related efforts to limit greenhouse gas emissions by reducing fossil fuel use) with updated data on control measures. Understanding the relative importance of each of these drivers through different emissions scenarios (Table E12) will be important for differentiating the influence of





Minamata-related legislation, ambitious end-of-pipe control measures, and net-zero climate targets, on levels of Hg in
atmosphere and ocean.

The simulations A6.1-A6.3 are designed to provide information on the relative impact of changes in atmospheric concentrations of mercury resulting from changes in anthropogenic emissions, environmental factors, and combined influences. Because each model setup is different, it will be important in these simulations to clearly differentiate how the impacts of past vs. future emissions are treated, the latter of which will also influence long-term cycling, as well as the
source of future environmental conditions. For those scenarios that do not account of the time-varying legacy contribution from future scenarios, they will reflect a lower bound. As the impact of different sources adds linearly in the global mercury cycle, it will also be possible to derive source-receptor relationships useful for policy and future evaluation from the additional atmospheric simulations for the present (an example of such a strategy is presented by De Simone et al., (2020)). In this study, we can update the legacy emission estimates for the atmospheric models by off-line coupling to the ocean and
making use of mass balance models. For the marine model simulations, O5.1-O5.3 will mirror the atmospheric setup. To allow for the identification of significant trends, the simulation will cover a timespan of 30 years.

Scenario A2.1, the perturbation simulation with no anthropogenic emissions, will also be informative in understanding and bounding the potential for anthropogenic actions to affect mercury concentrations. Examining model response to these scenarios, particularly with coupled and multimedia simulations, will provide important uncertainty bounds on future policy-
relevant analysis. Quantifying mercury response to zeroing out emissions will be an important metric through which to compare model processes, similar to the Zero Emissions Commitment (ZEC) concept used for earth system models for $CO_2$ (MacDougall et al., 2022).

In all future scenarios, it will also be important to account for variability (especially with respect to future climate influences). While single year constant meteorological comparison is appropriate for evaluating impacts of near-term
emission changes, multiple years or even decades of simulation will be required to account for natural variability in the climate system and its influence on atmospheric chemical processes (Garcia-Menendez et al., 2017; Giang et al., 2018). In addition, differences among models reflect systematic uncertainties in key processes underlying the mercury cycle. Moving forward, model intercomparisons and model analysis in combination with increasing data constraints, including through the activities proposed above, will help narrow key uncertainties that hamper current ability to project future scenarios.

**10. Summary**

The Multi-Compartment Hg Modeling and Analysis Project (MCHgMAP), an international ensemble modeling initiative to inform the effectiveness evaluations of the Minamata Convention on Mercury and Convention on Long-Range Transboundary Air Pollution, has been introduced. The MCHgMAP is intended to address the key policy questions of these multi-lateral environmental agreements (MEAs) using up-to-date Hg modeling capacity: (1) What are the contributions of
anthropogenic emissions and releases and other Hg sources to current Hg levels observed in air, biota, humans, and other





media?, (2) How have these contribution levels changed over time and over the timeline of the MEA?, (3) How do the contribution levels and their trends vary geographically at the global scale?, (4) What are the contributions of anthropogenic emissions and releases and other drivers to the temporal trends in observed Hg levels across global regions? and (5) How are observed Hg levels expected to change in the future?

This overview paper is an important preparatory stage of the MCHgMAP, which will be followed by individual publications on different aspects of the study. Sect. 1 to 4 provide a synthesis of what is currently known and not known about the estimates of mercury primary and secondary emissions, observations (direct monitoring and natural archives) and single to multi-compartmental mercury models, and how these advances can be exploited to develop mechanistic understanding of the past, present and future fate of environmental mercury. The project aims at improving comparability across the participating

models by employing a consensus set of emissions and environmental conditions (where possible), and constraining model evaluation with common observation datasets. Methodologies for developing harmonized model drivers including historical and future primary and secondary emissions as well as biochemical fields that can be used in core and optional simulations are described. The Sect. 5 and 6 describe the rationale, design, implementation, and evaluation of multi-model experiments. The subsequent Sect. 7 to 9 present analysis of modeling results, i.e., detection and attribution of spatial patterns and

temporal trends, identification of uncertainties, and future projections, to address the policy questions. Additionally, all sections provide recommendations for the areas of mercury research (monitoring and modeling) that are needed to improve effectiveness evaluations of the Hg MEAs going forward.

A challenge of analyzing the fate of emitted mercury is that it can recycle between the atmosphere, land, and ocean (i.e., secondary emissions and releases), and as a result, past and present emissions can continue to affect the environment on

timescales of decades to centuries. There have been significant developments in marine and land Hg observations and modeling since previous ensemble Hg modeling studies which were mainly focused on atmospheric processes. Availing new advances, the MCHgMAP utilizes a coordinated simulation approach between single medium (atmosphere, land, and ocean) and multi-media Hg models to allow and measure the influence of changes in secondary Hg exchanges on environmental Hg cycling. The MCHgMAP *baseline* and *perturbation* ensemble simulations are designed to robustly analyze global and

regional environmental Hg levels, i.e., detection and attribution of spatial gradients and temporal trends, understanding of global mass balance between sources, sinks and burdens, and future projections under different scenarios of implementation of the Minamata and other MEAs and changing environmental conditions. Additional model experiments, *uncertainty* and *idealized*, are designed to assess the sensitivity of modeled levels and trends to their drivers and underlying processes to guide future developments of emission inventories, and field and lab studies investigating environmental mercury. Finally,

the model experiments are prioritized into three tiers - core, optional and future simulations - to ensure a systematic and comprehensive analysis and flexibility in the plan to encourage participation of a variety of mercury models from the scientific community.



**Appendix A. The policy questions**

The coordinated multi-media multi-model simulations in MCHgMAP are designed to analyze current and future levels and
temporal trends of environmental Hg on global and regional scales to inform the effectiveness evaluations of the MC and
LRTAP Conventions. The MCHgMAP activities are targeted to address the following policy questions (divided into five
categories) in alignment with the objectives outlined in the Minamata monitoring guidance UNEP/MC/COP.4/INF/12
(Guidance on monitoring mercury and mercury compounds to support the effectiveness evaluation of the Minamata
Convention | Minamata Convention on Mercury, 2023) and the data analysis plan of the MC Open-Ended Science Group
(UNEP/MC/COP.5/INF/24). The first set of questions is aimed at developing and/or examining the levels and trends of
environmental Hg drivers, i.e., primary and secondary Hg emissions and releases, and environmental conditions. The next
four categories of policy questions address the analysis of environmental Hg spatiotemporal patterns and trends, i.e., their
detection, attribution, process understanding and future projections. In the following list, "mercury levels" refer to Hg
concentrations and fluxes.


**A.   *Driving sources and environmental conditions of environmental mercury***

1.   What are current levels and temporal trends of primary Hg emissions and releases from anthropogenic and natural
sources across the globe?
2.   What are current levels and temporal trends of secondary Hg emissions and releases from biomass burning, soils
and oceans across the globe?
3.   What are current levels and temporal trends of environmental variables related with Hg processes across the globe?
4.   What are the fractional contributions of changes in anthropogenic Hg emissions and releases and environmental
conditions to changes in secondary emissions and releases of Hg over the period circa 2010 to present across the
globe?
5.   How are anthropogenic Hg emissions and releases expected to change in future under different scenarios of
implementation of the Minamata and other Conventions across global regions?
6.   How are secondary Hg emissions and releases from biomass burning, soils and oceans expected to change in future
under changing anthropogenic Hg emissions and releases and environmental conditions across global regions?


**B.   *Detection of spatiotemporal features of environmental mercury***

7.   What are the spatiotemporal patterns and trends of Hg levels in air and ocean across the globe?
8.   What is the spatial representativeness of current levels and temporal trends of Hg observed in air and ocean?





9.  Are the spatial patterns and temporal trends of Hg levels observed in air and ocean consistent with the anthropogenic emissions inventories/inventory data on the global scale?

10.  How are spatial patterns and temporal trends of Hg levels in air and ocean related with patterns and trends of other pollutants or environmental variables on the global scale?

*C.  Attribution of spatiotemporal features of environmental mercury*

11.  What are the fractional contributions of primary (anthropogenic and natural) and secondary Hg emissions and releases to current Hg levels observed in air and ocean across global regions?

12.  How have the fractional contributions of primary (anthropogenic and natural) and secondary Hg emissions and
releases to levels of Hg observed in air and ocean changed over the period circa 2010 to present across global regions?

13.  What is the fractional contribution of anthropogenic Hg emissions influenced by the Convention to current Hg levels observed in air and ocean across global regions?

14.  How has the fractional contribution of anthropogenic Hg emissions influenced by the Convention to Hg levels
observed in air and ocean changed over the period circa 2010 to present across global regions?

15.  What are the fractional contributions of changes in primary anthropogenic and natural Hg emissions and releases and environmental conditions to changes in Hg levels observed in air and ocean over the period circa 2010 to present across global regions?

16.  What is the fractional contribution of changes in anthropogenic Hg emissions influenced by the Convention to
changes in Hg levels observed in air and ocean over the period circa 2010 to present across global regions?

17.  In each world region, how much of the contribution of changes in primary anthropogenic Hg emissions to changes in Hg levels observed in air over the period circa 2010 to present are due to emissions within the region versus outside the region?

*D.  Process understanding of environmental mercury*

18.  How consistent are the spatial patterns and temporal trends of Hg levels observed in air and ocean with estimates from current mechanistic models across global regions?

19.  How sensitive are the spatial patterns and temporal trends of Hg levels in air and ocean to various primary and
secondary emissions and releases and environmental conditions across global regions?

20.  How sensitive are the spatial patterns and temporal trends of Hg in air and ocean to various Hg processes across global regions?





*E. Future projections of environmental mercury*


21. How are Hg levels in air and ocean expected to change in future under different scenarios of implementation of the Minamata and other Conventions across global regions?

**22.** How are Hg levels in air and ocean expected to change in future under changing environmental conditions across global regions?


**Appendix B. Harmonized air-sea flux formulation**

The exchange of mercury between the ocean and the atmosphere plays an important role in the cycling of mercury in the global environment. Available parameterizations of mercury air-water exchange used in modelling and measurement studies still vary considerably indicating an incomplete understanding of the process and lack of direct measurements. We here

present the parameterizations recommended for use in the MCHgMAP multi-model experiments.

The two-layer (film) model by Liss and Slater (1974), which is the traditional parameterization used to determine the Hg air-water gas exchange flux is recommended. It is possible to neglect the gas-side part of the transfer velocity in the model as the gas-side conductance for Hg is much bigger than on the water-side ($k_w << k_a H'$) (Nerentorp Mastromonaco et al., 2017a).

We further recommend the parameterization of the gas transfer velocity ($k_w$) by Nightingale et al. (2000a) based on a comprehensive multiple-tracer study. In addition, the Nightingale et al. (2000a) parameterization represents the middle range of published parameterizations (Fig B1). Recent micrometeorological measurements by Osterwalder et al. (2021) provide estimates of $k_w$ directly determined for Hg. However, this is the first time this method has been used for Hg flux evaluation and more measurements are needed to validate the result. Therefore, the application of this alternative parameterization is

recommended only for the uncertainty analysis.





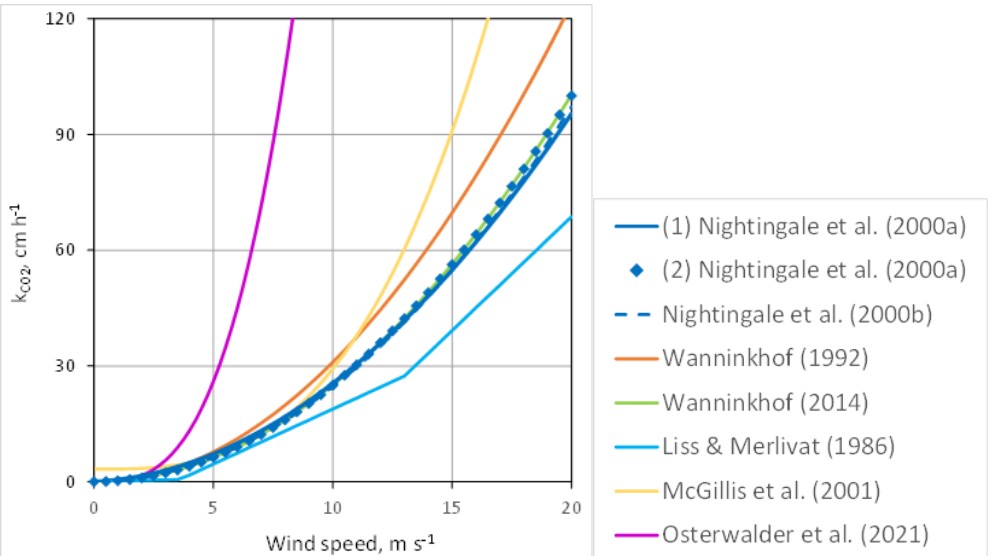

**Figure B1. Wind speed dependence of air-water gas transfer velocities from various parameterizations often applied in Hg modelling and measurement studies scaled to a Schmidt number of $CO_2$ at 20°C ($k_{CO2}$). Nightingale et al. (2000a)**
**2615    parameterizations 1 and 2 (in solid dark blue) are both commonly used in Hg work and give almost identical results. The dark blue dotted line refers to Nightingale et al. (2000b), the orange line to Wanninkhof (1992), the green line to Wanninkhof (2014), the light blue line to Liss and Merlivat (1986), the yellow line to McGillis et al. (2001), and the pink line to Osterwalder et al. (2021).**

The temperature dependence of the gas transfer velocity of Hg is defined by the ratio of the Schmidt numbers of $CO_2$ and Hg (Schmidt number is a ratio of water kinematic viscosity to the diffusion coefficient of a gas). The biggest differences between available parametrizations for the Schmidt numbers of $CO_2$ and Hg relate to the Schmidt number of $CO_2$ and the diffusion coefficient of Hg(0). We find that the Schmidt number of $CO_2$ from Poissant et al. (2000) does not seem to be based on reliable sources and recommend the Schmidt number of $CO_2$ from Wanninkhof (2014) (Fig. B2). For the diffusion

coefficient of Hg(0), we recommend the newest experimentally determined approximation from Kuss (2014). Parameterizations of water kinematic viscosities by Kestin et al. (1978) and Sharqawy et al. (2010; fresh and saltwater versions) do not differ significantly from each other. Of the two available parameterisations of the water kinematic viscosity for seawater (Wanninkhof, 1992; Sharqawy et al., 2010), we suggest the latter to be more universal since it provides values for a wide range of salinity. It is further consistent with the parameterisation of the Schmidt number of $CO_2$ from

Wanninkhof (2014). Fig. B3 shows the temperature dependence on the transfer velocity for parameterizations recommended compared to some alternative combinations.



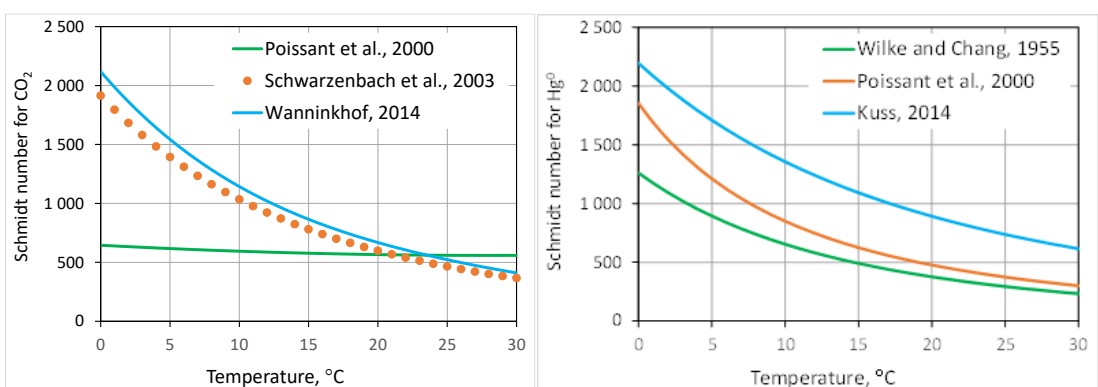

**Figure B2. A) Temperature dependence of the Schmidt number for CO₂ in seawater (from Poissant et al. (2000) in green, Schwarzenbach et al. (2003) in orange, and Wanninkhof (2014) in blue). Data from Schwarzenbach et al. (2003) are added for comparison, B) Temperature dependence of the Schmidt number of Hg(0) in seawater (from Wilke and Chang (1955) in green, Poissant et al. (2000) in orange, and Kuss (2014) in blue).**

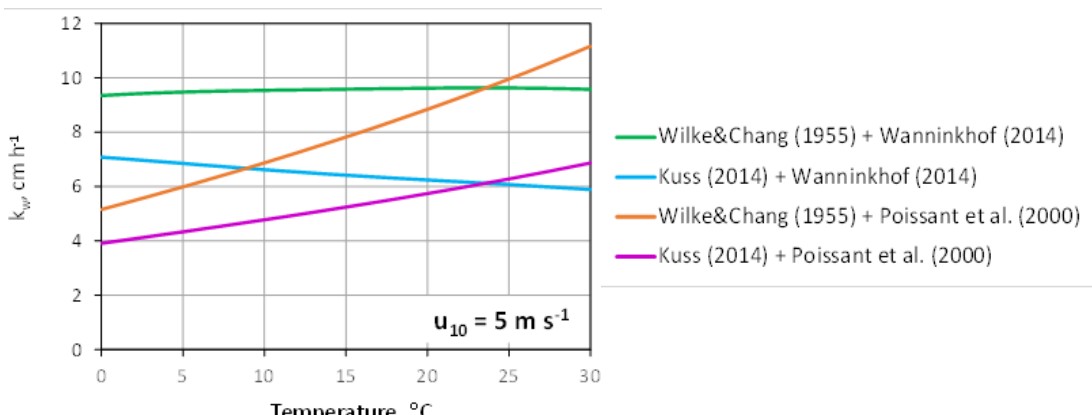

**Figure B3. Temperature dependence of air-seawater gas transfer velocity for Hg(0) (from Wilke and Chang (1955) and Wanninkhof (2014) in green, from Kuss (2014) and Wanninkhof (2014) in blue, from Wilke and Chang (1955) and Poissant et al. (2000) in orange, and from Kuss (2014) and Poissant et al. (2000) in pink).**

The equations below outline all recommendations for Hg(0) air-water exchange parameterisations for the model experiments under MCHgMAP. Parameterization of Hg(0) exchange flux between air and seawater (S = 35 g/kg) approach:

1. $F_{Hg} = k_w \left( c_w - \frac{c_a}{H'} \right)$ [ng m$^{-2}$ h$^{-1}$]
   $c_a$ and $c_w$ are concentrations of Hg(0) in air and water [ng m$^{-3}$].

2. $H' = exp \left( -\frac{2404.3}{T} + 6.92 \right)$ (Andersson et al., 2008a)
   $T$ is the water temperature [K].

3. $k_w = (0.222u_{10}^2 + 0.333u_{10}) \left( \frac{Sc_{Hg}}{Sc_{CO2}} \right)^{-0.5} \times 10^{-2}$, [m h$^{-1}$] (Nightingale et al., 2000a)
   $u_{10}$ is wind speed at 10 m height [m s$^{-1}$].

4. $Sc_{CO2}^{sw} = a_1 + a_2t + a_3t^2 + a_4t^3 + a_5t^4$ (Wanninkhof, 2014)





$a_1 = 2116.8$, $a_2 = -136.25$, $a_3 = 4.7353$, $a_4 = -0.092307$, $a_5 = 0.0007555$

$t$ is water temperature [°C].

5.  $Sc_{Hg}^{sw} = v_{sw}/D_{Hg}^{sw}$ , $v_{sw} = \eta_{sw}/\rho_{sw}$ (Sharqawy et al., 2010)

$\eta_{sw} = \eta_{fw}(1.064 + 6.067 \times 10^{-4}t - 2.753 \times 10^{-6}t^2)$, [g cm$^{-1}$ s$^{-1}$]

$\eta_{fw} = 4.2844 \times 10^{-4} + (0.0157(t + 64.993)^2 - 9.13)^{-1}$, [g cm$^{-1}$ s$^{-1}$]

$\rho_{sw} = 1.028 - 4.970 \times 10^{-5}t - 5.575 \times 10^{-6}t^2$, [g cm$^{-3}$]

$t$ is water temperature [°C];

$D_{Hg}^{sw} = 0.0011 \, exp(-1330.2/T)$, [cm$^2$ s$^{-1}$] (Kuss, 2014)

$T$ is the water temperature [K].

**Appendix C. Geographic regions for multi-model analysis**

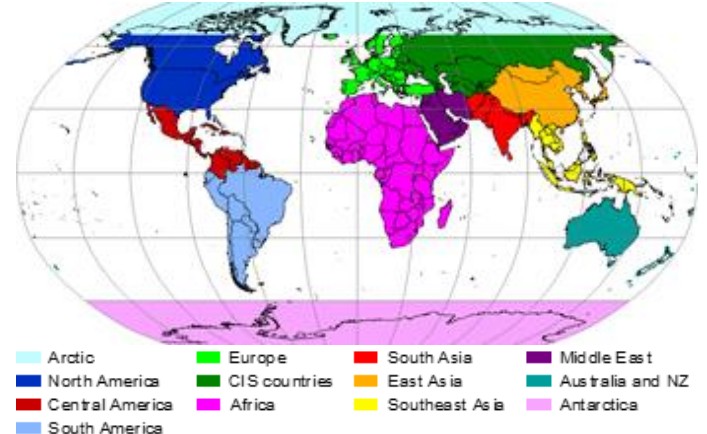

**Figure C1. Geographical receptor regions for atmospheric model analysis.**

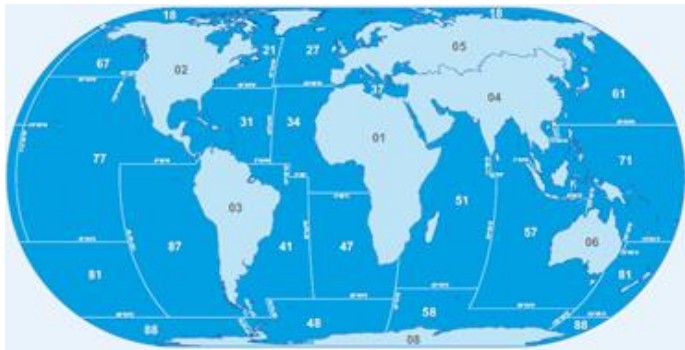

**Figure C2. Aquatic receptor regions for ocean model analysis (FAO major fishing areas,
https://www.fao.org/fishery/en/area/search).**

| Code | Fishing area | Code | Fishing area | Code | Fishing area |
|------|--------------|------|--------------|------|--------------|





| 18 | Arctic Sea | 47 | Southeast Atlantic | 71 | Western Central Pacific |
|----|-----------|----|----|----|----|
| 21 | Northwest Atlantic | 48 | Antarctic Atlantic | 77 | Eastern Central Pacific |
| 27 | Northeast Atlantic | 51 | Western Indian Ocean | 81 | Southwest Pacific |
| 31 | Western Central Atlantic | 57 | Eastern Indian Ocean | 87 | Southeast Pacific |
| 34 | Eastern Central Atlantic | 58 | Antarctic Indian Ocean | 88 | Antarctic Pacific |
| 37 | Mediterranean and Black Sea | 61 | Northwest Pacific | | |
| 41 | Southwest Atlantic | 67 | Northeast Pacific | | |


**Table C1. Geographical regions for model analysis**

| Type | List / description |
|------|-------------------|
| Geographical regions (Fig. C1) | 1. Arctic<br>2. North America<br>3. Central America<br>4. South America<br>5. Europe<br>6. CIS countries<br>7. Africa<br>8. Middle East<br>9. South Asia<br>10. East Asia<br>11. Southeast Asia<br>12. Australia and New Zealand<br>13. Antarctica |
| Land cover types (for each geographical region) | 1. Fresh water bodies<br>2. High vegetation<br>3. Low vegetation<br>4. Crops<br>5. Urban areas<br>6. Bare lands / ice cover |
| Aquatic regions (FAO major fishing areas, Fig. C2) | 1. Arctic Sea<br>2. Northwest Atlantic<br>3. Northeast Atlantic<br>4. Western Central Atlantic<br>5. Eastern Central Atlantic<br>6. Mediterranean and Black Sea<br>7. Southwest Atlantic<br>8. Southeast Atlantic<br>9. Antarctic Atlantic<br>10. Western Indian Ocean<br>11. Eastern Indian Ocean<br>12. Antarctic Indian Ocean<br>13. Northwest Pacific<br>14. Northeast Pacific<br>15. Western Central Pacific<br>16. Eastern Central Pacific<br>17. Southwest Pacific<br>18. Southeast Pacific<br>19. Antarctic Pacific |
| Affected regions | Shelf Sea / Coastal regions |





## Appendix D. File format for data submission

The following formatting rules are required for data submission:


1) All netCDF files should adhere to COARDS conventions (https://geos-chem.readthedocs.io/en/latest/geos-chem-shared-docs/supplemental-guides/coards-guide.html). These conventions suggest appropriate attributes (name, units, etc.) for the dimensions of a netCDF file (including time, lev, lat, and lon). One can use the script isCoards (https://github.com/geoschem/netcdf-scripts/blob/main/scripts/isCoards) provided by the GEOS-Chem support team to verify if your file is COARDS compliant (ignore errors related to _bnds variables or GCHP-specific errors). See for instructions: https://geos-chem.readthedocs.io/en/latest/geos-chem-shared-docs/supplemental-guides/netcdf-guide.html#determining-if-a-netcdf-file-is-coards-compliant.


2) netCDF files should be concatenated to reduce the number of files that need to be opened and make I/O more efficient. We have suggested using one file per output variable per month or year, requiring concatenation of individual files. Yearly files are requested for monthly-averaged data; monthly files are requested for daily-averaged data and the diurnal cycle averaged variables.


3) All netCDF files should be provided with global attributes that describe information about the institution, model experiment, model version and type, contact person, and additional comments.

4) All data should be stored in `float` precision, where possible (we don't want to store too much data, so `double` precision is unnecessary).

5) netCDF files should be deflated (compressed) to reduce their size. This can be done with the command: `nccopy - d1 in_file.nc out_file.nc`


6) A README file should be included detailing the data source, contents, etc. It would be helpful to include expected results in this README file, especially for input emissions files (i.e., global emissions totals for each species or source.

7) Where possible, include scripts used to produce the data. For example, if an input emissions file is being submitted, include the script where this emissions data is created.


8) Each variable should be provided in a separate file containing all time steps for a year (for monthly averages) or month (for daily or diurnal averages). The files should be constructed according to the convention:

```
mchgmap_<ModelName>_<ExperimentName>_<VariableName>_<VerticalCoordinateType>_<Perio
d>_<Frequency>.nc
```


where:

`<ModelName>` should be chosen so that the model's name, model version, and possibly the institution can be identified. No underscores (_) are allowed in `<ModelName>`, use hyphens (-) instead. Restrict `<ModelName>` to max 20 characters.

`<ExperimentName>` should correspond to the model experiment codes from Table F8-10.

`<VariableName>` should correspond to the output variable short names.


`<VerticalCoordinateType>` is Surface or ModelLevel from the output spreadsheet.

`<Period>` is the year and/or month of the data, in format `YYYY` (monthly data) or `YYYYMM` (daily data).

`<Frequency>` refers to `timeinvariant`, `monthly`, `daily`, or `diurnal`.

## Appendix E. Anthropogenic mercury emissions

### E1. Historical anthropogenic emissions


**Table E1. Overview of anthropogenic atmospheric Hg emission inventories.**

| Region | Dataset | Methodology | Year | Grid accuracy | Speciation | Uncertainty (Estimated | References |
|--------|---------|-------------|------|---------------|------------|------------------------|------------|



| | | | | | | year)* | |
|---|---|---|---|---|---|---|---|
| Global | AMAP | -Sector-specific emission factor<br>-Sectoral activity data by nation | 1990-2005 | 1°×1° (1990)<br>0.5°×0.5° (1995-2005) | Hg(0), Hg(II)$_g$, Hg(II)$_p$ | NA | Pacyna and Pacyna, 2002(2002); Pacyna et al. (2006a); AMAP/UNEP (2008) |
| Global | AMAP/GMA | -Emission factor based on mass-balance approach<br>-Sectoral activity data by nation | 2010, 2015 | 0.5°×0.5° (2010)<br>0.25°×0.25° (2015***) | Hg(0), Hg(II)$_g$, Hg(II)$_p$ | −10% to +27% (2015) | AMAP/UNEP, 2013; AMAP/UN Environment, 2019; Steenhuisen and Wilson (2015, 2019) |
| Global | EDGAR | -Emission factor based on mass-balance approach<br>-Sectoral activity data by nation | 1970-2012 | 0.1°×0.1° | Hg(0), Hg(II)$_g$, Hg(II)$_p$ | −26% to +33% (2012) | Muntean et al. (2014, 2018) |
| Global | STREETS** | -Emission factor based on transformed normal distribution function<br>-Sectoral activity data by nation or region | 2000-2015 | 1°×1° | Hg(0), Hg(II)$_g$, Hg(II)$_p$ | −20% to +44% (2015) | Streets et al. (2011, 2019a, b) |
| Global | WHET | Updated STREET dataset with country-specific estimates | 1990, 2000, 2010 | 1°×1° | Hg(0), Hg(II)$_g$, Hg(II)$_p$ | −33% to +60% (2010) | Zhang et al. (2016b) |
| North America | NA | Coupling US TRI dataset with Canadian dataset | 1990 | NA | Hg(0), Hg(II)$_g$, Hg(II)$_p$ | −12% to +22% (1990) | Walcek et al. (2003) |
| America | US NEI | Facility reporting | 1970-2017 (triennial) | NA | THg | NA | (2022a) |
| America | US TRI | Facility reporting | 1987-2021 (Annual) | NA | THg | NA | US EPA (2022b) |
| Canada | Canada NPRI | Facility reporting | 1994-2021 (Annual) | NA | THg | NA | NPRI (2022) |
| China | Abacas-EI-Hg | -Emission factor based on mass-balance approach<br>-Facility-level activity data and pollution control in key sectors | 1978-2017 | 0.25°×0.3125° | Hg(0), Hg(II)$_g$, Hg(II)$_p$ | −19% to +21% (2017) | Wu et al. (2016); Zhang et al. (2015a); Liu et al. (2019a) |
| EU | NA | -Sector-specific emission factor<br>-Sectoral activity data by nation | 1980-2000 | 0.5°×0.5° | Hg(0), Hg(II)$_g$, Hg(II)$_p$ | −26 to +33% (2000) | Pacyna et al. (2006b) |
| India | NA | -Sector-specific emission factor<br>-Sectoral activity data | 2010 | NA | THg | −80 to +300% | Chakraborty et al. (2013) |
| Thailand | NA | -Emission factor based on mass-balance approach<br>-Facility-level activity data | 2018 | 0.1°×0.1° | Hg(0), Hg(II)$_g$, Hg(II)$_p$ | −8% to +8% | Thao et al. (2021) |
| Korea | NA | -Emission factor based on mass-balance approach<br>-Facility-level activity data | 2007 | NA | Hg(0), Hg(II)$_g$, Hg(II)$_p$ | −20% to +20% | Kim et al. (2010) |
| Australia | NA | -Industrial sources: obtained from national emission inventory<br>-Other sources: Sector-specific emission factor and activity data | 2000-2019 | Distributed emissions at 0.25° resolution and point-source emissions at 0.1° resolution | Hg(0), Hg(II)$_g$, Hg(II)$_p$ | −35% to +35% | MacFarlane et al. (2022) |
| 70 countries (Available on Dec 21, 2022) | MIA | -Emission factor based on UNEP toolkit or local field experiment<br>-Sectoral-specific activity data | Vary by country | NA | THg | NA | UNEP (2022a) |
| 20 countries (Available on Dec 21, 2022) | NR | -Sectoral-specific emission factor and activity data | Vary by country | NA | THg | NA | UNEP (2022b) |
| 44 countries (Available on Dec 21, | PRTR | -Facility reporting | Vary by country | THg | NA | NA | UNECE (2022) |





| | | | | | |
|---|---|---|---|---|---|
| Chemicals | NA | NA | 26.4 | NA | 31.8 |
| Dental | NA | NA | 21.5 | NA | 16.1 |
| Transport | * | 17.4 | NA | * | NA |
| Vinyl-chloride monomer | NA | NA | NA | 58.3 | NA |

Note: NA– Not available; * - included in coal/oil/gas combustion 'other'

**Table E3. Spatial point source and proxy rasters applied in different anthropogenic Hg emission inventories. Minamata Convention target sectors are shown in bold.**

| Source sector | Emissions dataset | Spatial proxy raster | Sector emissions spatially distributed using spatial proxy (%) | Sector emissions spatially distributed using point source information (%) | References |
|---|---|---|---|---|---|
| **Coal-fired power plants** | EDGARv4 1970-2012 | N/A (Point sources: CARMA coal) | 0.0 | 100.0 | Janssens-Maenhout et al. (2019); Muntean et al. (2014, 2018) |
| | AMAP 1990 | Global population 1990 (CGEIC) | [2] | [2] | AMAP (1998) |
| | AMAP 1995-2005 | CARMA_Power plants | [3] | [3] | Pacyna et al. (2010, 2003) |
| | AMAP/UN Env. 2010, 2015 | CARMA_coal | 1.8[4] | 98.2[4, 5] | (Steenhuisen and Wilson, 2019; AMAP/ UN Environnement, 2019) |
| Gas fired power plants | EDGARv4 1970-2012 | Point sources: CARMA gas and in-house EDGAR population proxy | 2.6 | 97.4 | Janssens-Maenhout et al. (2019); Muntean et al. (2014, 2018) |
| | AMAP/UN Env. 2010, 2015 | CARMA oil/gas | 2.1[4] | 97.9[4, 5] | (Steenhuisen and Wilson, 2019; AMAP/ UN Environnement, 2019) |
| Oil burning power plants | EDGARv4 1970-2012 | Point sources: CARMA oil and in-house EDGAR population proxy | 0.5 | 99.5 | Janssens-Maenhout et al. (2019); Muntean et al. (2014, 2018) |
| | AMAP/UN Env. 2010, 2015 | CARMA oil/gas | 6.0[4] | 94.0[4, 5] | (Steenhuisen and Wilson, 2019; AMAP/ UN Environnement, 2019) |
| Autoproducers power plants | EDGARv4 1970-2012 | Point sources: CARMA autoproducers and in-house EDGAR population proxy | 0.5 | 99.5 | Janssens-Maenhout et al. (2019); Muntean et al. (2014, 2018) |

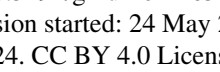


| Biomass power plants[1] | AMAP/UN Env. 2010, 2015 | CARMA generic industry | 100[4] | 0.0[4, 5] | (Steenhuisen and Wilson, 2019; AMAP/ UN Environnement, 2019) |
|---|---|---|---|---|---|
| Agricultural waste burning | EDGARv4 1970-2012 | Crops | 100 | 0.0 | Janssens-Maenhout et al. (2019); Muntean et al. (2014, 2018) |
| **Cement production** | EDGARv4 1970-2012 | N/A | 0.1 | 99.9 | Janssens-Maenhout et al. (2019); Muntean et al. (2014, 2018) |
| | AMAP 1990 | Global population 1990 (CGEIC) | [2] | [2] | AMAP (1998) |
| | AMAP 1995-2005 | Urban population | [3] | [3] | Pacyna et al. (2010, 2003) |
| | AMAP/UN Env. 2010, 2015 | CARMA generic industry | 0.7[4] | 99.3[4, 5] | (Steenhuisen and Wilson, 2019; AMAP/ UN Environnement, 2019) |
| Chlor-alkali production | EDGARv4 1970-2012 | N/A | 0.0 | 100.0 | Muntean et al. (2014, 2018) |
| | AMAP 1990 | Global population 1990 (CGEIC) | [2] | [2] | AMAP (1998) |
| | AMAP 1995-2005 | Urban population | [3] | [3] | AMAP/UNEP (2008); Pacyna et al. (2010) |
| | AMAP/UN Env. 2010, 2015 | CARMA generic industry | 0.5[4] | 99.5[4, 5] | (Steenhuisen and Wilson, 2019; AMAP/ UN Environnement, 2019) |
| **Non-ferrous metal production** | EDGARv4 1970-2012 (Zinc, Lead, Copper) | N/A | 0.0 | 100.0 | Janssens-Maenhout et al. (2019); Muntean et al. (2014, 2018) |
| | AMAP 1990 | Global population 1990 (CGEIC) | [2] | [2] | AMAP (1998) |
| | AMAP 1995-2005 | Urban population | [3] | [3] | Pacyna et al. (2010, 2003) |
| | AMAP/UN Env. 2010, 2015 | CARMA generic industry | 0.3[4] | 99.7[4, 5] | (Steenhuisen and Wilson, 2019; AMAP/ UN Environnement, 2019) |
| **Gold, ASGM** | EDGARv4 1970-2012 | Point sources: Global gold mines and in-house EDGAR population proxy | 2.7 | 97.3 | Muntean et al. (2014, 2018) |
| | AMAP/UN Env. 2010, 2015 | ASGM proxy | 100.0[4] | 0.0[4, 5] | (Steenhuisen and Wilson, 2019; AMAP/ UN Environnement, 2019) |



| | | | | | |
|---|---|---|---|---|---|
| | AMAP 1995-2005 | Gold deposits | [3] | [3] | Pacyna et al. (2010, 2003) |
| **Gold, large-scale** | EDGARv4 1970-2012 | Point sources: Global gold mines and in-house EDGAR population proxy) | 2.7 | 97.3 | Muntean et al. (2014, 2018) |
| | AMAP/UN Env. 2010, 2015 | Gold (mines) | 73.2[4] | 26.8[4, 5] | (Steenhuisen and Wilson, 2019; AMAP/ UN Environnement, 2019) |
| | AMAP 1990 | Global population 1990 (CGEIC) | [2] | [2] | AMAP (1998) |
| | AMAP 1995-2005 | Gold deposits | [3] | [3] | Pacyna et al. (2010, 2003) |
| **Mercury production** | EDGARv4 1970-2012 | Point sources: Global mercury mines and in-house EDGAR population proxy | 0.5 | 99.5 | Muntean et al. (2014, 2018) |
| | AMAP 1990 | Global population 1990 (CGEIC) | [2] | [2] | AMAP (1998) |
| | AMAP 1995-2005 | Urban population | [3] | [3] | Pacyna et al. (2010, 2003) |
| | AMAP/UN Env. 2010, 2015 | CARMA generic industry | 96.6[4] | 0.4[4, 5] | (Steenhuisen and Wilson, 2019; AMAP/ UN Environnement, 2019) |
| Oil refining | EDGARv4 1970-2012 | N/A | 0.5 | 99.5 | Janssens-Maenhout et al. (2019); Muntean et al. (2014, 2018) |
| | AMAP/UN Env. 2010, 2015 | CARMA generic industry | 9.9[4] | 90.1[4, 5] | Steenhuisen and Wilson, 2019 |
| | AMAP 1990 | Global population 1990 (CGEIC) | [2] | [2] | AMAP (1998) |
| Iron and steel production | AMAP 1995-2005 | Urban population | [3] | [3] | Pacyna et al. (2010, 2003) |
| | EDGARv4 1970-2012 (Crude steel production) | N/A | 0.4 | 99.6 | Janssens-Maenhout et al. (2019); Muntean et al. (2014, 2018) |
| | EDGARv4 1970-2012 (Sinter production) | N/A | 0.5 | 99.5 | Janssens-Maenhout et al. (2019); Muntean et al. (2014, 2018) |
| | EDGARv4 1970-2012 (Pig iron production) | N/A | 0.2 | 99.8 | Janssens-Maenhout et al. (2019); Muntean et al. (2014, 2018) |
| | AMAP/UN Env. | CARMA generic | 0.2[4] | 99.8[4, 5] | (Steenhuisen and Wilson, 2019; |





| | 2010, 2015 | industry | | | AMAP/ UN Environnement, 2019) |
|---|---|---|---|---|---|
| Secondary iron and steel production | AMAP/UN Env. 2010, 2015 | CARMA generic industry | 92.6[4] | 0.4[4,5] | (Steenhuisen and Wilson, 2019; AMAP/ UN Environnement, 2019) |
| Vinyl-chloride monomer production | AMAP/UN Env. 2010, 2015 | CARMA generic industry | 100.0[4] | 0.0[4,5] | (Steenhuisen and Wilson, 2019; AMAP/ UN Environnement, 2019) |
| Aluminium production | AMAP/UN Env. 2010, 2015 | CARMA generic industry | 4.3[4] | 95.7[4,5] | (Steenhuisen and Wilson, 2019; AMAP/ UN Environnement, 2019) |
| Cremation | AMAP/UN Env. 2010, 2015 | Population (2015) | 96.9[4] | 3.1[4] | Janssens-Maenhout et al. (2019); Muntean et al. (2014, 2018) |
| | AMAP 1990 | Global population 1990 (CGEIC) | [2] | [2] | AMAP (1998) |
| | AMAP 1995-2005 | Population | [3] | [3] | Pacyna et al. (2010, 2003) |
| Stationary combustion (domestic or residential) | EDGARv4 1970-2012 | In-house EDGAR population proxy | 100.0 | 0.0 | Janssens-Maenhout et al. (2019); Muntean et al. (2014, 2018) |
| | AMAP 1990 | Global population 1990 (CGEIC) | [2] | [2] | AMAP (1998) |
| | AMAP 1995-2005 | Population | [3] | [3] | Pacyna et al. (2010, 2003) |
| | AMAP/UN Env. 2010, 2015 | CARMA generic industry | 100.0[4] | 0.0[4,5] | (Steenhuisen and Wilson, 2019; AMAP/ UN Environnement, 2019) |
| Stationary combustion for industrial | EDGARv4 1970-2012 | Point sources and in-house EDGAR population proxy | 55.9 | 44.1 | Janssens-Maenhout et al. (2019); Muntean et al. (2014, 2018) |
| | AMAP 1990 | Global population 1990 (CGEIC) | [2] | [2] | AMAP (1998) |
| | AMAP 1995-2005 | Population | [3] | [3] | Pacyna et al. (2010, 2003) |
| | AMAP/UN Env. 2010, 2015 | CARMA generic industry | 100.0[4] | 0.0[4,5] | (Steenhuisen and Wilson, 2019; AMAP/ UN Environnement, 2019) |
| Fuel production: transformation industry | EDGARv4 1970-2012 | Point sources and in-house EDGAR population proxy | 10.9 | 89.1 | Janssens-Maenhout et al. (2019); Muntean et al. (2014, 2018) |
| Road transport | EDGARv4 1970-2012 | Line sources: in-house EDGAR proxy based on | 100.0 | 0.0 | Janssens-Maenhout et al. (2019); Muntean et al. (2014, 2018) |





| | | OpenStreetMap | | | |
|---|---|---|---|---|---|
| Shipping | EDGARv4 1970-2012 | Line sources: in-house EDGAR proxy based on Long Range Identification and Tracking (LRIT) and Wang et al. (2007) | 100.0 | 0.0 | Janssens-Maenhout et al. (2019); Muntean et al. (2014, 2018) |
| Municipal solid waste incineration | AMAP/UN Env. 2010, 2015 | Urban population 2015 | 8.6[4] | 91.4[4,5] | (Steenhuisen and Wilson, 2019; AMAP/ UN Environnement, 2019) |
| | AMAP 1990 | Global population 1990 (CGEIC) | [2] | [2] | AMAP (1998) |
| | AMAP 1995-2005 | Population | [3] | [3] | Pacyna et al. (2010, 2003) |
| | EDGARv4 1970-2012 | Point sources (in-house EDGAR proxy based on EPRTR) and in-house EDGAR population proxy | 94.5 | 5.5 | Janssens-Maenhout et al. (2019); Muntean et al. (2014, 2018) |
| Hazardous wastes incineration | AMAP/UN Env. 2010, 2015 | Population (2015) | 99.8[4] | 0.2[4,5] | (Steenhuisen and Wilson, 2019; AMAP/ UN Environnement, 2019) |
| | AMAP 1995-2005 | Population | [3] | [3] | Pacyna et al. (2010, 2003) |

Notes:

[1] In EDGARv4 1970-2012, the mercury emissions from biomass combustion in power generation are distributed using the proxies of coal, gas, oil and autoproducers[7].

[2] AMAP_1990 was distributed using the 1990 global population and a limited number of large emission point sources.

[3] AMAP 1995-2005 was distributed using an increasing number of point sources over time. The remaining (diffuse) emissions were distributed over several different proxies. A percentage of point source and spatial proxy distribution would only be valid for a specific year and can therefore not be given for the 1995-2005 period.

[4] Percentages for 2015 emissions distribution.

[5] AMAP/UN Environment utilizes point source emission information derived from national emissions inventories as well as from additional (open) sources as described in Steenhuisen and Wilson (2019, 2015).

[6] Combustion emissions from fuel use during the manufacture of secondary and tertiary products from solid fuels including production of charcoal (IPCC 2006 classification 1.A.1.c).

[7] Activities which generate electricity/heat wholly or partly for their own use, as an activity that supports their primary activity (EDGAR 1.A.1.a).

When all emissions (100%) of a given sector are distributed using point source information, consequently no proxy raster has been used. Hence the proxy raster is listed as N/A.





Minamata target sectors (in bold)
- coal-fired power plants
- coal-fired industrial boilers
•   nonferrous metal smelting (zinc, lead, copper and large scale-gold production)
- waste incineration
- cement clinker production
- ASGM
- Mercury (primary) production


**Table E4. Speciation profile used in AMAP/GMA, WHET and EDGAR (%) global Hg emissions inventories.**

| Sectors/Fuel | AMAP, 2005 (AMAP/UNEP, 2008) | | | WHET (Zhang et al., 2016b) | | | EDGAR (Muntean et al., 2018) | | | AMAP/GMA, 2015 (Steenhuisen and Wilson, 2019) | | |
|---|---|---|---|---|---|---|---|---|---|---|---|---|
| | $Hg(0)$ | $Hg(II)_g$ | $Hg(II)_p$ | $Hg(0)$ | $Hg(II)_g$ | $Hg(II)_p$ | $Hg(0)$ | $Hg(II)_g$ | $Hg(II)_p$ | $Hg(0)$ | $Hg(II)_g$ | $Hg(II)_p$ |
| **Solid biomass** | **50** | **40** | **10** | **80** | **15** | **5** | **74.4** | **4.8** | **20.8** | **75** | **5** | **20** |
| **Liquid and gas** | **50** | **40** | **10** | **80** | **15** | **5** | **50** | **40** | **10** | **50** | **40** | **10** |
| **Coal, CFPP, bituminous** | **50** | **40** | **10** | **80** | **15** | **5** | **NA** | **NA** | **NA** | **80** | **5** | **15** |
| -ESP+wetFGD | NA | NA | NA | NA | NA | NA | 84.99 | 14.76 | 0.25 | NA | NA | NA |
| -FF+wetFGD | NA | NA | NA | NA | NA | NA | 78 | 21.1 | 0.9 | NA | NA | NA |
| -FF+dryFGD | NA | NA | NA | NA | NA | NA | 78 | 21.1 | 0.9 | NA | NA | NA |
| -ESP | NA | NA | NA | NA | NA | NA | 62.5 | 37 | 0.5 | NA | NA | NA |
| -FF | NA | NA | NA | NA | NA | NA | 50 | 49.5 | 0.5 | NA | NA | NA |
| -SNCR+ESP | NA | NA | NA | NA | NA | NA | 64.4 | 34 | 1.6 | NA | NA | NA |
| -SCR+ESP+wetFGD | NA | NA | NA | NA | NA | NA | 71 | 28.9 | 0.1 | NA | NA | NA |
| -no EoP | NA | NA | NA | NA | NA | NA | 56 | 34 | 10 | NA | NA | NA |
| **Coal, CFPP, lignite** | **50** | **40** | **10** | **80** | **15** | **5** | **NA** | **NA** | **NA** | **80** | **5** | **15** |
| -ESP+ wetFGD | NA | NA | NA | NA | NA | NA | 84.99 | 14.76 | 0.25 | NA | NA | NA |
| -ESP+ dryFGD | NA | NA | NA | NA | NA | NA | 84.99 | 14.76 | 0.25 | NA | NA | NA |
| -FF+ESP | NA | NA | NA | NA | NA | NA | 50 | 49.5 | 0.5 | NA | NA | NA |
| -ESP | NA | NA | NA | NA | NA | NA | 62.5 | 37 | 0.5 | NA | NA | NA |
| -SCR+ESP+wetFGD | NA | NA | NA | NA | NA | NA | 71 | 28.9 | 0.1 | NA | NA | NA |
| -no EoP | NA | NA | NA | NA | NA | NA | 56 | 34 | 10 | NA | NA | NA |
| **Coal, CFPP, sub-bituminous** | **50** | **40** | **10** | **80** | **15** | **5** | **NA** | **NA** | **NA** | **80** | **5** | **15** |
| -ESP+wetFGD | NA | NA | NA | NA | NA | NA | 84.99 | 14.76 | 0.25 | NA | NA | NA |





| | | | | | | | | | | | | |
|---|---|---|---|---|---|---|---|---|---|---|---|---|
| -FF+dryFGD | NA | NA | NA | NA | NA | NA | 78 | 21.1 | 0.9 | NA | NA | NA |
| -ESP+dryFGD | NA | NA | NA | NA | NA | NA | 84.99 | 14.76 | 0.25 | NA | NA | NA |
| -FF | NA | NA | NA | NA | NA | NA | 50 | 49.5 | 0.5 | NA | NA | NA |
| -ESP | NA | NA | NA | NA | NA | NA | 62.5 | 37 | 0.5 | NA | NA | NA |
| -SCR+ESP+wetFGD | NA | NA | NA | NA | NA | NA | 71 | 28.9 | 0.1 | NA | NA | NA |
| -no EoP | NA | NA | NA | NA | NA | NA | 56 | 34 | 10 | NA | NA | NA |
| **Combustion in residential** | **50** | **40** | **10** | **80** | **15** | **5** | **NA** | **NA** | **NA** | **80** | **15** | **5** |
| -coal | NA | NA | NA | NA | NA | NA | 83.1 | 6.9 | 10 | NA | NA | NA |
| -solid biomass | NA | NA | NA | NA | NA | NA | 74.4 | 4.8 | 20.8 | NA | NA | NA |
| -liquid/gas | NA | NA | NA | NA | NA | NA | 50 | 40 | 10 | NA | NA | NA |
| **Cement production** | **80** | **15** | **5** | **80** | **15** | **5** | **67.6** | **28.4** | **4** | **35** | **65** | **0** |
| **Iron and steel** | **80** | **15** | **5** | **80** | **15** | **5** | **32.1** | **62.9** | **5** | **35** | **65** | **0** |
| **Copper smelting** | **80** | **15** | **5** | **80** | **15** | **5** | **44.5** | **50.5** | **5** | **45** | **50** | **5** |
| **Lead smelting** | **80** | **15** | **5** | **80** | **15** | **5** | **49.7** | **45.3** | **5** | **45** | **50** | **5** |
| **Zinc smelting** | **80** | **15** | **5** | **80** | **15** | **5** | **37.7** | **57.4** | **5** | **45** | **50** | **5** |
| **Waste incineration:** | **20** | **60** | **20** | **20** | **60** | **20** | **NA** | **NA** | **NA** | **95** | **0** | **5** |
| -agricultural | NA | NA | NA | NA | NA | NA | 96 | 0 | 4 | NA | NA | NA |
| -municipal | NA | NA | NA | NA | NA | NA | 17.7 | 82 | 0.3 | NA | NA | NA |
| -industrial | NA | NA | NA | NA | NA | NA | 2.8 | 97.2 | 0 | NA | NA | NA |
| **Mercury production** | **80** | **20** | **0** | **80** | **20** | **0** | **80** | **20** | **0** | **80** | **15** | **0** |
| **Artisanal and small-scale gold production** | **100** | **0** | **0** | **100** | **0** | **0** | **100** | **0** | **0** | **100** | **0** | **0** |
| **Chlor-alkali industry** | **70** | **30** | **0** | **70** | **30** | **0** | **70** | **30** | **0** | **100** | **0** | **0** |

NA: Not available



## E2. Future anthropogenic emissions

**Table E5. Abbreviations used in Future Emissions Tables.**

| | |
|---|---|
| ASGM | Artisanal and small-scale mining |
| BAS | Baseline: business as usual for energy/climate policy |
| BAU | Business as usual: continuation of current air pollution and climate policy |
| CLE | Current legislation |
| CLIM | Climate mitigation scenario for energy/activities |
| EC | Emission control |
| ETP | Energy Technology Perspectives |
| GMA'18 | Global Mercury Assessment 2018 (AMAP/ UN Environnement, 2019)2024-05-23 16:50:00 |
| GMA'13 | Global Mercury Assessment 2013 (AMAP/UNEP, 2013) |
| GHG | Greenhouse gases |
| MFR | Maximum (technologically) feasible reduction |
| MINA | Minamata policy scenario |
| NAP | National Action Plan |
| NFME | Non-ferrous metals |
| NP | No Policy |
| PM | Particulate matter |
| RED/BAN | Reduction or ban of Hg use in activity |
| SRES | Special Report on Emission Scenarios |
| VCM | Vinyl chloride monomer production |
| WEO | World Energy Outlook |

**Table E6. Scenario design for future Hg emissions: considerations and resulting scenarios.**

| | Considerations | Scenario Options |
|---|---|---|
| **Energy projections** | o Energy scenarios<br>o Represent future development of energy systems<br>o Take into account energy and climate policy<br>o Usually also include projections of (energy-intensive) industries | o **BAS**: Baseline scenario assuming current energy & climate policy into the future<br>o **CLIM**: More ambitious climate scenarios, e.g., taking into consideration net-zero pathways |
| **Air quality policy** | o Assumptions on PM, $SO_2$ and NOx control further influence future Hg emission levels and speciation through co-benefits<br>o Assumptions on Hg-specific policy, e.g., implementation of the Minamata Convention | o **NOC**: No policy scenario provides a baseline of emission without any assumed interventions<br>o **CLE**: Current legislation scenario reflecting current Hg, PM, $SO_2$ and NOx policy<br>o **MINA**: Minamata policy scenario; reflects full adoption of Minamata Hg reduction targets in relevant sectors, including National Action Plans (NAPs) for ASGM<br>o **coMFR**: Maximum (technologically) feasible reduction scenario; reflects full application of the most efficient air pollution control |



| | | |
|---|---|---|
| | | methods, regardless of cost |
| | | o **HgMFR**: Maximum (technologically) feasible reduction of Hg using Hg-specific technologies, regardless of cost |

**Table E7. Data sources and other considerations for building future Hg emission scenarios.**

| | Considerations | Options |
|---|---|---|
| **Base year emission inventory** | o Gridding of emissions (i.e., spatial distribution)<br>o Definition of sectors must be consistent with available activity projections (e.g., separation of emissions from fuels and other input materials in industrial processes)<br>o Calibration to existing inventories and statistics | o EDGAR4vtox2 (This inventory will be used for future projections by TAPS model)<br>o AMAP/UN Environment 2013 (GAINS v3 (Rafaj et al., 2013) is calibrated to this inventory)<br>o GMA'18 (UNEP, 2019) (GAINS v4 is calibrated to this inventory)<br>o STREETS |
| **Source of activity projections** | o Global or regional consistency<br>o Granularity<br>o Time steps<br>o Sectoral coverage, e.g., availability of projections for Hg-specific sectors like ASGM, generated waste, VCM production, etc.<br>o Timing of climate policies, projections of population growth, poverty, etc. | o IEA World Energy Outlook<br>o MIT's EPPA model<br>o Other integrated assessment models<br>o Regional energy models<br>o Own projections |
| **Choice of projected sectors** | o Sector alignment with baseline emissions inventory | o Maximum number of available sectors<br>o Focus on sectors explicitly covered by the Minamata Convention (see **Table E10**). |
| **Representation of air pollutant control technologies** | o Implicit – e.g., represented as variable emission factors for a selected activity that might lower over time, as technology or efficiency improvements are made | o Global: e.g., TAPS model (not yet implemented) |
| | o Explicit – e.g., represented as a distinct choice of control options per modelling region, which leads to a lowering of the unabated emission factor of an activity to a final abated emission factor | o Global: GAINS model (mirrors the approach in GMA'18)<br>o Regional: CAME model (China) |
| **Time steps** | o Present only "end point" of a scenario or equally spaced time steps? | o Yearly steps up to 2050<br>o 5-year steps up to 2050 (e.g., GAINS model)<br>o Only present endpoint (e.g., Pacyna et al. (2016)) |






**Table E8. Global projections of mercury emissions in literature.**

BAS - Baseline energy/activity scenario; CLIM - Climate mitigation scenario for energy/activities; CLE - Current legislation for Hg pollution control; MFR - Maximum feasible Hg reduction; SRES - Special Report on Emission Scenarios; WEO - World Energy Outlook; ETP - Energy Technology Perspectives

| Scale | Reference | Base year | Projection year | Inventory used | Policy scenarios – Activities/Energy/ Hg control | | Speciation | Emissions gridded? |
|---|---|---|---|---|---|---|---|---|
| **Global** | Streets et al. (2009) | 2006 | 2050 | STREETS | IPCC SRES scenarios: (1) A1B (2) A2 (3) B1 (4) B2 | | Hg(0) Hg(II)$_g$ Hg(II)$_p$ | Yes |
| **Global** | Rafaj et al. (2013) | 2010 | 2050 | Own / calibrated to UNEP/ AMAP 2013 | IEA WEO 2012, supplemented with own projections for gold, caustic soda production, other Hg emissions | | Hg(0) Hg(II)$_g$ Hg(II)$_p$ | No |
| | | | | | (1) BAS | (a) CLE | | |
| | | | | | | (b) MFR | | |
| | | | | | (2) CLIM | (a) CLE | | |
| | | | | | | (b) MFR | | |
| **Global** | Pacyna et al. (2010) | 2005 | 2020 | Own | (1) BAS: Business as usual (2) Policy scenario (Hg focus) (3) Deep Green scenario (Hg focus) | | Hg(0) Hg(II)$_g$ Hg(II)$_p$ | Yes |
| **Global** | Lei et al. (2014) | 2000 | 2050 | STREETS, Pacyna et al. 2006 | IPCC SRES scenarios: (1) A1FI (2) A1B (3) B1 | | | Yes |
| **Global** | Pacyna et al. (2016) | 2010 | 2035 | AMAP/UNEP 2013 | Own, using a variety of data sources for different activities; (1) CLE (2) New policies, 450ppm $CO_2$ (3) MFR | | Hg(0) Hg(II)$_g$ Hg(II)$_p$ | Yes |


**Table E9. Regional projections of mercury emissions in literature.**

APCD - Air Pollution Control Device; BAS - Baseline energy/activity scenario; BAU - Business as usual scenario; CCS – Carbon Capture & Storage; CLIM - Climate mitigation scenario for energy/activities; CLE - Current legislation for Hg pollution control; HgCD – Hg Control Device; MFR - Maximum feasible Hg reduction; SRES - Special Report on Emission Scenarios; WEO - IEA World Energy Outlook


| Scale | Reference | Base year | Projection year | Inventory used | Policy scenarios – Activities/Energy Hg control | Speciation | Emissions gridded? |
|---|---|---|---|---|---|---|---|
| **Europe** | Pacyna et al. (2006a) | 2000 | 2020 | Own | Own projections (1) BAU + Policy scenario (2) BAU + Deep Green scenario | Hg(0) Hg(II)$_g$ Hg(II)$_p$ | Yes |



| | | | | | | (Hg focus) | | |
|---|---|---|---|---|---|---|---|---|
| **Poland** | Glodek et al. (2010) | 2005 | 2020 | Hlawczika et al. (2006) as reported to EMEP | Own projections<br>(1) BAU + Extended controls<br>(2) BAU + MFR | | THg | No |
| **India** | Chakraborty et al. (2013) | 2001 | 2020 | Own | Own projections<br>(1) BAU + Phase-out of intentional Hg applications | | THg | No |
| **Europe** | Rafaj et al. (2014) | 2010 | 2050 | Own (GAINS) | EU ReMix renewable energy model<br>(1) BAS + CLE<br>(2) max. renewable power + CLE | | Hg(0)<br>Hg(II)$_g$<br>Hg(II)$_p$ | Yes |
| **China, India** | Giang et al. (2015) | 2010 | 2050 | GMA'13 (UNEP, 2013) | IPCC SRES scenarios: | | | Yes |
| | | | | | A1B | Minamata Flexible<br>Minamata Strict | Hg(0)<br>Hg(II)$_g$<br>Hg(II)$_p$ | |
| | | | | | B1 | Minamata Flexible<br>Minamata Strict | | |
| **China** | Zhao et al. (2015) | 2012 | 2030 | Own | Own projections + WEO2012<br>(1) No future policy<br>(2) New policy (based on WEO2012)<br>(3) New Policy + APCDs | | Hg(0)<br>Hg(II)$_g$<br>Hg(II)$_p$ | Yes |
| **China** | Ancora et al. (2016) | 2010 | 2030 | Own | Own projections<br>(1) BAU<br>(2) Minamata Low<br>(3) Minamata High<br>(4) Minamata Medium | | | No |
| **China** | Wu et al. (2018a) | 2015 | 2030 | Own | Own<br>(1) CLE<br>(2) Climate Policy + Minamata | Technical paths to ELVs of 15 / 5 / 2 µg/m$^3$ | Hg(0)<br>Hg(II)$_g$<br>Hg(II)$_p$ | No |
| **China** | Wu et al. (2018c) | 2015 | 2030 | Own | Own<br>(1) CLE<br>(2) Climate Policy + Minamata | (a) BAU<br>(b) Extended emission control<br>(c) Accelerated emission control | Hg(0)<br>Hg(II)$_g$<br>Hg(II)$_p$ | Yes |
| **China** | Mulvaney et al. (2020) | 2012 | 2030 | EDGAR.v4 tox2 | China Regional Energy Model (C-REM) | | Hg(0)<br>Hg(II)$_g$<br>Hg(II)$_p$ | Yes |





| | | | | | (1) No climate or Hg policy | Minamata APCDs | | |
| | | | | | (2)-(4) Climate scenarios | | | |
| **China** | Guo et al. (2022) | 2020 | 2060 | Own | (a) coal washing (b) APCD (c) operating hours (d) plant lifetime | | THg | Yes |
| **8 Copper-producing countries** | Yamamoto et al. (2023) | 2020 | 2050 | Own | (1)-(3) Varying copper stock levels | (a) – (d) Varying Cu recycling rates and HgCD efficiency | THg | No |
| **India** | Vishwanathan et al. (2023) | 2020(?) | 2070 | Own | (1) BAS (2) CLIM + CCS (3) CLIM + renewables | (a) No HgCD (b) cheapest HgCD (c) best HgCD | THg | No |



**Table E10. Anthropogenic emission sectors covered by different future global emissions scenario modelling efforts in literature.**

ASGM - Artisanal and small-scale gold mining; EC - Emission control; Red/BAN - Reduction or ban; VCM - Vinyl chloride monomer production

| | Base year | Projection year | Combustion – Coal: Power (EC) | Coal: Industrial (EC) | Coal: Residential | Oil: Industrial | Biomass: Domestic | Biomass: Residential | Industry – Pig iron | Steel | Cement (EC) | Chlor-alkali (Red/BAN) | VCM (Red/BAN) | Smelting: Cu (EC) | Smelting: Pb (EC) | Smelting: Zn (EC) | Smelting: Hg (Red/BAN) | Al | Large-scale Au (EC) | Oil refining | Other (e.g., glass, brick) | ASGM (Red/BAN) | Waste – Hg containing products: Incineration (EC) | Disposal (Red/BAN) | General waste (EC) | Cremation | Transport |
|---|---|---|---|---|---|---|---|---|---|---|---|---|---|---|---|---|---|---|---|---|---|---|---|---|---|---|---|
| **Minamata target →** | | | | | | | | | | | | | | | | | | | | | | | | | | | |
| Streets et al. (2009) | 2006 | 2050 | x | | x | x | x | | | x | x | | | x | x | x | x | | | | | x | | | | | |
| Pacyna et al. (2010) | 2005 | 2020 | | | | | | | x | x | x | x | | x | x | x | | | | | coke | | x | | | | |
| Rafaj et al. (2013) | 2010 | 2050 | x | x | x | x | x | x | x | x | x | x | x | | x | | x | x | x | x | x | x | x | | x | x | x |
| Lei et al. (2014) | 2010 | 2050 | x | x | x | x | | | | x | x | x | x | | x | x | | | | | | | x | | | | |
| Pacyna et al. (2016) | 2010 | 2035 | x | x | x | x | | | | x | | x | x | x | x | x | x | x | x | | | x | x | x | | x | |

Options for future Hg emission scenario modelling:

| | Base year | Projection year | Power | Industrial | Residential | Industrial | Domestic | Residential | Pig iron | Steel | Cement | Chlor-alkali | VCM | Cu | Pb | Zn | Hg | Al | Large-scale Au | Oil refining | Other | ASGM | Incineration | Disposal | General waste | Cremation | Transport |
|---|---|---|---|---|---|---|---|---|---|---|---|---|---|---|---|---|---|---|---|---|---|---|---|---|---|---|---|
| Option 1: all available | 2015 | 2050 | x | x | x | x | x | x | x | x | x | x | x | x | x | x | x | x | x | x | x | x | x | x | x | x | x |
| Option 2: Minamata | 2015 | 2050 | x | x | | | | | | | x | x | x | x | x | x | x | | x | | | x | x | x | x | | |




**Table E11. Proposed setup for GAINS v4 and TAPS future Hg emission scenarios.**

| | GAINS v4 | TAPS |
|---|---|---|
| **Model name** | Greenhouse Gas and Air Pollution Interaction and Synergies model v4 | Tool for Air Pollution Scenarios |
| **Associated research group** | Pollution Management Group, International Institute of Applied Systems Analysis (IIASA) | Selin Group, Massachusetts Institute of Technology (MIT) |
| **Base year – projection year (time steps)** | 2015 – 2050 (5-year steps) | 2015 – 2050 (5-year steps) |
| **Emission inventory** | AMAP/UN-Environment 2018 | EDGAR |
| **Activity projections** | Energy projections: Wide range of climate and air quality scenarios such as WEO2022<br><br>Hg-specific: Conservative activity projections for caustic soda production, VCM production and ASGM | EPPA v7 model |
| **Sectoral coverage** | Large sectoral coverage, especially of the Convention-relevant sectors (see Table E10) | EDGAR sectors (with a few aggregations) |
| **Regional resolution** | 183 GAINS regions, presented in 17 aggregated WEO2022 regions | 18 EPPA regions |
| **Hg policy Scenarios** | Current legislation (**CLE**):<br>▪ Reflects current policy for mercury<br>▪ Reflects current policy PM, SO2 and greenhouse gases<br>Minamata policy scenario (**MINA**):<br>▪ Reflects full adoption of Minamata Hg reduction targets in relevant sectors<br>Maximum (technologically) feasible reduction scenario (**MFR**):<br>▪ Reflects minimum anthropogenic emissions, based on full application of the most efficient technologies, regardless of cost constraints. | Combination of EPPA v7 scenarios (e.g., Paris Forever, Paris 2° C, Paris 1.5°C, Accelerated Actions) with Hg emission scenarios |
| **Representation of Hg control options** | Explicit – Hg-specific control technologies have been updated between 2021 and 2022, allowing for combining co-benefits from PM and $SO_2$ abatement with Hg-specific removal technologies. | Implicit – e.g., represented as variable emission factors for a selected activity that might lower over time, as technology or efficiency improvements are made |
| **Gridding** | Under implementation – resolution: 0.5°×0.5° globally, 0.1°×0.1° for some urban areas | 0.1°×0.1° |






**Table E12. Proposed future anthropogenic Hg emission scenarios.**

| Activity projections: Energy, population, climate policy, industry, Hg-specific activities | # of regions | Air pollutant control policy: PM/SO$_2$ control | Mercury-specific policy: Hg control | Scenario ID |
|---|---|---|---|---|
| Baseline (**BAS**) Scenario: current policy | 15 world regions | NOC | NOC | 00_BAS_NOC |
| | | CLE | CLE | 01_BAS_CLE |
| | | | CLE + Minamata NAPs | 02_BAS_MINA |
| | | coMFR | CLE | 03_BAS_coMFR |
| | | | HgMFR | 04_BAS_HgMFR |
| Announced Policies (**CLIM1**) Scenario: including announced climate targets | 15 world regions | CLE | CLE | 05_CLIM1_CLE |
| Net-Zero Emissions (**CLIM2**) Scenario | Global | CLE | CLE | 06_CLIM2_CLE |
| | | coMFR | HgMFR | 07_CLIM2_HgMFR |

**Additional details on the GAINS model**

IIASA's Greenhouse Gas and Air Pollution Interactions and Synergies (GAINS) model has been used to evaluate air
pollution and greenhouse gas policies for several decades (Amann et al., 2011). As a multi-pollutant model, GAINS includes
emission factors and control strategies for SO$_2$, PM, NH$_3$, N$_2$O, NOx, CH$_4$, CO$_2$, CO, VOC, F-gases and Hg from
anthropogenic emission sources. Currently, 55 emission sectors are combined with 15 activity types are associated with Hg
emissions factors, leading to a nuanced description of Hg emissions. The model relies on exogenous activity inputs from
integrated assessment models, extended by additional mercury-specific activity inputs. It contains a bottom-up inventory of
pollution control options for each pollutant, on the basis of its 182 regions on the global level, which for many regions are
coordinated in consultation with local experts. Mercury co-benefits from existing PM and SO2 control are calculated
automatically for all applicable combustion and industry sectors and additional mercury control strategies can be added,
based on current and future policy. Results are presented in 5-year steps up to 2050.

The following paragraphs describe the most current Hg-GAINS scenarios, for which publication is planned in 2023.

***Inventory used and activity projections:*** The World Energy Outlook scenarios for 2022 are currently implemented in
GAINS and have been used as activity for the most recent Hg scenarios. Data on ASGM, caustic soda production, Hg
mining as well as VCM were derived from the Global Mercury Assessment, USGS mineral yearbooks and similar sources.
ASGM and VCM production are kept constant until 2050, representing a conservative estimate which can be used to
illustrate policy changes clearly. The ratio of bodies cremated is kept constant until 2050, but cremation activities are driven
by population growth.

***Emission factors:*** Unabated mercury emission factors in GAINS are derived from different literature sources and the Global
Mercury assessment, aggregated (e.g., non-ferrous metal sector) and disaggregated (e.g., iron and steel sectors) as
appropriate to the model resolution. Emission factors for the combustion sectors are documented in Rafaj et al. (2013),
others have been developed based on information in the GMA'18 (Global Mercury Assessment 2018, 2020). Unabated



emission factors are constant through all model years. Own emission factors were developed for the waste sector. GAINSv4 was calibrated to the 2015 emissions in GMA'18 (highest transparency), all changes are documented.

***Air pollution control policies in GAINS:*** The global mercury scenarios make use of the most up-to-date PM and SO$_2$ control policies from the recently implemented WEO2022 scenarios, utilizing both the Current Legislation (CLE) and the cost-optimal Maximum Feasible Reduction (MFR) scenarios for PM and SO$_2$ control. Hg co-benefits for the PM/SO$_2$ CLE and MFR are calculated within the model for all types of combustion and industry sectors. Where relevant legislation exists, for example in the EU, Hg-specific control options are then added to the co-benefit control strategy. Where a Hg-specific control

strategy overlaps with existing PM/SO$_2$, a hierarchy of controls is implemented whereby the most efficient controls are assumed to be implemented in the most technologically advanced installations. Additional Hg-specific controls include e.g. the EU directive on Large Combustion Plants concerning waste incineration, phase-out of caustic soda production using mercury cells, and all Minamata National Action Plans for ASGM reduction published until December 2022.

Results from the scenarios will be reported following the regional resolution of the IEA World Energy Outlook 2022, which uses 15 world regions: (1) North America excl. United states, (2) United States, (3) Central and South America excl. Brazil, (4) Brazil, (5) Europe excl. European Union, (6) European Union, (7) Africa excl. South Africa, (8) South Africa, (9) Middle East, (10) Eurasia, (11) Asia Pacific excl. China, India, Japan, Southeast Asia, (12) China, (13) India, (14) Japan, (15) Southeast Asia.

**Appendix F. Models, drivers, simulations, and outputs**

**Table F1. 3D global and regional atmospheric Hg models.**

| Name | Model type | Horizontal resolution | Vertical resolution | Gas-phase chemistry | Aqueous chemistry | Reference |
|---|---|---|---|---|---|---|
| **Global models** | | | | | | |
| GISS-CTM | Atmospheric | 8° × 10° | 9 levels, top 10 hPa | O$_3$ | Yes | Shia et al. (1999) |
| CAM-Chem | Multi-media | 0.9° × 1.25° | 32 levels, top 2 hPa | O$_3$, OH | Yes | Lei et al. (2013); Zhang and Zhang (2022) |
| ECHMERIT | Atmospheric | 2.8° × 2.8° | 19 levels, top 10 hPa | O$_3$, OH | Yes | Jung et al. (2009) |
| GEM-MACH-Hg | Atmospheric (global and regional) | 0.5° × 0.5° | 58 levels, top 7 hPa | OH, bromine species | No | Dastoor et al. (2021); Zhou et al. (2021) |
| GEOS-Chem | Atmospheric | 2° × 2.5° | 47 levels, top 0.01 hPa | Br, OH, O$_3$, Cl, NO$_2$, CO | Yes | Shah et al. (2021) |
| GLEMOS | Atmospheric | 1° × 1° | 20 levels, top 10 hPa | O$_3$, OH, Br | Yes | Travnikov and Ilyin (2009) |
| WACCM | Atmospheric | 1.9° × 2.5° | 88 levels, top 6 ×10$^{-6}$ hPa (140 km) | Cl, OH, Br, O$_3$ | No | Saiz-Lopez et al. (2022) |





| Regional models | | | | | | |
|---|---|---|---|---|---|---|
| CMAQ-newHg-Br | Atmospheric | 12 × 12 km | 35 vertical layers | O₃, OH, H₂O₂, Cl₂, Cl, Br | Yes | Ye et al. (2018a) |
| WRF-Chem | Atmospheric | 100 × 100 km | 72 levels up to 50 hPa | Br, Cl, OH | No | Gencarelli et al. (2014); Ahmed et al. (2023) |
| GEOS-Chem nested | Atmospheric | 0.5 × 0.625° or 0.25 × 0.3125° | 47 levels, top 0.01 hPa | Br, OH, O₃ | Yes | Wang et al. (2014a); Zhang et al. (2012) |
| WRF-GC-Hg | Atmospheric | 50 × 50 km or 25 × 25 km | 47 levels, top 0.01 hPa | Br, OH, O3 | Yes | Xu et al. (2022) |

**Table F2. 3D global and regional ocean Hg Models.**

| Name | Model type | Ocean | Biogeo-chemistry | Horizontal resolution | Vertical resolution | Hg species | Chemistry scheme | Reference |
|---|---|---|---|---|---|---|---|---|
| FATE-Hg | Global | FATE | n/a | 1.0° | 55 | Hg(0), Hg(II), MMHg, DMHg | Kawai et al. (2020) | Kawai et al. (2020) |
| OGSTM-BFM-Hg | Regional | NEMO | OGSTM-BFM | 1/16° | 70 | Hg(0), Hg(II), MMHg, DMHg, 4 phytoplankton MMHg, 4 zooplankton MMHg | Rosati et al. (2022) | Rosati et al. (2022) |
| MITgcm | Global | MITgcm | Darwin | LLC90 grid with a nominal resolution of 1°×1° | 50 levels with depths 10-500 m | Hg(0), Hg(II), Hg(II)ₚ, MMHg, DMHg, and MMHgP in water; MMHg in 6 phytoplankton and 2 zooplankton | *** (see comment below table) | Zhang et al. (2019c) |
| MERCY | Global | ICON-O | ECOSMO | 7 to 148 km unstructured triangular grid | z- or z*-levels with depths of 15-200 m | 14 Hg species in water; 21 in biota (3 phytoplankton, 2 zooplankton, 1 macrobenthos, 2 fish) | MERCY | Bieser et al. (2023) |

**\*\*\*** The inorganic Hg chemistry in MITgcm-Hg includes photochemical and biological redox conversions between dissolved Hg(0) and Hg(II) in the oceanic mixed layer and biologically mediated redox reactions in the subsurface ocean. The dark oxidation of dissolved Hg(0) in the mixed layer is also included. The formation rates of methylated Hg are parameterized according to microbial remineralization of organic carbon. The photochemical and dark demethylation processes are included. The rate constants of photochemical reactions are scaled by the shortwave radiation intensity attenuated by ocean pigments (chlorophyll and dissolved organic carbon). The dark demethylation rate is calculated as a function of seawater temperature. CH₃Hg uptake by
phytoplankton is simulated with volume concentration factors as a function of cell size and seawater DOC concentration. The trophic transfer of CH₃Hg to zooplankton is modelled with the processes including zooplankton grazing, excretion, and mortality.

**Table F3. Multi-media mass-balance models. The numbers in parentheses give the number of boxes per media.**

| Name | Number of Compartments | Media | Mercury Species | Spatial Scope | References |
|---|---|---|---|---|---|





| Global Biogeochemical Box Model (GBC) | 7 | Atmosphere (1) Ocean (3) Terrestrial (3) | THg | Global | Amos et al. (2013, 2014, 2015) |
|---|---|---|---|---|---|
| Six Box Biogeochemical Cycle Model | 6 | Atmosphere (2) Ocean (3) Terrestrial (2) | THg | Global | Selin (2014) |
| Atm-Hg_3boxmodel | 3 | Atmosphere (3) | Hg(0), Hg(II) | Hemispheric | Feinberg et al. (2022) |
| Arctic Ocean Hg and MeHg Mass Budget Model | 2 | Ocean (2) | Hg(0), Hg(II), MMHg, DMHg | Regional (Arctic Ocean) | Soerensen et al. (2016a); Schartup et al. (2022) |
| WorM$^3$ | 5 | Atmosphere (1) Soil (1) Vegetation (1) Ocean (2) | Hg(0), Hg(II)$_g$, Hg(II)$_p$, Hg(II)$_{aq}$ | Global | Qureshi et al. (2011) |

**Table F4. Exposure-risk models.**

| Research Scope | Types of Sources | Exposure Pathways | Data Sources of Hg in Food | Risk Endpoint | References |
|---|---|---|---|---|---|
| Region | Coal-fired power plant | Fish Ingestion | Survey Data | Adult Paresthesia | Fthenakis et al. (1995) |
| Region (nearby the plant) | MSW gasification plant | Inhalation, Soil ingestion, Dermal contact and Diet | X | Hazard Index (HI) | Lonati and Zanoni (2013) |
| Region (China) | China Atmospheric Hg Emission | Diet | Obtained from Previous Studies by Authoritative Scientific Research | IQ and Fatal Heart Attacks (FHA) | Chen et al. (2019) |
| Region (China) | Coal-fired power plants retrofitting measures | Diet (10 kinds of food) | Obtained from previous studies | IQ and Fatal Heart Attacks | Li et al. (2020b) |
| Global | Artisanal and Small-scale Gold Mining (ASGM) | Seafood, Freshwater fish, and Rice | Food Intake Inventory and Literature-Collected | IQ and Fatal Heart Attacks and Monetized Impact | Pang et al. (2022) |





| Global | Global Atmospheric Mercury Emissions | Seafood, Freshwater fish and Rice | Food Intake Inventory and Literature-Collected | IQ and Fatal Heart Attacks and Monetized global Impact | Zhang et al. (2021) |
|---|---|---|---|---|---|
| Region (Pacific Ocean) | Anthropogenic Mercury Release | | X | Pacific Ocean Tuna MeHg Concentration | Médieu et al. (2022) |
| Global | | Rice | FAO data and Peer-reviewed Publications | IQ and Fatal Heart Attacks | Liu et al. (2019b) |

**Table F5. Global atmospheric model drivers. External and internal atmospheric model drivers. Different options for harmonizing drivers for the multi-model exercise are given in the last column, the option for the baseline simulation being highlighted in bold.**

| Drivers | GEOS-Chem | GEM-MACH-Hg | GLEMOS | ECHMERIT | CAM-Chem/Hg | Driver options for simulations |
|---|---|---|---|---|---|---|
| **Meteorological** | | | | | | |
| Meteorological data | Offline. MERRA-2 or GEOS-FP reanalysis | Online. Global Environmental Multi-scale (GEM) model | Offline. WRF driven by the operational analysis data from ECMWF | Online. Fifth-generation Atmospheric General Circulation Model ECHAM5 | Offline. NCEP/DOE AMIP II reanalysis | 1) **Model-specific** 2) Using the same base reanalysis (when possible) |
| **Chemical** | | | | | | |
| Oxidant concentrations | Offline. Archived oxidant concentrations (Wang et al., 2021b) | Offline. Time-varying oxidant concentrations are from GEM-MACH (Makar et al., 2018); the Br concentrations are derived from satellite observations | Offline. The Br, O3 and OH concentrations are imported from GEOS-Chem. | Online. The gas phase chemical mechanism is based on the CBMZ mechanism (Zaveri and Peters, 1999); The tropospheric aqueous phase mechanism is derived from MECCA (Sander et al., 2005) | Online for ozone, OH, sulphate, and chlorine based on the internal model; Monthly averaged Br concentrations derived from GEOS-Chem Hg model. | 1) **Model-specific** 2) Harmonized dataset (GEOS-Chem) 3) Harmonized dataset (CAM-Chem) |
| Aerosol data for Hg(II) partitioning | Offline. Archived aerosol concentrations | Offline. Time-varying concentrations from GEM-MACH (Makar et al., 2018) | Off-line. Imported from GEOS-Chem | None | Online but assume 50/50 partitioning of Hg(II) (Holmes et al., 2010). | 1) **Model specific** 2) Harmonized dataset (GEOS-Chem) |
| J-values | Fast-JX v7.0a implemented in GEOS-Chem (Eastham et al., 2014) | JVAL14-MESSy based formulation (Sander et al., 2014) | Hg(II)$_g$ photo-reduction rates are pre-calculated by CAM-Chem (Saiz-Lopez et al., 2018, 2020) | Fast-J photolysis mechanism (Wild et al., 2000) | A combined (online-lookup table) approach (Lamarque et al., 2012) | 1) **Model-specific** |
| **Geophysical** | | | | | | |
| Land cover type | Olson 2001 land map (73 land-use categories) with LAI data from | 15 land-use categories | 17 land-use categories (MODIS) | Land use characteristics are directly transferred from ECHAM5 | Community Land Model (Oleson et al., 2010) | 1) **Model-specific** 2) MODIS land cover dataset (MCD12Q1, https://doi.org/10.506 |





| | | | | | | |
|---|---|---|---|---|---|---|
| | MODIS | | | | | 7/MODIS/MCD12Q 1.006)<br>3) MODIS LAI (GEOS-Chem) |
| **Emissions** | | | | | | |
| Anthropogenic Hg emissions | Various | Various | Various | Various | Various | **E2010/2020[1]** |
| Geogenic Hg emissions | Geogenic emissions distributed according to the locations of mercury mines (Selin et al., 2007) | | | | | **1) Harmonized geogenic emissions inventory (Section 3.1)**<br>2) GEOS-Chem emission files |
| Secondary Hg emissions (ocean) | Air-sea exchange of prescribed ocean Hg(0) concentration field (Horowitz et al., 2017) | Ocean emissions are determined by air–sea Hg(0) exchange scheme | Prescribed fluxes from seawater proportional to the primary production in seawater and based on global estimates (Lamborg et al., 2002; Mason, 2009) | Hg(0) ocean fluxes are calculated using the two-layer gas exchange model (Liss and Slater, 1974) | Ocean emissions are determined by a simplified air-sea exchange scheme | **1) Time-varying (2010–2020) sea surface Hg concentrations from multi-model ocean Hg average; harmonized air-sea exchange scheme (Appendix B)**<br>2) Model-specific<br>3) Mass balance models for scaling legacy emissions magnitude and trend |
| Secondary Hg emissions (land) | The method of Selin et al. (2008) | Natural sources and reemissions of previously deposited Hg from land are based on formulations by Gbor et al. (2007) and Shetty et al. (2008). | Prescribed fluxes from soil based on global estimates (Lamborg et al., 2002; Mason, 2009). On-line calculated prompt reemission from snow. | Soils and vegetation emissions are calculated offline and derived from the EDGAR/POET emission inventory (Granier et al., 2005) | Land sources include emissions from soil and vegetation, plus rapid reemissions of deposited mercury | **1) Time-varying soil reemissions based on global soil Hg concentration dataset and air-soil reemissions modelling (Section 3.4)**<br>2) Model-specific<br>3) Mass balance models for scaling legacy emissions magnitude and trend |
| Wildfire Hg emissions | Hg emissions factors are applied to an emission inventory, including: Global Fire Emissions Database (GFED) v4.1s (van der Werf et al., 2017), Quick Fire Emissions Dataset (QFEDv2), or Fire Inventory from NCAR (FINN) | Fire INventory from NCAR (FINN), and Hg emission factors (Fraser et al., 2018) | Fire INventory from NCAR (FINN). Hg emission factors from Andreae (2019) | Various | From the monthly means from the IPCC estimate of biomass burned and the IMAGE projection of managed forests | 1) **Harmonized Wildfire emissions inventory (Section 3.3)**<br>2) GFED based emissions<br>3) FINN based emissions |

[1]E2010/2020: Updated spatially distributed global anthropogenic emissions inventories from 2010 to 2020 as available for the study (see Section 3.2.1).





**Table F6. Ocean model drivers. External and internal marine model environmental drivers that could be the cause of trends in regional or global ocean Hg concentration during the past and coming decades. References are focused on prior model studies on the impact of these drivers (regional to global context).**

| Drivers | MERCY | MITgcm | Possibility of harmonizing drivers for simulations | References |
|---|---|---|---|---|
| **Atmospheric ensemble input** | | | | |
| Atmospheric Hg(0) concentration | CMAQ-Hg/ ICON-ART | GEOS-Chem | atmospheric model ensemble | Schröter et al. (2018) |
| Hg deposition | CMAQ-Hg/ ICON-ART | GEOS-Chem | atmospheric model ensemble | Schröter et al. (2018) |
| **Rivers** | | | | |
| River freshwater discharge | HD | Liu et al. (2021b) | Liu et al. (2021b) | Hagemann and Dümenil (1997); Liu et al. (2021b) |
| Hg in river discharge | HydroPy | Liu et al. (2021b) | Liu et al. (2021b) | Stacke and Hagemann (2021); Liu et al. (2021b) |
| Nutrients in river discharge | HydroPy | NEWS | NEWS | Stacke and Hagemann (2010); Mayorga et al. (2010)5/23/2024 4:50:00 PM |
| TOC in river discharge | GRDC | NEWS | NEWS | Mayorga et al. (2010) |
| **Atmosphere** | | | | |
| Precipitation | ERA5 | ERA5 | ERA5 | Hersbach et al. (2020) |
| Wind speed | ERA5 | ERA5 | ERA5 | Hersbach et al. (2020) |
| Surface shortwave radiation | ERA5 | ERA5 | ERA5 | Hersbach et al. (2020) |
| Atmospheric N deposition | CAMS | n/a | CAMS | Inness et al. (2019); Schröter et al. (2018) |
| Atmospheric Fe deposition | CAMS | Mahowald et al. (2009) | CAMS | Inness et al. (2019); Schröter et al. (2018) |
| Air-sea exchange | modeled variable (MERCY) | modelled variable | Nightingale et al (2000a) | Kuss et al. (2014; 2009); Nightingale et al. (2000a); Andersson et al. (2008a); Schwarzenbach et al. (2003); Liss (1973); Liss and Slater (1974) |
| **Ocean** | | | | |
| Bathymetry | GEBCO | ECCO v4 | GEBCO | GEBCO Compilation Group (2022) |
| Coast lines | GEBCO | ECCO v4 | GEBCO | GEBCO Compilation Group (2022) |
| Salinity | modelled variable (ICON-O) | modelled variable (MITgcm) | WOA | Boyer et al. (2018) |
| Sea water temperature | modelled variable (ICON-O) | modelled variable (MITgcm) | WOA | Boyer et al. (2018) |
| Nutrients in water (N,P,Si,Fe) | modelled variable (ECOMSO) | modelled variable (MITgcm) | n/a | Boyer et al. (2018) |
| Sea ice extend | modelled variable (ICON-O) | modelled variable (MITgcm) | n/a | Korn et al. (2022) |
| Tides | modelled variable (ICON-O) | n/a | n/a | Egbert and Erofeeva (2002) |
| Primary production | modelled variable (ECOSMO) | Darwin | n/a | Daewel and Schrum (2013) ; Daewel et al. (2019) |
| Biota growth, feeding, and mortality rates | modelled variable (ECOSMO) | Darwin | n/a | Daewel and Schrum (2013) ; Daewel et al. (2019) |





| Light attenuation | modelled variable (MERCY) | Darwin | n/a | Bieser et al. (2023) |
|---|---|---|---|---|
| Particle settling | modelled variable (ICON-O) | Darwin | n/a | Schröter et al. (2018) |
| Bioaccumulation | modelled variable (MERCY) | modelled variable | n/a | Bieser et al. (2023) |
| Sediments and resuspension | modelled variable (MERCY) | n/a | n/a | Bieser et al. (2023) |

**Table F7. Multi-media mass balance box-model drivers.**

| **Drivers** | |
|---|---|
| **External: Hg emissions and releases** | |
| Natural | Geogenic |
| Anthropogenic | Emissions to air |
| | Releases to land and water |
| **Internal: time-invariant rate coefficients of Hg exchange between compartments** | |
| Atmosphere | Hg(II) deposition to ocean |
| | Hg(0) deposition to ocean |
| | Hg(II) deposition to land |
| | Hg(0) deposition to land |
| Surface ocean | Hg(0) ocean evasion |
| | Particle settling to subsurface ocean |
| | Water transfer to subsurface ocean |
| Subsurface ocean | Particle settling to deep ocean |
| | Water transfer to surface ocean |
| | Water transfer to deep ocean |
| Deep ocean | Burial to deep sediments |
| | Water transfer to subsurface ocean |
| Fast terrestrial pool | Evasion due to respiration of organic carbon |
| | Photochemical reemission of deposited Hg |
| | Biomass burning |
| | Transfer to slow pool |
| | Transfer to armored pool |
| | River runoff to surface ocean |
| Slow soil pool | Evasion due to respiration of organic carbon |
| | Biomass burning |
| | Transfer to fast pool |
| | River runoff to surface ocean |
| Armored soil pool | Evasion due to respiration of organic carbon |





| | Biomass burning |
|---|---|
| | Transfer to fast pool |
| | River runoff to surface ocean |

**Table F8. Atmospheric multi-model ensemble simulations.**

| Simulation | Purpose | Description | Driver option | Time range | Priority[*] |
|---|---|---|---|---|---|
| **Baseline** | | | | | |
| A1 | Spatial and temporal trends of Hg levels (baseline) | Global-scale simulations of Hg atmospheric dispersion using the state-of-the-art model configuration and unified emissions inventories | Defaults (bold in **Table F5**) | 2010-2020 | 1 |
| **Source Contributions[1]** | | | | | |
| A2.1 | Relative contribution of anthropogenic emissions | Perturbation simulation with no anthropogenic emissions | Ant. = 0 | 2010-2020 | 1 |
| A2.2 | Relative contribution of geogenic emissions | Perturbation simulation with no geogenic emissions | Geo. = 0 | 2010-2020 | 1 |
| A2.3 | Relative contribution of land legacy emissions | Perturbation simulation with no legacy emissions from soil | L_land = 0 | 2010-2020 | 1 |
| A2.4 | Relative contribution of wildfire emissions | Perturbation simulation with no emissions from wildfires | BB = 0 | 2010-2020 | 1 |
| A2.5 | Relative contributions of ASGM emissions | Perturbation simulation with no ASGM emissions | ASGM = 0 | 2010-2020 | 1 |
| A2.6 | Relative contributions of MC Annex D sources | Perturbation simulation with no emissions from MC Annex D sources | Annex D = 0 | 2010-2020 | 1 |
| **Trend analysis – identifying drivers of trends** | | | | | |
| a. **Contributions of various drivers to trends[2]** | | | | | |
| A3a.1 (**Nature Simulation**) | Contributions of anthropogenic emissions and environmental drivers on changes of Hg levels | Perturbation simulation with fixed 2010 anthropogenic emissions | E2010/2020 = 2010 All others = varying | 2010-2020 | 1 |
| A3a.2 | Contribution of ASGM on changes of Hg levels | Perturbation simulation with ASGM emissions fixed at 2010 | ASGM = 2010 All others = varying | 2010-2020 | 1 |
| A3a.3 | Contribution of Annex D on changing Hg levels | Perturbation simulation with MC Annex D emissions fixed at 2010 | Annex D = 2010 All others = varying | 2010-2020 | 1 |
| A3a.4 | Contribution of geogenic emissions on changes of Hg levels | Perturbation simulation with fixed 2010 geogenic emissions | Geogenic = 2010 All others = varying | 2010-2020 | 1 |
| b. **Effects of changes in various drivers to trends** | | | | | |
| A3b.1 | Effect of transport on changes in Hg levels (inert tracer experiment) | Baseline simulation with inert tracer emitted with the same spatial distribution as Hg, but all chemistry and deposition turned off (exponential decay imposed with 6-month lifetime) | Meteorology = varying All emissions fields fixed with 2010 spatial distribution (multi-model mean from A1) | 2010-2020 | 2 |
| A3b.2 | Effect of wildfire emissions on changes of Hg levels | Perturbation simulation with varying wildfire emissions and all other drivers fixed | Wildfire emissions = varying All others = 2010 | 2010-2020 | 2 |



| A3b.3 | Effect of terrestrial legacy emissions on changes of Hg levels | Perturbation simulation with varying terrestrial legacy emissions and all other drivers fixed | Land reemissions = varying All others = 2010 | 2010-2020 | 2 |
|---|---|---|---|---|---|
| A3b.4 | Effect of ocean concentrations on changes of Hg levels | Perturbation simulation with varying oceanic concentrations and all other drivers fixed | Ocean reemission = varying All others = 2010 | 2010-2020 | 2 |
| A3b.5 | Effect of land surface conditions on changes of Hg levels | Perturbation simulation with varying land surface conditions (land cover, LAI) and all other drivers fixed | LAI/land cover = varying All others = 2010 | 2010-2020 | 2 |
| A3b.6 | Effect of chemical reactants on changes of Hg levels | Perturbation simulation with varying air concentrations of major chemical reactants (Br, OH, Cl, $O_3$, …) and $PM_{2.5}$ and all other drivers fixed | Reactants = varying All others = 2010 | 2010-2020 | 2 |
| A3b.7 | Effect of anthropogenic emissions speciation on changes of Hg levels | Perturbation simulation with varying anthropogenic emissions speciation ratios for $Hg(II)_g$:$Hg(II)_p$:$Hg(0)$ and all other drivers fixed | E2010/2020= varying $Hg(II)_g$:$Hg(II)_p$:$Hg(0)$ ratio All others = 2010 | 2010-2020 | 2 |
| c. | **Contributions of HTAP regions (Appendix C) anthropogenic emissions changes to trends[2]** | | | | |
| A3c.1 | Contribution of Arctic emissions on changes of Hg levels | Perturbation simulation with fixed 2010 emissions from the Arctic | Arctic = 2010 All others = varying | 2010-2020 | 2 |
| A3c.2 | Contribution of North American emissions on changes of Hg levels | Perturbation simulation with fixed 2010 emissions from North America | N. America = 2010 All others = varying | 2010-2020 | 2 |
| A3c.3 | Contribution of Central American emissions on changes of Hg levels | Perturbation simulation with fixed 2010 emissions from Central America | C. America = 2010 All others = varying | 2010-2020 | 2 |
| A3c.4 | Contribution of South American emissions on changes of Hg levels | Perturbation simulation with fixed 2010 emissions from South America | S. America = 2010 All others = varying | 2010-2020 | 2 |
| A3c.5 | Contribution of European emissions on changes of Hg levels | Perturbation simulation with fixed 2010 emissions from Europe | Europe = 2010 All others = varying | 2010-2020 | 2 |
| A3c.6 | Contribution of CIS countries emissions on changes of Hg levels | Perturbation simulation with fixed 2010 emissions from CIS countries | CIS countries = 2010 All others = varying | 2010-2020 | 2 |
| A3c.7 | Contribution of African emissions on changes of Hg levels | Perturbation simulation with fixed 2010 emissions from Africa | Africa = 2010 All others = varying | 2010-2020 | 2 |
| A3c.8 | Contribution of South Asian emissions on changes of Hg levels | Perturbation simulation with fixed 2010 emissions from South Asia | S. Asia = 2010 All others = varying | 2010-2020 | 2 |
| A3c.9 | Contribution of East Asian emissions on changes of Hg levels | Perturbation simulation with fixed 2010 emissions from East Asia | E. Asia = 2010 All others = varying | 2010-2020 | 2 |
| A3c.10 | Contribution of Southeast Asian emissions on changes of Hg levels | Perturbation simulation with fixed 2010 emissions from Southeast Asia | SE. Asia = 2010 All others = varying | 2010-2020 | 2 |
| A3c.11 | Contribution of Middle Eastern emissions on changes of Hg levels | Perturbation simulation with fixed 2010 emissions from the Middle East | Middle East = 2010 All others = varying | 2010-2020 | 2 |
| A3c.12 | Contribution of Australia and NZ emissions on changes of Hg levels | Perturbation simulation with fixed 2010 emissions from Australia and NZ | Australia and NZ = 2010 All others = varying | 2010-2020 | 2 |





| | Sensitivity analysis – Uncertainty experiments | | | | |
|---|---|---|---|---|---|
| A4.1 | Uncertainties due to anthropogenic emissions data[†] | Sensitivity simulation with second alternative anthropogenic emissions inventory | Ant. = E2010/2020_2[†] | 2010-2020 | 2 |
| A4.2 | Uncertainties due to anthropogenic emissions data[†] | Sensitivity simulation with third alternative anthropogenic emissions inventory | Ant. = E2010/2020_3[†] | 2010-2020 | 2 |
| A4.3 | Uncertainties due to anthropogenic emissions data[†] | Sensitivity simulation with mosaic inventory with regional emissions inventories (including the Tsinghua inventory for China) substituted inside global inventory | Ant. = E_mos# | 2010–2020 | 2 |
| A4.4 | Uncertainties due to legacy Hg emissions magnitude and trend | Sensitivity simulation with ocean surface concentrations magnitude and trend based on mass balance modelling results | Ocean = scaled by mass balance model results | 2010-2020 | 2 |
| A4.5 | Uncertainties due to legacy Hg emissions magnitude and trend | Sensitivity simulation with land legacy emissions magnitude and trend based on mass balance modelling results | Land = scaled by mass balance model results | 2010-2020 | 2 |
| A4.6 | Uncertainties due to air-sea exchange processes | Sensitivity simulation with alternative parametrization for air-sea exchange | Using air-sea exchange parametrization from Nightingale or Wanninkhof (depending what was used for A1) | 2010-2020 | 2 |
| A4.7 | Uncertainties due to soil Hg emissions | Sensitivity simulation with alternative soil Hg emissions dataset | Each model uses their own in-house soil emissions parametrization or dataset here | 2010-2020 | 2 |
| A4.8 | Uncertainties due to model resolution | Sensitivity simulation with enhanced model spatial resolution | Model resolution increased | 2010-2020 | 2 |
| | Sensitivity analysis – Idealized experiments | | | | |
| A5.1 | Sensitivity of anthropogenic emissions speciation | Sensitivity simulation with all anthropogenic emissions as Hg(0) | E2010/2020= released with speciation ratios 100:0:0 Hg(0):Hg(II)$_g$:Hg(II)$_p$ | 2015 | 2 |
| A5.2 | Sensitivity of anthropogenic emission seasonality | Sensitivity simulation with imposed seasonality of anthropogenic Hg emissions | e.g., Hg emissions maximum during winter | 2015 | 2 |
| A5.3 | Sensitivity of atmospheric chemistry | Sensitivity simulation with perturbed Hg oxidation rates | Hg(0) oxidation rates enhanced +50% | 2015 | 2 |
| A5.4 | Sensitivity of atmospheric chemistry | Sensitivity simulation with perturbed Hg(II) aqueous photoreduction rates | Hg(II) aqueous reduction enhanced +50% | 2015 | 2 |
| A5.6 | Sensitivity of atmospheric chemistry | Sensitivity simulation with perturbed Hg(II)$_g$-Hg(II)$_p$ gas-particle partitioning | Hg(II)$_p$ partitioning rate enhanced +50% | 2015 | 2 |
| A5.7 | Sensitivity of Hg(0) dry deposition velocities | Sensitivity simulation with perturbed Hg(0) dry deposition velocities to land | Hg(0) dry deposition velocity to land doubled | 2015 | 2 |
| A5.8 | Sensitivity of models due to vegetation | Sensitivity simulation with vegetation increased by a certain factor (idealizing future greening) | LAI increased by a factor of 50% | 2015 | 2 |
| A5.9 | Sensitivity of models to air-sea exchange processes | Sensitivity simulation with alternative parametrization for air-sea exchange based on Baltic Sea measurements | Using air-sea exchange parametrization from Osterwalder et al. (2021) | 2015 | 2 |
| A5.10 | Sensitivity of models to sea ice | Sensitivity simulation with all sea ice removed | Sea ice = 0 | 2015 | 2 |
| | Future scenarios[3] | | | | |





| | | | | | |
|---|---|---|---|---|---|
| A6.1 **(Future baseline simulation)** | Future changes of Hg levels due to cumulative effect of anthropogenic emissions and environmental factors | Simulations of future Hg levels using scenarios of anthropogenic emissions, projected environmental conditions and estimates for legacy/geogenic emissions | | 2020-2050 | 3 |
| A6.2 | Effect of anthropogenic emissions on future changes of Hg levels | Perturbation simulation with varying anthropogenic emissions[3] | E2010/2020 = varying all others = 2020 | 2020-2050 | 3 |
| A6.3 | Effect of environmental conditions on changes of Hg levels | Perturbation simulation with varying environmental conditions and legacy emissions[3] | E2010/2020= 2020 all others = varying | 2020-2050 | 3 |


[*]Priority ranking: (1) "Core simulations", expected to be performed by all global models; (2) "Optional simulations", expected to be performed by as many models as possible; and (3) "Future simulations" to be conducted by a multi-modeling study following the modeling and analysis study based on retrospective simulations (i.e., Priority ranking 1 and 2).

[†]Simulations with the other emission inventories will depend on availability of these inventories at the time of the multi-

model project (see Section 3.2.1 for discussion of inventories).

[#]Abbreviations: E_mos = Mosaic inventory: E2010/2020 + available regional emissions inventories for time period

[1] The contribution of a source type to Hg levels is obtained by subtraction of results of the perturbation simulation from results of the baseline simulation (Simulation A1). The contribution of oceanic Hg reemissions is calculated by subtracting all other source simulations (A2.1-A2.4) from baseline (A1).

[2] The contribution of a factor on changes of Hg levels is obtained by subtraction of results of the perturbation simulation from results of the baseline simulation (Simulation A1).

[3] The contribution of a factor on future changes of Hg levels is obtained by subtraction of results of the perturbation simulation from results of the future baseline simulation (Simulation A6.1).






**Table F9. Ocean multi-model ensemble simulations.**

| Simulations | Purpose | Description | Related atmospheric scenario | Driver option | Time range | Priority* |
|---|---|---|---|---|---|---|
| **1. Baseline** | | | | | | |
| O1 **(base case)** | Spatial and temporal trends of Hg levels | Global-scale simulations of oceanic Hg transport using the state-of-the-art model configuration, with upper boundary conditions (i.e., atmospheric Hg concentrations and deposition fluxes) from atmospheric model ensemble results. | A1 | Harmonized default drivers as described in Table F6 | 2010-2020 | 1 |
| **2. Source Contributions** | | | | | | |
| O2.1 | Relative contribution of anthropogenic emissions | Perturbation run with no anthropogenic emissions. | A2.1 | Ant. = 0 | 2010-2020 | 1 |
| O2.2 | Relative contribution of geogenic emissions | Perturbation run with no geogenic emissions. | A2.2 | Geo. = 0 | 2010-2020 | 1 |
| O2.3 | Relative contribution of riverine Hg inflow | Perturbation runs with no riverine Hg influx. | A1 | Rivers = 0 | 2010-2020 | 1 |
| O2.4 | Impact of food web Hg uptake on marine Hg cycling | Model runs with and without food web interactions. | A1 | Bioacc. = 0. | 2010-2020 | 1 |
| O2.5 | Relative contributions of ASGM emissions. | Perturbation simulation with no ASGM emissions. | A2.5 | ASGM = 0 | 2010-2020 | 1 |
| O2.6 | Relative contributions of MC Annex D sources | Perturbation simulation with no emissions from MC Annex D sources | A2.6 | Annex D = 0 | 2010-2020 | 1 |
| **3. Trend analysis** - identifying drivers of trends | | | | | | |
| O3.1 **(Nature simulation)** | Contributions of anthropogenic emissions and environmental drivers on changes of Hg levels. | Perturbation simulation with fixed 2010 anthropogenic emissions | A3a.1 | E2010/2020 = 2010 | 2010-2020 | 1 |



| O3.2 | Effect of inter-annual variability of meteorological drivers. | Perturbation run with varying meteorological boundary conditions and all other drivers fixed. | A3b.1 | Meteorological fields varying. All others = 2010 | 2010-2020 | 2 |
|------|------|------|------|------|------|------|
| O3.3 | Contribution of Arctic emissions on changes of Hg levels | Perturbation simulation with fixed 2010 emissions from the Arctic | A3c.1 | Arctic = 2010 All others = varying | 2010-2020 | 2 |
| O3.4 | Contribution of North American emissions on changes of Hg levels | Perturbation simulation with fixed 2010 emissions from North America | A3c.2 | North America = 2010 All others = varying | 2010-2020 | 2 |
| O3.5 | Contribution of Central American emissions on changes of Hg levels | Perturbation simulation with fixed 2010 emissions from Central America | A3c.3 | Central America = 2010 All others = varying | 2010-2020 | 2 |
| O3.6 | Contribution of South American emissions on changes of Hg levels | Perturbation simulation with fixed 2010 emissions from South America | A3c.4 | South America = 2010 All others = varying | 2010-2020 | 2 |
| O3.7 | Contribution of European emissions on changes of Hg levels | Perturbation simulation with fixed 2010 emissions from Europe | A3c.5 | Europe = 2010 All others = varying | 2010-2020 | 2 |
| O3.8 | Contribution of CIS countries' emissions on changes of Hg levels | Perturbation simulation with fixed 2010 emissions from CIS countries | A3c.6 | CIS = 2010 All others = varying | 2010-2020 | 2 |
| O3.9 | Contribution of African emissions on changes of Hg levels | Perturbation simulation with fixed 2010 emissions from Arica | A3c.7 | Africa = 2010 All others = varying | 2010-2020 | 2 |
| O3.10 | Contribution of South Asian emissions on changes of Hg levels | Perturbation simulation with fixed 2010 emissions from South Asia | A3c.8 | South Asia = 2010 All others = varying | 2010-2020 | 2 |
| O3.11 | Contribution of East Asian emissions on changes of Hg level | Perturbation simulation with fixed 2010 emissions from East Asia | A3c.9 | East Asia = 2010 All others = varying | 2010-2020 | 2 |
| O3.12 | Contribution of Southeast Asian emissions on changes of Hg level | Perturbation simulation with fixed 2010 emissions from Southeast Asia | A3c.10 | Southeast Asia = 2010 All others = varying | 2010-2020 | 2 |





| O3.13 | Contribution of Middle Eastern emissions on changes of Hg levels | Perturbation simulation with fixed 2010 emissions from the Middle East | A3c.11 | Middle East = 2010 All others = varying | 2010-2020 | 2 |
|---|---|---|---|---|---|---|
| O3.14 | Contribution of Australia and NZ emissions on changes of Hg levels | Perturbation simulation with fixed 2010 emissions from Australia and NZ | A3c.12 | Australia and NZ = 2010 All others = varying | 2010-2020 | 2 |
| **4. Uncertainty analysis** | | | | | | |
| O4.1 | Uncertainties due to atmospheric deposition data † | Sensitivity runs with alternative atmospheric deposition data from the second, alternative emissions inventory | A4.1 | Ant. = E2010/2020_2 † | 2010-2020 | 2 |
| O4.2 | Uncertainties due to atmospheric deposition data † | Sensitivity runs with alternative atmospheric deposition data from third, alternative emissions inventory | A4.2 | Ant. = E2010/2020_3 † | 2010-2020 | 2 |
| O4.3 | Uncertainties due to anthropogenic emissions data † | Sensitivity simulation with mosaic inventory with regional emissions inventories (including the Tsinghua inventory for China) substituted inside global inventory | A4.3 | Ant. = E_mos# | 2010-2020 | 2 |
| O4.4 | Uncertainties due to oceanic Hg chemistry | Sensitivity runs with alternative chemical mechanisms and partitioning | A1 | Use an alternative chemistry scheme | 2010-2020 | 2 |
| O4.5 | Uncertainties due to air-sea exchange processes | Sensitivity runs with alternative parametrizations of air-sea exchange | A4.6 | Using air-sea exchange parametrization from Nightingale or Wanninkhof (depending on what was used for O1) | 2010-2020 | 2 |
| O4.6 | Uncertainties due to air-sea exchange processes | Sensitivity simulation with alternative parametrization for air-sea exchange based on Baltic Sea measurements | A5.8 | Using air-sea exchange parametrization from Osterwalder21 | 2010-2020 | 2 |
| O4.7 | Uncertainties due to model resolution | Sensitivity simulation with enhanced model spatial resolution | A4.8 | Model resolution increased | 2010-2020 | 2 |
| **5. Future scenarios** | | | | | | |

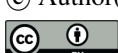



| O5.1<br><br>**(Future baseline simulation)** | Future changes of Hg levels due to the cumulative effect of anthropogenic emissions and environmental factors | Simulations of future oceanic Hg levels using forcing from climate models and scenarios of future anthropogenic emissions, and estimates for riverine and geogenic emissions (2020-2050) | A6.1 | Future emission scenarios and climate model forcing | 2020-2050 | 3 |
|---|---|---|---|---|---|---|
| O5.2 | Effect of anthropogenic emissions on future changes of Hg levels | Perturbation run with varying anthropogenic emissions. | A6.2 | Future emission scenario all other drivers = 2020 | 2020-2050 | 3 |
| O5.3 | Effect of meteorological/ocean physical state on future changes of Hg levels | Perturbation run with fixed meteorological boundary conditions from climate models | A6.3 | Climate model forcing All other drivers = 2020 | 2020-2050 | 3 |

\* Priority ranking: (1) "Core simulations", expected to be performed by all global models; (2) "Optional simulations", expected to be performed by as many models as possible; and (3) "Future simulations" to be conducted by a multi-modelling study following the modelling and analysis study based on retrospective simulations (i.e., Priority ranking 1 and 2).

\# Abbreviations: E_mos = Mosaic inventory: E2010/2020 + available regional emissions inventories for time period

† Simulations with the other emission inventories will depend on availability of these inventories at the time of the multi-model project (see Section 3.2.1 for discussion of inventories).







**Table F10. Mass balance model simulations. All simulations will be performed with the model ensemble (see Sect. 5.3) to account for uncertainties in Hg emissions, rate constants and fluxes. In addition, all the simulations listed here are "core simulations".**

| Simulation | Purpose | Description | Time range[1] |
|---|---|---|---|
| **Baseline** | | | |
| M1 | Temporal trends in Hg levels and legacy emissions | Rates constructed using "passing" model configurations (see Fig. 9) | 1500 – 2020 (simulation and analysis) |
| **Secondary emission and reservoir mass trends [2010 – 2020]** | | | |
| M2-M11 | Pre-YYYY legacy | Primary emissions eliminated as of year YYYY (YYYY = 2010 to 2019) | 1500 – 2020 (simulation); 2010 – 2020 (analysis) |
| **Source attribution** | | | |
| M12 | Temporal trends in Hg levels and legacy emissions attributable to Annex D emissions to *air* [all-time] | Calculate legacy emissions and reservoir Hg masses attributable to emissions from Annex D source categories. Simulate entire ensemble for each of [lower, central, upper] estimates of category emission. Repeat for available emissions inventories. | 1500 – 2020 (simulation); 2010 – 2020 (analysis) |
| M13 | Temporal trends in Hg levels and legacy emissions attributable to ASGM emissions to *air* [all-time] | Calculate legacy emissions and reservoir Hg masses attributable to emissions from ASGM. Simulate entire ensemble for each of [lower, central, upper] estimates of category emission. Repeat for available emissions inventories. | 1500 – 2020 (simulation); 2010 – 2020 (analysis) |
| M14 | Temporal trends in Hg levels and legacy emissions attributable to all other emissions to *air* [all-time] | Calculate legacy emissions and reservoir Hg masses attributable to emissions from all source categories other than Annex D and ASGM emissions. Simulate entire ensemble for each of [lower, central, upper] estimates of category emission. Repeat for available emissions inventories. | 1500 – 2020 (simulation); 2010 – 2020 (analysis) |
| M15 | Temporal trends in Hg levels and legacy emissions attributable to Annex D releases to *land and water* [all-time] | Calculate legacy emissions and reservoir Hg masses attributable to emissions from Annex D source categories. Simulate entire ensemble for each of [lower, central, upper] estimates of category emission. Repeat for available emissions inventories. | 1500 – 2020 (simulation); 2010 – 2020 (analysis) |
| M16 | Temporal trends in Hg levels and legacy emissions attributable to ASGM releases to *land and water* [all-time] | Calculate legacy emissions and reservoir Hg masses attributable to emissions from ASGM. Simulate entire ensemble for each of [lower, central, upper] estimates of category emission. Repeat for available emissions inventories. | 1500 – 2020 (simulation); 2010 – 2020 (analysis) |
| M17 | Temporal trends in Hg levels and legacy emissions attributable to all other releases to *land and water* [all-time] | Calculate legacy emissions and reservoir Hg masses attributable to emissions from all source categories other than Annex D and ASGM emissions. Simulate entire ensemble for each of [lower, central, upper] estimates of category emission. Repeat for available emissions inventories. | 1500 – 2020 (simulation); 2010 – 2020 (analysis) |
| **Future scenarios** | | | |
| M18 | Hg levels and legacy emissions under future *anthropogenic* emission trajectory | Calculate legacy emission and reservoir Hg mass trend [2020 - 2050] under future anthropogenic emission scenario(s) | 1500 – 2050 (simulation); 2020 – 2050 (analysis) |

[1] All simulations begin from the analytical steady state calculated for model rates + geogenic emissions.






**Table F11. Atmospheric model simulation output variables.**

| Type | Variables | Temporal frequency |
|---|---|---|
| **3D variables** | | |
| Mixing ratio of Hg species | Hg(0), Hg(II)$_g$, Hg(II)$_p$ | Monthly |
| Chemical reaction fluxes for Hg species | Gross oxidation of Hg(0), gross reduction of Hg(II)$_g$ to Hg(0) | Monthly |
| Meteorological parameters | Air pressure, air temperature, specific humidity, air density, volume of a grid cell | Monthly |
| **2D variables** | | |
| Volume mixing ratio of Hg species (surface layer) | Hg(0), Hg(II)$_g$, Hg(II)$_p$ | Daily and monthly-averaged diurnal cycle |
| Mass concentration of Hg species (surface layer) | Hg(0), Hg(II)$_g$, Hg(II)$_p$ | Daily |
| Vertical column density of Hg species | Hg(0), Hg(II)$_g$, Hg(II)$_p$ | Daily |
| Wet deposition | THg | Daily |
| Dry deposition | Hg(0), Hg(II)$_g$, Hg(II)$_p$ | Daily |
| Mixing ratio of major reactants | Br, OH, Cl, O$_3$, PM$_{2.5}$, … | Daily |
| Emissions | Net oceanic flux of Hg(0), land reemissions of Hg(0) | Daily |
| Meteorological parameters (surface layer) | Temperature, precipitation, surface pressure, wind speed at 10 m height | Daily |
| Land surface parameters | Fraction of snow coverage in a grid cell, fraction of sea ice coverage in a grid cell | Daily |
| Land surface parameters | Model orography, fraction of high vegetation coverage in a grid cell, fraction of low vegetation coverage in a grid cell, fraction of bare land in a grid cell, fraction of water in a grid cell | Time-invariant |





**Table F12. Ocean model simulation output variables.**

| Type | Variables | Temporal frequency |
|---|---|---|
| **3D variables** | | |
| Concentration of Hg species in water | THg, Hg(0), Hg(II), MMHg, DMHg, DGM | Monthly/Annual |
| Concentration of Hg species on particles | THg, Hg(II), MeHg | |
| Concentration of Hg species in biota | THg, MeHg | |
| Concentration of Hg species on biota | THg, MeHg | |
| **2D variables** | | |
| Air-sea exchange | Hg(0), DMHg | Daily/Monthly/Annual |
| Surface gaseous Hg | Hg(0), DMHg, DGM | |
| Atmospheric concentrations (input) | Hg(II) | |
| Atmospheric deposition (input) | Hg(II) | Monthly /Annual |
| Hg loads in sediments | THg | |
| Net sedimentation | THg | |
| **Secondary variables** | | |
| Nutrients | N, P, Si, Fe | Monthly/Annual |
| Carbon cycle | Total, C, DOC, POC<br><br>Total = DOC + POC + biota<br><br>Net primary productivity | |
| Biological variables | Phytoplankton, Zooplankton, Macrobenthos | |
| Physical variables | velocity, salinity, temperature, wind speed, air temperature, surface shortwave radiation | |




*Data availability*. The anthropogenic emissions datasets analyzed in this study are available publicly [EDGAR: https://edgar.jrc.ec.europa.eu/dataset_4tox2 (Muntean et al., 2018); AMAP: https://www.amap.no/mercury-emissions and https://doi.org/10.34894/SZ2KOI (Steenhuisen and Wilson, 2019); Streets: https://ftp.as.harvard.edu/gcgrid/geos-
chem/data/ExtData/HEMCO/MERCURY (Streets et al., 2019b)].

*Author contributions*. AD led the overall design of the study and writing of the article with contributions from all authors, listed in alphabetical order with lead authors identified in **bold**: **AD**, **HA**, **JB**, **FB**, **BE**, **AF**, XF, **BG**, CG, YH, IMH, II, TK, **JK**, **CJL**, **IL**, **RM**, **DM**, MM, PR, EMR, **AR**, **NES**, FDS, **ALS**, FS, **OT**, SW, **XW**, SW, RW, **QW**, **YZ**, JZ, **WZ**, **SZ.**


*Competing interests.* The authors declare that they have no conflict of interest.

*Acknowledgements.* We thank J. Fisher for review and comments on the manuscript.

*Financial support.* QW was supported by the National Natural Science Foundation of China (No. 42394094, No. 2222604). AF was supported by the Swiss National Science Foundation (P2EZP2_195424), the US National Science Foundation (#1924148), and Horizon Europe (Marie Skłodowska-Curie grant agreement 101103544). Contribution of OT was funded by the EU GMOS-Train project of the European Union's Horizon 2020 research and innovation programme under the Marie Sklodowska-Curie grant agreement no. 860497 and co-funded by the ARRS Research Programme P1-0143 and J1-3033
(IsoCont). The views expressed are those of the authors and do not necessarily represent those of their employers or funding agencies.

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
