# Peer review of "The Multi-Compartment Hg Modeling and Analysis Project (MCHgMAP): Mercury modeling to support international environmental policy"

_Geoscientific Model Development, 2024_

## Author Comment (AC3)

Dear Editor,

Thank you very much for guiding the review of our manuscript. Thoughtful comments and suggestions from the three reviewers and Dr. Panagos are truly appreciated!  We have accordingly revised the manuscript.  Point-by-point responses to reviewers and Dr. Panagos are embedded in their comments below (italicized). We have also uploaded all emission datasets used in the paper to GMD trustable permanent archival repositories as required; the data availability section has been updated.

Thank you again for your guidance and support for the publication of our manuscript in GMD.

Best regards,

Ashu Dastoor and coauthors

**RC1: Anonymous Referee #1**

This paper presents the ambitious plan of the Multi-Compartment Hg Modeling and Analysis Project (MCHgMAP) for a multi-model ensemble approach to supporting the Minamata Convention as it seeks to assess the degree to which international efforts are achieving the convention's goals of reducing Hg in the environment. Since anthropogenic emissions to the atmosphere that lead to transboundary air pollution are the subject of another international treaty, the Convention on Long-Range Transboundary Air Pollution consideration is given to those policy goals as well.

The scope of the task envisioned by the MCHgMAP is huge. The length and the depth of the paper is commensurate to that task. Despite the paper's great length, and the complexity of the issues that it brings up, the paper has a clear structure that keeps many parts of the paper from getting in the way of each other. The text is also commendably well-written, with just a few places where some clarification is needed. The survey of the literature is similarly well-done, building confidence that the knowledge base going into the modeling exercise draws on much of the relevant literature – even though that literature is developing rapidly – which is a reflection of how the understanding of Hg cycling continues to develop.

*We thank the reviewer for the appreciation of the paper and insightful comments and suggestions, contributing to improving the paper.*

I have two major concerns, as well as some smaller points for the authors to consider.

1. The first of the major concerns regards the value of providing examples of similar efforts to use multi-model ensembles to support policy efforts. As a reader feeling overwhelmed by the scope of what is to be attempted, and trying to get a clearer vision of where the overall process presented in this paper is heading, it would be helpful to be given other examples where MME's have been applied. (I presume the work of the IPCC has such examples.) So please try to present what is envisaged for supporting the Minamata Convention in relation to what has been achieved in modeling other global scale pollution issues to support the policy process.

   *We have added following paragraph in the introduction taking reviewer's suggestion:*

*"Mechanistic models provide an effective framework for integrating and analyzing information across a scientific system and exploring uncertainties. In recent decades, coordinated model ensembles, such as the continuous climate Earth system modelling initiative Coupled Model Intercomparison Project (CMIP) (Eyring et al., 2016), have become essential tool to tackling key environmental science questions on its anthropogenic influences by facilitating multi-model experimental design, data (drivers and observations) development, numerical and infrastructure solutions and delivery of quantitative responses for the benefit of the research community and national and international assessments informing policy. The MCHgMAP (Multi-Compartment Hg Modeling and Analysis Project) is a multi-phase international multi-model initiative intended to consolidate variety of Hg models in coordinated ensembles supported by observational evaluation to develop better understanding of past, present and future mercury cycling and support the mercury MEAs such as the MC and LRTAP. Mercury models are susceptible to various sources of uncertainty, including defining the natural state, anthropogenic and environmental (physical and biochemical) forcings, process mechanisms and parameters and model formulations; challenging questions remain regarding how to design and interpret mercury MME within the constraints of current scientific understanding and computational power. The overarching goal of this discussion paper is to review recent advances in mercury science (sources, processes and models) and propose a synergistic multi-model experimental design to simulate and analyze short- and long-term response of mercury cycling to anthropogenic mercury and other forcings and determine scientific gaps and uncertainties. A novel feature of the MCHgMAP MME design is an approach to couple atmospheric, land, marine and multi-media mass balance models to perform harmonized simulations of Hg cycling across environmental compartments that account for the changing secondary Hg emissions and releases. The MCHgMAP is envisaged as a long-term initiative, each phase focusing on specific science or policy areas. The objectives of the first phase of the MCHgMAP are to analyze current global mercury spatial patterns and temporal trends to inform the first cycle of the MC EE as well as laying the groundwork for subsequent studies. This paper provides the required information to produce a consistent set of mercury model simulations that can be presently scientifically exploited to support the first cycle of the MC EE. Additionally, individual studies documenting the preparation of observational and emission (past and future) datasets along with assessment of their uncertainties are expected to be published, supporting the MCHgMAP initiative (GMD/ACP/BG inter-journal Special Issue 2023:* [https://gmd.copernicus.org/articles/special_issue1294.html)](https://gmd.copernicus.org/articles/special_issue1294.html).*"*

2. My other suggestion concerns Section 8 where I am missing treatment of the Terrestrial-Hydrological modeling. Earlier in the paper, three classes of models are presented (Atmosphere, Ocean and Terrestrial-Hydrological models (Sections 2.1-2.3, and the "Contemporary Cycle" in Figure 1.). These all seem to have a place in the lower part of Fig 1, the "Long-Term Cycle" of "Multi-Media Mass Balance models". But when one gets to "Section 8 Modeling analysis: process-understanding", there are two sections, 8.1 on atmospheric models and 8.2 on ocean models. The issues addressed for these two classes of models would have been appropriate for

Terrestrial-Hydrological Models as well. More generally, I find the presentation of the Terrestrial-Hydrological modeling as hard to understand fitting into the global scope of the MCHgMAP, since there are so many more potential compartments in the terrestrial-hydrological modeling domain compared to the ocean or atmosphere.

*Section 8 proposes model experiments to analyze the uncertainties and gaps in process understanding pertaining to abiotic global environmental Hg cycling that can be achieved via currently available global scale Hg models, i.e., 3D atmospheric and ocean models.  As the reviewer has noted, coupling of dynamic terrestrial-hydrological modeling to global Hg cycling is currently lacking in this work, primarily due to lack of such models that can be utilized on global scale. Recognizing this modeling gap, we have added the following two paragraphs in Section 5 (Coordinated multi-modeling design) and Section 8 (Modeling analysis: process understanding), respectively, to clarify the modeling plan, and highlight the importance and future needs of terrestrial Hg modeling.*

*"Terrestrial ecosystems play a major role in retaining and mobilizing deposited Hg across air-land-ocean interface; thus, accounting for the variations in terrestrial Hg occurring due to changes in anthropogenic emissions is important to global Hg cycling.  Additionally, alterations of land cover, biogeochemical and hydrological characteristics and meteorological conditions (driven by climate change and other anthropogenic influences) are influencing net terrestrial exchange of Hg with the atmosphere and downstream aquatic systems. Schaefer et al. (2020) estimated that thawing circumpolar arctic permafrost regions can lead to Hg reemissions comparable to the current global anthropogenic emissions by 2200 under high emission climate scenario. As noted in Section 2, spatially explicit air-terrestrial-hydrological modeling presently remains a major research challenge largely due to the biogeochemical heterogeneity and complexity of terrestrial ecosystems (Bishop et al. 2020). The influences of changes in meteorology (such as temperature, precipitation, soil moisture and snow cover), primary productivity and wildfires will be modeled in this work via surface-atmosphere parameterizations in atmospheric models using inputs from weather models and satellite-based products. Also, the changes in terrestrial legacy Hg emissions and releases driven by long-term changes in anthropogenic emissions are considered based on multi-media mass balance modeling. However, other important global change impacts such as related with permafrost thaw and hydrological modifications are not incorporated in the first set of model simulations.   Model scenarios to further investigate the effects of global change on environmental Hg levels will be developed under future scenarios in a later phase of the project utilizing current advances in terrestrial Hg."*

*"Limited progress on large scale process-based terrestrial-hydrological modeling is recognized as an outstanding science gap here. Further research to monitor and model terrestrial Hg is critical to determining lifetime and attribution (to anthropogenic and natural sources) of Hg in major terrestrial compartments and its coupling with atmospheric and marine Hg. Bishop et al. (2020)*

*reviewed recent advances in terrestrial Hg cycling and recommended increased routine analyses of terrestrial Hg and process studies using isotopes, micrometeorological and microbial techniques to characterize spatiotemporal variations across atmosphere-terrestrial-water interface. Also, increased measurements of Hg levels and mobilization properties in contaminated sites (e.g., industrial and mining operations including ASGM) are required to adequately represent their role in releasing Hg to the atmosphere and global surface waters. Sonke et al. (2023) recommended monitoring of Hg concentrations in major regional rivers and their tributaries to assess the integrated influences of global change on terrestrial Hg cycling. In the following sections, atmospheric and oceanic model simulations probing Hg process understanding are discussed."*

*Bishop, K., Shanley, J. B., Riscassi, A., de Wit, H. A., Eklöf, K., Meng, B., Mitchell, C., Osterwalder, S., Schuster, P. F., Webster, J., Zhu, W.: Recent advances in understanding and measurement of mercury in the environment: Terrestrial Hg cycling. Science of The Total Environment, 721, 137647 10.1016/j.scitotenv.2020.137647, 2020.*

*Schaefer, K., Elshorbany, Y., Jafarov, E., Schuster, P. F., Striegl, R. G., Wickland, K. P., and Sunderland, E. M.: Potential impacts of mercury released from thawing permafrost, Nat. Commun., 11, 1–6, https://doi.org/10.1038/s41467-020-18398-5, 2020.*

*Sonke, J. E., Angot, H., Zhang, Y., Poulain, A., Björn, E., and Schartup, A.: Global change effects on biogeochemical mercury cycling, Ambio, 52, 853–876, https://doi.org/10.1007/s13280-023-01855-y, 2023.*

Below are more minor points for the authors to consider:

1. Line 1615. There will be perturbation modeling and sensitivity modeling. It seemed hard for me to understand the need to differentiate these, since the two seem so closely related.

   *While the experimental methodology for the perturbation and sensitivity experiments is similar, the purpose is different. As defined, the perturbation experiments are designed to quantify source and process attribution of Hg levels and trends. Whereas the sensitivity experiments are designed to quantify uncertainties and sensitivity of model formulations and inputs to Hg levels and trends. The two terms serve the purpose of distinguishing these two categories of experiments in the paper.*

2. I have a sense that a major challenge for global Hg modeling is the apparent discrepancy between increasing anthropogenic emission inventories during recent decades, and the regional decline in atmospheric Hg over Europe. If this is indeed the case in the eyes of the authors, it would be nice to see this lifted up as an example of the type of issue that the MCHgMAP will address.

*Yes, this is a major challenge. We have added following text and reference in Section 7.1.2 (Temporal trends):*

*"In contrast to the decline in atmospheric Hg concentrations in recent decades at measurement sites in the Northern Hemisphere (NH), global anthropogenic emission inventories report increases in Hg emissions (e.g., AMAP/ UN Environnement, 2019). Atmospheric Hg models have been useful tools to assess the drivers of Hg trends and evaluate the accuracy of emission assumptions and biogeochemical cycle parameterizations. Zhang et al., (2016b) conducted GEOS-Chem simulations between 1990 and 2010 and compared with available observations globally and suggested potential refinements in both the magnitude and speciation of anthropogenic Hg emissions. Since that time, more Hg observations have been collected, continuously providing us with a better picture of global Hg trends. At the same time, trend detection and attribution methods for atmospheric compounds have become more sophisticated (Chang et al., 2021). Feinberg et al. (2024) reanalyzed the observed Hg(0) trend in recent decades and detected decline in most regions of the NH; applying comprehensive modeling, the authors suggest that the most likely driver for the declining Hg(0) trend in NH is decline in anthropogenic emissions (currently not reflected in global inventories). The factors contributing to the discrepancy between the trends in observed Hg(0) levels and global emission inventories will be further investigated using the proposed MME simulations."*

*Feinberg, N. E. Selin, C. F. Braban, K.-L. Chang, D. Custódio, D. A. Jaffe, K. Kyllönen, M. S. Landis, S. R. Leeson, W. Luke, K. M. Molepo, M. Murovec, M. G. Nerentorp Mastromonaco, K. A. Pfaffhubern, J. Rüdiger, G.-R. Sheu, and V. L. St.Louis: Unexpected anthropogenic emission decreases explain recent atmospheric mercury concentration declines. Proceedings of the National Academy of Sciences, 121(42), e2401950121, https://doi.org/10.1073/pnas.2401950121, 2024.*

3. In section 2.3, a number of terrestrial hydrological models are mentioned. I missed one published in Environmental Science and Technology that seems appropriate to include: Eklof, Karin, et al. "Parsimonious model for simulating total mercury and methylmercury in boreal streams based on riparian flow paths and seasonality." *Environ. Sci. Technol.* 49.13 (2015): 7851-7859.

   *We have included this reference in section 2.3 as follows:*

   *"Eklöf et al. (2015) developed various simple dynamic RIM (Riparian Profile Flow-Concentration Integration Model) models to simulate THg and MeHg in Boreal streams in Sweden demonstrating their ability to reproduce the observed variability in Hg and identify key drivers; such approaches can be utilized in upscaling terrestrial models."*

*Eklöf, K., Kraus, A., Futter, M., Schelker, J., Meili, M., Boyer, E. W., and Bishop, K.: Parsimonious model for simulating total mercury and methylmercury in boreal streams based on riparian flow paths and seasonality. Environ. Sci. Technol. 49.13, 7851-7859, 2015.*

4. Lines 1212-1218. Here the authors state that "It is currently not possible to determine the origin (anthropogenic/geogenic) or period of origin of the primary emission of Hg that are currently being reemitted from the ocean based on traditional measurements of Hg concentrations (for an explanation on how the Hg(0) evasion flux is determined see Sect. 4.4 and Appendix B). However, both box and 3D models can be used to gain insights into the extent of the Hg(0) evasion flux from the ocean, the fraction that different sources or historic periods contribute, and how concentrations will change in response to changes in Hg ocean inputs as a result of measures under the Hg MEAs.". This would seem like great place to mention the use of Hg isotopes to resolve these issues as Jiskra et al. did in their 2021 Nature publication. That article is cited in the first paragraph of the papers introduction (line 75), and one other place, but not in relation to where it is most relevant, specifically in this section at line 1212-1218. I am also concerned that the power of isotopes is not going to be brought to bear in the MCHgMAP as fully as they could be. But I expect that is something that will have to be wrestled with when looking at the results of the MCHgMAP.

*We thank the reviewer for pointing out the oversight of properly addressing the recent development in the Hg isotopes field in the section on ocean reemissions/air-sea exchange. We have now included text on this aspect in section 3.5 (Ocean reemissions) and section 4.4 (Air-sea exchange). The importance of using Hg isotope to determine observational constraints is furthermore already included in other sections on the observations like section 4.2 on Air-land Flux Exchange and section 4.5 on Environmental Archives. While we can use insights from Hg isotope studies to constrain fluxes across the air-sea interface, including Hg isotopes as variables in models is an issue for future development. This is currently most relevant for atmospheric models, that are more developed and where more isotopic measurements are available. Sections on future development of atmospheric and mass balance models also reflect on the importance of Hg isotopes for future work in sections 7.1.1.3 and 7.3. With the new additions to section 3.5 and 4.4 we feel that the manuscript adequately highlights the use of Hg isotopes in constraining sources and fluxes for this and future iterations of MCHgMAP/effectiveness evaluation.*

5. In the reference list there are three references where the first author is Li, C. But there are actually two different people involved. So good that their names are distinguished…

*Yes, the two different authors are distinguished in the reference list.*

6. Line 1506 begins a new paragraph. It was difficult to understand that this new paragraph had started taking about lake sediments. (The previous paragraph was about tree-rings).

*The previous paragraph explains the types of Hg records assimilated by different environmental archives. Next few paragraphs explain the methodology for collecting Hg records from each archive, starting from lake sediments. We have rephased the first sentence of this paragraph for the sake of clarity as follows:*

*"There are numerous published Hg sediment records spanning the past 100-200 years due to the ease of collecting "short cores" using gravity corers and the availability of $^{210}$Pb dating to derive ages and sediment accumulation rates (g m$^{-2}$ y$^{-1}$) with $^{137}$Cs dating often also used as an independent tracer to validate the $^{210}$Pb chronology (Blais et al., 1995)."*

**RC2: Anonymous Referee #2**

This reviewer commends the authors' endeavor to have produced such an ambitious proposal on multi-medium modeling and analysis to support the Minamata Convention and CLRTAP. The document is well structured for the most part, and the literature is fairly up to date.

*We thank the reviewer for the appreciation of the paper and insightful comments and suggestions, contributing to improving the paper.*

Below are a couple of main comments.

1. This document is essentially a proposal, which should be clearly reflected in the title, and in the same vein, the approach or the set of modeling experiments proposed were not exactly "developed", which implied evaluation or demonstration of validity and effectiveness.

   *We think that the paper title is suitable. The overview paper describes the development of multi-model ensemble methodology and experimental design that will be followed to achieve the model evaluation and Minamata effectiveness results. It is common to publish development of model experimental design such as the CMIP multi model ensemble experiments.*

2. The goal, objectives and the time periods for the present day and the future scenario should be presented upfront, rather than being buried two-thirds of the way through the document.

   *As suggested, we have added goal, objectives and simulation time periods in the introduction as follows:*
   *"The objective of MME experiments is to simulate the evolution of primary and secondary sources of Hg emissions and releases in the environment: (1) to generate spatially-resolved global Hg levels (concentrations and fluxes), filling monitoring gaps; (2) to detect their spatial gradients and temporal trends; (3) to attribute the levels and trends to emission sources and environmental drivers; (4) to quantify the impact of MC on Hg levels and trends; (5) to quantify uncertainty of model results; (6) to develop insights into future Hg cycling under different scenarios of implementation of the MC and other MEAs; and (7) to improve the understanding of environmental Hg processes and model representations. Since anthropogenic mercury emission inventories as well as observations are better defined starting around 2010, the first phase of multi-model simulations are proposed for the period 2010 – 2020 using harmonized mercury*

*emissions and releases to analyze Hg temporal trends from the period beginning prior to the MC (circa 2010) and extending to as close as possible to the availability of datasets. In addition, future projections of Hg levels from 2020 to 2050 are proposed using available future Hg emissions scenarios (Brocza et al. 2024)."*

3.  Did I miss whether the physical input data for all models were to be harmonized? For example, interannual variability in meteorology can significantly impact transport, deposition, chemistry, and emissions/releases. To quantify the impact of changing emissions, the influence of physical parameters needs to be assessed.

    *Physical input data such as meteorology will not be harmonized; this is mentioned in Table F5 (Global atmospheric model drivers) and respective sections. Often physical inputs are produced by weather forecasting models or simulated online within the same model such as GEM-MACH-Hg (one of the global models being used in MCHgMAP). It is usually challenging to use harmonized physical variables in all models. However, as noted by the reviewer, we will assess the influence of interannual variability in physical variables on Hg temporal trends (see "nature simulation" A3a.1 in Table F8: Atmospheric multi-model ensemble simulations); this assessment will include examination of the sensitivity of simulated natural state of Hg trend (i.e., without the influence of changing primary anthropogenic emissions) to the uncertainty in natural parameters by making use of the spread between models.*

4.  The difference between "perturbation" and "sensitivity" runs appeared to be rather subtle (L1615-1621). The latter seemed to encompass the former. They might need to be better defined.

    *While the experimental methodology for the perturbation and sensitivity experiments is similar, the purpose is different.  As defined, the perturbation experiments are designed to quantify source and process attribution of Hg levels and trends.  Whereas the sensitivity experiments are designed to quantify uncertainties and sensitivity of model formulations and inputs to Hg levels and trends.  The two terms serve the purpose of distinguishing these two categories of experiments in the paper. We think, above is clear in our definitions.*

5.  Sections 5 and 7 included not only atmospheric and ocean models but also other environmental mass balance models. Sections 6 and 8 appeared to include atmospheric and ocean models only.

    *The simulations, evaluation, and uncertainty analysis of mass balance modeling is discussed in one section (Sect. 5.3).  Since atmospheric and ocean models are 3D distributed models, extensive validation and uncertainty analysis using direct observations is possible for these models, which is detailed in sections 6 and 8.*

6.  Provide a glossary of terminology used in the document.

*We have moved the terminology definitions to the beginning of the paper as suggested later by the reviewer. We think this is sufficient.*

Next are specific comments:

1. Abstract: L66, L67: Instead of "is developed", it might be more appropriate to say "is proposed".

   *Revised as suggested.*

2. L156: What is the difference between "emission" and "release"?

   *In mercury science literature, emissions and releases are understood as "releases to air" and "releases to land and water", respectively. For the sake of clarity, we have rephrased the first sentence of the introduction as follows:*
   *"The presence and levels of mercury (Hg) in environmental matrices of concern are associated with its primary atmospheric emissions (to air) and releases (to land and water) from anthropogenic and geogenic sources and environmental residence time."*

3. L201-202: For regional models, BCs can be the most dominant factors in determining GEM concentrations.

   *This sentence is meant to present the modeling approaches used for inter-media Hg exchange in models.*

4. L202: There is a "constant" ring to "prescribe", but that's not exactly how emission fluxes are handled in some models.

   *The sentence presents two options used in atmospheric models, which are prescribed emission fluxes or formulations based on prescribed surface concentrations of Hg from land and oceans.*

5. L224-226: A most recent version of CMAQ-new-Hg-Br v2 included a newly developed gas-particle partitioning scheme with most up-to-date gas-phase redox reactions (Wu et al., 2024, JAMES, https://doi.org/10.1029/2023MS003823), which is probably one of the latest breakthroughs in atmospheric Hg modeling.

   *We have added the reference to this latest version of CMAQ-new-Hg Br v2 and have updated the text to reflect the latest updates:*
   *"In parallel to global models, regional 3D Hg models have also been developed such as the Community Multiscale Air Quality (CMAQ-Hg) model (Bullock and Brehme, 2002), which in its latest development CMAQ-newHg-Br v2 includes a new scheme for gas-particle partitioning of mercury, replacing earlier empirical parametrizations (Wu et al., 2024) (see Table F1)."*

*Wu, L., Mao, H., Ye, Z., Dibble, T. S., Saiz-Lopez, A., and Zhang, Y.: Improving Simulation of Gas-Particle Partitioning of Atmospheric Mercury Using CMAQ-newHg-Br v2, J Adv Model Earth Syst, 16, e2023MS003823, https://doi.org/10.1029/2023MS003823, 2024.*

6. L394: What does the term "secondary Hg emission" refer to? Legacy emission? Release? Or chemically formed?

   *We have moved these definitions at the beginning of the paper at the top of Section 2 after introduction as suggested by the reviewer below.*

7. L472-477: These definitions should be presented upfront, instead of being buried here after having already used such terms in the preceding text. Or provide a glossary as suggested in one of my main comments.

   *These definitions have been moved to the beginning of the paper as suggested.*

8. L779-L784: This part is confusing. It was said that 2010 should be used as the baseline year. Then later, in scenario design, the period of 2010-2020 was said to be the present-day period.

   *We have replaced the words "baseline year" with "starting year" in this sentence for the sake of clarity.*

9. L846: What is the "mass-flow" approach? Add a reference.

   *We have added the definition for mass-flow approach and a reference in this sentence as suggested:*
   *"To evaluate the impact of control measures, the mercury mass-flow approach (i.e., tracking mercury mass from sources through various stages of a given human activity process to its final emission or release into the environment (Hui et al. 2017)) is the most suitable method."*

   *Hui, M. L.; Wu, Q. R.; Wang, S. X.; Liang, S.; Zhang, L.; Wang, F. Y.; Lenzen, M.; Wang, Y. F.; Xu, L. X.; Lin, Z. T.; Yang, H.; Lin, Y.; Larssen, T.; Xu, M.; Hao, J. M., Mercury flows in China and global drivers. Environ. Sci. Technol. 2017, 51, 222-231.*

10. L944: Under "Scenario design", there's no scenario design. It's unclear what was meant to be shown there.

    *"Scenario design" explains one of the areas (the time period and spatiotemporal resolution of the future emission scenarios) where the published "future Hg emissions inventories", summarized in Tables E8 and E9, differ from each other. For clarity, we have revised the title to "Scenario time period and resolution".*

11. L1124-1127: On what was this statement based? This statement is valid only if those parameterizations were correct, but are they? It seems unclear which causes larger uncertainty: the air-surface exchange models or scaling up measured fluxes, unless work is done to compare the two.

*We agree with the reviewer that at the current state of the understating soil Hg evasion, it is difficult to compare uncertainties between modeling estimates and scale-up calculation using observational data. Thus, we have revised the text as follows:*
*"Using air-surface exchange models is another approach to estimate Hg reemission from terrestrial ecosystems. In contrast to scaling up field measurements, the air-surface exchange model is based on parameterization of physicochemical factors that influence Hg reemission fluxes at various temporal and spatial scales."*

12. L1298-1299: It's such a popular result. The dry deposition flux of Hg° may exceed that of HgII, but it doesn't make the former more "important" than the latter. The question is how much of the deposited Hg° gets oxidized.

*We have removed this sentence from here, since $Hg^0$ exchange processes with land and ocean surfaces are discussed in later sections.*

13. L1584: Present: 2010-2020; Future: Spell out the simulation period. It was said to be "up to 2050" - unclear.

*We have revised the sentence to indicate the simulation time period for future scenarios as below:*
*"Future projections of Hg levels in air and ocean from 2020 to 2050 will be simulated using the proposed set of future Hg emissions scenarios (see Sect. 3.2.2)."*

14. L1605-1613: Simulation objectives: Came too late and buried.

*We have revised the introduction to include the simulation objectives.*

15. L1625: What is "an analogous strategy"?

*These sentences have been revised for clarity as follows:*
*"Below we describe the proposed approach for the coordination of MME simulations between atmospheric, oceanic, and mass-balance models in the context of analyzing recent Hg levels and trends. Coupled atmosphere-ocean simulations for future scenarios will be performed using an analogous strategy as used to conduct the simulations for the recent Hg trends."*

16. L1807: Why interpolate twice?

    *The model results will not be interpolated twice. For the purpose of inter-model comparison, all model results will be converted to a common latitude-longitude grid. Whereas, for the purpose of model evaluation with station data, model results will be directly interpolated to station locations.*

17. L1948-1949: "for different years of the period 2010-2020 to reveal their temporal changes and causes" - It should probably just be said as "interannual variability and its driving factors".

    *We have revised the sentence as suggested:*
    *"The analysis will be performed for different years of the period 2010-2020 to reveal their interannual variability and its driving factors."*

18. L1950: "On the annual basis and for different seasons" – Awkward wording.

    *The wording has been revised to "as annual and seasonal averages."*

19. L2271: "Coupled offline": If it's coupled, then it's online. If it's offline, then it's decoupled.

    *We have revised the statements as follows:*
    *"Finally, in this study independent models for atmosphere and ocean will be coupled through sequential iterations (see Section 5.1). In future EE efforts, fully interactive multi-media models should be developed and applied."*

**RC3: Anonymous Referee #3**

This manuscript provides a very ambitious and comprehensive description of the current status of global multi-compartment modelling of mercury and the key anthropogenic and environmental processes that are necessary to describe in the modelling. The initiative and the ambitious approach is driven by a strong policy need and thus represents an important step to provide a scientifically sound approach to answer policy-relevant questions and to provide support for developing common and realistic expectations of effects of emission reductions as well as for setting common and realistic policy targets. Descriptions of the relevant policy frameworks (Minamata Convention, LRTAP convention) are included as are the direct policy related questions posed by these frameworks.

The manuscript provides an excellent state-of-the-art review of mercury (occurrence, transformations, trends etc.) in the atmosphere and oceans, and to some extent for terrestrial systems, and provides a basis for designing the multi-compartment modelling which is described in the following sections. Uncertainties and lack of knowledge are also included.

A main part of the manuscript contains a description of the plans for modelling activities and how the results will be evaluated. This is quite lengthy but necessary to provide the scientific community to engage, comment/criticize and potentially contribute to the planned activities.

In summary, the opinion of this reviewer is that the publication of this manuscript will make an important contribution to the scientific understanding and development of models for global mercury cycling, and to supporting policy efforts to reduce mercury emissions and impacts globally.

*We thank the reviewer for the appreciation of the paper and insightful comments and suggestions, contributing to improving the paper.*

Some specific comments/suggestions are:

1. An important factor not described in detail is the importance of "permanent" sinks for mercury (oceanic sediments, deep soil layers etc.), i.e. what is the current status of knowledge of these sinks, are they permanent and how can they be described in the models? From a policy perspective, a common understanding of the global permanent sinks and how important they are for setting the pace of reducing environmental mercury burdens in a scenario of emission reductions would be beneficial.

   *We thank the reviewer for raising the policy question regarding the role of permanent sinks in reducing environmental mercury burden in an emission reduction scenario. Permanent Hg sinks are indeed important to the recovery of Hg cycling in surface environments. Permanent Hg burial is mentioned at the beginning of Section 2 (Environmental mercury models) in the revised manuscript; also, these sinks are discussed in the context of multi-media mass balance models (Sections 2.4, 2.6), multi-media mass balance simulations (Section 5.3) and Hg records in environmental natural archives (Section 4.5). In MCHgMAP, we account for the permanent burial in terrestrial and oceanic ecosystems and long-term changes in environmental Hg cycling by making use of mass balance models (Section 5.3); we define the inter-media mass transfer rates in mass balance models based on the detailed 3D atmospheric and ocean models and observations. In addition, we use trends from long term Hg records from environmental archives such as lake sediments, ice cores, peats and tree rings (Section 4.5) to constrain the mass balance models.*
   *In addition to the proposed long-term mass balance Hg modeling simulations, we have added additional simulation (see simulation M19, Table F10) to investigate the time scale for the full recovery of global Hg cycling under a zero anthropogenic emissions and releases scenario.*

2. In general, the current understanding of terrestrial mercury cycling/storage and factors that influence the mobility/releases of mercury to the aquatic compartments are very limited in comparison to the atmospheric and (to some extent) oceanic cycling and, consequently, the possibilities for modelling are limited. This is also reflected in the initial state-of-the-art sections. Nevertheless, a description of how mercury in the terrestrial compartment will be handled in the modelling activities would be beneficial. Soil-air, and vegetation-air exchange is covered but

releases of soil-bound mercury to surface waters are not. For instance, land-use change (e.g. forestry) and climate (increased precipitation) are factors that can influence mercury (and methylmercury) burdens in freshwater and coastal systems and thus cause increased mercury concentrations in fish and seafood, the primary exposure route for most population groups.

*As noted by the reviewer, analysis of the impact of changes in terrestrial-hydrological Hg on global Hg cycling is currently limited in this work, primarily due to lack of such models that can be utilized on global scale. Recognizing this modeling gap, we have added the following two paragraphs in Section 5 (Coordinated multi-modeling design) and Section 8 (Modeling analysis: process understanding), respectively, to clarify the modeling plan, and highlight the importance and future needs of terrestrial Hg modeling.*

*"Terrestrial ecosystems play a major role in retaining and mobilizing deposited Hg across air-land-ocean interface; thus, accounting for the variations in terrestrial Hg occurring due to changes in anthropogenic emissions is important to global Hg cycling.  Additionally, alterations of land cover, biogeochemical and hydrological characteristics and meteorological conditions (driven by climate change and other anthropogenic influences) are influencing net terrestrial exchange of Hg with the atmosphere and downstream aquatic systems. Schaefer et al. (2020) estimated that thawing circumpolar arctic permafrost regions can lead to Hg(0) reemissions comparable to the current global anthropogenic emissions by 2200 under high emission climate scenario. As noted in Section 2, spatially explicit air-terrestrial-hydrological modeling presently remains a major research challenge largely due to the biogeochemical heterogeneity and complexity of terrestrial ecosystems (Bishop et al. 2020). The influences of changes in meteorology (such as temperature, precipitation, soil moisture and snow cover), primary productivity and wildfires will be modeled in this work via surface-atmosphere parameterizations in atmospheric models using inputs from weather models and satellite-based products. Also, the changes in terrestrial legacy Hg emissions and releases driven by long-term changes in anthropogenic emissions are considered based on multi-media mass balance modeling. However, other important global change impacts such as related with permafrost thaw and hydrological modifications are not incorporated in the first set of model simulations.   Model scenarios to further investigate the effects of global change on environmental Hg levels will be developed under future scenarios in a later phase of the project utilizing current advances in terrestrial Hg."*

*"Limited progress on large scale process-based terrestrial-hydrological modeling is recognized as an outstanding science gap here. Further research to monitor and model terrestrial Hg is critical to determining lifetime and attribution (to anthropogenic and natural sources) of Hg in major terrestrial compartments and its coupling with atmospheric and marine Hg. Bishop et al. (2020) reviewed recent advances in terrestrial Hg cycling and recommended increased routine analyses of terrestrial Hg and process studies using isotopes, micrometeorological and microbial techniques to characterize spatiotemporal variations across atmosphere-terrestrial-water*

*interface. Also, increased measurements of Hg levels and mobilization properties in contaminated sites (e.g., industrial and mining operations including ASGM) are required to adequately represent their role in releasing Hg to the atmosphere and global surface waters. Sonke et al. (2023) recommended monitoring of Hg concentrations in major regional rivers and their tributaries to assess the integrated influences of global change on terrestrial Hg cycling. In the following sections, atmospheric and oceanic model simulations probing Hg process understanding are discussed."*

*Bishop, K., Shanley, J. B., Riscassi, A., de Wit, H. A., Eklöf, K., Meng, B., Mitchell, C., Osterwalder, S., Schuster, P. F., Webster, J., Zhu, W.: Recent advances in understanding and measurement of mercury in the environment: Terrestrial Hg cycling. Science of The Total Environment, 721, 137647 10.1016/j.scitotenv.2020.137647, 2020.*

*Schaefer, K., Elshorbany, Y., Jafarov, E., Schuster, P. F., Striegl, R. G., Wickland, K. P., and Sunderland, E. M.: Potential impacts of mercury released from thawing permafrost, Nat. Commun., 11, 1–6, https://doi.org/10.1038/s41467-020-18398-5, 2020.*

*Sonke, J. E., Angot, H., Zhang, Y., Poulain, A., Björn, E., and Schartup, A.: Global change effects on biogeochemical mercury cycling, Ambio, 52, 853–876, https://doi.org/10.1007/s13280-023-01855-y, 2023.*

3. A large part of the manuscript concerns the ambitious plans for modelling activities and evaluation of results. As mentioned above, this is very positive, but it would also be of interest for the reader (and for potential contributors) to understand if these plans are fixed and funded or if they are more of a long-term ambition, e.g. a brief description of the MCHgMAP initiative.

   *The MCHgMAP is not a fixed initiative; the project is expected to evolve and continue in phases to address different objectives. To address reviewer's concern, we have added following paragraph in the introduction to clarify the MCHgMAP initiative:*

   *"Mechanistic models provide an effective framework for integrating and analyzing information across a scientific system and exploring uncertainties. In recent decades, coordinated model ensembles, such as the continuous climate Earth system modelling initiative Coupled Model Intercomparison Project (CMIP) (Eyring et al., 2016), have become essential tool to tackling key environmental science questions on its anthropogenic influences by facilitating multi-model experimental design, data (drivers and observations) development, numerical and infrastructure solutions and delivery of quantitative responses for the benefit of the research community and national and international assessments informing policy. The MCHgMAP (Multi-Compartment Hg Modeling and Analysis Project) is a multi-phase international multi-model initiative intended to consolidate variety of Hg models in coordinated ensembles supported by observational evaluation to develop better understanding of past, present and future mercury cycling and*

*support the mercury MEAs such as the MC and LRTAP. Mercury models are susceptible to various sources of uncertainty, including defining the natural state, anthropogenic and environmental (physical and biochemical) forcings, process mechanisms and parameters and model formulations; challenging questions remain regarding how to design and interpret mercury MME within the constraints of current scientific understanding and computational power. The overarching goal of this discussion paper is to review recent advances in mercury science (sources, processes and models) and propose a synergistic multi-model experimental design to simulate and analyze short- and long-term response of mercury cycling to anthropogenic mercury and other forcings and determine scientific gaps and uncertainties. A novel feature of the MCHgMAP MME design is an approach to couple atmospheric, land, marine and multi-media mass balance models to perform harmonized simulations of Hg cycling across environmental compartments that account for the changing secondary Hg emissions and releases. The MCHgMAP is envisaged as a long-term initiative, each phase focusing on specific science or policy areas. The objectives of the first phase of the MCHgMAP are to analyze current global mercury spatial patterns and temporal trends to inform the first cycle of the MC EE as well as laying the groundwork for subsequent studies. This paper provides the required information to produce a consistent set of mercury model simulations that can be presently scientifically exploited to support the first cycle of the MC EE. Additionally, individual studies documenting the preparation of observational and emission (past and future) datasets along with assessment of their uncertainties are expected to be published, supporting the MCHgMAP initiative (GMD/ACP/BG inter-journal Special Issue 2023: https://gmd.copernicus.org/articles/special_issue1294.html)."*

**CC1: Dr. Panos Panagos**

This is a very interesting manuscript which addresses the whole Hg cycle in all media.

As expert in soils, I would recommend that the Hg content in soils is better addressed. Reading the manuscript, you tend to explain the Hg content in soils due to natural factors while there is an important contribution by anthropogenic sources. In EU, we have done a very detailed assessment of Hg content in topsoils trying to explain both the natural and the anthropogenic drivers.

This can be done taking into account 22,000 measured samples taken with LUCAS topsoil survey in Europe. More insights can be given by the recent work Ballabio et al 2019. A spatial assessment of mercury content in the European Union topsoil.

In this assessment, authors can consider the findings of the large EU study as this is the largest harmonized and recent measured Hg campaign:

- large proportion of atmospheric Hg that is bound to soil organic matter.

- In addition, the findings indicate that soil Hg is not only bound to organic matter but associated with iron oxide and clay minerals.

- Hg accumulation increases with pH, starting from a pH of 4 and reaching a maximum at a pH of 6.5.

- Parent material also increase Hg. Highest Hg content are found in mineralized regions characterized by subduction zones and volcanic deposits.

- High correlation with NDVI - dense vegetation (higher concentration in forests and grasslands compared to croplands).

- Elevated temperatures reduce activation energy required to release Hg from soil or vegetation and subsequently increase Hg volatilisation rates to the atmosphere

- However, the outliers are explained by anthropogenic activities. 1% of the samples show very high Hg values due to proximity to mining sites. Particular high Hg values have been found close to Almaden (spain), Monte Amiata (Italy) and Idrija mine in Slovenia.

- Another source of Hg is represented by coal combustion in power plants in the European Union.

- Also, sampling points close to active or past Chrol-alkali plants have shown very high Hg values.

Important to notice the impact of erosion and sediment removal to transfer Hg in aquatic systems (oceans, river-basins, lakes).

Future Projections for Hg in topsoils are mainly positive as the decrease of coal combustion and the better treatment in Chrol alkali plants have positive effects. In addition, specific control technologies and legal binding regulations (in the European Union, Soil Monitoring Law, Mercury Regulation) will have positive impact in reducing Hg concentrations.

Many thanks for taking those remarks into account.

*We thank Dr. Panagos for the appreciation of the paper and insightful comments and suggestions, contributing to improving the paper.*

*Indeed, the observational Hg data through the LUCAS topsoil survey in Europe represents the most comprehensive analysis of soil Hg content in Europe. We recognize that it is difficult to address global modeling needs with the level of spatial trend and resolution as presented in Ballabio et al. (2021). That said, we concur with Dr. Panagos's insights and have added additional discussion with adequate references in Section 3.4 (Soil reemissions):*

*"Field measurements suggest that soil Hg concentration and solar irradiance are the dominant driving factors for Hg evasion from bare land and polluted soils, while other factors can influence the Hg fluxes at various degrees depending on the environmental conditions at the measurement sites (Carpi and Lindberg, 1997; Lindberg et al., 2007; Gustin et al., 2008; Agnan et al., 2016; Zhu et al., 2016). Therefore, data on Hg concentration in the topsoil are essential for reliable estimates of Hg reemission from soils in various ecosystems. Efforts to quantify global soil Hg concentrations have been made through empirical*

*modeling, such as the Hg/carbon ratio, linear regression, and nonlinear models with the estimation of 240–1,100 Gg Hg stored within the top 20 or 30 cm of soil layers worldwide (Liu et al., 2023; Smith-Downey et al., 2010; Wang et al., 2019). These estimates are, however, associated with large uncertainties since the spatial variability of soil Hg is controlled by multiple natural factors such as climate conditions, soil physiochemical properties and primary productivity as well as anthropogenic activities (Ballabio et al., 2021). In general, most global soil Hg models lack adequate representation of Hg contaminated regions and do not include a comprehensive set of parameters that influence air-soil exchange fluxes because measurements often only provide a subset of the parameters (Guo et al., 2024; Wang et al., 2019). Extensive topsoil Hg surveys in China, U.S. and Europe analyzed factors that influence soil Hg concentration and spatial distribution and have reported hotspots of soil Hg distribution (e.g., mining and industrial pollution regions) (Ballabio et al., 2021; Panagos et al., 2021; Olson et al., 2022; Wang et al., 2019). These assessments and measurements are valuable in developing and constraining models to attain a more representative estimate and distribution of global soil Hg content."*

*Ballabio, C., M. Jiskra, S. Osterwalder, P. Borrelli, L. Montanarella, and P. Panagos (2021), A spatial assessment of mercury content in the European Union topsoil, Sci Total Environ, 769, 144755, doi:10.1016/j.scitotenv.2020.144755.*

*Guo, W., et al. (2024), Warming-Induced Vegetation Greening May Aggravate Soil Mercury Levels Worldwide, Environ Sci Technol, doi:10.1021/acs.est.4c01923.*

*Liu, Y. R., L. Guo, Z. Yang, Z. Xu, J. Zhao, S. H. Wen, M. Delgado-Baquerizo, and L. Chen (2023), Multidimensional Drivers of Mercury Distribution in Global Surface Soils: Insights from a Global Standardized Field Survey, Environ Sci Technol, 57(33), 12442-12452, doi:10.1021/acs.est.3c04313.*

*Olson, C. I., B. M. Geyman, C. P. Thackray, D. P. Krabbenhoft, M. T. Tate, E. M. Sunderland, and C. T. Driscoll (2022), Mercury in soils of the conterminous United States: patterns and pools, Environmental Research Letters, 17(7), doi:ARTN 07403010.1088/1748-9326/ac79c2.*

*Panagos, P., Jiskra, M., Borrelli, P., Liakos, L., and Ballabio, C.: Mercury in European topsoils: Anthropogenic sources, stocks and fluxes, Environ. Res., 111556, https://doi.org/10.1016/j.envres.2021.111556, 2021.*

*Smith-Downey, N. V., E. M. Sunderl, and D. J. Jacob (2010), Anthropogenic impacts on global storage and emissions of mercury from terrestrial soils: Insights from a new global model, Journal of Geophysical Research Atmospheres, 115(G3), 227-235.*

*Wang, X., W. Yuan, C. J. Lin, L. Zhang, H. Zhang, and X. Feng (2019), Climate and Vegetation As Primary Drivers for Global Mercury Storage in Surface Soil, Environ Sci Technol, 53(18), 10665-10675, doi:10.1021/acs.est.9b02386.*

---

## Author Response (AR2)

Dear Editor,

Thank you very much for guiding the review of our manuscript. The new suggestion is appreciated! We have accordingly revised the manuscript. Please find our response below.

Thank you again for your guidance and support for the publication of our manuscript in GMD.

Best regards,

Ashu Dastoor and coauthors

**Referee's comment:**

The manuscript provides a comprehensive state-of-art review of multi-compartment mercury (Hg) models and their key inputs and constraints, and proposes the ambitious MCHgMAP multi-model research initiative for the effectiveness evaluations of the Minamata Convention (MC) and the Convention on Long-Range Transboundary Air Pollution (LRTAP). The manuscript is well organized and provides useful information to broad audience in the global Hg community. The authors have also done a great job revising the manuscript according to the referees' comments. Overall in my opinion, the manuscript is acceptable for publication on Geoscientific Model Development after minor revision. I have one more comment for the authors to consider:

Section 4.1: Although reactive Hg monitoring is not prioritized for the effectiveness evaluation, concentrations of reactive Hg are crucial constraints for the atmospheric models. Recent advances in reactive Hg measurements should be briefly summarized in this section. The following literatures could be utilized.

**Response:** We thank the reviewer for appreciating the paper and for providing an excellent suggestion (and relevant papers) to include reactive Hg advances in section 4.1. We have accordingly added the following text and references in this section on the topic of reactive Hg (i.e., oxidized Hg or Hg(II) = Hg(II)g + Hg(II)p, these are the terms used in the paper).

*"For constraining atmospheric chemistry in models, robust and high temporal-resolution measurements of oxidized Hg (gaseous and particulate) concentrations and characterization of its chemical compounds under all environmental conditions are needed. However, currently deployed instruments for Hg(II) observations have significant artifacts (Gustin et al., 2021; Dunham-Cheatham et al. 2023). KCl-coated denuders, the predominant method to measure Hg(II)g, has been shown to be biased low by up to 50%; its collection efficiencies for various Hg(II) compounds depend on environmental conditions and local chemistry. Hg(II)p is typically collected on filters, such as quartz fiber and polytetrafluoroethylene (PTFE), which are found to also collect Hg(II)g.*

*Recent advances in oxidized Hg measurement methods such as Dual-channel system (DCS) and The Reactive Mercury Active System (RMAS) (e.g., Lyman et al. 2020; Elgiar et al. 2024) allow for the development and deployment of new Hg(II) measurement systems in the field, which can improve the measurement accuracy and elucidate Hg(II) compounds involved in atmospheric chemistry across space and time. Gustin et al. (2024) recently reviewed the advances and limitations of current ambient Hg*

*measurements and suggested that future work should focus on the development of following: better surfaces for collecting oxidized Hg compounds, analytical methods to characterize Hg chemistry, methods for differentiating between Hg(II)g and Hg(II)p, and high time-resolution calibrated measurement systems."*

Gustin, M. S., Dunham-Cheatham, S. M., Lyman, S., Horvat, M., Gay, D. A., Gačnik, J., Gratz, L., Kempkes, G., Khalizov, A., Lin, C.-J., Lindberg, S. E., Lown, L., Martin, L., Mason, R. P., MacSween, K., Nair, S. V., Nguyen, L. S. P., O'Neil, T., Sommar, J., Weiss-Penzias, P., Zhang, L., and Živković, I.: Measurement of atmospheric mercury: current limitations and suggestions for paths forward, Environ. Sci. Technol., 58(29), 12853–12864, 2024.

Elgiar, T. R., Lyman, S. N., Andron, T. D., Gratz, L., Hallar, A. G., Horvat, M., Vijayakumaran Nair, S., O'Neil, T., Volkamer, R., and Zivkovic, I.: Traceable calibration of atmospheric oxidized mercury measurements, Environ. Sci. Technol., 58(24), 10706–10716, 2024.

Dunham-Cheatham, S. M.; Lyman, S.; Gustin, M. S. Comparison and calibration of methods for ambient reactive mercury quantification. *Sci. Total Environ. 856*, 159219, 2023.

Gustin, M. S., Dunham-Cheatham, S. M., Huang, J., Lindberg, S., and Lyman, S. N.: Development of an understanding of reactive mercury in ambient air: a review, Atmosphere, 12(1), 73, 2021.

Lyman, S. N., Gratz, L. E., Dunham-Cheatham, S. M., Gustin, M. S., and Luippold, A.: Improvements to the accuracy of atmospheric oxidized mercury measurements, Environ. Sci. Technol., 54(21), 13379–13388, 2020.